# Statistical-Computational Tradeoffs in High-Dimensional Single Index Models

**Lingxiao Wang**[*]      **Zhuoran Yang**[†]      **Zhaoran Wang**[‡]

## Abstract

We study the statistical-computational tradeoffs in a high dimensional single index model $Y = f(X^\top \beta^*) + \epsilon$, where $f$ is unknown, $X$ is a Gaussian vector and $\beta^*$ is $s$-sparse with unit norm. When $\mathsf{Cov}(Y, X^\top \beta^*) \neq 0$, [43] shows that the direction and support of $\beta^*$ can be recovered using a generalized version of Lasso. In this paper, we investigate the case when this critical assumption fails to hold, where the problem becomes considerably harder. Using the statistical query model to characterize the computational cost of an algorithm, we show that when $\mathsf{Cov}(Y, X^\top \beta^*) = 0$ and $\mathsf{Cov}(Y, (X^\top \beta^*)^2) > 0$, no computationally tractable algorithms can achieve the information-theoretic limit of the minimax risk. This implies that one must pay an extra computational cost for the nonlinearity involved in the model.

## 1  Introduction

A single index model (SIM) specifies that the response $Y$ and the covariate $X$ satisfy $Y = f(X^\top \beta^*) + \epsilon$, where $\beta^* \in \mathbb{R}^d$ is an unknown parameter, $f \colon \mathbb{R} \to \mathbb{R}$ is an unknown link function, and $\epsilon \in \mathbb{R}$ is a random noise. This model extends linear regression by incorporating the unknown link function, offers additional modeling flexibility and robustness to model misspecification. SIMs are extensively studied in the literature, with wide applications such as time-series [17], survival analysis [35], and quantile regression [56].

Given $n$ i.i.d. observations of this model, the primary focus is to estimate the parametric component $\beta^*$ without knowing the exact form of $f$. When $\beta^*$ is estimated accurately, $f$ can be fitted via univariate nonparametric regression. Recently, there is growing research interest in recovering $\beta^*$ in the high-dimensional setting where the dimensionality $d$ is much larger than the sample size $n$ and $\beta^*$ is sparse. When $Y$ and $X^\top \beta^*$ have nonzero correlation, [43, 44] propose to estimate $\beta^*$ by fitting an $\ell_1$-regularized linear model, i.e., Lasso [50], directly using $Y$ and $X$. More interestingly, they also establish similar theoretical guarantees as those for the linear model. Specifically, they show that the Lasso estimator is consistent as long as the sample size is of the order $s \log d$, where $s$ is the number of nonzero entries in $\beta^*$. Moreover, this sample complexity result is known to be optimal in the sense that it attains the information-theoretical lower bound [46, 53], and the proposed estimator can be obtained efficiently using convex optimization. However, the Lasso approach fails when $Y$ and $X^\top \beta^*$ are uncorrelated, which is the case when the link function is symmetric. A prominent example is phase retrieval [10, 11], where $f$ is known to be either the absolute value or quadratic function. For sparse phase retrieval, $s \log d$ sample complexity is only attained by the empirical risk minimizer

---

[*]Northwestern University; `lingxiaowang2022@u.northwestern.edu`

[†]Princeton University; `zy6@princeton.edu`

[‡]Northwestern University; `zhaoranwang@gmail.com`

[33], which searches over all $\binom{d}{s}$ possible support sets of $\beta$, and is thus computationally intractable. In addition, various efficient estimators are proposed based on convex relaxation or projected gradient descent [8, 13], whose consistency is only shown when the sample size is of the order $s^2 \log d$. Thus, there seems an interesting tradeoff between the statistical optimality and computational efficiency, i.e., there is a gap between the optimal statistical performance achieved by the family of computationally efficient estimators and that attained by all possible estimators. In sparse phase retrieval, such a gap is conjectured to be fundamental [8] and is also observed in SIMs where $f$ is symmetric [42, 48, 62].

This intriguing phenomenon motivates the following two questions: (i) How does the unknown link function affect the statistical and computational aspects of learning SIMs in high dimensions? (ii) Are the gap observed in symmetric links intrinsic and cannot be eliminated by more sophisticated algorithm design and analysis?

For the first question, we introduce the notions of first- and second-order Stein's associations which characterize the dependence between $Y$ and $X^\top \beta^*$ of two different orders. We differentiate two types of link functions: (i) $f$ with nonzero Stein's associations and (ii) $f$ with zero first-order and nonzero second-order Stein's association. These two classes capture the functions considered in [43, 44] and [42, 48, 62] respectively. More importantly, we establish the statistical-computational barrier under an oracle computational model [16, 18, 19, 54], which is an abstraction of computations made by algorithms that interact with data. Specifically, we study the signal detection problem where the link function is defined as a continuous interpolation of two link functions of different types. We establish information-theoretical and computational lower bounds for the minimum signal strength required for successful detection and also propose algorithms that yield matching upper bounds. Moreover, we characterize the gap between signal strengths for learning SIMs under limited and unlimited computational budgets and display the evolution of this gap as the link function transits from one type to the other.

**Main Contribution.** Our contribution is three-fold. First, we introduce the first- and second-order Stein's associations, which bring a general characterization of the link functions considered in the literature. Second, for the detection problem, we establish nearly tight information-theoretical and computational lower bounds under the framework of oracle model, which exhibit the statistical price paid for achieving computational efficiency in learning SIMs. Third, we also construct algorithms which yield matching upper bounds. Our results also imply a similar computational barrier for parameter estimation, thus providing a positive answer to the open problem raised in [8].

**Related Work.** There is a huge body of literature on single-index models in the low-dimensional setting. See, for example, [25, 27, 29, 39] and the references therein. For high-dimensional SIMs, when $Y$ and $X^\top \beta^*$ have a nonzero correlation, [22, 23, 26, 40, 41, 43, 44, 58] study the statistical rates of Lasso-type estimators, which are shown to achieve both statistical accuracy and computational efficiency. In contrast, [42, 49, 61, 62] study SIMs which are generalizations of sparse phase retrieval [8].

In addition, the statistical query model is proposed by [30] and further extended by [15, 18–20] for studying the computational complexity of planted clique, random satisfiability problems, stochastic convex optimization, and Gaussian mixture model. In addition, based on a slightly modified version, [16, 34, 54, 63] establish the statistical-computational tradeoffs in statistical problems including sparse PCA, high-dimensional mixture models, weakly supervised learning, and graph structure inference. Among them, our work is mostly related to [16], which validates the computational barrier in phase retrieval with absolute value link function by drawing the connection to mixture of regression models. In comparison, we tackle SIMs directly, which takes phase retrieval as a particular case. More importantly, by interpolating the two sub-classes of SIMs, we obtain the full spectrum of phase transitions, which shed new light on the open problem raised in [8].

Furthermore, there is a massive body of literature on understanding the computational barriers of statistical models. Besides our oracle model approach, there are two other popular means of attacking such problems. The first one is based on polynomial-time reductions from the conjectured computationally challenging problems to statistical problems of interest. See, e.g., [3–7, 9, 12, 21, 24, 37, 57] and the references therein. Second method constructs a sequence of sum-of-squares convex relaxations that are increasingly tighter based on semidefinite programming [1, 2, 14, 28, 31, 36, 38, 45, 55]. Although this approach is free of hardness conjectures, their computational barriers only hold for the restricted family of convex relaxation algorithms.

## 2 Background

In this section, we first introduce the single index model and the associated signal detection problem. We then introduce the statistical query model, which quantifies the computational cost of an algorithm that interacts with data and is later used to establish the main results.

### 2.1 Statistical Model

We consider the single index model

$$Y = f(X^\top \beta^*) + \epsilon, \tag{2.1}$$

where $X \sim N(0, I_d)$ is the covariate, $Y$ is the response, $\beta^* \in \mathbb{R}^d$ is the unknown parameter of interest, $\epsilon \sim N(0, \sigma^2)$ is the noise, and $f \colon \mathbb{R} \to \mathbb{R}$ is the unknown link function. Given $n$ independent realizations $\{z_i = (y_i, x_i)\}_{i \in [n]}$ of this model, our goal is to estimate $\beta^*$ under the assumption that $\beta^*$ is s-sparse, $s \ll n$, and $d \gg n$.

[43] estimate $\beta^*$ by exploiting the covariance structure $\mathsf{Cov}(Y, X^\top \beta^*)$. When such a structure is unavailable, that is, $\mathsf{Cov}(Y, X^\top \beta^*) = 0$, [42, 62] estimate $\beta^*$ by exploiting $\mathsf{Cov}[Y, (X^\top \beta^*)^2]$. However, the resulting estimators require a higher sample complexity than the estimators that are based on $\mathsf{Cov}(Y, X^\top \beta^*)$. To understand such a gap in sample complexity, we consider more general settings under a unified framework. The key of this framework is the following Stein's identities [47]. Let $X \sim N(0, I_d)$ be the standard Gaussian distribution and $Y = h(X)$. If the expectation $\mathbb{E}[\nabla h(X)]$ exists, the first-order Stein's identity takes the form

$$\mathbb{E}\big[\nabla h(X)\big] = \mathbb{E}[YX]. \tag{2.2}$$

Let $Y = h(X)$, where $h$ is twice differentiable. If the expectation $\mathbb{E}[\nabla^2 h(X)]$ exists, the second-order Stein's identity takes the form

$$\mathbb{E}\big[\nabla^2 h(X)\big] = \mathbb{E}\big[Y \cdot (XX^\top - I_d)\big]. \tag{2.3}$$

The above identities show that the covariance structures $\mathsf{Cov}(Y, X^\top \beta^*)$ and $\mathsf{Cov}[Y, (X^\top \beta^*)^2]$ are pivotal in the estimation of the model defined in (2.1) [59, 60]. Specifically, following from (2.2) with $h(X) = f(X^\top \beta^*) + \epsilon$, it holds that $\mathbb{E}[YX] = \mathbb{E}[f'(X^\top \beta^*, \epsilon)] \cdot \beta^*$, where we denote by $f'$ the derivative of $f$ with respect to the first coordinate. In other words, $\mathbb{E}[YX]$ recovers $\beta^*$ up to a scaling under the assumption that $\mathsf{Cov}(Y, X^\top \beta^*) \neq 0$. Meanwhile, following from (2.3) with $h(X) = f(X^\top \beta^*) + \epsilon$, it holds that

$$\mathbb{E}[Y \cdot XX^\top] = \mathbb{E}\big[f''(X^\top \beta^*, \epsilon)\big] \cdot \beta^* \beta^{*\top} + \mathbb{E}[Y] \cdot I_d.$$

In other words, $\beta^*$ is the leading eigenvector of $\mathbb{E}[Y \cdot XX^\top]$ under the assumption that $\mathsf{Cov}[Y, (X^\top \beta^*)^2] > 0$. We define the following covariance structures, which play important roles in the estimation of $\beta^*$ in the model in (2.1) with unknown link function $f$.

**Definition 2.1** (First-order and second-order Stein's associations)**.** Let $\psi$ be a twice differentiable transformation from $\mathbb{R}$ to $\mathbb{R}$ and $Y$ be the response of $X$ under the model in (2.1). We define the first- and second-order Stein's association between $Y$ and $X^\top \beta^*$ as

$$S_1(Y) = \mathsf{Cov}(Y, X^\top \beta^*), \quad S_2(Y, \psi) = \mathsf{Cov}\big[\psi(Y), (X^\top \beta^*)^2\big],$$

respectively, where $\psi$ is called the marginal transformation.

In the following, we introduce classes of link functions of interest. We consider the following two classes of link functions,

$$\mathcal{C}_1 = \big\{ f : \mathsf{Cov}\big(f(X^\top \beta^*), X^\top \beta^*\big) / \|\beta^*\|_2^2 = 1 \big\},$$
$$\mathcal{C}_2 = \big\{ f : \mathsf{Cov}\big(f(X^\top \beta^*), X^\top \beta^*\big) = 0 \big\}. \tag{2.4}$$

The function class $\mathcal{C}_1$ is a class of normalized link functions. Following from the first-order Stein's identity in (2.2), it holds that

$$\mathsf{Cov}\big(f(X^\top \beta^*), X^\top \beta^*\big) = \mathbb{E}\big[f'(X^\top \beta^*)\big] \cdot \|\beta^*\|_2^2.$$

In other words, the definition of $\mathcal{C}_1$ in (2.4) equivalently requires the link function $f \in \mathcal{C}_1$ to satisfy $\mathbb{E}[f'(X^\top \beta^*)] = 1$.

For any twice differentiable marginal transformation $\psi$, we define $\mathcal{C}(\psi)$ as the class of link functions $f$ such that

$$\mathcal{C}(\psi) = \big\{ f : \mathsf{Cov}\big[\psi(Y), (X^\top \beta^*)^2\big] / \|\beta^*\|_2^4 \geq 1 \text{ for } Y = f(X^\top \beta^*) + \epsilon \big\}. \tag{2.5}$$

The definition of $\mathcal{C}(\psi)$ is a generalization of the misspecified phase retrieval model studied by [42, 62] with additive noise. By allowing marginal transformations of $Y$, such a class also covers the linear regression model as a special case.

Note that in (2.5), we require the covariance structure $\mathsf{Cov}[\psi(Y), (X^\top \beta^*)^2]$ to have a magnitude comparable to $\|\beta^*\|_2^4$. Without any loss of generality, such a requirement specifies the scaling of the marginal transformation $\psi$ and the corresponding link function $f \in \mathcal{C}(\psi)$. To see this, note that it holds from the second-order Stein's identity in (2.3) that

$$\mathsf{Cov}\big[\psi(Y), (X^\top \beta^*)^2\big] = \mathbb{E}\big[D^2 \psi\big(f(X^\top \beta^*) + \epsilon\big)\big] \cdot \|\beta^*\|_2^4,$$

where $D$ is the differentiation operator with respect to $X^\top \beta^*$. In other words, (2.5) equivalently requires the link function $f \in \mathcal{C}(\psi)$ to satisfy $\mathbb{E}[D^2 \psi(f(X^\top \beta^*) + \epsilon)] \geq 1$.

For $\psi(y) = y$, the function class $\mathcal{C}(\psi)$ defined in (2.5) reduces to the misspecified phase retrieval models considered by [42, 62] with additive noise. For $\psi(y) = y^2$, $\mathcal{C}(\psi)$ characterizes the linear regression model, the mixed regression model, and various phase retrieval models, including $Y = (X^\top \beta^*)^2 + \epsilon$ and $Y = |X^\top \beta^*| + \epsilon$, up to normalizations. In particular, $\mathcal{C}(\psi)$ also characterizes a class of one-hidden-layer neural networks with Rectified Linear Units (ReLU) activation function. For a neural network with two neurons in the hidden layer, where the parameters in the first layer are $\beta^*$ and $-\beta^*$, and the parameter in the second layer is $(1, 1) \in \mathbb{R}^2$, we have

$$Y = \max\{X^\top \beta^*, 0\} + \max\{-X^\top \beta^*, 0\} + \epsilon = |X^\top \beta^*| + \epsilon,$$

which is captured by $\mathcal{C}(\psi)$ with $\psi(y) = y$ or $\psi(y) = y^2$ up to normalizations.

Throughout this paper, we focus on the marginal transformations $\psi$ such that $\mathcal{C}(\psi) \cap \mathcal{C}_1 \neq \varnothing$ and $\mathcal{C}(\psi) \cap \mathcal{C}_2 \neq \varnothing$, where the function classes $\mathcal{C}_1$, $\mathcal{C}_2$, and $\mathcal{C}(\psi)$ are defined in (2.4) and (2.5). Such a class of marginal transformations $\psi$ enables us to study the phase transition between $f_1 \in \mathcal{C}(\psi) \cap \mathcal{C}_1$ and $f_2 \in \mathcal{C}(\psi) \cap \mathcal{C}_2$. As an example, we consider $\psi(y) = y$. It holds that $f_1 \in \mathcal{C}(\psi) \cap \mathcal{C}_1$ for $f_1(X^\top \beta^*) = X^\top \beta^* + (X^\top \beta^*)^2$, and $f_2 \in \mathcal{C}(\psi) \cap \mathcal{C}_2$ for $f_2(X^\top \beta^*) = (X^\top \beta^*)^2$. In other words, it holds that $\mathcal{C}(\psi) \cap \mathcal{C}_1 \neq \varnothing$ and $\mathcal{C}(\psi) \cap \mathcal{C}_2 \neq \varnothing$ for $\psi(y) = y$. With link functions $f_1 \in \mathcal{C}(\psi) \cap \mathcal{C}_1$ and $f_2 \in \mathcal{C}(\psi) \cap \mathcal{C}_2$, we introduce the following statistical model of interest,

$$Y = \begin{cases} f_1(X^\top \beta^*) + \epsilon, & \text{with probability } \alpha, \\ f_2(X^\top \beta^*) + \epsilon, & \text{with probability } 1 - \alpha, \end{cases} \tag{2.6}$$

where $\epsilon \sim N(0, \sigma^2)$, $X \sim N(0, I_d)$, and $\beta^*$ is $s$-sparse. We assume that $f_1$ and $f_2$ are unknown, and $\psi$ is known a priori. In (2.6), the mixture probability $\alpha$ controls the magnitude of the first-order Stein's association $S_1(Y)$ defined in Definition 2.1, which characterizes a notion of linearity between the response $Y$ and the index $X^\top \beta^*$.

Let $z_i = (y_i, x_i)$ be $n$ independent observations of (2.6) with $n \ll d$, we aim at detecting the existence of a nonzero parameter $\beta^*$, that is, testing the following hypotheses,

$$H_0 \colon \beta^* = 0 \quad \text{versus} \quad H_1 \colon \beta^* \neq 0. \tag{2.7}$$

In what follows, we assume that $s$ is a known integer and $\sigma^2$ is an unknown constant. Meanwhile, to address the identifiability issue, we assume that $\|\beta^*\|_2$ is fixed.

The difficulty of the testing problem in (2.7) is characterized by the signal-to-noise ratio (SNR), which is defined as $\kappa(\beta^*, \sigma) = \|\beta^*\|_2^2 / \sigma^2$. Moreover, to characterize the minimum required SNR, we consider the following parameter spaces corresponding to the null and alternative hypotheses,

$$\mathcal{G}_0 = \big\{(\beta^*, \sigma) \in \mathbb{R}^{d+1} \colon \beta^* = 0\big\},$$
$$\mathcal{G}_1(s, \gamma_n) = \big\{(\beta^*, \sigma) \in \mathbb{R}^{d+1} \colon \|\beta^*\|_0 = s, \kappa(\beta^*, \sigma) \geq \gamma_n\big\}, \tag{2.8}$$

where $\{\gamma_n\}_{n=1}^\infty$ is a nonnegative sequence. For notational simplicity, we denote by $\theta^* = (\beta^*, \sigma)$ and $\mathbb{P}_{\theta^*}^n$ the joint distribution of $\{z_i\}_{i=1}^n$, which are generated by the model in (2.6) with the parameter of interest $\theta^*$ and nuisance parameters $f_1$, $f_2$, and $\psi$. For any function $\phi$ that maps $\mathbf{z} = (z_1, \ldots, z_n) \in \mathbb{R}^{(d+1) \times n}$ to $\{0, 1\}$, the worst-case risk for testing $H_0 \colon \theta \in \mathcal{G}_0$ versus $H_1 \colon \theta^* \in \mathcal{G}_1(s, \gamma_n)$ is defined as the sum of the maximum type-I and type-II errors,

$$R_n(\phi; \mathcal{G}_0, \mathcal{G}_1) = \sup_{\theta^* \in \mathcal{G}_0} \mathbb{P}_{\theta^*}(\phi = 1) + \sup_{\theta^* \in \mathcal{G}_1} \mathbb{P}_{\theta^*}(\phi = 0). \tag{2.9}$$

Correspondingly, the minimax risk is defined as

$$R_n^*(\mathcal{G}_0, \mathcal{G}_1) = \inf_\phi \sup_{f_1, f_2, \psi} R_n(\phi; \mathcal{G}_0, \mathcal{G}_1), \tag{2.10}$$

where we take the supreme over the nuisance parameters $f_1$, $f_2$, and $\psi$ of models in (2.6), and the infimum over the function $\phi$. We further define the minimax separation rate in the following.

**Definition 2.2** (Minimax separation rate [32, 51]). A sequence $\{\gamma_n^*\}_{n=1}^\infty$ is called the minimax separation rate if

(i) given any sequence $\{\gamma_n\}_{n=1}^\infty$ with $\gamma_n = o(\gamma_n^*)$, it holds that $\liminf_{n\to\infty} R_n^*(\mathcal{G}_0, \mathcal{G}_1(s, \gamma_n)) = 1$,

(ii) given any sequence $\{\gamma_n\}_{n=1}^\infty$ with $\gamma_n = \Omega(\gamma_n^*)$, it holds that $\lim_{n\to\infty} R_n^*(\mathcal{G}_0, \mathcal{G}_1(s, \gamma_n)) = 0$.

The minimax separation rate characterizes the minimum SNR that guarantees the existence of an asymptotically powerful test. Therefore, it captures the difficulty of the hypothesis testing problem in (2.7).

## 2.2   Oracle Computational Model

In what follows, we introduce an oracle computational model that quantifies the computational cost of an algorithm. Our model follows from the one considered in [16, 54], which slightly extends the statistical query model originally proposed in [18–20, 30].

**Definition 2.3** (Statistical query model). A statistical oracle $r$ responds to a given query function $q$ with $Z_q$, which is a random variable in $\mathbb{R}$. We define $\mathcal{Q} \subseteq \{q : \mathbb{R}^{d+1} \to [-M, M]\}$ as the space consisting of all the query functions.

We define an algorithm $\mathscr{A}$ as the iterative process that queries a given statistical oracle with query functions in $\mathcal{Q}_A \subseteq \mathcal{Q}$ but does not access the data directly. We denote by $\mathcal{A}(T)$ the set of algorithms that query the statistical oracle $T$ rounds, where $T$ is called the oracle complexity. We denote by $\mathcal{R}[\xi, n, T, \eta(\mathcal{Q}_{\mathscr{A}})]$ the set of statistical oracles $r$ such that

$$\mathbb{P}\left(\bigcap_{q \in \mathcal{Q}_{\mathscr{A}}} \left\{ \left| Z_q - \mathbb{E}[q(Z)] \right| \leq \tau_q \right\} \right) \geq 1 - 2\xi, \tag{2.11}$$

where $Z_q$ is the response of the statistical oracle $r$, $Z = (Y, X)$ is the random variable following the underlying statistical model, $\xi \in [0, 1)$ is the tail probability, and $\tau_q$ is the tolerance parameter given by

$$\tau_q = \frac{\left[\eta(\mathcal{Q}_{\mathscr{A}}) + \log(1/\xi)\right] \cdot M}{n} \bigvee \sqrt{\frac{2\left[\eta(\mathcal{Q}_{\mathscr{A}}) + \log(1/\xi)\right] \cdot \left(M^2 - \{\mathbb{E}[q(Y, X)]\}^2\right)}{n}}. \tag{2.12}$$

Here the parameter $\eta(\mathcal{Q}_{\mathscr{A}})$ is the logarithmic measure of the capacity of $\mathcal{Q}_{\mathscr{A}}$. For a countable $\mathcal{Q}_{\mathscr{A}}$, we have $\eta(\mathcal{Q}_{\mathscr{A}}) = \log(|\mathcal{Q}_{\mathscr{A}}|)$. For an uncountable $\mathcal{Q}_{\mathscr{A}}$, the magnitude $\eta(\mathcal{Q}_{\mathscr{A}})$ can be the Vapnik-Chervonenkis dimension or the metric entropy.

The intuition behind Definition 2.3 is to separate the algorithm from the dataset. Under this definition, the algorithms we consider are blackbox systems that access the necessary information from a statistical oracle. The definition of the statistical oracle $r \in \mathcal{R}[\xi, n, T, \eta(\mathcal{Q}_{\mathscr{A}})]$ is a generalization of the sample average. Note that it holds that

$$M^2 - \{\mathbb{E}[q(Y, X)]\}^2 \geq \mathsf{Var}\big[q(Y, X)\big]. \tag{2.13}$$

If the response $z_q$ of the statistical oracle is the sample mean of $n$ independent realizations of $q(Z)$, then (2.11) follows from Bernstein's inequality coupled with a uniform concentration argument over $\mathcal{Q}_{\mathscr{A}}$, where the variance term is replaced by its upper bound in (2.13) [16].

To capture the computational difficulty of the hypothesis testing problem in (2.7), we introduce the following definition of computational minimax separation risk, which is an analog of the minimax separation risk defined in (2.10) with an additional constraint on the oracle complexity. We consider the algorithms $\mathscr{A} \in \mathcal{A}(T)$ associated with the statistical oracle $r \in \mathcal{R}[\xi, n, T, \eta(\mathcal{Q}_{\mathscr{A}})]$, and denote by $\mathcal{H}(\mathscr{A}, r)$ the set of all the test functions based on $\mathscr{A} \in \mathcal{A}(T)$, which queries $r \in \mathcal{R}[\xi, n, T, \eta(\mathcal{Q}_{\mathscr{A}})]$ $T$ rounds. We define the risk for test function $\phi \in \mathcal{H}(\mathscr{A}, r)$ as

$$\bar{R}_n(\phi; \mathcal{G}_0, \mathcal{G}_1) = \sup_{\theta^* \in \mathcal{G}_0} \bar{\mathbb{P}}_{\theta^*}(\phi = 1) + \sup_{\theta^* \in \mathcal{G}_1} \bar{\mathbb{P}}_{\theta^*}(\phi = 0). \tag{2.14}$$

Correspondingly, we define the computational minimax risk as

$$\bar{R}_n^*(\mathcal{G}_0, \mathcal{G}_1; \mathscr{A}, r) = \inf_{\phi \in \mathcal{H}(\mathscr{A}, r)} \sup_{f_1, f_2, \psi} \bar{R}_n(\phi; \mathcal{G}_0, \mathcal{G}_1) \qquad (2.15)$$

The probability $\bar{\mathbb{P}}_{\theta^*}$ in the above formulation is taken over the distribution of responses from the statistical oracle $r$ under the model in (2.6) with the parameter of interest $\theta^*$ and nuisance parameter $f_1$, $f_2$, and $\psi$. We introduce the following definition of computational minimax separation rate [18, 19, 54].

**Definition 2.4** (Computational minimax separation rate). A sequence $\{\bar{\gamma}_n^*\}_{n=1}^\infty$ is called the computational minimax separation rate if

(i) given any sequence $\{\gamma_n\}_{n=1}^\infty$ with $\gamma_n = o(\bar{\gamma}_n^*)$, for any $\eta$ and any $\mathscr{A} \in \mathcal{A}(d^\eta)$, there exists a statistical oracle $r \in \mathcal{R}[\xi, n, d^\mu, \eta(\mathcal{Q}_{\mathscr{A}})]$ such that

$$\liminf_{n \to \infty} \bar{R}_n^*(\mathcal{G}_0, \mathcal{G}_1(s, \gamma_n); \mathscr{A}, r) = 1,$$

(ii) given any sequence $\{\gamma_n\}_{n=1}^\infty$ with $\gamma_n = \Omega(\bar{\gamma}_n^*)$, there exists an algorithm $\mathscr{A} \in \mathcal{A}(d^\eta)$ with some absolute constant $\eta$ such that it holds for any statistical oracle $r \in \mathcal{R}[\xi, n, d^\mu, \eta(\mathcal{Q}_{\mathscr{A}})]$ that

$$\lim_{n \to \infty} \bar{R}_n^*(\mathcal{G}_0, \mathcal{G}_1(s, \gamma_n); \mathscr{A}, r) = 0.$$

In the following section, we give the explicit forms of $\gamma_n^*$ and $\bar{\gamma}_n^*$. In particular, when the link function $f$ deviates from class $\mathcal{C}_1(\psi)$, a gap between $\bar{\gamma}_n^*$ and $\gamma_n^*$ arises, which characterizes the computational cost to pay for the lack of first-order Stein's association defined in Definition 2.1.

# 3 Main Results

In this section, we lay out the theoretical results. For the hypothesis testing problem in (2.7), we establish the information-theoretic and computational lower bounds by constructing a worst-case hypothesis testing problem. We further establish upper bounds that attain these lower bounds up to logarithmic factors, which is deferred to §A. These lower and upper bounds together characterize the statistical-computational tradeoff. Finally, we show that such a tradeoff in hypothesis testing implies similar computational barriers in parameter estimation.

## 3.1 Lower Bounds

In what follows, we present lower bounds of the minimax and computational minimax separation rates defined in Definitions 2.2 and 2.4, respectively. For the hypothesis testing problem in (2.7) with parameter spaces defined in (2.8), we have the following proposition that characterizes its information-theoretic difficulty.

**Proposition 3.1.** We assume that $\beta^*$ in (2.6) is sparse such that $s = o(d^{1/2-\delta})$ for some positive absolute constant $\delta$. For

$$\gamma_n = o\left(\sqrt{\frac{s \log d}{n}} \bigwedge \frac{1}{\alpha^2} \cdot \frac{s \log d}{n}\right), \qquad (3.1)$$

it holds that $\liminf_{n \to \infty} R_n^*[\mathcal{G}_0, \mathcal{G}_1(s, \gamma_n)] \geq 1$. In other words, any test for the hypothesis testing problem in (2.7) and (2.8) is asymptotically powerless.

*Proof.* See §B.1 for a detailed proof. $\qquad \square$

It follows from Proposition 3.1 that any sequence satisfying (ii) of Definition 2.2 is asymptotically lower bounded by any sequence that satisfies (3.1). As a result, it holds that

$$\gamma_n^* = \Omega\left(\sqrt{\frac{s \log d}{n}} \bigwedge \frac{1}{\alpha^2} \cdot \frac{s \log d}{n}\right), \qquad (3.2)$$

where $\gamma_n^*$ is the minimax separation rate defined in Definition 2.2. Based on (3.2) and the upper bound in Theorem A.2, which is deferred to §A, up to logarithmic factors, the minimax separation

rate defined in Definition 2.2 takes the form

$$\gamma_n^* = \sqrt{\frac{s \log d}{n}} \bigwedge \frac{1}{\alpha^2} \cdot \frac{s \log d}{n}. \tag{3.3}$$

The following theorem establishes a lower bound of the computational minimax separation rate defined in Definition 2.4.

**Theorem 3.2.** We assume that $\beta^*$ in (2.6) is sparse such that $s = o(d^{1/2-\delta})$ for some positive absolute constant $\delta$. For any positive absolute constant $\mu$ and $\mathscr{A} \in \mathcal{A}(d^\mu)$ with

$$\gamma_n = o\left(\left\{\sqrt{\frac{s^2}{n}} \bigwedge \frac{1}{\alpha^2} \cdot \frac{s}{n}\right\} \bigvee \gamma_n^*\right), \tag{3.4}$$

there exists a statistical oracle $r \in \mathcal{R}[\xi, n, d^\mu, \eta(\mathcal{Q})]$ such that $\liminf_{n \to \infty} \bar{R}_n^*(\mathcal{G}_0, \mathcal{G}_1; \mathscr{A}, r) \geq 1$. In other words, any computational tractable test for the hypothesis testing problem in (2.7) and (2.8) is asymptotically powerless.

*Proof.* See §B.2 for a detailed proof. □

It follows from Theorem 3.2 that any sequence satisfying (ii) of Definition 2.4 is asymptotically lower bounded by any sequence that satisfies (3.4). As a result, it holds that

$$\bar{\gamma}_n^* = \Omega\left(\left\{\sqrt{\frac{s^2}{n}} \bigwedge \frac{1}{\alpha^2} \cdot \frac{s}{n}\right\} \bigvee \gamma_n^*\right), \tag{3.5}$$

where $\gamma_n^*$ and $\bar{\gamma}_n^*$ are the minimax and computational minimax separation rates defined in Definitions 2.2 and 2.4, respectively. Based on (3.5) and the upper bound in Theorem A.3, which is deferred to §A, up to logarithmic factors, the computational minimax separation rate defined in Definition 2.4 takes the form

$$\bar{\gamma}_n^* = \sqrt{\frac{s^2}{n}} \bigwedge \frac{1}{\alpha^2} \cdot \frac{s \log d}{n}. \tag{3.6}$$

### 3.2 Phase Transition

In what follows, we characterize the phase transition in the minimax and computational minimax separation rates when the mixture probability $\alpha$ transits from zero to one. We categorize the phase transition into the following regimes in terms of $\alpha$.

1. For $0 < \alpha \leq ((\log d)^2/n)^{1/4}$, our results show that $\gamma_n^* = \sqrt{s \log d/n}$ and $\bar{\gamma}_n^* = \sqrt{s^2/n}$. For $\gamma_n = o(\sqrt{s \log d/n})$ , any test for the hypothesis testing problem in (2.7) is asymptotically powerless. For $\gamma_n = \Omega(\sqrt{s \log d/n})$ and $\gamma_n = o(\sqrt{s^2/n})$, any asymptotically powerful test for (2.7) is computationally intractable with superpolynomial oracle complexity defined in Definition 2.3. For $\gamma_n = \Omega(\sqrt{s^2/n})$, there exists an asymptotically powerful test that is computationally tractable with polynomial oracle complexity. In this regime, the gap between the computational minimax separation rate $\bar{\gamma}_n^*$ and the minimax separation rate $\gamma_n^*$ is invariant to $\alpha$.

2. For $(\log^2 d/n)^{1/4} \leq \alpha \leq (s \log d/n)^{1/4}$, our results show that $\gamma_n^* = \sqrt{s \log d/n}$ and $\bar{\gamma}_n^* = 1/\alpha^2 \cdot s \log d/n$. For $\gamma_n = o(\sqrt{s \log d/n})$, any test is asymptotically powerless. For $\gamma_n = \Omega(\sqrt{s \log d/n})$ and $\gamma_n = o(1/\alpha^2 \cdot s \log d/n)$, any asymptotically powerful test for (2.7) is computationally intractable. For $\gamma_n = \Omega(1/\alpha^2 \cdot s \log d/n)$, there exists an asymptotically powerful test that is computationally tractable. In this regime, a larger $\alpha$ implies a smaller gap between $\bar{\gamma}_n^*$ and $\gamma_n^*$.

3. For $(s \log d/n)^{1/4} < \alpha \leq 1$, our results show that $\gamma_n^* = \bar{\gamma}_n^* = 1/\alpha^2 \cdot s \log d/n$. For $\gamma_n = o(1/\alpha^2 \cdot s \log d/n)$, any test for the hypothesis testing problem in (2.7) is asymptotically powerless, whereas for $\gamma_n = \Omega(1/\alpha^2 \cdot s \log d/n)$, there exists an asymptotically powerful test that is computationally tractable. In this regime, the gap between $\gamma_n^*$ and $\bar{\gamma}_n^*$ vanishes.

By the normalization specified following (2.7), the mixture probability $\alpha$ characterizes the first-order Stein's association of the model under the alternative hypothesis. Therefore, the phase transition

implies that when the first-order Stein's association attains its maximum, which corresponds to $\alpha = 1$, the gap between the computational minimax separation rate $\bar{\gamma}_n^*$ and the minimax separation rate $\gamma_n^*$ vanishes, whereas when the first-order Stein's association vanishes, which corresponds to $\alpha = 0$, the gap between the computational minimax separation rate $\bar{\gamma}_n^*$ and the minimax separation rate $\gamma_n^*$ attains its maximum. In other words, the lack of the first-order Stein's association leads to an extra price of computational cost.

## 3.3  Implication for Parameter Estimation

For the model in (2.6), our result on the computational minimax separation rate in §A implies computational barriers in the estimation of $\beta^*$, which is established in the following theorem.

**Theorem 3.3.** For the estimation of $\beta^*$ in (2.6) with

$$n = o\left( \frac{s^2}{\gamma_n^2} \bigwedge \frac{s \log d}{\gamma_n \cdot \alpha^2} \right), \tag{3.7}$$

where $\gamma_n = \|\beta^*\|^2/\sigma^2$, it holds that, for any positive absolute constant $\mu$ and algorithm $\mathscr{A} \in \mathcal{A}(T)$ that gives $\widehat{\beta}$ within oracle complexity $T = O(d^\mu)$, there exists a statistical oracle $r \in \mathcal{R}[\xi, n, T, \eta(\mathcal{Q})]$ such that

$$\bar{\mathbb{P}}\big( \|\widehat{\beta} - \beta^*\|_2 \geq \sigma \|\beta^*\|_2^{-1} \cdot \gamma_n/4 \big) \geq C, \tag{3.8}$$

where $C$ is a positive absolute constant.

*Proof.* See §B.5 for a detailed proof. $\qquad\square$

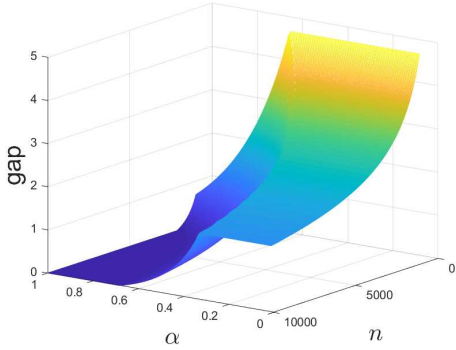

Figure 1: Phase transition in the gap between minimax separation rate and computational minimax seperation rate: (i) for $0 < \alpha \leq ((\log d)^2/n)^{1/4}$, the gap is invariant to $\alpha$. (ii) for $(\log^2 d/n)^{1/4} \leq \alpha \leq (s \log d/n)^{1/4}$, a larger $\alpha$ implies a smaller gap. (iii) for $(s \log d/n)^{1/4} < \alpha \leq 1$, the gap vanishes.

For $\alpha = 0$, the estimation of $\beta^*$ in (2.6) reduces to the sparse phase retrieval problem. For simplicity of discussion, let $\gamma_n = \|\beta^*\|_2^2/\sigma^2$ be a constant in the following discussions. Theorem 3.3 implies that for $n = o(s^2)$, any computationally tractable estimator is statistically inconsistent in the sense that $\|\widehat{\beta} - \beta^*\|_2 \geq C$ holds with at least constant probability. [8] construct a computational tractable estimator for sparse phase retrieval with the quadratic link function $Y = |X^\top \beta^*|^2 + \epsilon$. The estimator by [8] is statistically consistent under the assumption that $n \geq C(1 + \sigma^2/\|\beta^*\|_2^4) \cdot s^2 \log d$. Similar phenomenon arises in misspecified sparse phase retrieval studied by [42], although their work is slightly more general, in the sense that they consider $f(X^\top \beta^*, \epsilon)$ as the link function. The estimator by [42] requires $n \geq Cs^2 \log d$ to be statistically consistent. Both [8] and [42] conjecture that their requirements on the sample size cannot be relaxed for computationally tractable estimators. Theorem 3.3 confirms this conjecture for the sparse phase retrieval problem under the statistical query model defined in Definition 2.3.

For $\alpha = 1$, the requirement for a computationally tractable estimator to be statistically consistent becomes $n \geq Cs \log d$. Such a sample size requirement agrees with the information-theoretic lower bound. [43] construct a computationally tractable estimator of $\beta^*$, which requires the sample size $n \geq Cs \log(d/s)$ to be statistically consistent. It follows from Theorem 3.3 that such a requirement is necessary.

For $0 < \alpha < 1$, we observe a phase transition in the required sample size in terms of $\alpha$, which is similar to the phase transition of the computational minimax separation rates. For $0 < \alpha \leq \sqrt{\gamma_n \log d/s}$, the requirement becomes $n \geq Cs^2$. For $\sqrt{\gamma_n \log d/s} \leq \alpha \leq 1$, the requirement becomes $n \geq Cs \log d/\alpha^2$. In this regime, a larger $\alpha$ implies a smaller sample size required for a computationally tractable estimator to be statistically consistent.

