[Supplementary Material 1]

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

## References

[1] Barak, B., Hopkins, S., Kelner, J., Kothari, P. K., Moitra, A. and Potechin, A. (2019). A nearly tight sum-of-squares lower bound for the planted clique problem. *SIAM Journal on Computing*, **48** 687–735.

[2] Barak, B. and Steurer, D. (2014). Sum-of-squares proofs and the quest toward optimal algorithms. *arXiv preprint arXiv:1404.5236*.

[3] Berthet, Q. and Rigollet, P. (2013). Computational lower bounds for sparse PCA. *arXiv preprint arXiv:1304.0828*.

[4] Berthet, Q., Rigollet, P. et al. (2013). Optimal detection of sparse principal components in high dimension. *The Annals of Statistics*, **41** 1780–1815.

[5] Brennan, M. and Bresler, G. (2019). Optimal average-case reductions to sparse pca: From weak assumptions to strong hardness. *arXiv preprint arXiv:1902.07380*.

[6] Brennan, M., Bresler, G. and Huleihel, W. (2018). Reducibility and computational lower bounds for problems with planted sparse structure. In *Conference On Learning Theory*.

[7] Brennan, M., Bresler, G. and Huleihel, W. (2019). Universality of computational lower bounds for submatrix detection. *arXiv preprint arXiv:1902.06916*.

[8] Cai, T. T., Li, X., Ma, Z. et al. (2016). Optimal rates of convergence for noisy sparse phase retrieval via thresholded wirtinger flow. *The Annals of Statistics*, **44** 2221–2251.

[9] Cai, T. T., Liang, T. and Rakhlin, A. (2017). Computational and statistical boundaries for submatrix localization in a large noisy matrix. *The Annals of Statistics*, **45** 1403–1430.

[10] Candes, E. J., Eldar, Y. C., Strohmer, T. and Voroninski, V. (2015). Phase retrieval via matrix completion. *SIAM review*, **57** 225–251.

[11] Candes, E. J., Strohmer, T. and Voroninski, V. (2013). Phaselift: Exact and stable signal recovery from magnitude measurements via convex programming. *Communications on Pure and Applied Mathematics*, **66** 1241–1274.

[12] Chen, Y. and Xu, J. (2014). Statistical-computational tradeoffs in planted problems and submatrix localization with a growing number of clusters and submatrices. *arXiv preprint arXiv:1402.1267*.

[13] d'Aspremont, A., Ghaoui, L. E., Jordan, M. I. and Lanckriet, G. R. (2005). A direct formulation for sparse pca using semidefinite programming. In *Advances in neural information processing systems*.

[14] Deshpande, Y. and Montanari, A. (2015). Improved sum-of-squares lower bounds for hidden clique and hidden submatrix problems. In *Conference on Learning Theory*.

[15] Diakonikolas, I., Kane, D. M. and Stewart, A. (2017). Statistical query lower bounds for robust estimation of high-dimensional gaussians and gaussian mixtures. In *2017 IEEE 58th Annual Symposium on Foundations of Computer Science (FOCS)*. IEEE.

[16] Fan, J., Liu, H., Wang, Z. and Yang, Z. (2018). Curse of heterogeneity: Computational barriers in sparse mixture models and phase retrieval. *arXiv preprint arXiv:1808.06996*.

[17] Fan, J. and Yao, Q. (2008). *Nonlinear time series: nonparametric and parametric methods*. Springer Science & Business Media.

[18] Feldman, V., Grigorescu, E., Reyzin, L., Vempala, S. S. and Xiao, Y. (2017). Statistical algorithms and a lower bound for detecting planted cliques. *Journal of the ACM (JACM)*, **64** 8.

[19] Feldman, V., Guzmán, C. and Vempala, S. (2017). Statistical query algorithms for mean vector estimation and stochastic convex optimization. In *Proceedings of the Twenty-Eighth Annual ACM-SIAM Symposium on Discrete Algorithms*. SIAM.

[20] Feldman, V., Perkins, W. and Vempala, S. (2018). On the complexity of random satisfiability problems with planted solutions. *SIAM Journal on Computing*, **47** 1294–1338.

[21] Gao, C., Ma, Z., Zhou, H. H. et al. (2017). Sparse cca: Adaptive estimation and computational barriers. *The Annals of Statistics*, **45** 2074–2101.

[22] Goldstein, L., Minsker, S. and Wei, X. (2018). Structured signal recovery from non-linear and heavy-tailed measurements. *IEEE Transactions on Information Theory*, **64** 5513–5530.

[23] Goldstein, L. and Wei, X. (2018). Non-gaussian observations in nonlinear compressed sensing via stein discrepancies. *Information and Inference: A Journal of the IMA*, **8** 125–159.

[24] Hajek, B., Wu, Y. and Xu, J. (2015). Computational lower bounds for community detection on random graphs. In *Conference on Learning Theory*.

[25] Han, A. K. (1987). Non-parametric analysis of a generalized regression model: the maximum rank correlation estimator. *Journal of Econometrics*, **35** 303–316.

[26] Han, F., Ji, H., Ji, Z., Wang, H. et al. (2017). A provable smoothing approach for high dimensional generalized regression with applications in genomics. *Electronic Journal of Statistics*, **11** 4347–4403.

[27] Härdle, W. K., Müller, M., Sperlich, S. and Werwatz, A. (2012). *Nonparametric and semiparametric models*. Springer Science & Business Media.

[28] Hopkins, S. B., Kothari, P. K., Potechin, A., Raghavendra, P., Schramm, T. and Steurer, D. (2017). The power of sum-of-squares for detecting hidden structures. In *Symposium on Foundations of Computer Science*. IEEE.

[29] Horowitz, J. L. (2009). *Semiparametric and nonparametric methods in econometrics*, vol. 12. Springer.

[30] Kearns, M. (1998). Efficient noise-tolerant learning from statistical queries. *Journal of the ACM (JACM)*, **45** 983–1006.

[31] Kothari, P. K. and Mehta, R. (2018). Sum-of-squares meets nash: lower bounds for finding any equilibrium. In *Symposium on Theory of Computing*. ACM.

[32] Le Cam, L. (1956). On the asymptotic theory of estimation and testing hypotheses. In *Proceedings of the Third Berkeley Symposium on Mathematical Statistics and Probability, Volume 1: Contributions to the Theory of Statistics*. The Regents of the University of California.

[33] Lecué, G. and Mendelson, S. (2013). Minimax rate of convergence and the performance of erm in phase recovery. *arXiv preprint arXiv:1311.5024*.

[34] Lu, H., Cao, Y., Lu, J., Liu, H. and Wang, Z. (2018). The edge density barrier: Computational-statistical tradeoffs in combinatorial inference. In *International Conference on Machine Learning*.

[35] Lu, X., Chen, G., Singh, R. S. and K. Song, P. X. (2006). A class of partially linear single-index survival models. *Canadian Journal of Statistics*, **34** 97–112.

[36] Ma, T. and Wigderson, A. (2015). Sum-of-squares lower bounds for sparse pca. In *Advances in Neural Information Processing Systems*.

[37] Ma, Z., Wu, Y. et al. (2015). Computational barriers in minimax submatrix detection. *The Annals of Statistics*, **43** 1089–1116.

[38] Meka, R., Potechin, A. and Wigderson, A. (2015). Sum-of-squares lower bounds for planted clique. In *Symposium on Theory of computing*. ACM.

[39] Nelder, J. A. and Wedderburn, R. W. (1972). Generalized linear models. *Journal of the Royal Statistical Society: Series A (General)*, **135** 370–384.

[40] Neykov, M., Lin, Q. and Liu, J. S. (2015). Signed support recovery for single index models in high-dimensions. *arXiv preprint arXiv:1511.02270*.

[41] Neykov, M., Liu, J. S. and Cai, T. (2016). $\ell_1$-regularized least squares for support recovery of high dimensional single index models with gaussian designs. *Journal of Machine Learning Research*, **17** 1–37.

[42] Neykov, M., Wang, Z. and Liu, H. (2016). Agnostic estimation for misspecified phase retrieval models. *Advances in Neural Information Processing Systems*.

[43] Plan, Y. and Vershynin, R. (2016). The generalized lasso with non-linear observations. *IEEE Transactions on Information Theory*, **62** 1528–1537.

[44] Plan, Y., Vershynin, R. and Yudovina, E. (2016). High-dimensional estimation with geometric constraints. *Information and Inference: A Journal of the IMA*, **6** 1–40.

[45] Potechin, A. (2017). Sum of squares lower bounds from symmetry and a good story. *arXiv preprint arXiv:1711.11469*.

[46] Raskutti, G., Wainwright, M. J. and Yu, B. (2011). Minimax rates of estimation for high-dimensional linear regression over $\ell_q$-balls. *IEEE transactions on information theory*, **57** 6976–6994.

[47] Tan, Y. S. (2017). Sparse phase retrieval via sparse pca despite model misspecification: A simplified and extended analysis. *arXiv preprint arXiv:1712.04106*.

[48] Thrampoulidis, C. and Rawat, A. S. (2017). Lifting high-dimensional nonlinear models with gaussian regressors. *arXiv preprint arXiv:1712.03638*.

[49] Tibshirani, R. (1996). Regression shrinkage and selection via the Lasso. *Journal of the Royal Statistical Society: Series B (Methodological)*, **58** 267–288.

[50] Tsybakov, A. B. (2009). Introduction to nonparametric estimation. revised and extended from the 2004 french original. translated by vladimir zaiats.

[51] Vershynin, R. (2010). Introduction to the non-asymptotic analysis of random matrices. *arXiv preprint arXiv:1011.3027*.

[52] Verzelen, N. et al. (2012). Minimax risks for sparse regressions: Ultra-high dimensional phenomenons. *Electronic Journal of Statistics*, **6** 38–90.

[53] Wang, Z., Gu, Q. and Liu, H. (2015). Sharp computational-statistical phase transitions via oracle computational model. *arXiv preprint arXiv:1512.08861*.

[54] Wang, Z., Gu, Q. and Liu, H. (2016). On the statistical limits of convex relaxations. In *International Conference on Machine Learning*.

[55] Wu, T. Z., Yu, K. and Yu, Y. (2010). Single-index quantile regression. *Journal of Multivariate Analysis*, **101** 1607–1621.

[56] Wu, Y. and Xu, J. (2018). Statistical problems with planted structures: Information-theoretical and computational limits. *arXiv preprint arXiv:1806.00118*.

[57] Yang, Z., Balasubramanian, K. and Liu, H. (2017). High-dimensional non-Gaussian single index models via thresholded score function estimation. In *Proceedings of the 34th International Conference on Machine Learning-Volume 70*. JMLR. org.

[58] Yang, Z., Balasubramanian, K. and Liu, H. (2017). High-dimensional non-gaussian single index models via thresholded score function estimation. *Proceedings of Machine Learning Research*.

[59] Yang, Z., Balasubramanian, K. and Liu, H. (2017). On stein's identity and near-optimal estimation in high-dimensional index models. *arXiv preprint arXiv:1709.08795*.

[60] Yang, Z., Balasubramanian, K., Wang, P. Z. and Liu, H. (2017). Estimating high-dimensional non-gaussian multiple index models via Stein's lemma. In *Advances in Neural Information Processing Systems*.

[61] Yang, Z., Yang, L. F., Fang, E. X., Zhao, T., Wang, Z. and Neykov, M. (2017). Misspecified nonconvex statistical optimization for phase retrieval. *arXiv preprint arXiv:1712.06245*.

[62] Yi, X., Wang, Z., Yang, Z., Caramanis, C. and Liu, H. (2016). More supervision, less computation: Statistical-computational tradeoffs in weakly supervised learning. *Advances in Neural Information Processing Systems*.

# A Upper bounds

In this section, we establish upper bounds that attain the lower bounds obtained in Proposition 3.1 and Theorem A.2 up to logarithmic factors. Based on the lower bounds and upper bounds, we obtain the minimax and computational minimax separation rates defined in Definitions 2.2 and 2.4, respectively.

Recall that the hypothesis testing problem in (2.7) takes the form

$$H_0\colon Y = \epsilon_0 \quad \text{versus} \quad H_1\colon Y = \begin{cases} f_1(X^\top \beta^*) + \epsilon, & \text{with probability } \alpha, \\ f_2(X^\top \beta^*) + \epsilon, & \text{with probability } 1 - \alpha. \end{cases} \quad \text{(A.1)}$$

Here $\epsilon$ is a Gaussian noise with variance $\sigma^2$ and $\epsilon_0$ is a noise such that the variances of $Y$ under the null and alternative hypotheses are the same. Besides, $f_1 \in \mathcal{C}_1 \cap \mathcal{C}(\psi)$ and $f_2 \in \mathcal{C}_2 \cap \mathcal{C}(\psi)$ are two unknown link functions, where $\mathcal{C}_1(\psi)$, $\mathcal{C}_2(\psi)$, and $\mathcal{C}(\psi)$ are defined in (2.4) and (2.5). Meanwhile, we set $X \sim N(0, I_d)$ and $\beta^*$ to be $s$-sparse. For the simplicity of the following discussions, we restrict to the set of $\beta^*$ such that $\beta^* = \rho \cdot v^*$, where $v^* \in \bar{\mathcal{G}}(s) = \{v \in \{-1, 0, 1\}^d : \|v\|_0 = s\}$. We further define

$$\bar{\mathcal{G}}_1(s, \gamma_n) = \big\{ (\beta^*, \sigma) \in \mathbb{R}^{d+1} \colon \beta^* = \rho \cdot v^*, v^* \in \bar{\mathcal{G}}(s), \kappa(\beta^*, \sigma) \geq \gamma_n \big\}.$$

We highlight the fact that such a restricted parameter set is sufficient to characterize the difficulty of the hypothesis testing problem in (2.7), and defer the proof of the general case to §D.

Let $Z = (Y, X)$ and $\mathbb{P}_0$, $\mathbb{P}_{v^*}$ be the distributions of $Z$ under the null and alternative hypotheses, respectively. We introduce the following assumption on $Y$ and $\psi(Y)$ under the alternative hypothesis, which regulates the tail and moment of $Y$ and $\psi(Y)$.

**Assumption A.1.** We assume that $Y$ and $\psi(Y)$ have bounded fourth moments. We further assume that under the alternative hypothesis, $Y$ and $\psi(Y)$ have desired tail bounds in the form of

$$\mathbb{P}_{v^*}(|Y| \geq R) \leq C \exp(-R^\nu), \quad \mathbb{P}_{v^*}(|\psi(Y)| \geq R) \leq C' \exp(-R^\nu), \quad \text{(A.2)}$$

which holds for a sufficiently large $R$ and positive absolute constants $C$, $C'$, and $\nu$.

Assumption A.1 is required only for the upper bounds. It is needed to construct bounded query functions defined in Definition 2.3. Such an assumption is a mild regularity condition in the sense that it holds for the linear regression model and most of the phase retrieval models. For instance, let $(Y, X)$ be generated by the mixed regression model and $\psi(Y) = Y^2$. Then $Y$ follows the mixture of Gaussian distributions. Therefore, $Y$ has bounded fourth moment and Gaussian tail, and $\psi(Y) = Y^2$ is sub-exponential under the alternative hypothesis with bounded fourth moment. Hence, the tail bound stated in (A.2) holds for $Y$ and $\psi(Y)$ with $\nu = 1$. Similar arguments hold for the linear regression model and the phase retrieval models $Y = |X^\top \beta^*| + \epsilon$ and $Y = (X^\top \beta^*)^2 + \epsilon$.

In what follows, we design the test function $\phi$ based on the first-order and second-order Stein's identities in (2.2) and (2.3). Following from (2.5), it holds that $S_2(Y, \psi) \geq \|\beta^*\|_2^4$ under the alternative hypothesis. It then follows from the second-order Stein's identity in (2.3) that $\mathbb{E}_{\mathbb{P}_{v^*}}[\psi(Y) \cdot (XX^\top - I)] \succeq \beta^* \beta^{*\top}$ under the alternative hypothesis. Meanwhile, under the null hypothesis, $\psi(Y)$ is independent of $X$. Therefore, it holds that

$$\mathbb{E}_{\mathbb{P}_{v^*}}\big[v^\top \psi(Y) \cdot (XX^\top - I)v\big] \geq (v^\top \beta^*)^2, \quad \mathbb{E}_{\mathbb{P}_0}\big[\psi(Y) \cdot (XX^\top - I)\big] = 0. \quad \text{(A.3)}$$

Meanwhile, following from (2.4), it holds that $\mathbb{E}[Y_1 X] = \beta^*$ with $Y_1 = f_1(X^\top \beta^*, \epsilon)$. Therefore, it follows from the first-order Stein's identity in (2.2) that

$$\mathbb{E}_{\mathbb{P}_{v^*}}[v^\top YX] = \alpha \cdot v^\top \beta^*, \quad \mathbb{E}_{\mathbb{P}_0}[YX] = 0. \quad \text{(A.4)}$$

We introduce the following query functions,

$$q_{1,v}(Y, X) = \psi(Y) \cdot \big[s^{-1}(v^\top X)^2 - 1\big] \cdot \mathbb{1}\big\{|\psi(Y)| \leq (R \log n)^{1/\nu}\big\} \cdot \mathbb{1}\big\{|v^\top X| \leq R \cdot \sqrt{s \log n}\big\},$$

$$q_{2,v}(Y, X) = Y \cdot (s^{-1/2} v^\top X) \cdot \mathbb{1}\big\{|Y| \leq (R \log n)^{1/\nu}\big\} \cdot \mathbb{1}\big\{|v^\top X| \leq R \cdot \sqrt{s \log n}\big\}. \quad \text{(A.5)}$$

We denote by $\bar{Z}_{1,v}$ and $\bar{Z}_{2,v}$ the responses of the statistical oracle to query functions $q_{1,v}$ and $q_{2,v}$, as defined in Definition 2.3. We define the test functions $\phi_1$ and $\phi_2$ as

$$\phi_1 = \mathbb{1}\Big\{ \sup_{v \in \bar{\mathcal{G}}(s)} \bar{Z}_{1,v} \geq \tau_1 \Big\}, \quad \phi_2 = \mathbb{1}\Big\{ \sup_{v \in \bar{\mathcal{G}}(s)} \bar{Z}_{2,v} \geq \tau_2 \Big\}, \quad \text{(A.6)}$$

where we set the thresholds $\tau_1$ and $\tau_2$ to be

$$\tau_1 = C R^{2+1/\nu} \cdot (\log n)^{1+1/\nu} \cdot \sqrt{\frac{s \log d}{n}}, \quad \tau_2 = C' R^{1+1/\nu} \cdot (\log n)^{1/2+1/\nu} \cdot \sqrt{\frac{s \log d}{n}}. \quad \text{(A.7)}$$

Here $C$ and $C'$ are absolute constants (which are specified in §B.3). We define the test function as $\phi = \phi_1 \vee \phi_2$. The following theorem characterizes an upper bound for the minimax separation rate by quantifying the SNR for $\phi$ to be asymptotically powerful, which attains the information-theoretic lower bound in Proposition 3.1 up to logarithmic factors.

**Theorem A.2.** We consider the hypothesis testing problem in (A.1) under Assumption A.1. For

$$\gamma_n = \Omega\left((\log n)^{1+1/\nu} \cdot \sqrt{\frac{s \log d}{n}} \bigwedge \frac{(\log n)^{1+2/\nu}}{\alpha^2} \cdot \frac{s \log d}{n}\right), \tag{A.8}$$

it holds that $R_n(\phi; \mathcal{G}_0, \bar{\mathcal{G}}_1) = O(1/d)$. In other words, $\phi$ is asymptotically powerful.

*Proof.* See §B.3 for a detailed proof. □

It follows from Theorem A.2 that any sequence satisfying (i) of Definition 2.2 is asymptotically upper bounded by any sequence that satisfies (A.8). As a result, it holds that

$$\gamma_n^* = o\left((\log n)^{1+1/\nu} \cdot \sqrt{\frac{s \log d}{n}} \bigwedge \frac{(\log n)^{1+2/\nu}}{\alpha^2} \cdot \frac{s \log d}{n}\right). \tag{A.9}$$

Based on (3.2) and (A.9), up to logarithmic factors, the minimax separation rate defined in Definition 2.2 takes the form

$$\gamma_n^* = \sqrt{\frac{s \log d}{n}} \bigwedge \frac{1}{\alpha^2} \cdot \frac{s \log d}{n}. \tag{A.10}$$

Note that the query functions in (A.5) have exponential oracle complexity, since searching over the parameter set $\bar{\mathcal{G}}(s)$ requires querying the statistical oracle $T = \binom{d}{s} \cdot 2^s$ rounds. To construct a computationally tractable test, we design query functions that access each entry $X_j$ of $X$,

$$q_{1,j}(Y, X) = \psi(Y) \cdot (X_j^2 - 1) \cdot \mathbb{1}\{|\psi(Y)| \leq (R \log n)^{1/\nu}\} \cdot \mathbb{1}\{|X_j| \leq R\sqrt{\log n}\}, \quad j \in [d]$$

$$q_{2,j}(Y, X) = Y \cdot X_j \cdot \mathbb{1}\{|Y| \leq (R \log n)^{1/\nu}\} \cdot \mathbb{1}\{|X_j| \leq R\sqrt{\log n}\}, \quad j \in [d]. \tag{A.11}$$

We denote by $\bar{Z}_{1,j}$ and $\bar{Z}_{2,j}$ the responses of the statistical oracle to the query functions $q_{1,j}$ and $q_{2,j}$, as defined in Definition 2.3 . We define the test functions $\widetilde{\phi}_1$ and $\widetilde{\phi}_2$ as

$$\widetilde{\phi}_1 = \mathbb{1}\left\{\sup_{j \in [d]} \bar{Z}_{1,j} \geq \widetilde{\tau}_1\right\}, \quad \widetilde{\phi}_2 = \mathbb{1}\left\{\sup_{j \in [d]} \bar{Z}_{2,j} \geq \widetilde{\tau}_2\right\} \bigvee \mathbb{1}\left\{\inf_{j \in [d]} \bar{Z}_{2,j} \leq -\widetilde{\tau}_2\right\}, \tag{A.12}$$

where we set the thresholds $\widetilde{\tau}_1$ and $\widetilde{\tau}_2$ to be

$$\widetilde{\tau}_1 = CR^{2+1/\nu}(\log n)^{1+1/\nu} \cdot \sqrt{\frac{\log d}{n}}, \quad \widetilde{\tau}_2 = C'R^{1+1/\nu}(\log n)^{1/2+1/\nu} \cdot \sqrt{\frac{\log d}{n}}. \tag{A.13}$$

Finally, we define the test function to be $\widetilde{\phi} = \widetilde{\phi}_1 \vee \widetilde{\phi}_2$. By the definition of $\phi_1$ and $\phi_2$ in (A.12), the test function $\widetilde{\phi}$ is computationally tractable with query complexity $T = 2d$. The following theorem characterizes an upper bound for the computational minimax separation rate, which attains the computational lower bound in Theorem 3.2 up to logarithmic factors.

**Theorem A.3.** We consider the hypothesis testing problem in (A.1) under Assumption A.1. For

$$\gamma_n = \Omega\left((\log n)^{1+1/\nu} \cdot \sqrt{\frac{s^2 \log d}{n}} \bigwedge \frac{(\log n)^{1+2/\nu}}{\alpha^2} \cdot \frac{s \log d}{n}\right), \tag{A.14}$$

it holds that $\bar{R}_n(\widetilde{\phi}; \mathcal{G}_0, \bar{\mathcal{G}}_1) = O(1/d)$. In other words, $\widetilde{\phi}$ is asymptotically powerful.

*Proof.* See §B.4 for a detailed proof. □

It follows from Theorem A.3 that any sequence satisfying (i) of Definition 2.4 is asymptotically upper bounded by any sequence that satisfies (A.14). As a result, it holds that

$$\bar{\gamma}_n^* = o\left((\log n)^{1+1/\nu} \cdot \sqrt{\frac{s^2 \log d}{n}} \bigwedge \frac{(\log n)^{1+2/\nu}}{\alpha^2} \cdot \frac{s \log d}{n}\right). \tag{A.15}$$

Based on (3.5) and (A.15), up to logarithmic factors, the computational minimax separation rate defined in Definition 2.4 takes the form

$$\bar{\gamma}_n^* = \sqrt{\frac{s^2}{n}} \bigwedge \frac{1}{\alpha^2} \cdot \frac{s \log d}{n}. \tag{A.16}$$

## B  Proof of Main Results

In this section, we lay out the proofs of the main results in §3 and §A.

### B.1  Proof of Proposition 3.1

*Proof.* We have the following lower bound of minimax risk,

$$R_n^*(\mathcal{G}_0, \mathcal{G}_1) = \inf_\phi \sup_{f_1, f_2, \psi} R_n(\phi; \mathcal{G}_0, \mathcal{G}_1) \geq \inf_\phi R_n(\phi; \mathcal{G}_0, \mathcal{G}_1)$$

$$= \inf_\phi \Big\{ \sup_{\theta^* \in \mathcal{G}_0} \mathbb{P}_{\theta^*}(\phi = 1) + \sup_{\theta^* \in \mathcal{G}_1} \mathbb{P}_{\theta^*}(\phi = 0) \Big\}.$$

where the first inequality is obtained by restricting $f_1$, $f_2$, and $\psi$ in the testing problem in (2.7) as follows. We set $\psi(y) = y^2$ and the sample $\{z_i\}_{i \in [n]}$ to be generated from a mixture of the linear regression model $Y_1 = f_1(X^\top \beta^*) + \epsilon = X^\top \beta^* + \epsilon$ and the mixed regression model $Y_2 = f_2(X^\top \beta^*) + \epsilon = \eta \cdot X^\top \beta^* + \epsilon$. Here we set $\epsilon \sim N(0, \sigma^2)$ and $\eta$ to be a Rademacher random variable, which is independent of both $X$ and $\epsilon$. Since $S_1(Y_1) = \|\beta^*\|_2^2$, $S_1(Y_2) = 0$, and $S_2(Y_1, \psi) = S_2(Y_2, \psi) = 2\|\beta^*\|_2^4$, we have $f_1 \in \mathcal{C}_1 \cap \mathcal{C}(\psi)$ and $f_2 \in \mathcal{C}_2 \cap \mathcal{C}(\psi)$, where $\mathcal{C}_1$, $\mathcal{C}_2$, and $\mathcal{C}(\psi)$ are defined in (2.4) and (2.5).

We further restrict the parameter space of $\theta^* = (\beta^*, \sigma)$ as follows. Let $\beta^* \in \{\beta = \rho \cdot \mathrm{v} \colon \mathrm{v} \in \mathcal{G}(s)\}$, where $\rho$ is a positive constant and $\mathcal{G}(s) = \{\mathrm{v} \in \{0,1\}^d \colon \|\mathrm{v}\|_0 = s\}$. Therefore, the original hypothesis testing problem is reduced to

$$H_0 \colon Y = \epsilon_0 \quad \text{versus} \quad H_1 \colon Y = \begin{cases} X^\top \beta^* + \epsilon, & \text{with probability } \alpha, \\ \eta \cdot X^\top \beta^* + \epsilon, & \text{with probability } 1 - \alpha, \end{cases} \tag{B.1}$$

where under $H_0$ we have $\epsilon_0 \sim N(0, \sigma^2 + s\rho^2)$ and under $H_1$ we have $\epsilon \sim N(0, \sigma^2)$. We denote by $\mathbb{P}_0$ and $\mathbb{P}_{\mathrm{v}^*}$ the probability distributions of $Z = (Y, X)$ under the null and alternative hypotheses with $\beta^* = \rho \cdot \mathrm{v}^*$, respectively. In addition, we define $\overline{\mathbb{P}} = |\mathcal{G}(s)|^{-1} \sum_{\mathrm{v} \in \mathcal{G}(s)} \mathbb{P}_{\mathrm{v}}^n$, where we use the superscript $n$ to denote the $n$-fold product probability measure. By Neyman-Pearson lemma, we have

$$R_n^*(\mathcal{G}_0, \mathcal{G}_1) \geq \inf_\phi \big[ \mathbb{P}_0^n(\phi = 1) + \overline{\mathbb{P}}(\phi = 0) \big] = 1 - 1/2 \cdot \mathbb{E}_{\mathbb{P}_0^n} \big[ |\mathrm{d}\overline{\mathbb{P}}/\mathrm{d}\mathbb{P}_0^n - 1| \big]$$

$$\geq 1 - 1/2 \cdot \Big( \big( \mathbb{E}_{\mathbb{P}_0^n} \big[ \mathrm{d}\overline{\mathbb{P}}/\mathrm{d}\mathbb{P}_0^n \big] \big)^2 - 1 \Big)^{1/2}, \tag{B.2}$$

where the second inequality follows from the Cauchy-Schwarz inequality. In what follows, we show that $\mathbb{E}_{\mathbb{P}_0^n} [\mathrm{d}\overline{\mathbb{P}}/\mathrm{d}\mathbb{P}_0^n]^2 = 1 + o(1)$ under the condition in (3.1), which implies $\liminf_{n \to \infty} R_n^*(\mathcal{G}_0, \mathcal{G}_1) \geq 1 - o(1)$ by (B.2). Note that on the right-hand side of (B.2), we have

$$\big( \mathbb{E}_{\mathbb{P}_0^n} \big[ \mathrm{d}\overline{\mathbb{P}}/\mathrm{d}\mathbb{P}_0^n \big] \big)^2 = \frac{1}{|\mathcal{G}(s)|^2} \sum_{\mathrm{v}, \mathrm{v}' \in \mathcal{G}(s)} \mathbb{E}_{\mathbb{P}_0^n} \Big[ \frac{\mathrm{d}\mathbb{P}_{\mathrm{v}}^n}{\mathrm{d}\mathbb{P}_0^n} \frac{\mathrm{d}\mathbb{P}_{\mathrm{v}'}^n}{\mathrm{d}\mathbb{P}_0^n} (Z_1, \ldots, Z_n) \Big], \tag{B.3}$$

where $Z_i$ are independent copies of $Z = (Y, X)$. The following lemma establishes an upper bound of the right-hand side of (B.3).

**Lemma B.1.** For any $\mathrm{v}_1, \mathrm{v}_2 \in \mathcal{G}(s)$, if $s\rho^2 = o(1)$, it holds that

$$\mathbb{E}_{\mathbb{P}_0} \Big[ \frac{\mathrm{d}\mathbb{P}_{\mathrm{v}_1}}{\mathrm{d}\mathbb{P}_0} \frac{\mathrm{d}\mathbb{P}_{\mathrm{v}_2}}{\mathrm{d}\mathbb{P}_0} (Z) \Big] \leq \cosh\Big( \frac{2\rho^2 \langle \mathrm{v}_1, \mathrm{v}_2 \rangle}{\sigma^2 + s\beta^2} \Big) + \alpha^2 \sinh\Big( \frac{2\rho^2 \langle \mathrm{v}_1, \mathrm{v}_2 \rangle}{\sigma^2 + s\rho^2} \Big). \tag{B.4}$$

*Proof.* See §C.1 for a detailed proof. ☐

Following from Lemma B.1, it holds that

$$\mathbb{E}_{\mathbb{P}_0} \Big[ \frac{\mathrm{d}\mathbb{P}_{\mathrm{v}_1}^n}{\mathrm{d}\mathbb{P}_0^n} \frac{\mathrm{d}\mathbb{P}_{\mathrm{v}_2}^n}{\mathrm{d}\mathbb{P}_0^n} (Z_1, \ldots, Z_n) \Big] = \Big( \mathbb{E}_{\mathbb{P}_0} \Big[ \frac{\mathrm{d}\mathbb{P}_{\mathrm{v}_1}}{\mathrm{d}\mathbb{P}_0} \frac{\mathrm{d}\mathbb{P}_{\mathrm{v}_2}}{\mathrm{d}\mathbb{P}_0} (Z) \Big] \Big)^n$$

$$\leq \Big[ \cosh\Big( \frac{2\rho^2 \langle \mathrm{v}_1, \mathrm{v}_2 \rangle}{\sigma^2 + s\rho^2} \Big) + \alpha^2 \sinh\Big( \frac{2\rho^2 \langle \mathrm{v}_1, \mathrm{v}_2 \rangle}{\sigma^2 + s\rho^2} \Big) \Big]^n, \tag{B.5}$$

where $Z_i$ are independent copies of $Z = (Y, X)$. The following lemma by [62] establishes an upper bound of the right-hand side in (B.5).

566 **Lemma B.2** ([62]). *For any $x \geq 0$ and $0 \leq k \leq 1$, we have,*
$$\cosh(x) + k \sinh(x) \leq \exp(2kx) \vee \cosh(2x).$$

567 *Proof.* See the appendix of [62] for a detailed proof. □

568 Following from (B.3), (B.5), and Lemma B.2, we conclude
$$\left( \mathbb{E}_{\mathbb{P}_0^n} \left[ d\bar{\mathbb{P}}/d\mathbb{P}_0^n \right] \right)^2 \leq \frac{1}{|\mathcal{G}(s)|^2} \sum_{v_1, v_2 \in \mathcal{G}(s)} \left[ \exp\left( \frac{4\alpha^2 \rho^2 \langle v_1, v_2 \rangle}{\sigma^2 + s\rho^2} \right) \vee \cosh\left( \frac{4\rho^2 \langle v_1, v_2 \rangle}{\sigma^2 + s\rho^2} \right) \right]^n. \quad (B.6)$$

569 The following lemma shows that the right-hand side of (B.6) is of order $1 + o(1)$.

570 **Lemma B.3** ([62]). *For*
$$\gamma_n = o\left( \sqrt{\frac{s \log d}{n}} \wedge \frac{1}{\alpha^2} \cdot \frac{s \log d}{n} \right),$$

571 *if $s = o(d^{1/2 - \delta})$ for some absolute constant $\delta > 0$, it then holds that*
$$\frac{1}{|\mathcal{G}(s)|^2} \sum_{v_1, v_2 \in \mathcal{G}(s)} \left[ \exp\left( \frac{4\alpha^2 \rho^2 \langle v_1, v_2 \rangle}{\sigma^2 + s\rho^2} \right) \bigvee \cosh\left( \frac{4\rho^2 \langle v_1, v_2 \rangle}{\sigma^2 + s\rho^2} \right) \right]^n = 1 + o(1). \quad (B.7)$$

572 *Proof.* See §C.2 for a detailed proof. □

573 Combining Lemma B.3 and (B.6), we conclude that for $\gamma_n = o(\sqrt{s \log d / n} \wedge 1/\alpha^2 \cdot$
574 $s \log d / n)$, it holds that $(\mathbb{E}_{\mathbb{P}_0^n}[d\bar{\mathbb{P}}/d\mathbb{P}_0^n])^2 - 1 = o(1)$. Then following from (B.2), we have
575 $\liminf_{n \to \infty} R_n^*(\mathcal{G}_0, \mathcal{G}_1) \geq 1$, which concludes the proof of Proposition 3.1. □

## B.2 Proof of Theorem 3.2

577 *Proof.* It follows from Definition 2.2 that for $\gamma_n = o(\gamma_n^*)$, any hypothesis testing problem in
578 (2.7) is asymptotically powerless. It remains to show that for $\gamma_n = o(\sqrt{s^2/n} \wedge 1/\alpha^2 \cdot s/n)$, any
579 computationally tractable test is asymptotically powerless. First, we restrict the original estimation
580 problem to the following hypothesis testing problem,
$$H_0: Y = \epsilon \quad \text{versus} \quad H_1: Y = \begin{cases} X^\top \beta^* + \epsilon, & \text{with probability } \alpha \\ \eta \cdot X^\top \beta^* + \epsilon, & \text{with probability } 1 - \alpha \end{cases}. \quad (B.8)$$

581 In (B.8), we restrict $\beta^*$ to the set $\beta^* \in \{\rho \cdot v : v \in \mathcal{G}(s)\}$ with $\mathcal{G}(s) = \{v \in \{0,1\}^d : \|v\|_0 = s\}$.
582 We set $\epsilon \sim N(0, \sigma^2 + s\rho^2)$ under $H_0$ and $\epsilon \sim N(0, \sigma^2)$ under $H_1$ so that straightforward tests based
583 on mean and variance are not able to detect the existence of a nonzero parameter $\beta^*$.

584 By restricting the parameter space, we obtain a lower bound for the minimax risk. Recall that we
585 denote by $\bar{\mathbb{P}}_0$ and $\bar{\mathbb{P}}_v$ the distributions of $Z_q$, which denotes the response of the oracle to the query $q$
586 when the true distributions of the data are $\mathbb{P}_0$ and $\mathbb{P}_v$, correspondingly. We have
$$\bar{R}_n^*[\mathcal{G}_0, \mathcal{G}_1; \mathscr{A}, r] \geq \inf_{\phi \in \mathcal{H}(\mathscr{A}, r)} \left\{ \bar{\mathbb{P}}_0(\phi = 1) + \sup_{v \in \mathcal{G}(s)} \bar{\mathbb{P}}_v(\phi = 0) \right\}. \quad (B.9)$$

587 To show that any computationally tractable test is asymptotically powerless, it suffices to show that
588 the right-hand side of (B.9) is asymptotically lower bounded by one. By Theorem 4.2 of [53], we
589 know that this holds true if
$$T \cdot \sup_{q \in \mathcal{Q}} |\mathcal{C}(q)| / |\mathcal{G}(s)| = o(1),$$

590 where $\mathcal{C}(q)$ is defined as
$$\mathcal{C}(q) = \left\{ v \in \mathcal{G}(s) : \left| \mathbb{E}_{\mathbb{P}_v}[q(Z)] - \mathbb{E}_{\mathbb{P}_0}[q(Z)] \right| > \tau_q \right\}.$$

591 Here $\tau_q$ is the tolerance parameter defined in Definition 2.3, with $(Y, X)$ following $\mathbb{P}_v$. The following
592 lemma shows that $T \cdot \sup_{q \in \mathcal{Q}} |\mathcal{C}(q)| / |\mathcal{G}(s)| = o(1)$ if $\gamma_n$ is sufficiently small.

593 **Lemma B.4** ([53]). *For $s = o(d^{1/2 - \delta})$, $T = O(d^\mu)$, and*
$$\gamma_n = o\left( \frac{s^2}{n} \bigwedge \frac{1}{\alpha^2} \cdot \frac{s}{n} \right),$$

it holds that

$$T \cdot \sup_{q \in \mathcal{Q}} |\mathcal{C}(q)|/|\mathcal{G}(s)| = o(1). \tag{B.10}$$

*Proof.* See §C.3 for a detailed proof. $\qquad\square$

By combining Theorem 4.2 of [53] and Lemma B.4, we conclude that the right-hand side of (B.9) is asymptotically lower bounded by one. Therefore, it holds that $\liminf_{n\to\infty} \bar{R}_n^*[\mathcal{G}_0, \mathcal{G}_1; \mathscr{A}, r] \geq 1$, which concludes the proof of Theorem 3.2. $\qquad\square$

## B.3 Proof of Theorem A.2

*Proof.* Recall that we denote by $Z = (Y, X)$ and $\mathbb{P}_0$, $\mathbb{P}_{v^*}$ the distributions of $Z$ under the null and alternative hypotheses with $\beta^* = \rho \cdot v^*$, respectively. For the hypothesis testing problem in (A.1), the following lemma characterizes the expectations of the query functions defined in (A.5).

**Lemma B.5.** For any $v, v^* \in \bar{\mathcal{G}}(s)$ and

$$\gamma_n = \Omega\bigg( (\log n)^{1+1/\nu} \cdot \sqrt{\frac{s \log d}{n}} \bigwedge \frac{(\log n)^{1+2/\nu}}{\alpha^2} \cdot \frac{s \log d}{n} \bigg),$$

it holds that

$$\mathbb{E}_{\mathbb{P}_0}\big[ q_{1,v}(Y, X) \big] \leq 1/n, \quad \mathbb{E}_{\mathbb{P}_0}\big[ q_{2,v}(Y, X) \big] \leq 1/n. \tag{B.11}$$

In addition, it holds that

$$\mathbb{E}_{\mathbb{P}_{v^*}}\big[ q_{1,v^*}(Y, X) \big] \geq s\rho^2/2 \ \text{ if } \ \gamma_n = \Omega\bigg( (\log n)^{1+1/\nu} \cdot \sqrt{\frac{s \log d}{n}} \bigg),$$

$$\mathbb{E}_{\mathbb{P}_{v^*}}\big[ q_{2,v^*}(Y, X) \big] \geq \sqrt{\alpha^2 s\rho^2}/2 \ \text{ if } \ \gamma_n = \Omega\bigg( \frac{(\log n)^{1+2/\nu}}{\alpha^2} \cdot \frac{s \log d}{n} \bigg). \tag{B.12}$$

*Proof.* See §C.4 for a detailed proof. $\qquad\square$

In what follows, we establish an upper bound of the risk of $\phi = \phi_1 \vee \phi_2$. Recall that we define the test functions $\phi_1$ and $\phi_2$ in (A.6) with parameters

$$\tau_1 = CR^{2+1/\nu} \cdot (\log n)^{1+1/\nu} \cdot \sqrt{\frac{s \log d}{n}}, \quad \tau_2 = C'R^{1+1/\nu} \cdot (\log n)^{1/2+1/\nu} \cdot \sqrt{\frac{s \log d}{n}}. \tag{B.13}$$

where $C$ and $C'$ are absolute constants. Note that the total number of query functions $\{q_{1,v}\}_{v \in \mathcal{G}(s)}$ and $\{q_{2,v}\}_{v \in \mathcal{G}(s)}$ is $|\mathcal{Q}_\phi| = 2 \cdot \binom{d}{s} \cdot 2^s$. Therefore, following from (2.12) with $\xi = 1/d$, for sufficiently large $d$ and $n$, it holds that

$$\tau_{q_{1,v}} \leq C_0 R^{2+1/\nu} (\log n)^{1/2+1/\nu} \cdot \sqrt{\frac{s \log d}{n}}, \quad \tau_{q_{2,v}} \leq C_1 R^{1+1/\nu} (\log n)^{1/2+1/\nu} \cdot \sqrt{\frac{s \log d}{n}}, \tag{B.14}$$

where $\tau_{q_{1,v}}$ and $\tau_{q_{2,v}}$ are the tolerance parameters of $q_{1,v}$ and $q_{2,v}$ defined in Definition 2.3, and $C_0$, $C_1$ are positive absolute constants. We fix $C$ and $C'$ in (B.13) such that $\tau_1 \geq \tau_{q_{1,v}} + 1/n$ and $\tau_2 \geq \tau_{q_{2,v}} + 1/n$. Recall that we denote by $\bar{Z}_{1,v}$ and $\bar{Z}_{2,v}$ the responses of the statistical oracle to the query functions $q_{1,v}$ and $q_{2,v}$. Further recall that we denote by $\bar{\mathbb{P}}_0$ and $\bar{\mathbb{P}}_{v^*}$ the distributions of response of the statistical oracle to the query functions when the true distribution of the data is $\mathbb{P}_0$ and $\mathbb{P}_{v^*}$. Following from Lemma B.5, it holds for any $v \in \mathcal{G}(s)$ and $i \in \{1, 2\}$ that

$$\bar{\mathbb{P}}_0\big( \bar{Z}_{i,v} \geq \tau_i \big) \leq \bar{\mathbb{P}}_0\Big( \big| \bar{Z}_{i,v} - \mathbb{E}_{\mathbb{P}_0}\big[ q_{i,v}(Y, X) \big] \big| \geq \tau_{q_{i,v}} \Big).$$

Based on (2.11) with $\xi = 1/d$, it holds for $i \in \{1, 2\}$ that

$$\bar{\mathbb{P}}_0(\phi_i = 1) = \bar{\mathbb{P}}_0\bigg( \sup_{v \in \mathcal{G}(s)} \bar{Z}_{i,v} > \tau_i \bigg)$$

$$\leq \bar{\mathbb{P}}_0\bigg( \bigcup_{v \in \mathcal{G}(s)} \big\{ \big| \bar{Z}_{i,v} - \mathbb{E}_{\mathbb{P}_0}\big[ q_{i,v}(Y, X) \big] \big| > \tau_{q_{i,v}} \big\} \bigg) \leq 2/d. \tag{B.15}$$

Recall that we define $\phi = \phi_1 \vee \phi_2$. Therefore, we obtain from (B.15) that
$$\bar{\mathbb{P}}_0(\phi = 1) \leq \bar{\mathbb{P}}_0(\phi_1 = 1) + \bar{\mathbb{P}}_0(\phi_2 = 1) = 4/d. \tag{B.16}$$
In other words, the type-I error of $\phi$ is upper bounded by $4/d$. It remains to upper bound the type-II error of $\phi$. Following from the lower bound of SNR in (A.8), it holds that either $s\rho^2/4 \geq \tau_1$ or $\sqrt{\alpha^2 s \rho^2}/4 \geq \tau_2$ for a sufficiently large $n$. Following from Lemma B.5, if $s\rho^2/4 \geq \tau_1$, it holds that
$$\bar{\mathbb{P}}_{v^*}\big(\bar{Z}_{1,v^*} \leq \tau_1\big) \leq \bar{\mathbb{P}}_{v^*}\big(\bar{Z}_{1,v^*} \leq \mathbb{E}_{\mathbb{P}_{v^*}}\big[q_{1,v^*}(Y,X)\big] - \tau_1\big)$$
$$\leq \bar{\mathbb{P}}_{v^*}\Big(\big|\bar{Z}_{1,v^*} - \mathbb{E}_{\mathbb{P}_{v^*}}\big[q_{1,v^*}(Y,X)\big]\big| \geq \tau_{q_{1,v^*}}\Big), \tag{B.17}$$
where the last inequality holds since $\tau_1 > \tau_{q_{1,v^*}}$. Therefore, it follows from (2.11) with $\xi = 1/d$ that
$$\bar{\mathbb{P}}_{v^*}(\phi_1 = 0) = \bar{\mathbb{P}}_{v^*}\Big(\sup_{v \in \mathcal{G}(s)} \bar{Z}_{1,v} < \tau_1\Big) \leq \bar{\mathbb{P}}_{v^*}(\bar{Z}_{1,v^*} < \tau_1)$$
$$\leq \bar{\mathbb{P}}_{v^*}\Big(\big|\bar{Z}_{1,v^*} - \mathbb{E}_{\mathbb{P}_{v^*}}\big[q_{1,v^*}(Y,X)\big]\big| > \tau_{q_{1,v^*}}\Big) \leq 2/d. \tag{B.18}$$
Similarly, following from Lemma B.5, if $\sqrt{\alpha^2 s \rho^2}/4 \geq \tau_2$, it holds that,
$$\bar{\mathbb{P}}_{v^*}(\phi_2 = 0) = \bar{\mathbb{P}}_{v^*}\Big(\sup_{v \in \mathcal{G}(s)} \bar{Z}_{2,v} < \tau_2\Big) \leq \bar{\mathbb{P}}_{v^*}(\bar{Z}_{2,v^*} < \tau_2)$$
$$\leq \bar{\mathbb{P}}_{v^*}\Big(\big|\bar{Z}_{2,v^*} - \mathbb{E}_{\mathbb{P}_{v^*}}\big[q_{2,v^*}(Y,X)\big]\big| > \tau_{q_{2,v^*}}\Big) \leq 2/d, \tag{B.19}$$
where the last inequality holds since $\tau_2 > \tau_{q_{2,v^*}}$. Note that (B.18) and (B.19) holds for any $(\beta^*, \sigma) \in \bar{\mathcal{G}}_1(s, \gamma_n)$ if (A.8) holds. Therefore, by combining (B.18) and (B.19), we have
$$\sup_{(\beta^*,\sigma) \in \bar{\mathcal{G}}_1(s,\gamma_n)} \bar{\mathbb{P}}_{v^*}(\phi = 0) \leq \sup_{(\beta^*,\sigma) \in \bar{\mathcal{G}}_1(s,\gamma_n)} \big\{\bar{\mathbb{P}}_{v^*}(\phi_1 = 0) \wedge \bar{\mathbb{P}}_{v^*}(\phi_2 = 0)\big\} \leq 2/d. \tag{B.20}$$
In other words, the type-II error of $\phi$ is upper bounded by $2/d$. By combining (B.16) and (B.20), we conclude that if (A.8) holds, the risk for $\phi$ is of order $O(1/d)$, which completes the proof of Theorem A.2. $\qquad\square$

## B.4 Proof of Theorem A.3

*Proof.* The proof is similar to that of Theorem A.2 in §B.3. Recall that we denote by $Z = (Y, X)$ and $\mathbb{P}_0, \mathbb{P}_{v^*}$ the distributions of $Z$ under the null and alternative hypotheses with $\beta^* = \rho \cdot v^*$, respectively. The following lemma characterizes the expectations of the query functions defined in (A.11).

**Lemma B.6.** For any $v^* \in \bar{\mathcal{G}}(s)$ and
$$\gamma_n = \Omega\bigg((\log n)^{1+1/\nu} \cdot \sqrt{\frac{s^2 \log d}{n}} \bigwedge \frac{(\log n)^{1+2/\nu}}{\alpha^2} \cdot \frac{s \log d}{n}\bigg),$$
it holds that
$$\sup_{j \in [d]} \mathbb{E}_{\mathbb{P}_0}\big[q_{1,j}(Y,X)\big] \leq 1/n, \quad \sup_{j \in [d]} \mathbb{E}_{\mathbb{P}_0}\big[q_{2,j}(Y,X)\big] \leq 1/n. \tag{B.21}$$
In addition, it holds that
$$\sup_{j \in [d]} \mathbb{E}_{\mathbb{P}_{v^*}}\big[q_{1,j}(Y,X)\big] \geq \rho^2/2 \ \text{if} \ \gamma_n = \Omega\bigg((\log n)^{1+1/\nu} \cdot \sqrt{\frac{s^2 \log d}{n}}\bigg),$$
$$\sup_{j \in [d]} \big|\mathbb{E}_{\mathbb{P}_{v^*}}\big[q_{2,j}(Y,X)\big]\big| \geq \alpha\rho/2 \ \text{if} \ \gamma_n = \Omega\bigg(\frac{(\log n)^{1+2/\nu}}{\alpha^2} \cdot \frac{s \log d}{n}\bigg). \tag{B.22}$$

*Proof.* See §C.5 for a detailed proof. $\qquad\square$

In what follows, we upper bound the risk of the test function $\widetilde{\phi} = \widetilde{\phi}_1 \vee \widetilde{\phi}_2$. Recall that we define the test functions $\widetilde{\phi}_1$ and $\widetilde{\phi}_2$ in (A.11) with parameters
$$\widetilde{\tau}_1 = CR^{2+1/\nu} \cdot (\log n)^{1+1/\nu} \cdot \sqrt{\frac{\log d}{n}}, \quad \widetilde{\tau}_2 = C'R^{1+1/\nu} \cdot (\log n)^{1/2+1/\nu} \cdot \sqrt{\frac{\log d}{n}}, \tag{B.23}$$
where $C, C'$ are absolute constants. Note that the total number of query functions $\{q_{1,j}\}_{j \in [d]}$ and $\{q_{2,j}\}_{j \in [d]}$ is $|\mathcal{Q}_{\widetilde{\phi}}| = 2d$. Therefore, following from Definition 2.3 with $\xi = 1/d$, for sufficiently

large $d$ and $n$, the tolerance parameters of $q_{1,j}$ and $q_{2,j}$ are upper bounded as follows,

$$\tau_{q_{1,j}} \leq C_0' R^{2+1/\nu} (\log n)^{1/2+1/\nu} \cdot \sqrt{\frac{\log d}{n}}, \quad \tau_{q_{2,j}} \leq C_1' R^{1+1/\nu} (\log n)^{1/2+1/\nu} \cdot \sqrt{\frac{\log d}{n}},$$
(B.24)

where $C_0'$ and $C_1'$ are positive absolute constants. We fix $C$ and $C'$ in (B.13) such that $\widetilde{\tau}_1 \geq \tau_{q_{1,j}} + 1/n$ and $\widetilde{\tau}_2 \geq \tau_{q_{2,j}} + 1/n$. Recall that we denote by $\bar{Z}_{1,j}$ and $\bar{Z}_{2,j}$ the responses of the statistical oracle to the query functions $q_{1,j}$ and $q_{2,j}$, respectively. Further recall that we denote by $\bar{\mathbb{P}}_0$ and $\bar{\mathbb{P}}_{v^*}$ the distributions of response of the statistical oracle to the query functions when the true distribution of the data is $\mathbb{P}_0$ and $\mathbb{P}_{v^*}$. Following from Lemma B.6, for any $j \in [d]$ and $i \in \{1, 2\}$, it holds that

$$\bar{\mathbb{P}}_0\big(\bar{Z}_{i,j} \geq \widetilde{\tau}_1\big) \leq \bar{\mathbb{P}}_0\Big(\big|\bar{Z}_{i,j} - \mathbb{E}_{\mathbb{P}_0}[q_{i,j}(Y, X)]\big| \geq \tau_{q_{i,j}}\Big).$$

Based on (2.11) with $\xi = 1/d$, it holds for $i \in \{1, 2\}$ that

$$\bar{\mathbb{P}}_0(\widetilde{\phi}_i = 1) = \bar{\mathbb{P}}_0\left(\sup_{j \in [d]} \bar{Z}_{i,j} > \widetilde{\tau}_i\right)$$

$$\leq \bar{\mathbb{P}}_0\left(\bigcup_{j \in [d]} \left\{\big|\bar{Z}_{i,j} - \mathbb{E}_{\mathbb{P}_0}[q_{i,j}(Y, X)]\big| > \tau_{q_{i,j}}\right\}\right) \leq 2/d, \qquad \text{(B.25)}$$

Recall that we define $\widetilde{\phi} = \widetilde{\phi}_1 \vee \widetilde{\phi}_2$. Therefore, we obtain from (B.25) that

$$\bar{\mathbb{P}}_0(\widetilde{\phi} = 1) \leq \bar{\mathbb{P}}_0(\widetilde{\phi}_1 = 1) + \bar{\mathbb{P}}_0(\widetilde{\phi}_2 = 1) = 4/d. \qquad \text{(B.26)}$$

In other words, the type-I error of $\widetilde{\phi}$ is upper bounded by $4/d$. It remains to upper bound the type-II error of $\phi$. Following from the lower bound on SNR in (A.14), it holds that either $\rho^2/4 \geq \widetilde{\tau}_1$ or $\alpha\rho/4 \geq \widetilde{\tau}_2$ with a sufficiently large $n$. For any $v^* \in \bar{\mathcal{G}}(s)$, let $j^* = \text{argmax}_{j \in [d]} \mathbb{E}_{\mathbb{P}_{v^*}}[q_{1,j}(Y, X)]$. Following from Lemma B.5, if $\rho^2/4 \geq \widetilde{\tau}_1$, it holds that

$$\bar{\mathbb{P}}_{v^*}\big(\bar{Z}_{1,j^*} \leq \widetilde{\tau}_1\big) \leq \bar{\mathbb{P}}_{v^*}\Big(\bar{Z}_{1,j^*} \leq \mathbb{E}_{\mathbb{P}_{v^*}}[q_{1,j^*}(Y, X)] - \widetilde{\tau}_1\Big)$$

$$\leq \bar{\mathbb{P}}_{v^*}\Big(\big|\bar{Z}_{1,j^*} - \mathbb{E}_{\mathbb{P}_{v^*}}[q_{1,j^*}(Y, X)]\big| \geq \tau_{q_{1,j^*}}\Big), \qquad \text{(B.27)}$$

where the last inequality holds since $\widetilde{\tau}_1 > \tau_{q_{1,j^*}}$. Therefore, we conclude from (2.11) with $\xi = 1/d$ that

$$\bar{\mathbb{P}}_{v^*}(\widetilde{\phi}_1 = 0) = \bar{\mathbb{P}}_{v^*}\left(\sup_{j \in [d]} \bar{Z}_{1,j} < \widetilde{\tau}_1\right) \leq \bar{\mathbb{P}}_{v^*}(\bar{Z}_{1,j^*} < \widetilde{\tau}_1)$$

$$\leq \bar{\mathbb{P}}_{v^*}\Big(\big|\bar{Z}_{1,j^*} - \mathbb{E}_{\mathbb{P}_{v^*}}[q_{1,j^*}(Y, X)]\big| > \tau_{q_{1,j^*}}\Big) \leq 2/d. \qquad \text{(B.28)}$$

Similarly, for any $v^* \in \bar{\mathcal{G}}(s)$, let $k^* = \text{argmax}_{j \in [d]} \mathbb{E}_{\mathbb{P}_{v^*}}[q_{2,j}(Y, X)]$ and $\ell^* = \text{argmin}_{j \in [d]} \mathbb{E}_{\mathbb{P}_{v^*}}[q_{2,j}(Y, X)]$. Following from Lemma B.5, if $\alpha\rho/4 \geq \widetilde{\tau}_2$, it holds that either $\mathbb{E}[q_{2,k^*}(Y, X)] \geq \alpha\rho/2$ or $\mathbb{E}[q_{2,\ell^*}(Y, X)] \leq -\alpha\rho/2$. If it holds that $\mathbb{E}_{\mathbb{P}_v^*}[q_{2,k^*}(Y, X)] \geq \alpha\rho/2 \geq 2\widetilde{\tau}_2$, we have

$$\bar{\mathbb{P}}_{v^*}(\widetilde{\phi}_2 = 0) \leq \bar{\mathbb{P}}_{v^*}\left(\sup_{j \in [d]} \bar{Z}_{2,j} < \widetilde{\tau}_2\right) \leq \bar{\mathbb{P}}_{v^*}(\bar{Z}_{2,k^*} < \widetilde{\tau}_2)$$

$$\leq \bar{\mathbb{P}}_{v^*}\Big(\big|\bar{Z}_{2,k^*} - \mathbb{E}_{\mathbb{P}_v}[q_{2,k^*}(Y, X)]\big| > \tau_{q_{2,k^*}}\Big) \leq 2/d, \qquad \text{(B.29)}$$

where the last inequality holds since $\widetilde{\tau}_2 > \tau_{q_{2,k^*}}$. If it holds that $\mathbb{E}_{\mathbb{P}_v^*}[q_{2,\ell^*}(Y, X)] \leq -\alpha\rho/2 \leq -2\widetilde{\tau}_2$, we have

$$\bar{\mathbb{P}}_{v^*}(\widetilde{\phi}_2 = 0) \leq \bar{\mathbb{P}}_{v^*}\left(\inf_{j \in [d]} \bar{Z}_{2,j} > -\widetilde{\tau}_2\right) \leq \bar{\mathbb{P}}_{v^*}(\bar{Z}_{2,\ell^*} > -\widetilde{\tau}_2)$$

$$\leq \bar{\mathbb{P}}_{v^*}\Big(\big|\bar{Z}_{2,\ell^*} - \mathbb{E}_{\mathbb{P}_v}[q_{2,\ell^*}(Y, X)]\big| > \tau_{q_{2,\ell^*}}\Big) \leq 2/d, \qquad \text{(B.30)}$$

where the last inequality holds since $\widetilde{\tau}_2 > \tau_{q_{2,\ell^*}}$. Note that (B.28), (B.29), and (B.30) holds for any $(\beta^*, \sigma) \in \bar{\mathcal{G}}_1(s, \gamma_n)$ if (A.14) holds. Therefore, by combining (B.28), (B.29), and (B.30), we have

$$\sup_{(\beta^*, \sigma) \in \bar{\mathcal{G}}_1(s, \gamma_n)} \bar{\mathbb{P}}_{v^*}(\widetilde{\phi} = 0) \leq \sup_{(\beta^*, \sigma) \in \bar{\mathcal{G}}_1(s, \gamma_n)} \big\{\bar{\mathbb{P}}_{v^*}(\widetilde{\phi}_1 = 0) \wedge \bar{\mathbb{P}}_{v^*}(\widetilde{\phi}_2 = 0)\big\} \leq 2/d. \qquad \text{(B.31)}$$

In other words, the type-II error of $\phi$ is upper bounded by $2/d$. By combining (B.26) and (B.31), we conclude that if (A.14) holds, the risk for $\widetilde{\phi}$ is of order $O(1/d)$, which completes the proof of Theorem A.3. □

## B.5 Proof of Theorem 3.3

*Proof.* We prove by contradiction in the following. We assume that there exist an absolute constant $\eta$ and an algorithm $\mathscr{A} \in \mathcal{A}(T)$ with $T = O(d^\eta)$ that estimates $\beta^*$ in (2.6), such that for any given oracle $r \in \mathcal{R}[\xi, n, T, \eta(\mathcal{Q})]$, it holds that

$$\bar{\mathbb{P}}\big(\|\widehat{\beta} - \beta^*\|_2^2 / \sigma^2 \geq \gamma_n/16\big) = o(1), \tag{B.32}$$

where $\widehat{\beta}$ is the estimator of $\beta^*$. In other words, it holds that $\|\widehat{\beta} - \beta^*\|_2^2/\sigma^2 \leq \gamma_n/16$ with probability $1 - o(1)$. Recall that we set $\|\beta^*\|^2/\sigma^2 = \gamma_n$. Based on (B.32), it holds with probability $1 - o(1)$ that

$$\|\widehat{\beta} + \beta^*\|_2^2 \leq (\|\widehat{\beta} - \beta^*\|_2 + 2\|\beta^*\|_2)^2 \leq 2\|\widehat{\beta} - \beta^*\|_2^2 + 8\|\beta^*\|_2^2 \leq (1/8 + 8) \cdot \sigma^2 \gamma_n. \tag{B.33}$$

Combining (B.32) and (B.33), it follows from the Cauchy-Schwartz inequality that

$$\big|\|\widehat{\beta}\|_2^2 - \|\beta^*\|_2^2\big|^2 = \big|(\widehat{\beta} - \beta^*)^\top (\widehat{\beta} + \beta^*)\big|^2 \leq \|\widehat{\beta} - \beta^*\|_2^2 \cdot \|\widehat{\beta} + \beta^*\|_2^2 \leq 5/8 \cdot \sigma^4 \gamma_n^2, \tag{B.34}$$

which holds with probability $1 - o(1)$. In what follows, we construct an asymptotically powerful test with $T = O(d^\eta)$ query complexity for the hyppthesis testing problem in (2.7). We set $\phi = \mathbb{1}\{\|\widehat{\beta}\|_2^2 \geq \gamma_n/5\}$, where $\widehat{\beta}$ is the estimator of $\beta^*$ given the algorithm $\mathscr{A}$. Following from (B.32), it holds with probability $1 - o(1)$ that $\|\widehat{\beta}\|_2^2/\sigma^2 \leq \gamma_n/16$ under the null hypothesis with $\beta^* = 0$. Meanwhile, following from (B.34), it holds with probability $1 - o(1)$ that $\|\widehat{\beta}\|_2^2/\sigma^2 \geq \gamma_n/5$ under the alternative hypothesis with $\beta^* \neq 0$ and $\|\beta^*\|^2/\sigma^2 = \gamma_n$. In other words, $\phi$ is asymptotically powerful and computationally tractable with $\gamma_n = o(\sqrt{s^2/n} \wedge 1/\alpha^2 \cdot s \log d/n)$, which contradicts the computational minimax separation rate in (A.16). □

# C  Proof of Lemmas

In this section, we lay out the proof of the lemmas in §B.

## C.1  Proof of Lemma B.1

*Proof.* It follows from the model in (B.1) that under the alternative hypothesis,

$$Z = (Y, X) \sim \alpha \cdot N\big(0, \Sigma(v)\big) + \frac{1-\alpha}{2} \cdot N\big(0, \Sigma(v)\big) + \frac{1-\alpha}{2} \cdot N\big(0, \Sigma(-v)\big),$$

$$\sim \frac{1+\alpha}{2} \cdot N\big(0, \Sigma(v)\big) + \frac{1-\alpha}{2} \cdot N\big(0, \Sigma(-v)\big),$$

where $\Sigma(v)$ is the covariance matrix

$$\Sigma(v) = \begin{bmatrix} \sigma^2 + s\rho^2 & \rho v^\top \\ \rho v & I_d \end{bmatrix} \in \mathbb{R}^{(d+1) \times (d+1)}. \tag{C.1}$$

Meanwhile, we have $Z = (Y, X) \sim N(0, \Sigma_0)$ under the null hypothesis, where we denote by $\Sigma_0 = \Sigma(0)$. Recall that we denote by $\mathbb{P}_v$ and $\mathbb{P}_0$ the distributions of $Z$ under the alternative and null hypotheses, respectively. Therefore, it holds that

$$\frac{d\mathbb{P}_v}{d\mathbb{P}_0}(Z) = \frac{1+\alpha}{2} \cdot \sqrt{\frac{\det(\Sigma_0)}{\det(\Sigma(v))}} \cdot \exp\left(-\frac{Z\big(\Sigma^{-1}(v) - \Sigma_0^{-1}\big)Z^\top}{2}\right)$$

$$+ \frac{1-\alpha}{2} \cdot \sqrt{\frac{\det(\Sigma_0)}{\det(\Sigma(-v))}} \cdot \exp\left(-\frac{Z\big(\Sigma^{-1}(-v) - \Sigma_0^{-1}\big)Z^\top}{2}\right), \tag{C.2}$$

where we denote by $\Sigma^{-1}(v)$ the inverse matrix of $\Sigma(v)$. We denote by $\xi$ the Bernoulli random variable with distribution

$$\mathbb{P}(\xi = 1) = \frac{1+\alpha}{2}, \quad \mathbb{P}(\xi = -1) = \frac{1-\alpha}{2}. \tag{C.3}$$

Therefore, it follows from (C.2) that

$$\frac{d\mathbb{P}_v}{d\mathbb{P}_0}(Z) = \mathbb{E}_\xi\left[\sqrt{\frac{\det(\Sigma_0)}{\det(\Sigma(\xi v))}} \cdot \exp\left(-\frac{Z\big(\Sigma^{-1}(\xi v) - \Sigma_0^{-1}\big)Z^\top}{2}\right)\right]. \tag{C.4}$$

Following from (C.4), for $v_1$ and $v_2$ in $\mathcal{G}(s)$, we have

$$\mathbb{E}_{\mathbb{P}_0}\left[\frac{d\mathbb{P}_{v_1}}{d\mathbb{P}_0}\frac{d\mathbb{P}_{v_2}}{d\mathbb{P}_0}(Z)\right] = \mathbb{E}_{\mathbb{P}_0}\mathbb{E}_{\xi_1,\xi_2}\left[\frac{\det(\Sigma_0)}{\sqrt{\det\big(\Sigma(\xi_1 v_1)\big) \cdot \det\big(\Sigma(\xi_2 v_2)\big)}}\right. \tag{C.5}$$
$$\left.\cdot \exp\left(-1/2 \cdot Z^\top\big(\Sigma^{-1}(\xi_1 v_1) + \Sigma^{-1}(\xi_1 v_2) - 2\Sigma_0^{-1}\big)Z\right)\right],$$

where $\xi_1$ and $\xi_2$ are independent copies of $\xi$ defined in (C.3). In what follows, we calculate the right-hand side of (C.5) by invoking Fubini's theorem. We first calculate the right-hand side of (C.5) by integrating under $\mathbb{P}_0$ and obtain that

$$\mathbb{E}_{\mathbb{P}_0}\left[\exp\left(-1/2 \cdot Z^\top\big(\Sigma^{-1}(\xi_1 v_1) + \Sigma^{-1}(\xi_1 v_2) - 2\Sigma_0^{-1}\big)Z\right)\right]$$
$$= \frac{1}{\sqrt{(2\pi)^{d+1} \cdot \det(\Sigma_0)}} \cdot \int_{z \in \mathbb{R}^{d+1}} \exp\left(-1/2 \cdot z^\top\big(\Sigma^{-1}(\xi_1 v_1) + \Sigma^{-1}(\xi_1 v_2) - \Sigma_0^{-1}\big)z\right)d\mathbb{P}_0(z)$$
$$= \Big(\det\big(\Sigma^{-1}(\xi_1 v_1) + \Sigma^{-1}(\xi_1 v_2) - \Sigma_0^{-1}\big) \cdot \det(\Sigma_0)\Big)^{-1/2}. \tag{C.6}$$

By plugging (C.6) into (C.5), we obtain

$$\mathbb{E}_{\xi_1,\xi_2}\mathbb{E}_{\mathbb{P}_0}\left[\frac{\det(\Sigma_0)}{\sqrt{\det\big(\Sigma(\xi_1 v_1)\big) \cdot \det\big(\Sigma(\xi_2 v_2)\big)}} \cdot \exp\left(-1/2 \cdot Z^\top\big(\Sigma^{-1}(\xi_1 v_1) + \Sigma^{-1}(\xi_1 v_2) - 2\Sigma_0^{-1}\big)Z\right)\right]$$
$$= \mathbb{E}_{\xi_1,\xi_2}\left[\frac{\det(\Sigma_0)}{\sqrt{\det\big(\Sigma(\xi_1 v_1)\big) \cdot \det\big(\Sigma(\xi_2 v_2)\big)}} \cdot \Big(\det\big(\Sigma^{-1}(\xi_1 v_1) + \Sigma^{-1}(\xi_1 v_2) - \Sigma_0^{-1}\big)\det(\Sigma_0)\Big)^{-1/2}\right]$$
$$= \sqrt{\det(\Sigma_0)} \cdot \mathbb{E}_{\xi_1,\xi_2}\left[\det\big(\Sigma(\xi_1 v_1) + \Sigma(\xi_2 v_2) - \Sigma(\xi_1 v_1)\Sigma_0^{-1}\Sigma(\xi_2 v_2)\big)^{-1/2}\right]. \tag{C.7}$$

Meanwhile, by (C.1) it holds that $\det(\Sigma_0) = \sigma^2 + s\rho^2$ and

$$\Sigma(\xi_1 v_1) + \Sigma(\xi_2 v_2) - \Sigma(\xi_1 v_1) \cdot \Sigma_0^{-1} \cdot \Sigma(\xi_2 v_2)$$
$$= \begin{bmatrix} \sigma^2 + s\rho^2(1 - \xi_1\xi_2 \cdot v_1^\top v_2) & 0 \\ 0 & I_d - (\rho^2\xi_1\xi_2)/(\sigma^2 + s\rho^2) \cdot v_1 v_2^\top \end{bmatrix}. \tag{C.8}$$

Therefore, we are able to calculate the right-hand side of (C.7) explicitly. Combining (C.5) and (C.7) and apply Fubini's theorem, we obtain that

$$\mathbb{E}_{\mathbb{P}_0}\left[\frac{d\mathbb{P}_{v_1}}{d\mathbb{P}_0}\frac{d\mathbb{P}_{v_2}}{d\mathbb{P}_0}(Z)\right] = \mathbb{E}_{\xi_1,\xi_2}\left[1 - \frac{\rho^2\xi_1\xi_2}{\sigma^2 + s\rho^2} \cdot \langle v_1, v_2\rangle\right]. \tag{C.9}$$

Recall that $\xi_1$ and $\xi_2$ are independent copies of $\xi$ defined in (C.3), it then holds that

$$\mathbb{E}_{\mathbb{P}_0}\left[\frac{d\mathbb{P}_{v_1}}{d\mathbb{P}_0}\frac{d\mathbb{P}_{v_2}}{d\mathbb{P}_0}(Z)\right] = \frac{1 + \alpha^2(\sigma^2 + s\rho^2)^{-1}\rho^2 \cdot \langle v_1, v_2\rangle}{1 - (\sigma^2 + s\rho^2)^{-2}\rho^4 \cdot \langle v_1, v_2\rangle^2}. \tag{C.10}$$

Meanwhile, for $0 \le x < 1/2$ and $0 \le k \le 1$, we have

$$\frac{1 + kx}{1 - x^2} \le \cosh(2x) + k \cdot \sinh(2x).$$

Therefore, following from (C.10) with $s\rho^2 = o(1)$, we obtain that

$$\mathbb{E}_{\mathbb{P}_0}\left[\frac{d\mathbb{P}_{v_1}}{d\mathbb{P}_0}\frac{d\mathbb{P}_{v_2}}{d\mathbb{P}_0}(Z)\right] \le \cosh\left(\frac{2\rho^2 \cdot \langle v_1, v_2\rangle}{\sigma^2 + s\rho^2}\right) + \alpha^2 \cdot \sinh\left(\frac{2\rho^2 \cdot \langle v_1, v_2\rangle}{\sigma^2 + s\rho^2}\right), \tag{C.11}$$

which concludes the proof of Lemma B.1. □

 **C.2 Proof of Lemma B.3**

 *Proof.* In what follows, we establish the upper bound of the following sum,

$$S = \frac{1}{|\mathcal{G}(s)|^2} \sum_{v_1, v_2 \in \mathcal{G}(s)} \left[ \exp\left( \frac{4\alpha^2 \rho^2 \cdot \langle v_1, v_2 \rangle}{\sigma^2 + s\rho^2} \right) \bigvee \cosh\left( \frac{4\rho^2 \cdot \langle v_1, v_2 \rangle}{\sigma^2 + s\rho^2} \right) \right]^n. \qquad \text{(C.12)}$$

 In specific, we show that $S = 1 + o(1)$ if it holds that

$$\gamma_n = o\left( \sqrt{\frac{s \log d}{n}} \bigwedge \frac{1}{\alpha^2} \cdot \frac{s \log d}{n} \right).$$

 The proof strategy is similar to that of Theorem 3.1 by [62]. We define $\mathcal{V}(s)$ the class of index set as
 follows,

$$\mathcal{V}(s) = \{ \mathcal{S} \subseteq [d] : |\mathcal{S}| = s \}.$$

 We further denote by $\mathcal{S}_1$ and $\mathcal{S}_2$ two independent random variables, which are uniformly distributed
 over $\mathcal{V}(s)$ and

$$T = |\mathcal{S}_1 \cap \mathcal{S}_2|.$$

 We obtain from (C.12) the following upper bound of $S$,

$$S \leq \mathbb{E}_T \left[ \left\{ \exp\left( \frac{4\alpha^2 \rho^2 T}{\sigma^2 + s\rho^2} \right) \bigvee \cosh\left( \frac{4\rho^2 T}{\sigma^2 + s\rho^2} \right) \right\}^n \right]. \qquad \text{(C.13)}$$

 Let $\{\eta_i\}_{i \in [n]}$ be $n$ independent Rademacher random variables and $U$ be their sum. Following from
 (C.13) and the fact that $\cosh(x) = \mathbb{E}_{\eta_i}[\exp(\eta_i x)]$, we obtain

$$S \leq \mathbb{E}_T \left[ \exp\left( \frac{4n\alpha^2 \rho^2 T}{\sigma^2 + s\rho^2} \right) \bigvee \mathbb{E}_U \left[ \exp\left( \frac{4\rho^2 U T}{\sigma^2 + s\rho^2} \right) \right] \right]$$

$$= \mathbb{E}_T \mathbb{E}_U \left[ \exp\left( \frac{4n\alpha^2 \rho^2 T}{\sigma^2 + s\rho^2} \right) \bigvee \exp\left( \frac{4\rho^2 U T}{\sigma^2 + s\rho^2} \right) \right]. \qquad \text{(C.14)}$$

 We apply Fubini's theorem to calculate the right-hand side of (C.14). We first calculate the expectation
 with respect to $T$. Recall that we denote by $T = |\mathcal{S}_1 \cap \mathcal{S}_2|$. Therefore, it holds that

$$\mathbb{E}_T \left[ \exp\left( \frac{4n\alpha^2 \rho^2 T}{\sigma^2 + s\rho^2} \right) \bigvee \exp\left( \frac{4\rho^2 U T}{\sigma^2 + s\rho^2} \right) \right]$$

$$= \mathbb{E}_T \left[ \left\{ \exp\left( \frac{4n\alpha^2 \rho^2}{\sigma^2 + s\rho^2} \right) \bigvee \exp\left( \frac{4\rho^2 U}{\sigma^2 + s\rho^2} \right) \right\}^T \right]$$

$$\leq \sup_{\mathcal{S} \in \mathcal{V}(s)} \mathbb{E}_{\mathcal{S}_2} \left[ \left\{ \exp\left( \frac{4n\alpha^2 \rho^2}{\sigma^2 + s\rho^2} \right) \bigvee \exp\left( \frac{4\rho^2 U}{\sigma^2 + s\rho^2} \right) \right\}^{|\mathcal{S} \cap \mathcal{S}_2|} \right], \qquad \text{(C.15)}$$

 where the last inequality holds since $\mathcal{S}_1$ is uniformly distributed over $\mathcal{V}(s)$. We fix an arbitrary
 $\mathcal{S} \in \mathcal{V}(s)$ and denote by $|\mathcal{S} \cap \mathcal{S}_2| = \sum_{i \in \mathcal{V}} v_i$, where $\{v_i\}_{i \in \mathcal{V}}$ are random variables that takes value
 one if $i \in \mathcal{S} \cap \mathcal{S}_2$ and zero otherwise. Recall that $\mathcal{S}_2$ is uniformly distributed over $\mathcal{C}(s)$. Therefore,
 $v_i$ takes value one with probability $s/d$ and zero otherwise. Meanwhile, for $i \neq j$, $v_i$ and $v_j$ are
 negatively associated with each other. Thus, it holds that

$$\mathbb{E}_{\mathcal{S}_2} \left[ \left\{ \exp\left( \frac{4n\alpha^2 \rho^2}{\sigma^2 + s\rho^2} \right) \bigvee \exp\left( \frac{4\rho^2 U}{\sigma^2 + s\rho^2} \right) \right\}^{|\mathcal{S} \cap \mathcal{S}_2|} \right]$$

$$\leq \prod_{i \in \mathcal{V}} \mathbb{E}_{v_i} \left[ \left\{ \exp\left( \frac{4n\alpha^2 \rho^2}{\sigma^2 + s\rho^2} \right) \bigvee \exp\left( \frac{4\rho^2 U}{\sigma^2 + s\rho^2} \right) \right\}^{v_i} \right]$$

$$= \left( s/d \cdot \left[ \exp\left( \frac{4n\alpha^2 \rho^2}{\sigma^2 + s\rho^2} \right) \bigvee \exp\left( \frac{4\rho^2 U}{\sigma^2 + s\rho^2} \right) \right] + 1 - s/d \right)^s. \qquad \text{(C.16)}$$

Since the inequality in (C.16) holds for any $\mathcal{S} \in \mathcal{V}(s)$, it holds for the supreme over $\mathcal{V}(s)$. By plugging (C.16) into (C.15), we obtain that

$$\mathbb{E}_T\left[\exp\left(\frac{4n\alpha^2\rho^2 T}{\sigma^2 + s\rho^2}\right) \bigvee \exp\left(\frac{4\rho^2 UT}{\sigma^2 + s\rho^2}\right)\right]$$

$$\leq 1 + \sum_{k=1}^{s}\binom{s}{k}\left(\frac{s}{d}\right)^k \cdot \left[\exp\left(\frac{4n\alpha^2\rho^2}{\sigma^2 + s\rho^2}\right) \bigvee \exp\left(\frac{4\rho^2 U}{\sigma^2 + s\rho^2}\right) - 1\right]^k. \tag{C.17}$$

Finally, by combining (C.14) and (C.17), we obtain from Fubini's theorem that

$$S - 1 \leq \sum_{k=1}^{s}\binom{s}{k}\left(\frac{s}{d}\right)^k \cdot \mathbb{E}_U\left[\left\{\exp\left(\frac{4n\alpha^2\rho^2}{\sigma^2 + s\rho^2}\right) \bigvee \exp\left(\frac{4\rho^2 U}{\sigma^2 + s\rho^2}\right) - 1\right\}^k\right]$$

$$\leq \sum_{k=1}^{s}\binom{s}{k}\left(\frac{s}{d}\right)^k \cdot \left[\exp\left(\frac{4n\alpha^2\rho^2}{\sigma^2 + s\rho^2}\right) - 1\right]^k$$

$$+ \binom{s}{k}\left(\frac{s}{d}\right)^k \cdot \mathbb{E}_U\left[\left\{\exp\left(\frac{4\rho^2 U}{\sigma^2 + s\rho^2}\right) - 1\right\}^k \,\middle|\, U \geq n\alpha^2\right]. \tag{C.18}$$

It now suffices to show that the right-hand side of (C.18) is of order $o(1)$. The following lemma upper bounds the first term on the right-hand side of (C.18).

**Lemma C.1** ([62]). For $\gamma_n = s\rho^2/\sigma^2 = o(1/\alpha^2 \cdot s\log d/n)$, it holds that

$$\sum_{k=1}^{s}\binom{s}{k}\left(\frac{s}{d}\right)^k \cdot \left[\exp\left(\frac{4n\alpha^2\rho^2}{\sigma^2 + s\rho^2}\right) - 1\right]^k = o(1). \tag{C.19}$$

*Proof.* See §C.6 for a detailed proof. □

We denote by $Q = 4\rho^2 U/(\sigma^2 + s\rho^2)$. Note that $\exp(x) - 1 \leq 2x$ for $0 < x < 1$. Therefore, the following upper bound of the second term on the right-hand side of (C.18) holds,

$$\sum_{k=1}^{s}\binom{s}{k}\left(\frac{s}{d}\right)^k \cdot \mathbb{E}_U\left[\left\{\exp\left(\frac{4\rho^2 U}{\sigma^2 + s\rho^2}\right) - 1\right\}^k \,\middle|\, U \geq 0\right]$$

$$\leq \sum_{k=1}^{s}\left(\frac{s^2 e}{kd}\right)^k \cdot \mathbb{E}_U\left[(2|Q|)^k + \exp(k|Q|) \cdot \mathbb{1}\{|Q| \geq 1\}\right]$$

$$\leq \underbrace{\sum_{k=1}^{s}\mathbb{E}_U\left[\frac{2s^2 e|Q|}{kd}\right]^k}_{(i)} + \underbrace{\sum_{k=1}^{s}\left(\frac{s^2 e}{kd}\right)^k \cdot \mathbb{E}_U\left[\exp(k|Q|) \cdot \mathbb{1}\{|Q| \geq 1\}\right]}_{(ii)}. \tag{C.20}$$

The following Lemma establishes the upper bounds of terms (i) and (ii) in (C.20).

**Lemma C.2** ([62]). For $\gamma_n = s\rho^2/\sigma^2 = o(\sqrt{s\log d/n})$, it holds that

$$T_1 = \sum_{k=1}^{s}\mathbb{E}_U\left[\frac{2s^2 e|Q|}{kd}\right]^k = o(1),$$

$$T_2 = \sum_{k=1}^{s}\left(\frac{s^2 e}{kd}\right)^k \cdot \mathbb{E}_U\left[\exp(k|Q|) \cdot \mathbb{1}(|Q| \geq 1)\right] = o(1). \tag{C.21}$$

*Proof.* See §C.7 for a detailed proof. □

By combining (C.18) and (C.20), we obtain from Lemmas C.1 and C.2 that $S - 1 = o(1)$ for

$$\gamma_n = o\left(\sqrt{\frac{s\log d}{n}} \bigwedge \frac{1}{\alpha^2} \cdot \frac{s\log d}{n}\right),$$

which concludes the proof of Lemma B.3. □

 **C.3 Proof of Lemma B.4**

737 *Proof.* In what follows, we prove that $T \cdot \sup_{q \in \mathcal{Q}} |\mathcal{C}(q)|/|\mathcal{G}(s)| = o(1)$ under the assumptions of
738 Lemma B.4. Our proof strategy is similar to that of Theorem 5.3 by [53]. As $|\mathcal{G}(s)|$ is given, we focus
739 on upper bounding $|\mathcal{C}(q)|$. We first partition $\mathcal{C}(q)$ into two parts, namely, $\mathcal{C}_1(q)$ and $\mathcal{C}_2(q)$, where

$$\mathcal{C}_1(q) = \Big\{ \mathrm{v} \in \mathcal{G}(s) : \mathbb{E}_{\mathbb{P}_0}\big[q(Z)\big] - \mathbb{E}_{\mathbb{P}_{\mathrm{v}}}\big[q(Z)\big] > \tau_q \Big\},$$

740 and $\mathcal{C}_2(q) = \mathcal{C}(q) \backslash \mathcal{C}_1(q)$. It holds that

$$\sup_{q \in \mathcal{Q}} |\mathcal{C}(q)| \leq \sup_{q \in \mathcal{Q}} |\mathcal{C}_1(q)| + \sup_{q \in \mathcal{Q}} |\mathcal{C}_2(q)|. \tag{C.22}$$

741 We introduce the following distributions,

$$\mathbb{P}_{\mathcal{C}_1(q)} = \frac{1}{|\mathcal{C}_1(q)|} \sum_{\mathrm{v} \in \mathcal{C}_1(q)} \mathbb{P}_{\mathrm{v}}, \quad \mathbb{P}_{\mathcal{C}_2(q)} = \frac{1}{|\mathcal{C}_2(q)|} \sum_{\mathrm{v} \in \mathcal{C}_2(q)} \mathbb{P}_{\mathrm{v}}.$$

742 We further denote by

$$\bar{\mathcal{C}}_\ell(q, \mathrm{v}) = \operatorname*{argmax}_{\mathcal{C}} \bigg\{ \frac{1}{|\mathcal{C}|} \sum_{\mathrm{v}' \in \mathcal{C}} \mathbb{E}_{\mathbb{P}_0}\bigg[ \frac{\mathrm{d}\mathbb{P}_{\mathrm{v}}}{\mathrm{d}\mathbb{P}_0} \frac{\mathrm{d}\mathbb{P}_{\mathrm{v}'}}{\mathrm{d}\mathbb{P}_0}(X) \bigg] - 1 \,\bigg|\, |\mathcal{C}| = |\mathcal{C}_\ell(q)| \bigg\} \subseteq \mathcal{G}(s) \tag{C.23}$$

743 for $\ell \in \{1, 2\}$. It then holds that

$$D_{\chi^2}\big(\mathbb{P}_{\mathcal{C}_\ell(q)}, \mathbb{P}_0\big) = \mathbb{E}_{\mathbb{P}_0}\bigg[ \bigg( \frac{\mathrm{d}\mathbb{P}_{\mathcal{C}_\ell(q)}}{\mathrm{d}\mathbb{P}_0}(Z) - 1 \bigg)^2 \bigg] = \frac{1}{\mathcal{C}_\ell(q)} \sum_{\mathrm{v}, \mathrm{v}' \in \mathcal{C}_\ell(q)} \mathbb{E}_{\mathbb{P}_0}\bigg[ \frac{\mathrm{d}\mathbb{P}_{\mathrm{v}}}{\mathrm{d}\mathbb{P}_0} \frac{\mathrm{d}\mathbb{P}_{\mathrm{v}'}}{\mathrm{d}\mathbb{P}_0}(Z) \bigg] - 1$$

$$\leq \sup_{\mathrm{v} \in \mathcal{C}_\ell(q)} \frac{1}{|\mathcal{C}_\ell(q)|} \sum_{\mathrm{v}' \in \mathcal{C}_\ell(q)} \mathbb{E}_{\mathbb{P}_0}\bigg[ \frac{\mathrm{d}\mathbb{P}_{\mathrm{v}}}{\mathrm{d}\mathbb{P}_0} \frac{\mathrm{d}\mathbb{P}_{\mathrm{v}'}}{\mathrm{d}\mathbb{P}_0}(Z) \bigg] - 1$$

$$\leq \sup_{\mathrm{v} \in \mathcal{C}_\ell(q)} \frac{1}{|\mathcal{C}_\ell(q)|} \sum_{\mathrm{v}' \in \bar{\mathcal{C}}_\ell(q, \mathrm{v})} \mathbb{E}_{\mathbb{P}_0}\bigg[ \frac{\mathrm{d}\mathbb{P}_{\mathrm{v}}}{\mathrm{d}\mathbb{P}_0} \frac{\mathrm{d}\mathbb{P}_{\mathrm{v}'}}{\mathrm{d}\mathbb{P}_0}(Z) \bigg] - 1, \tag{C.24}$$

744 where the last inequality follows from the definition of $\bar{\mathcal{C}}_\ell(q, \mathrm{v})$ in (C.23). By Lemma B.1, it holds
745 that

$$\mathbb{E}_{\mathbb{P}_0}\bigg[ \frac{\mathrm{d}\mathbb{P}_{\mathrm{v}}}{\mathrm{d}\mathbb{P}_0} \frac{\mathrm{d}\mathbb{P}_{\mathrm{v}'}}{\mathrm{d}\mathbb{P}_0}(Z) \bigg] \leq \cosh\bigg( \frac{2\rho^2 \cdot \langle \mathrm{v}, \mathrm{v}' \rangle}{\sigma^2 + s\rho^2} \bigg) + \alpha^2 \cdot \sinh\bigg( \frac{2\rho^2 \cdot \langle \mathrm{v}, \mathrm{v}' \rangle}{\sigma^2 + s\rho^2} \bigg). \tag{C.25}$$

746 Combining (C.24) and (C.25), we conclude that

$$1 + D_{\chi^2}\big(\mathbb{P}_{\mathcal{C}_\ell(q)}, \mathbb{P}_0\big)$$

$$\leq \sup_{\mathrm{v} \in \mathcal{C}_\ell(q)} \bigg\{ \frac{1}{|\mathcal{C}_\ell(q)|} \sum_{\mathrm{v}' \in \bar{\mathcal{C}}_\ell(q, \mathrm{v})} \cosh\bigg( \frac{2\rho^2 \cdot \langle \mathrm{v}, \mathrm{v}' \rangle}{\sigma^2 + s\rho^2} \bigg) + \alpha^2 \cdot \sinh\bigg( \frac{2\rho^2 \cdot \langle \mathrm{v}, \mathrm{v}' \rangle}{\sigma^2 + s\rho^2} \bigg) \bigg\}. \tag{C.26}$$

747 In what follows, we calculate the sum on the right-hand side of (C.26). To achieve this, we calculate
748 the sum based on the value of $\langle \mathrm{v}, \mathrm{v}' \rangle$. We denote by

$$\mathcal{C}_j(\mathrm{v}) = \big\{ \mathrm{v}' \in \mathcal{G}(s) : \langle \mathrm{v}, \mathrm{v}' \rangle = s - j \big\}.$$

749 Then for any choice of $\ell$, $q$, and $\mathrm{v} \in \mathcal{C}_\ell(q)$, there exists an integer $k_\ell(q, \mathrm{v})$ such that

$$\bar{\mathcal{C}}_\ell(q, \mathrm{v}) = \mathcal{C}_0(\mathrm{v}) \cup \cdots \cup \mathcal{C}_{k_\ell(q, \mathrm{v})-1} \cup \mathcal{C}'_\ell(q, \mathrm{v}),$$

750 where $\mathcal{C}'_\ell(q, \mathrm{v}) = \bar{\mathcal{C}}_\ell(q, \mathrm{v}) \backslash \bigcup_{j=0}^{k_\ell(q, \mathrm{v})-1} \mathcal{C}_j(\mathrm{v})$. Note that we have

$$|\mathcal{C}'_\ell(q, \mathrm{v})| = |\mathcal{C}_\ell(q)| - \sum_{j=0}^{k_\ell(q, \mathrm{v})-1} |\mathcal{C}_j(\mathrm{v})| < |\mathcal{C}_{k_\ell(q, \mathrm{v})}(\mathrm{v})|.$$

751 Hence, the cardinality of $\bar{\mathcal{C}}_\ell(q, \mathrm{v})$ is between $\sum_{j=0}^{k_\ell(q, \mathrm{v})-1} |\mathcal{C}_j(\mathrm{v})|$ and $\sum_{j=0}^{k_\ell(q, \mathrm{v})} |\mathcal{C}_j(\mathrm{v})|$. Following
752 form (C.26), we have

$$1 + D_{\chi^2}\big(\mathbb{P}_{\mathcal{C}_\ell(q)}, \mathbb{P}_0\big) \leq \frac{\sum_{j=0}^{k_\ell(q, \mathrm{v})-1} h_\alpha(j) \cdot |\mathcal{C}_j(\mathrm{v})| + h_\alpha\big(k_\ell(q, \mathrm{v})\big) \cdot |\mathcal{C}'_\ell(q, \mathrm{v})|}{\sum_{j=0}^{k_\ell(q, \mathrm{v})-1} |\mathcal{C}_j(\mathrm{v})| + |\mathcal{C}'_\ell(q, \mathrm{v})|}, \tag{C.27}$$

where we denote by $h_\alpha(j)$ the right-hand side of (C.25) when $v' \in C_j(v)$. In other words, it holds that

$$h_\alpha(j) = \cosh\left(\frac{2\rho^2(s-j)}{\sigma^2 + s\rho^2}\right) + \alpha^2 \cdot \sinh\left(\frac{2\rho^2(s-j)}{\sigma^2 + s\rho^2}\right). \tag{C.28}$$

Note that $h_\alpha(j)$ is monotonically decreasing as $j$ increases. Therefore, it follows from (C.27) that

$$1 + D_{\chi^2}(\mathbb{P}_{C_\ell(q)}, \mathbb{P}_0) \leq \frac{\sum_{j=0}^{k_\ell(q,v)-1} h_\alpha(j) \cdot |C_j(v)|}{\sum_{j=0}^{k_\ell(q,v)-1} |C_j(v)|}. \tag{C.29}$$

Further note that $|C_j(v)| = \binom{s}{s-j}\binom{d-s}{j}$. Therefore, it holds that

$$|C_{j+1}(v)|/|C_j(v)| = (s-j)(d-s-j)/(j+1)^2 \geq d/2s^2,$$

where $j \in \{0, \ldots, s-1\}$, $v \in \mathcal{G}(s)$, and $s = o(d^{1/2-\delta})$. We denote by $\zeta = d/2s^2$, which satisfies $\zeta^{-1} = o(1)$ by the assumption that $s = o(d^{1/2-\delta})$. It then holds that

$$|C_\ell(q)| \leq \sum_{j=0}^{k_\ell(q,v)} |C_j(v)| \leq |C_s(v)| \cdot \sum_{j=0}^{k_\ell(q,v)} \zeta^{j-s}$$

$$\leq \frac{\zeta^{-(s-k_\ell(q,v))} \cdot |\mathcal{G}(s)|}{1 - \zeta^{-1}} \leq 2\zeta^{-(s-k_\ell(q,v))} \cdot |\mathcal{G}(s)|. \tag{C.30}$$

For any integer $k \geq 1$ and two positive sequences $\{w_i\}_{i=0}^\infty$ and $\{u_i\}_{i=0}^\infty$ such that $w_i/w_{i-1} \geq u_i/u_{i-1} > 1$, it holds that

$$\frac{\sum_{j=0}^k w_j \cdot h_\alpha(j)}{\sum_{i=0}^k w_j} \leq \frac{\sum_{j=0}^k u_j \cdot h_\alpha(j)}{\sum_{j=0}^k u_j}. \tag{C.31}$$

Therefore, by setting $w_j = |C_j(v)|$ and $u_j = \zeta^j$, we conclude from (C.29) and (C.31) that

$$1 + D_{\chi^2}(\mathbb{P}_{C_\ell(q)}, \mathbb{P}_0) \leq \frac{\sum_{j=0}^{k_\ell(q,v)-1} \zeta^j \cdot h_\alpha(j)}{\sum_{j=0}^{k_\ell(q,v)-1} \zeta^j} \tag{C.32}$$

$$= \left[\sum_{j=0}^{k_\ell(q,v)-1} \zeta^j \cdot \cosh\left(\frac{2\rho^2(s-j)}{\sigma^2 + s\rho^2}\right) + \alpha^2 \cdot \sinh\left(\frac{2\rho^2(s-j)}{\sigma^2 + s\rho^2}\right)\right] \Big/ \sum_{j=0}^{k_\ell(q,v)-1} \zeta^j$$

$$\leq \sum_{j=0}^{k_\ell(q,v)-1} \zeta^j \cdot \left\{\cosh\left(\frac{4\rho^2(s-j)}{\sigma^2 + s\rho^2}\right) \bigvee \exp\left(\frac{4\alpha^2\rho^2(s-j)}{\sigma^2 + s\rho^2}\right)\right\} \Big/ \sum_{j=0}^{k_\ell(q,v)-1} \zeta^j,$$

where the last inequality follows from Lemma B.1. In what follows, we denote by

$$f(j) = \cosh\left(\frac{4\rho^2(s-j)}{\sigma^2 + s\rho^2}\right), \quad g(j) = \exp\left(\frac{4\alpha^2\rho^2(s-j)}{\sigma^2 + s\rho^2}\right) \tag{C.33}$$

for notational simplicity. Note that

$$f(j-1)/f(j) \geq \cosh\left(\frac{4\rho^2}{\sigma^2 + s\rho^2}\right).$$

Therefore, it holds for $j \in \{0, 1, \ldots, k_\ell(q, v) - 1\}$ that

$$f(j) \leq f(k_\ell(q, v) - 1) \cdot \left\{\cosh\left(\frac{4\rho^2}{\sigma^2 + s\rho^2}\right)\right\}^{k_\ell(q,v)-j-1}. \tag{C.34}$$

Meanwhile, we have

$$g(j) = \exp\left(4\alpha^2\rho^2(s-j)\sigma^2 + s\rho^2\right) = g(k_\ell(q, v) - 1) \cdot \left\{\exp\left(\frac{4\alpha^2\rho^2}{\sigma^2 + s\rho^2}\right)\right\}^{k_\ell(q,v)-j-1}. \tag{C.35}$$

We denote by

$$\Gamma(s, \rho) = \exp\left(\frac{4\alpha^2\rho^2}{\sigma^2 + s\rho^2}\right) \bigvee \cosh\left(\frac{4\rho^2}{\sigma^2 + s\rho^2}\right). \tag{C.36}$$

Combining (C.34) and (C.35), we conclude that

$$f(j) \vee g(j) \leq \left\{f(k_\ell(q, v) - 1) \vee g(k_\ell(q, v) - 1)\right\} \cdot (\Gamma(s, \rho))^{k_\ell(q,v)-j-1}. \tag{C.37}$$

768 Following from (C.32) and (C.37), it holds that

$$1 + D_{\chi^2}(\mathbb{P}_{\mathcal{C}_\ell(q)}, \mathbb{P}_0)$$

$$\leq \left\{ f\big(k_\ell(q, \mathrm{v}) - 1\big) \vee g\big(k_\ell(q, \mathrm{v}) - 1\big) \right\} \cdot \frac{\sum_{j=0}^{k_\ell(q,\mathrm{v})-1} \zeta^j \cdot \big(\Gamma(s, \rho)\big)^{k_\ell(q,\mathrm{v})-j-1}}{\sum_{j=0}^{k_\ell(q,\mathrm{v})-1} \zeta^j}. \qquad (C.38)$$

769 By direct calculation, we obtain

$$\frac{\sum_{j=0}^{k_\ell(q,\mathrm{v})-1} \zeta^j \cdot \big(\Gamma(s, \rho)\big)^{k_\ell(q,\mathrm{v})-j-1}}{\sum_{j=0}^{k_\ell(q,\mathrm{v})-1} \zeta^j} = \frac{\zeta^{k_\ell(q,\mathrm{v})-1} \cdot \sum_{j=0}^{k_\ell(q,\mathrm{v})-1} \big(\Gamma(s, \rho)/\zeta\big)^{k_\ell(q,\mathrm{v})-j-1}}{\zeta^{k_\ell(q,\mathrm{v})-1} \cdot \sum_{j=0}^{k_\ell(q,\mathrm{v})-1} \zeta^{-(k_\ell(q,\mathrm{v})-j-1)}}$$

$$= \frac{1 - \big(\Gamma(s, \rho)/\zeta\big)^{k_\ell(q,\mathrm{v})}}{1 - \zeta^{-k_\ell(q,\mathrm{v})}} \cdot \frac{1 - \zeta^{-1}}{1 - \Gamma(s, \rho)/\zeta}. \qquad (C.39)$$

770 Note that $\Gamma(s, \rho) \geq 1$. Therefore, the following upper bound of the right-hand side of (C.39) holds,

$$\frac{1 - \big(\Gamma(s, \rho)/\zeta\big)^{k_\ell(q,\mathrm{v})}}{1 - \zeta^{-k_\ell(q,\mathrm{v})}} \cdot \frac{1 - \zeta^{-1}}{1 - \Gamma(s, \rho)/\zeta} \leq \frac{1 - \zeta^{-1}}{1 - \Gamma(s, \rho)/\zeta}. \qquad (C.40)$$

771 Combining (C.38), (C.39), and (C.40), we conclude that

$$1 + D_{\chi^2}(\mathbb{P}_{\mathcal{C}_\ell(q)}, \mathbb{P}_0) \leq \left\{ f\big(k_\ell(q, \mathrm{v}) - 1\big) \vee g\big(k_\ell(q, \mathrm{v}) - 1\big) \right\} \cdot \frac{1 - \zeta^{-1}}{1 - \Gamma(s, \rho)/\zeta}, \qquad (C.41)$$

772 where $f(j)$ and $g(j)$ are defined in (C.33). Meanwhile, by Lemma 4.5 of [53], it holds that

$$D_{\chi^2}(\mathbb{P}_{\mathcal{C}_\ell(q)}, \mathbb{P}_0) \geq \log(T/\xi)/n. \qquad (C.42)$$

773 We denote by $\tau^2$ the right-hand side of (C.42). Combining (C.41) and (C.42), we have

$$\tau^2 + 1 \leq \left\{ f\big(k_\ell(q, \mathrm{v}) - 1\big) \vee g\big(k_\ell(q, \mathrm{v}) - 1\big) \right\} \cdot \frac{1 - \zeta^{-1}}{1 - \Gamma(s, \rho)/\zeta}.$$

774 Therefore, one of the following inequalities holds,

$$(1 + \tau^2) \cdot \frac{1 - \Gamma(s, \rho)/\zeta}{1 - \zeta^{-1}} \leq g\big(k_\ell(q, \mathrm{v}) - 1\big) = \exp\left( \frac{4\alpha^2 \rho^2 \cdot \big(s - k_\ell(q, \mathrm{v}) + 1\big)}{\sigma^2 + s\rho^2} \right),$$

$$(1 + \tau^2) \cdot \frac{1 - \Gamma(s, \rho)/\zeta}{1 - \zeta^{-1}} \leq f\big(k_\ell(q, \mathrm{v}) - 1\big) \leq \exp\left( \frac{2\rho^4 \cdot \big(s - k_\ell(q, \mathrm{v}) + 1\big)^2}{(\sigma^2 + s\rho^2)^2} \right), \qquad (C.43)$$

775 where the second inequality holds because of the fact that $\cosh(x) \leq \exp(x^2/2)$. We take the
776 logarithm of (C.43) and obtain that one of the following inequalities holds,

$$\log(1 + \tau^2) + \log\left( \frac{1 - \zeta^{-1}}{1 - \Gamma(s, \rho)/\zeta} \right) \leq \frac{4\alpha^2 \rho^2 \cdot \big(s - k_\ell(q, \mathrm{v}) + 1\big)}{\sigma^2 + s\rho^2},$$

$$\log(1 + \tau^2) + \log\left( \frac{1 - \zeta^{-1}}{1 - \Gamma(s, \rho)/\zeta} \right) \leq \frac{2\rho^4 \cdot \big(s - k_\ell(q, \mathrm{v}) + 1\big)^2}{(\sigma^2 + s\rho^2)^2}. \qquad (C.44)$$

777 Following from the definition of $\Gamma(s, \rho)$ in (C.36), we have $\Gamma(s, \rho)/\zeta = o(1)$. By Taylor's expansion,
778 it holds that

$$\log\left( \frac{1 - \zeta^{-1}}{1 - \Gamma(s, \rho)/\zeta} \right) = \log\left( 1 - \zeta^{-1} \cdot \frac{1 - \Gamma(s, \rho)}{1 - \Gamma(s, \rho)/\zeta} \right) = O(\zeta^{-1}\rho^4 \vee \zeta^{-1}\alpha^2\rho^2). \qquad (C.45)$$

779 For $\gamma_n = s\rho^2/\delta^2 = o(\sqrt{s^2/n} \wedge 1/\alpha^2 \cdot s/n)$, where $\sigma^2$ is a constant, it holds that $\alpha^2\rho^2 \vee \rho^4 = o(1/n)$.
780 Hence, the right-hand side of (C.45) is negligible compared with $\log(1 + \tau^2)$. Then following form
781 (C.44), it holds that

$$s - k_\ell(q, \mathrm{v}) + 1 \geq \sqrt{\frac{(\sigma^2 + s\rho^2)^2 \cdot \log(1 + \tau^2)}{2\rho^4}} \bigwedge \sqrt{\frac{(\sigma^2 + s\rho^2) \cdot \log(1 + \tau^2)}{4\alpha^2\rho^2}}. \qquad (C.46)$$

Note that $\log(1 + \tau^2) \geq \tau^2/2 = \log(T/\xi)/(2n)$ for $\tau < 1$. Therefore, by combining (C.30) and (C.46), we conclude that

$$T \cdot \frac{\sup_{q \in \mathcal{Q}} |\mathcal{C}(q)|}{|\mathcal{G}(s)|} \tag{C.47}$$

$$\leq 4T \cdot \exp\left(-\log \zeta \cdot \left\{\sqrt{\frac{(\sigma^2 + s\rho^2)^2 \cdot \log(T/\xi)}{4n\rho^4}} - 1 \bigvee \sqrt{\frac{(\sigma^2 + s\rho^2) \cdot \log(T/\xi)}{8n\alpha^2\rho^2}} - 1\right\}\right).$$

Note that $\rho^4 \cdot n \vee \alpha^2\rho^2 \cdot n = o(1)$ for $s\rho^2/\sigma^2 = o(\sqrt{s^2/n} \wedge 1/\alpha^2 \cdot s/n)$. We choose an absolute constant $C > 0$ satisfying $\delta(C - 1) > \mu$, where $\mu$ and $\delta$ are absolute constants such that $T = O(d^\mu)$ and $s = o(d^{1/2-\delta})$. Then it holds for a sufficiently large $n$ that

$$\sqrt{\frac{(\sigma^2 + s\rho^2)^2 \cdot \log(T/\xi)}{4n\rho^4}} \bigvee \sqrt{\frac{(\sigma^2 + s\rho^2) \cdot \log(T/\xi)}{8n\alpha^2\rho^2}}$$

$$\geq \sqrt{\frac{(\sigma^2 + s\rho^2)^2 \cdot \log(1/\xi)}{4n\rho^4}} \bigvee \sqrt{\frac{(\sigma^2 + s\rho^2) \cdot \log(1/\xi)}{8n\alpha^2\rho^2}} \geq C. \tag{C.48}$$

Note that $\zeta = d/(2s^2) = \Omega(d^\delta)$ for $s = o(d^{1/2-\delta})$, where $\delta > 0$ is an absolute constant. Finally, combining (C.47) and (C.48), we obtain that for $T = O(d^\mu)$,

$$T \cdot \sup_{q \in \mathcal{Q}} |\mathcal{C}(q)|/|\mathcal{G}(s)| \leq \mathcal{O}(d^\mu \cdot \zeta^{-(C-1)}) = \mathcal{O}(d^{\mu-\delta(C-1)}) = o(1), \tag{C.49}$$

which concludes the proof of Lemma B.4. □

## C.4  Proof of Lemma B.5

*Proof.* In the following proof, we denote by $C$ and $C'$ absolute constants, the value of which may vary from lines to lines. We define the following unbounded query functions,

$$\widetilde{q}_{1,\mathrm{v}}(Y, X) = \psi(Y) \cdot \left[s^{-1}(\mathrm{v}^\top X)^2 - 1\right] \cdot \mathbb{1}\{|\psi(Y)| \leq (R \cdot \log n)^{1/\nu}\}, \quad \mathrm{v} \in \bar{\mathcal{G}}(s),$$

$$\widetilde{q}_{2,\mathrm{v}}(Y, X) = Y \cdot (s^{-1/2}\mathrm{v}^\top X) \cdot \mathbb{1}\{|Y| \leq (R \cdot \log n)^{1/\nu}\}, \quad \mathrm{v} \in \bar{\mathcal{G}}(s). \tag{C.50}$$

In the sequel, we first upper bound the difference between the query functions in (A.5) and the query functions in (C.50). We then characterize the two expectations $\mathbb{E}_{\mathbb{P}_\mathrm{v}}[q_{i,\mathrm{v}}(Y, X)]$ and $\mathbb{E}_{\mathbb{P}_0}[q_{i,\mathrm{v}}(Y, X)]$ using the corresponding expectations of $\widetilde{q}_{i,\mathrm{v}}(Y, X)$. Following from (A.5) and (C.50), it holds that

$$\widetilde{q}_{1,\mathrm{v}} - q_{1,\mathrm{v}} = \psi(Y) \cdot \left[s^{-1}(\mathrm{v}^\top X)^2 - 1\right] \cdot \mathbb{1}\{|\psi(Y)| \leq (R \cdot \log n)^{1/\nu}\} \cdot \mathbb{1}\{|\mathrm{v}^\top X| > R \cdot \sqrt{s \log n}\},$$

$$\widetilde{q}_{2,\mathrm{v}} - q_{2,\mathrm{v}} = Y \cdot (s^{-1/2}\mathrm{v}^\top X) \cdot \mathbb{1}\{|Y| \leq (R \cdot \log n)^{1/\nu}\} \cdot \mathbb{1}\{|\mathrm{v}^\top X| > R \cdot \sqrt{s \log n}\}. \tag{C.51}$$

Then following from the Cauchy-Schwartz inequality, it holds for $q_{1,\mathrm{v}}$ and $\widetilde{q}_{1,\mathrm{v}}$ that

$$\left|\mathbb{E}_{\mathbb{P}_0}\left[q_{1,\mathrm{v}}(Y, X) - \widetilde{q}_{1,\mathrm{v}}(Y, X)\right]\right|^2$$

$$\leq \left|\mathbb{E}_{\mathbb{P}_0}\left[\psi(Y) \cdot \left(s^{-1}(\mathrm{v}^\top X)^2 - 1\right)\right]\right|^2 \cdot \mathbb{P}_0\left(|\mathrm{v}^\top X| > R \cdot \sqrt{s \log n}\right). \tag{C.52}$$

Note that under $H_0$, $X \sim N(0, I_d)$ is the standard Gaussian distribution, which is independent of $Y$. Therefore, it holds that $\mathbb{E}_{\mathbb{P}_0}[(s^{-1}(X^\top \mathrm{v})^2 - 1)^2] = 2$. Then following from the Cauchy-Schwartz inequality, we obtain that

$$\left|\mathbb{E}_{\mathbb{P}_0}\left[\psi(Y) \cdot \left(s^{-1}(\mathrm{v}^\top X)^2 - 1\right)\right]\right|^2 \cdot \mathbb{P}_0\left(|\mathrm{v}^\top X| > R \cdot \sqrt{s \log n}\right)$$

$$\leq \mathbb{E}_{\mathbb{P}_0}\left[\psi^2(Y)\right] \cdot \mathbb{E}_{\mathbb{P}_0}\left[\left(s^{-1}(X^\top \mathrm{v})^2 - 1\right)^2\right] \cdot \mathbb{P}_0\left(|\mathrm{v}^\top X| > R \cdot \sqrt{s \log n}\right)$$

$$= C \cdot \mathbb{P}_0\left(|\mathrm{v}^\top X| > R \cdot \sqrt{s \log n}\right) \tag{C.53}$$

for a positive absolute constant $C$. Note that $X^\top \mathrm{v}/\sqrt{s} \sim N(0, 1)$ under the null hypothesis. Following from the tail bound of standard Gaussian distribution, it holds for any $t \geq 1$ that

$$\mathbb{P}_0\left(|X^\top \mathrm{v}/\sqrt{s}| \geq t\right) \leq 2\exp(-t^2/2). \tag{C.54}$$

Combining (C.52), (C.53), and (C.54), we obtain that

$$\left|\mathbb{E}_{\mathbb{P}_0}\left[q_{1,\mathrm{v}}(Y, X) - \widetilde{q}_{1,\mathrm{v}}(Y, X)\right]\right|^2 \leq C \cdot \mathbb{P}(|\mathrm{v}^\top X| > R \cdot s\sqrt{\log n})$$

$$\leq C \cdot \exp(-R^2 \cdot \log n/2). \tag{C.55}$$

In the following, we upper bound the distance between $q_{1,\mathrm{v}}(Y,X)$ and $\widetilde{q}_{1,\mathrm{v}}(Y,X)$ under $\mathbb{P}_{\mathrm{v}}$. Following from the Cauchy-Schwartz inequality, it holds that

$$
\begin{aligned}
\left|\mathbb{E}_{\mathbb{P}_{\mathrm{v}^*}}\big[q_{1,\mathrm{v}}(Y,X) - \widetilde{q}_{1,\mathrm{v}}(Y,X)\big]\right|^2 & \\
\leq \mathbb{E}_{\mathbb{P}_{\mathrm{v}^*}}\Big[\psi^2(Y) \cdot \big(s^{-1}(\mathrm{v}^\top X)^2 - 1\big)^2\Big] & \cdot \mathbb{P}_{\mathrm{v}^*}\big(|\mathrm{v}^\top X| > R \cdot \sqrt{s \log n}\big) \\
\leq \sqrt{\mathbb{E}_{\mathbb{P}_{\mathrm{v}^*}}\big[\psi^4(Y)\big] \cdot \mathbb{E}_{\mathbb{P}_{\mathrm{v}^*}}\Big[\big(s^{-1}(\mathrm{v}^\top X)^2 - 1\big)^4\Big]} & \cdot \mathbb{P}_{\mathrm{v}^*}\big(|\mathrm{v}^\top X| > R \cdot \sqrt{s \log n}\big). \quad \text{(C.56)}
\end{aligned}
$$

Note that under Assumption A.1, $\mathbb{E}_{\mathbb{P}_{\mathrm{v}^*}}[\psi^4(Y)]$ is upper bounded. Meanwhile, we have that $X^\top \mathrm{v}/\sqrt{s} \sim N(0,1)$. Therefore, it holds for an absolute constant $C$ that

$$
\left|\mathbb{E}_{\mathbb{P}_{\mathrm{v}^*}}\big[q_{1,\mathrm{v}}(Y,X) - \widetilde{q}_{1,\mathrm{v}}(Y,X)\big]\right|^2 \leq C \cdot \exp(-R^2 \cdot \log n/2). \quad \text{(C.57)}
$$

Similar arguments apply to $q_{2,\mathrm{v}}(Y,X)$ and $\widetilde{q}_{2,\mathrm{v}}(Y,X)$. Under the null hypothesis, it holds for an absolute constant $C'$ that

$$
\begin{aligned}
\left|\mathbb{E}_{\mathbb{P}_0}\big[q_{2,\mathrm{v}}(Y,X) - \widetilde{q}_{2,\mathrm{v}}(Y,X)\big]\right|^2 &\leq \mathbb{E}_{\mathbb{P}_0}[Y^2] \cdot \mathbb{E}_{\mathbb{P}_0}\big[s^{-1}(X^\top \mathrm{v})^2\big] \cdot \mathbb{P}\big(|\mathrm{v}^\top X| > R \cdot \sqrt{s \log n}\big) \\
&\leq C' \cdot \exp(-R^2 \cdot \log n/2), \quad \text{(C.58)}
\end{aligned}
$$

which also holds under the alternative hypothesis with distribution $\mathbb{P}_{\mathrm{v}^*}$. Therefore, following from (C.55), (C.57), and (C.58), it holds for a sufficiently large constant $R$ that

$$
\begin{aligned}
\left|\mathbb{E}_{\mathbb{P}_{\mathrm{v}^*}}\big[q_{1,\mathrm{v}}(Y,X) - \widetilde{q}_{1,\mathrm{v}}(Y,X)\big]\right| \vee \left|\mathbb{E}_{\mathbb{P}_0}\big[q_{1,\mathrm{v}}(Y,X) - \widetilde{q}_{1,\mathrm{v}}(Y,X)\big]\right| &\leq 1/n, \\
\left|\mathbb{E}_{\mathbb{P}_{\mathrm{v}^*}}\big[q_{2,\mathrm{v}}(Y,X) - \widetilde{q}_{2,\mathrm{v}}(Y,X)\big]\right| \vee \left|\mathbb{E}_{\mathbb{P}_0}\big[q_{2,\mathrm{v}}(Y,X) - \widetilde{q}_{2,\mathrm{v}}(Y,X)\big]\right| &\leq 1/n, \quad \text{(C.59)}
\end{aligned}
$$

which holds for any $\mathrm{v} \in \bar{\mathcal{G}}(s)$. In what follows, we characterize the expectations of $\widetilde{q}_{i,\mathrm{v}}(Y,X)$ under the null and alternative hypotheses for $i \in \{1,2\}$. We then obtain the desired bounds of $q_{i,\mathrm{v}}(Y,X)$ based on $\widetilde{q}_{i,\mathrm{v}}(Y,X)$. Note that under the null hypothesis, $Y$ is independent of $X$. Then, following from (C.50) and the fact that $X \sim N(0, I_d)$, it holds that

$$
\mathbb{E}_{\mathbb{P}_0}\big[\widetilde{q}_{1,\mathrm{v}}(Y,X)\big] = \mathbb{E}_{\mathbb{P}_0}\big[\widetilde{q}_{2,\mathrm{v}}(Y,X)\big] = 0. \quad \text{(C.60)}
$$

Following from (A.3), we have

$$
\begin{aligned}
s\rho^2 - \mathbb{E}_{\mathbb{P}_{\mathrm{v}^*}}\big[\widetilde{q}_{1,\mathrm{v}^*}(Y,X)\big] &\leq \mathbb{E}_{\mathbb{P}_{\mathrm{v}^*}}\Big[\psi(Y) \cdot \big(s^{-1}(\mathrm{v}^{*\top} X)^2 - 1\big) - \widetilde{q}_{1,\mathrm{v}}(Y,X)\Big] \quad \text{(C.61)} \\
&= \mathbb{E}_{\mathbb{P}_{\mathrm{v}^*}}\Big[\psi(Y) \cdot \big(s^{-1}(\mathrm{v}^{*\top} X)^2 - 1\big) \cdot \mathbb{1}\big\{|\psi(Y)| > (R \cdot \log n)^{1/\nu}\big\}\Big] \\
&\leq \sqrt{\mathbb{E}_{\mathbb{P}_{\mathrm{v}^*}}\Big[\psi^2(Y) \cdot \big(s^{-1}(\mathrm{v}^{*\top} X)^2 - 1\big)^2\Big]} \cdot \sqrt{\mathbb{P}_{\mathrm{v}^*}\big(|\psi(Y)| > (R \cdot \log n)^{1/\nu}\big)},
\end{aligned}
$$

where the last inequality follows from the Cauchy-Schwartz inequality. It then follows from Assumption A.1 that

$$
\mathbb{P}_{\mathrm{v}^*}\big(|\psi(Y)| > (R \cdot \log n)^{1/\nu}\big) \leq C \cdot \exp(-R \cdot \log n). \quad \text{(C.62)}
$$

Meanwhile, following from the Cauchy-Schwartz inequality, it holds that

$$
\mathbb{E}_{\mathbb{P}_{\mathrm{v}^*}}\Big[\psi^2(Y) \cdot \big(s^{-1}(\mathrm{v}^{*\top} X)^2 - 1\big)^2\Big] \leq \sqrt{\mathbb{E}_{\mathbb{P}_{\mathrm{v}^*}}\big[\psi^4(Y)\big] \cdot \mathbb{E}_{\mathbb{P}_{\mathrm{v}^*}}\Big[\big(s^{-1}(\mathrm{v}^{*\top} X)^2 - 1\big)^4\Big]}, \quad \text{(C.63)}
$$

which is upper bounded by an absolute constant. Combining (C.61), (C.62), and (C.63), if it holds that $s\rho^2/\sigma^2 = \Omega(\sqrt{s \log d/n})$, then for sufficiently large $n$ and constant $R$, we obtain that $1/n \leq s\rho^2/4$ and

$$
s\rho^2 - \mathbb{E}_{\mathbb{P}_{\mathrm{v}^*}}\big[\widetilde{q}_{1,\mathrm{v}}(Y,X)\big] \leq s\rho^2/4. \quad \text{(C.64)}
$$

In other words, it holds that $\mathbb{E}_{\mathbb{P}_{\mathrm{v}^*}}\big[\widetilde{q}_{1,\mathrm{v}}(Y,X)\big] \geq 3s\rho^2/4$. Similar arguments hold for the query function $\widetilde{q}_{2,\mathrm{v}}(Y,X)$. If it holds that $s\rho^2/\sigma^2 = \Omega(1/\alpha^2 \cdot s \log d/n)$, then for sufficiently large $n$ and constant $R$, we obtain that $1/n \leq \sqrt{\alpha^2 s\rho^2}/4$ and

$$
\mathbb{E}_{\mathbb{P}_{\mathrm{v}^*}}\big[\widetilde{q}_{2,\mathrm{v}}(Y,X)\big] \geq 3\sqrt{\alpha^2 s\rho^2}/4. \quad \text{(C.65)}
$$

Combining (C.59), (C.60), (C.64), and (C.65), it holds for sufficiently large $n$ and constant $R$ that

$$
\mathbb{E}_{\mathbb{P}_0}\big[q_{1,\mathrm{v}}(Y,X)\big] \leq 1/n, \quad \mathbb{E}_{\mathbb{P}_0}\big[q_{2,\mathrm{v}}(Y,X)\big] \leq 1/n.
$$

Furthermore, it holds for sufficiently large $n$ and constant $R$ that

$$
\begin{aligned}
\mathbb{E}_{\mathbb{P}_{\mathrm{v}^*}}\big[q_{1,\mathrm{v}^*}(Y,X)\big] &\geq s\rho^2/2, \ \ \text{if } s\rho^2/\sigma^2 = \Omega(\sqrt{s \log d/n}), \\
\mathbb{E}_{\mathbb{P}_{\mathrm{v}^*}}\big[q_{2,\mathrm{v}^*}(Y,X)\big] &\geq \sqrt{\alpha^2 s\rho^2}/2, \ \ \text{if } s\rho^2/\sigma^2 = \Omega(1/\alpha^2 \cdot s \log d/n),
\end{aligned}
$$

which concludes the proof of Lemma B.5. □

## C.5 Proof of Lemma B.6

*Proof.* In the following proof, we denote by $C$ and $C'$ absolute constants, the value of which may vary from lines to lines. We define the following unbounded query functions,

$$\widetilde{q}_{1,j}(Y, X) = \psi(Y) \cdot (X_j^2 - 1) \cdot \mathbb{1}\{|\psi(Y)| \leq (R \cdot \log n)^{1/\nu}\}, \quad j \in [d],$$

$$\widetilde{q}_{2,j}(Y, X) = Y X_j \cdot \mathbb{1}\{(|Y| \leq (R \cdot \log n)^{1/\nu}\}, \quad j \in [d]. \tag{C.66}$$

The proof is similar to the proof of Lemma B.5 in §C.4. Following from (C.66) and (A.11), it holds that

$$\left|\mathbb{E}_{\mathbb{P}_0}\left[\widetilde{q}_{1,j}(Y, X) - q_{1,j}(Y, X)\right]\right|^2 \leq \mathbb{E}_{\mathbb{P}_0}\left[\psi^2(Y) \cdot (X_j^2 - 1)^2\right] \cdot \mathbb{P}_0\left(|X_j| \geq R \cdot \sqrt{\log n}\right), \tag{C.67}$$

where the inequality follows from the Cauchy-Schwartz inequality. Under the null hypothesis, $Y$ is independent of $X$. Meanwhile, it holds that $X \sim N(0, I_d)$. Thus, we have $X_j \sim N(0, 1)$. Following from the Gaussian tail bound in (C.54), we have

$$\left|\mathbb{E}_{\mathbb{P}_0}\left[\widetilde{q}_{1,j}(Y, X) - q_{1,j}(Y, X)\right]\right|^2 \leq C \cdot \exp(-R^2 \cdot \log n / 2). \tag{C.68}$$

Therefore, for a sufficiently large constant $R$, the right-hand side of (C.68) is upper bounded by $1/n^2$. Under the alternative hypothesis, it follows from the Cauchy-Schwartz inequality that

$$\left|\mathbb{E}_{\mathbb{P}_{v^*}}\left[\widetilde{q}_{1,j}(Y, X) - q_{1,j}(Y, X)\right]\right|^2 \leq \mathbb{E}_{\mathbb{P}_{v^*}}\left[\psi^2(Y) \cdot (X_j^2 - 1)^2\right] \cdot \mathbb{P}_{v^*}\left(|X_j| \geq R \cdot \sqrt{\log n}\right) \tag{C.69}$$

$$\leq \sqrt{\mathbb{E}_{\mathbb{P}_{v^*}}\left[\psi^4(Y)\right] \cdot \mathbb{E}_{\mathbb{P}_{v^*}}\left[(X_j^2 - 1)^4\right]} \cdot \mathbb{P}_{v^*}\left(|X_j| \geq R \cdot \sqrt{\log n}\right).$$

Following from Assumption A.1, it holds that $\mathbb{E}_{\mathbb{P}_v}[\psi^4(Y)]$ is upper bounded under the alternative hypothesis. Meanwhile, it holds that $X_j \sim N(0, 1)$ under the alternative hypothesis. Therefore, for a sufficiently large constant $R$, the right-hand side of (C.69) is upper bounded by $1/n^2$.

For $q_{2,j}(X, Y)$, we follow similar arguments. By the Cauchy-Schwartz inequality, it holds under the null hypothesis that

$$\left|\mathbb{E}_{\mathbb{P}_0}\left[\widetilde{q}_{2,j}(Y, X) - q_{2,j}(Y, X)\right]\right|^2 \leq \mathbb{E}_{\mathbb{P}_0}[Y^2 X_j^2] \cdot \mathbb{P}_0\left(|X_j| \geq R \cdot \sqrt{\log n}\right). \tag{C.70}$$

Note that $Y$ is independent of $X$ and $X_j \sim N(0, 1)$ under the null hypothesis. Thus, following from the Gaussian tail bound, it holds for a sufficiently large constant $R$ that

$$\left|\mathbb{E}_{\mathbb{P}_0}\left[\widetilde{q}_{2,j}(Y, X) - q_{2,j}(Y, X)\right]\right|^2 \leq 1/n^2. \tag{C.71}$$

Meanwhile, it holds under the alternative hypothesis that

$$\left|\mathbb{E}_{\mathbb{P}_{v^*}}\left[\widetilde{q}_{1,j}(Y, X) - q_{1,j}(Y, X)\right]\right|^2 \leq \mathbb{E}_{\mathbb{P}_{v^*}}[Y^2 X_j^2] \cdot \mathbb{P}_{v^*}\left(|X_j| \geq R \cdot \sqrt{\log n}\right)$$

$$\leq \sqrt{\mathbb{E}_{\mathbb{P}_{v^*}}[Y^2] \cdot \mathbb{E}_{\mathbb{P}_{v^*}}[X_j^4]} \cdot \mathbb{P}_{v^*}\left(|X_j| \geq R \cdot \sqrt{\log n}\right), \tag{C.72}$$

where the above inequalities follow from the Cauchy-Schwartz inequality. Also, by Assumption A.1, it holds that $\mathbb{E}_{\mathbb{P}_{v^*}}[Y^4]$ is upper bounded under the alternative hypothesis. Therefore, the right-hand side of (C.72) is upper bounded by $1/n^2$ with a sufficiently large constant $R$. In conclusion, it holds for a sufficiently large constant $R$ that

$$\left|\mathbb{E}_{\mathbb{P}_0}\left[q_{1,j}(Y, X) - \widetilde{q}_{1,j}(Y, X)\right]\right| \vee \left|\mathbb{E}_{\mathbb{P}_v}\left[q_{1,j}(Y, X) - \widetilde{q}_{1,j}(Y, X)\right]\right| \leq 1/n,$$

$$\left|\mathbb{E}_{\mathbb{P}_0}\left[q_{2,j}(Y, X) - \widetilde{q}_{2,j}(Y, X)\right]\right| \vee \left|\mathbb{E}_{\mathbb{P}_v}\left[q_{2,j}(Y, X) - \widetilde{q}_{2,j}(Y, X)\right]\right| \leq 1/n. \tag{C.73}$$

It remains to characterize the expectations of $\widetilde{q}_{1,j}(Y, X)$ and $\widetilde{q}_{2,j}(Y, X)$ under the null and alternative hypotheses. Note that under the null hypothesis, it holds that $Y$ is independent of $X$ and $X_j \sim N(0, 1)$. Therefore, we have $\mathbb{E}_{\mathbb{P}_0}[X_j^2 - 1] = 0$ and $\mathbb{E}_{\mathbb{P}_0}[X_j] = 0$, which imply

$$\mathbb{E}_{\mathbb{P}_0}\left[\widetilde{q}_{1,j}(Y, X)\right] = \mathbb{E}_{\mathbb{P}_0}\left[\psi(Y) \cdot (X_j^2 - 1) \cdot \mathbb{1}\{|\psi(Y)| \leq (R \cdot \log n)^{1/\nu}\}\right] = 0,$$

$$\mathbb{E}_{\mathbb{P}_0}\left[\widetilde{q}_{2,j}(Y, X)\right] = \mathbb{E}_{\mathbb{P}_0}\left[Y X_j \cdot \mathbb{1}\{|Y| \leq (R \cdot \log n)^{1/\nu}\}\right] = 0. \tag{C.74}$$

Under the alternative hypothesis, it follows from (A.3) and (A.4) that

$$\mathbb{E}_{\mathbb{P}_{v^*}}\left[\psi(Y) \cdot (X_j^2 - 1)\right] \geq \rho^2 \mathrm{v}_j^{*2}, \quad \mathbb{E}_{\mathbb{P}_v}[Y X_j] = \alpha \rho \mathrm{v}_j^*, \tag{C.75}$$

where $v_j^* \in \{-1, 0, 1\}$ is the $j$-th entry of $v^* \in \bar{\mathcal{G}}(s)$. For the query function $q_{1,j}(Y, X)$, it holds that

$$\rho^2 v_j^{*2} - \mathbb{E}_{\mathbb{P}_{v^*}}\big[\widetilde{q}_{1,j}(Y, X)\big] \leq \mathbb{E}_{\mathbb{P}_{v^*}}\Big[Y^2(X_j^2 - 1) \cdot \mathbb{1}\big\{|Y| > (R \cdot \log n)^{1/\nu}\big\}\Big]$$

$$\leq \sqrt{\mathbb{E}_{\mathbb{P}_{v^*}}\big[Y^4(X_j^2 - 1)^2\big]} \cdot \sqrt{\mathbb{P}_{v^*}\big(|Y| > (R \cdot \log n)^{1/\nu}\big)}$$

$$\leq C \cdot \exp(-R \cdot \log n), \tag{C.76}$$

where $C$ is a positive absolute constant and the last inequality follows from Assumption A.1. We fix an index $k$ such that $v_k^* \neq 0$. Therefore, if $s\rho^2/\sigma^2 = \Omega(\sqrt{s \log d/n})$, it holds for a sufficiently large constant $R$ that

$$\rho^2 - \mathbb{E}_{\mathbb{P}_v}\big[\widetilde{q}_{1,k}(Y, X)\big] \leq \rho^2/4. \tag{C.77}$$

In other words, it holds that $\sup_{j \in [d]} \mathbb{E}_{\mathbb{P}_v}[\widetilde{q}_{1,j}(Y, X)] \geq 3\rho^2/4$. Similarly, we have

$$\rho v_j^* - \mathbb{E}_{\mathbb{P}_{v^*}}\big[\widetilde{q}_{1,j}(Y, X)\big] = \mathbb{E}_{\mathbb{P}_{v^*}}\Big[YX_j \cdot \mathbb{1}\big\{|Y| > (R \cdot \log n)^{1/\nu}\big\}\Big]. \tag{C.78}$$

Meanwhile, if $s\rho^2/\sigma^2 = \Omega(1/\alpha^2 \cdot s \log d/n)$, it holds for a sufficiently large constant $R$ that

$$\Big|\mathbb{E}_{\mathbb{P}_{v^*}}\Big[YX_j \cdot \mathbb{1}\big\{|Y| > (R \cdot \log n)^{1/\nu}\big\}\Big]\Big|$$

$$\leq \sqrt{\mathbb{E}_{\mathbb{P}_{v^*}}[Y^2 X_j^2]} \cdot \sqrt{\mathbb{P}_{v^*}\big(|Y| > (R \cdot \log n)^{1/\nu}\big)} \leq \alpha\rho/4. \tag{C.79}$$

Recall that $v_j^* \in \{-1, 0, 1\}$ is the $j$-th entry of $v^* \in \bar{\mathcal{G}}(s)$. Following from (C.78) and (C.79), we obtain that

$$\sup_{j \in [d]} \big|\mathbb{E}_{\mathbb{P}_{v^*}}\big[\widetilde{q}_{1,j}(Y, X)\big]\big| \geq 3\alpha\rho/4. \tag{C.80}$$

Combining (C.73), (C.74), (C.77), and (C.80), we conclude that for sufficiently large $n$ and constant $R$, it holds that

$$\sup_{j \in [d]} \mathbb{E}_{\mathbb{P}_0}\big[q_{1,j}(Y, X)\big] \leq 1/n, \quad \sup_{j \in [d]} \mathbb{E}_{\mathbb{P}_0}\big[q_{1,j}(Y, X)\big] \leq 1/n. \tag{C.81}$$

Moreover, for sufficiently large $n$ and constant $R$, it holds that

$$\sup_{j \in [d]} \mathbb{E}_{\mathbb{P}_v^*}\big[q_{1,j}(Y, X)\big] \geq \rho^2/2 \text{ if } s\rho^2/\sigma^2 = \Omega(\sqrt{s \log d/n}),$$

$$\sup_{j \in [d]} \mathbb{E}_{\mathbb{P}_v^*}\big[q_{2,j}(Y, X)\big] \geq \alpha\rho/2 \text{ if } s\rho^2/\sigma^2 = \Omega(1/\alpha^2 \cdot s \log d/n), \tag{C.82}$$

which concludes the proof of Lemma B.6. $\qquad\square$

## C.6 Proof of Lemma C.1

*Proof.* In what follows, we show that for $\gamma_n = s\rho^2/\sigma^2 = o(1/\alpha^2 \cdot s \log d/n)$, we have

$$T = \sum_{k=1}^{s} \binom{s}{k}\Big(\frac{s}{d}\Big)^k \cdot \exp\Big(\frac{4nk\alpha^2\rho^2}{\sigma^2 + s\rho^2}\Big) = o(1).$$

Note that if $\gamma_n = s\rho^2/\sigma^2 = o(1/\alpha^2 \cdot s \log d/n)$, it holds that $\rho^2/(\sigma^2 + s\rho^2) = o(1/\alpha^2 \cdot \log d/n)$, where $\sigma^2$ is a constant. Therefore, we have

$$\Big(\frac{s}{d}\Big)^k \cdot \exp\Big(\frac{4nk\alpha^2\rho^2}{\sigma^2 + s\rho^2}\Big) \leq \Big(\frac{s}{d}\Big)^k \cdot \exp(C \cdot k \log d) = (s \cdot d^{C-1})^k, \tag{C.83}$$

which holds for an arbitrary positive absolute constant $C$ and a sufficiently large $n$, respectively. Meanwhile, note that $s = o(d^{1/2-\delta})$ for an absolute constant $\delta > 0$ and $\binom{s}{k} \leq (es/k)^k$. By (C.83), it holds that

$$\binom{s}{k}\Big(\frac{s}{d}\Big)^k \leq (s^2 e/k \cdot d^{C-1})^k \leq (e/k \cdot d^{C-2\delta})^k. \tag{C.84}$$

Since $C$ is arbitrary, we fix $C \leq \delta$. Following from (C.84), we obtain that

$$T = \sum_{k=1}^{s} \binom{s}{k}\Big(\frac{s}{d}\Big)^k \cdot \exp\Big(\frac{4nk\alpha^2\rho^2}{\sigma^2 + s\rho^2}\Big) \leq \sum_{k=1}^{s}(e/k \cdot d^{C-2\delta})^k = o(1),$$

which concludes the proof of Lemma C.1. $\qquad\square$

## C.7 Proof of Lemma C.2

*Proof.* In the following proof, we denote by $C$, $C'$, and $C''$ absolute constants, the value of which may vary from lines to lines. We first show that for $\gamma_n = s\rho^2/\sigma^2 = o(\sqrt{s\log d/n})$, it holds that

$$T_1 = \sum_{k=1}^{s} \mathbb{E}_U\left[\left(\frac{2s^2eQ}{kd}\right)^k\right] = o(1),$$

where $Q = 4\rho^2 U/(\sigma^2 + s\rho^2)$. Recall that $U$ is the sum of $n$ independent Rademacher random variables with Orlicz $\psi_2$-norm equal to one. Therefore, it holds that $\|U\|_{\psi_2} \leq C\sqrt{n}$ for an absolute constant $C$. It then follows from the definition of Orlicz $\psi_2$-norm [51] that

$$\mathbb{E}_U\left[|Q|^k\right] \leq \left(\frac{\sqrt{k} \cdot 4\rho^2 \cdot \|U\|_{\psi_2}}{\sigma^2 + s\rho^2}\right)^k \leq \left(\frac{C\rho^2\sqrt{nk}}{\sigma^2 + s\rho^2}\right)^k. \tag{C.85}$$

Following from (C.85), it holds that

$$T_1 \leq \sum_{k=1}^{s} \mathbb{E}_U\left[\frac{2s^2e|Q|}{kd}\right]^k \leq \sum_{k=1}^{s}\left(Ce \cdot \frac{s^2\rho^2\sqrt{n}}{\sigma^2 d\sqrt{k}}\right)^k. \tag{C.86}$$

For $s\rho^2/\sigma^2 = o(\sqrt{s\log d/n})$ and $s = o(d^{1/2-\delta})$, it holds that

$$s\sqrt{n}/d \cdot s\rho^2/\sigma^2 = o(s/d \cdot \sqrt{s\log d}) = o(1). \tag{C.87}$$

Combining (C.86) and (C.87), we obtain that $T_1 = o(1)$. It remains to show that

$$T_2 = \sum_{k=1}^{s}\left(\frac{s^2e}{kd}\right)^k \cdot \mathbb{E}_U\left[\exp(k|Q|) \cdot \mathbb{1}\{|Q| \geq 1\}\right] = o(1).$$

By integration by parts, we have

$$\mathbb{E}\left[\exp(k|Q|) \cdot \mathbb{1}\{|Q| \geq 1\}\right] = \exp(k) \cdot \mathbb{P}(|Q| \geq 1) + \int_1^{\infty} k \cdot \exp(tk) \cdot \bar{F}_{|Q|}(t)\mathrm{d}t. \tag{C.88}$$

Note that $Q = 4\rho^2 U/(\sigma^2 + s\rho^2)$ is symmetric and sub-Gaussian with Orlicz $\psi_2$-norm upper bounded by $\|Q\|_{\psi_2} \leq C\rho^2\sqrt{n}/(\sigma^2 + s\rho^2)$ for an absolute constant $C$. Thus, it holds that

$$\mathbb{P}(Q \geq t) \leq C_1 \cdot \exp\left(-\frac{C_2 \cdot t^2(\sigma^2 + s\rho^2)^2}{\rho^4 n}\right), \tag{C.89}$$

where $C_1$ and $C_2$ are positive absolute constants. Then for the right-hand side of (C.88), it holds that

$$\int_1^{\infty} k \cdot \exp(tk) \cdot \bar{F}_{|Q|}(t)\mathrm{d}t$$

$$\leq C_1 k \cdot \exp\left(\frac{k^2\rho^4 n}{4C_2(\sigma^2 + s\rho^2)^2}\right) \cdot \int_1^{\infty} \exp\left(-\frac{C_2(\sigma^2 + s\rho^2)^2}{\rho^4 n} \cdot \left(t - \frac{k\rho^4 n}{2C_2(\sigma^2 + s\rho^2)}\right)^2\right)\mathrm{d}t$$

$$\leq Ck \cdot \exp\left(\frac{k^2\rho^4 n}{4C_2(\sigma^2 + s\rho^2)^2}\right) \cdot \frac{\rho^2\sqrt{n}}{\sigma^2 + s\rho^2}, \tag{C.90}$$

where $C$ is a positive absolute constant. Meanwhile, for $s\rho^2/\sigma^2 = o(\sqrt{s\log d/n})$, it holds for the right-hand side of (C.90) that

$$\exp\left(\frac{k^2\rho^4 n}{4C_2(\sigma^2 + s\rho^2)^2}\right) \cdot \frac{\rho^2\sqrt{n}}{\sigma^2 + s\rho^2} \leq C'\sqrt{\log d/s} \cdot \exp(C_0 k^2 \log d/s), \tag{C.91}$$

which holds for an arbitrary positive absolute constant $C_0$ and a sufficiently large $n$, respectively. Here $C'$ is a positive absolute constant. Combining (C.88), (C.90), and (C.91), we conclude that

$$T_2 = \sum_{k=1}^{s}\left(\frac{s^2e}{kd}\right)^k \cdot \mathbb{E}_U\left[\exp(k|Q|) \cdot \mathbb{1}\{|Q| \geq 1\}\right]$$

$$\leq C_1 \sum_{k=1}^{s}\left(\frac{s^2e^2}{kd}\right)^k + C''\sqrt{\log d/s} \cdot \sum_{k=1}^{s} k \cdot \left(\frac{s^2e^2}{kd} \cdot \exp(C_0 k \log d/s)\right)^k. \tag{C.92}$$

Note that $s = o(d^{1/2-\delta})$ for a positive absolute constant $\delta$. Thus, it holds that $s^2 e^2/(kd) = o(1)$ for $0 \le k \le s$, which implies that

$$\sum_{k=1}^{s} \left( \frac{s^2 e^2}{kd} \right)^k = o(1). \tag{C.93}$$

Meanwhile, it holds for any $1 \le k \le s$ that

$$\frac{s^2 e^2}{kd} \cdot \exp(C_0 k \log d/s) \le \frac{s^2 e^2}{kd} \cdot \exp(C_0 \log d) \le e^2/d^{2\delta - C_0}. \tag{C.94}$$

Since $C_0$ is arbitrary, we fix $C_0 > 2\delta$. It then holds for a positive absolute constant $C$ that

$$\sqrt{\log d/s} \cdot \sum_{k=1}^{s} k \cdot \left( \frac{s^2 e^2}{kd} \cdot \exp(C_0 k \log d/s) \right)^k \le C \cdot \sqrt{\log d/s} \cdot e^2/d^{2\delta - C_0} = o(1). \tag{C.95}$$

Combining (C.92), (C.93), and (C.95), we obtain that $T_2 = o(1)$, which concludes the proof of Lemma C.2. $\qquad\square$

# D  Upper Bounds for General Cases

In this section, we characterize the upper bounds for the hypothesis testing problem in (A.1) under the general setting. In specific, we consider the hypothesis testing problem that takes the form

$$H_0\colon Y = \epsilon_0 \quad \text{versus} \quad H_1\colon Y = \begin{cases} f_1(X^\top \beta^*) + \epsilon, & \text{with probability } \alpha, \\ f_2(X^\top \beta^*) + \epsilon, & \text{with probability } 1 - \alpha. \end{cases} \tag{D.1}$$

Here $\epsilon$ is a Gaussian noise with variance $\sigma^2$, $\epsilon_0$ is a noise such that the variances of $Y$ under the null and alternative hypotheses are the same. Besides, $f_1 \in \mathcal{C}_1 \cap \mathcal{C}(\psi)$ and $f_2 \in \mathcal{C}_2 \cap \mathcal{C}(\psi)$ are two unknown link functions, where $\mathcal{C}_1(\psi)$, $\mathcal{C}_2(\psi)$, and $\mathcal{C}(\psi)$ are defined in (2.4) and (2.5). Meanwhile, we set $X \sim N(0, I_d)$ and

$$(\beta^*, \sigma) \in \mathcal{G}_1(s, \gamma_n) = \left\{ (\beta^*, \sigma) \in \mathbb{R}^{d+1}\colon \|\beta^*\|_0 = s, \kappa(\beta^*, \sigma) \ge \gamma_n \right\} \tag{D.2}$$

under the alternative hypothesis, where $\kappa(\beta^*, \sigma) = \|\beta^*\|_2^2/\sigma^2$ is the SNR. We further denote by

$$\mathcal{H}(s, \gamma_n) = \left\{ \beta^* \in \mathbb{R}^d : \|\beta^*\|_2^2/\sigma^2 = s\rho^2/\sigma^2 \ge \gamma_n, \ \|\beta^*\|_0 = s \right\}. \tag{D.3}$$

We denote by $Z = (Y, X)$ and $\mathbb{P}_0$, $\mathbb{P}_{\beta^*}$ be the distributions of $Z$ under the null and alternative hypotheses, respectively. We assume that the Assumption A.1 holds. We denote by

$$\mathcal{V}(s) = \left\{ \mathcal{S} \in [d] : |\mathcal{S}| = s \right\}$$

the class of index sets. For each index set $\mathcal{S} \in \mathcal{V}(s)$, we denote by $\mathcal{B}(\mathcal{S})$ the $s$-sparse unit sphere that is supported on the index set $\mathcal{S}$. We further denote by $\mathcal{N}(\epsilon, \mathcal{S}) \subseteq \mathcal{B}(\mathcal{S})$ the minimum $\epsilon$-covering of the $s$-sparse unit sphere $\mathcal{B}(\mathcal{S})$. In other words, it holds for any $u \in \mathcal{B}(\mathcal{S})$ that $\|u - v\|_2 \le \epsilon$ for some $v \in \mathcal{N}(\epsilon, \mathcal{S})$. Meanwhile, $\mathcal{N}(\epsilon, \mathcal{S})$ attains the smallest cardinality among the sets that have such a property. It then holds that

$$|\mathcal{N}(\epsilon, \mathcal{S})| \le C_0 \cdot (1 + 2/\epsilon)^s, \tag{D.4}$$

where $C_0$ is a positive absolute constant. We define

$$\mathcal{N}(\epsilon) = \bigcup_{\mathcal{S} \in \mathcal{V}(s)} \mathcal{N}(\epsilon, \mathcal{S}). \tag{D.5}$$

Therefore, it holds that

$$|\mathcal{N}(\epsilon)| \le C_0 \cdot (1 + 2/\epsilon)^s \cdot \binom{d}{s}. \tag{D.6}$$

In what follows, we construct test functions based on $v \in \mathcal{N}(1/2)$. We introduce the following query functions for $v \in \mathcal{N}(1/2)$,

$$q_{1,v}(Y, X) = \psi(Y) \cdot \left[ (v^\top X)^2 - 1 \right] \cdot \mathbb{1}\left\{ |\psi(Y)| \le (R \log n)^{1/\nu} \right\} \cdot \mathbb{1}\left\{ |v^\top X| \le R \cdot \sqrt{\log n} \right\},$$

$$q_{2,v}(Y, X) = Y \cdot (v^\top X) \cdot \mathbb{1}\left\{ |Y| \le (R \log n)^{1/\nu} \right\} \cdot \mathbb{1}\left\{ |v^\top X| \le R \cdot \sqrt{\log n} \right\}. \tag{D.7}$$

We denote by $\bar{Z}_{1,v}$ and $\bar{Z}_{2,v}$ the responses of the statistical oracle to query functions $q_{1,v}$ and $q_{2,v}$, as defined in Definition 2.3. We define the test functions $\phi_1$ and $\phi_2$ as

$$\phi_1 = \mathbb{1}\left\{ \sup_{v \in \bar{\mathcal{G}}(s)} \bar{Z}_{1,v} \ge \tau_1 \right\}, \quad \phi_2 = \mathbb{1}\left\{ \sup_{v \in \bar{\mathcal{G}}(s)} \bar{Z}_{2,v} \ge \tau_2 \right\}, \tag{D.8}$$

where we set the thresholds $\tau_1$ and $\tau_2$ to be

$$\tau_1 = CR^{2+1/\nu} \cdot (\log n)^{1+1/\nu} \cdot \sqrt{\frac{s \log d}{n}}, \quad \tau_2 = C'R^{1+1/\nu} \cdot (\log n)^{1/2+1/\nu} \cdot \sqrt{\frac{s \log d}{n}}, \quad \text{(D.9)}$$

where $C$ and $C'$ are positive absolute constants that will be specified in §D.1. We define the test function as $\phi = \phi_1 \vee \phi_2$. Following from (D.6), the capacity of $\mathcal{Q}_\phi$ is upper bounded as follows,

$$|\mathcal{Q}_\phi| \le 2C_0 \cdot 5^s \cdot \binom{d}{s}. \quad \text{(D.10)}$$

The following theorem characterizes an upper bound for the minimax separation rate by quantifying the SNR for $\phi$ to be asymptotically powerful.

**Theorem D.1.** We consider the hypothesis testing problem in (D.1) under Assumption A.1. For

$$\gamma_n = \Omega\left( (\log n)^{1+1/\nu} \cdot \sqrt{\frac{s \log d}{n}} \bigwedge \frac{(\log n)^{1+2/\nu}}{\alpha^2} \cdot \frac{s \log d}{n} \right), \quad \text{(D.11)}$$

it holds that $R_n(\phi; \mathcal{G}_0, \mathcal{G}_1) = O(1/d)$. In other words, $\phi$ is asymptotically powerful.

*Proof.* See §D.1 for a detailed proof. $\square$

To construct a computationally tractable test, we define query functions as follows,

$$q_{1,j}(Y,X) = \psi(Y) \cdot (X_j^2 - 1) \cdot \mathbb{1}\{|\psi(Y)| \le (R \log n)^{1/\nu}\} \cdot \mathbb{1}\{|X_j| \le R\sqrt{\log n}\}, \quad j \in [d]$$

$$q_{2,j}(Y,X) = Y \cdot X_j \cdot \mathbb{1}\{|Y| \le (R \log n)^{1/\nu}\} \cdot \mathbb{1}\{|X_j| \le R\sqrt{\log n}\}, \quad j \in [d]. \quad \text{(D.12)}$$

We denote by $\bar{Z}_{1,j}$ and $\bar{Z}_{2,j}$ the responses of the statistical oracle to the query functions $q_{1,j}$ and $q_{2,j}$, as defined in Definition 2.3 . We define the test functions $\widetilde{\phi}_1$ and $\widetilde{\phi}_2$ as

$$\widetilde{\phi}_1 = \mathbb{1}\left\{ \sup_{j \in [d]} \bar{Z}_{1,j} \ge \widetilde{\tau}_1 \right\}, \quad \widetilde{\phi}_2 = \mathbb{1}\left\{ \sup_{j \in [d]} \bar{Z}_{2,j} \ge \widetilde{\tau}_2 \right\} \bigvee \mathbb{1}\left\{ \inf_{j \in [d]} \bar{Z}_{2,j} \le -\widetilde{\tau}_2 \right\}, \quad \text{(D.13)}$$

where we set the thresholds $\widetilde{\tau}_1$ and $\widetilde{\tau}_2$ to be

$$\widetilde{\tau}_1 = CR^{2+1/\nu}(\log n)^{1+1/\nu} \cdot \sqrt{\frac{\log d}{n}}, \quad \widetilde{\tau}_2 = C'R^{1+1/\nu}(\log n)^{1/2+1/\nu} \cdot \sqrt{\frac{\log d}{n}}. \quad \text{(D.14)}$$

We define the test function $\widetilde{\phi} = \widetilde{\phi}_1 \vee \widetilde{\phi}_2$. Therefore, the test function $\widetilde{\phi}$ is with capacity of query functions $|\mathcal{Q}_{\widetilde{\phi}}| = 2d$. The following theorem holds, which characterizes the minimum SNR required for the test function $\widetilde{\phi}$ to be asymptotically powerful.

**Theorem D.2.** We consider the hypothesis testing problem in (D.1) under Assumption A.1. For

$$\gamma_n = \Omega\left( (\log n)^{1+1/\nu} \cdot \sqrt{\frac{s^2 \log d}{n}} \bigwedge \frac{(\log n)^{1+2/\nu}}{\alpha^2} \cdot \frac{s \log d}{n} \right), \quad \text{(D.15)}$$

it holds that $\bar{R}_n(\widetilde{\phi}; \mathcal{G}_0, \mathcal{G}_1) = O(1/d)$. In other words, $\widetilde{\phi}$ is asymptotically powerful.

*Proof.* See §D.2 for a detailed proof. $\square$

## D.1 Proof of Theorem D.1

*Proof.* The proof is similar to that of Theorem A.2 in §B.3. Recall that we denote by $\mathbb{P}_0$ and $\mathbb{P}_{\beta^*}$ the distributions of $Z = (Y, X)$ under the null and alternative hypotheses, respectively. The following lemma holds, which characterizes the expection of $q_{1,\mathrm{v}}$ and $q_{2,\mathrm{v}}$ under the null and alternative hypotheses, respectively.

**Lemma D.3.** For any $\mathrm{v} \in \mathcal{N}(1/2)$, $\beta^* \in \mathcal{H}(s, \gamma_n)$, and

$$\gamma_n = \Omega\left( (\log n)^{1+1/\nu} \cdot \sqrt{\frac{s \log d}{n}} \bigwedge \frac{(\log n)^{1+2/\nu}}{\alpha^2} \cdot \frac{s \log d}{n} \right),$$

it holds that

$$\mathbb{E}_{\mathbb{P}_0}\big[q_{1,\mathrm{v}}(Y,X)\big] \le 1/n, \quad \mathbb{E}_{\mathbb{P}_0}\big[q_{2,\mathrm{v}}(Y,X)\big] \le 1/n. \quad \text{(D.16)}$$

In addition, it holds that

$$\sup_{v\in\mathcal{N}(1/2)} \mathbb{E}_{\mathbb{P}_{\beta^*}}\big[q_{1,v}(Y,X)\big] \geq s\rho^2/2 \ \text{ if } \gamma_n = \Omega\bigg((\log n)^{1+1/\nu}\cdot\sqrt{\frac{s\log d}{n}}\bigg),$$

$$\sup_{v\in\mathcal{N}(1/2)} \mathbb{E}_{\mathbb{P}_{\beta^*}}\big[q_{2,v}(Y,X)\big] \geq \sqrt{\alpha^2 s\rho^2}/2 \ \text{ if } \gamma_n = \Omega\bigg(\frac{(\log n)^{1+2/\nu}}{\alpha^2}\cdot\frac{s\log d}{n}\bigg). \qquad (\text{D.17})$$

*Proof.* See §D.3 for a detailed proof. $\qquad\square$

It now suffices to upper bound the risk of $\phi = \phi_1 \vee \phi_2$, where $\phi_1$ and $\phi_2$ are defined in (D.8). Recall that we define the threshold $\tau_1$ and $\tau_2$ as

$$\tau_1 = CR^{2+1/\nu}\cdot(\log n)^{1+1/\nu}\cdot\sqrt{\frac{s\log d}{n}}, \quad \tau_2 = C'R^{1+1/\nu}\cdot(\log n)^{1/2+1/\nu}\cdot\sqrt{\frac{s\log d}{n}}, \ (\text{D.18})$$

where $C$ and $C'$ are positive absolute constants. Note that for the test function $\phi$, the capacity of query functions is upper bounded in (D.10). Therefore, following from (2.12) with $\xi = 1/d$, it holds for a sufficiently large $n$ that

$$\tau_{q_{1,v}} \leq C_1 R^{2+1/\nu}(\log n)^{1/2+1/\nu}\cdot\sqrt{\frac{s\log d}{n}},$$

$$\tau_{q_{2,v}} \leq C_2 R^{1+1/\nu}(\log n)^{1/2+1/\nu}\cdot\sqrt{\frac{s\log d}{n}}, \qquad (\text{D.19})$$

where $\tau_{q_{1,v}}$ and $\tau_{q_{2,v}}$ are the tolerance parameters of $q_{1,v}$ and $q_{2,v}$ defined in Definition 2.3, and $C_1$, $C_2$ are positive absolute constants. We fix $C$ and $C'$ in (D.18) such that $\tau_1 \geq \tau_{q_{1,v}} + 1/n$ and $\tau_2 \geq \tau_{q_{2,v}} + 1/n$. The rest of the proof then follows a similar argument in §B.3. Recall that we denote by $\bar{Z}_{1,v}$ and $\bar{Z}_{2,v}$ the responses of the statistical oracle to the query functions $q_{1,v}$ and $q_{2,v}$. We denote by $\bar{\mathbb{P}}_0$ and $\bar{\mathbb{P}}_{\beta^*}$ the distributions of response of the statistical oracle to the query functions when the true distribution of the data is $\mathbb{P}_0$ and $\mathbb{P}_{\beta^*}$. Following from Lemma D.3, it holds for any $v\in\mathcal{N}(1/2)$ that

$$\bar{\mathbb{P}}_0\big(\bar{Z}_{i,v}\geq\tau_i\big)\leq\bar{\mathbb{P}}_0\Big(\big|\bar{Z}_{i,v}-\mathbb{E}_{\mathbb{P}_0}\big[q_{i,v}(Y,X)\big]\big|\geq\tau_{q_{i,v}}\Big), \quad i\in\{1,2\}.$$

Therefore, following from (2.11) with $\xi = 1/d$, we obtain

$$\bar{\mathbb{P}}_0(\phi_i = 1) = \bar{\mathbb{P}}_0\bigg(\sup_{v\in\mathcal{N}(1/2)}\bar{Z}_{i,v}>\tau_i\bigg)$$

$$\leq\bar{\mathbb{P}}_0\bigg(\bigcup_{v\in\mathcal{N}(1/2)}\Big\{\big|\bar{Z}_{i,v}-\mathbb{E}_{\mathbb{P}_0}\big[q_{i,v}(Y,X)\big]\big|>\tau_{q_{i,v}}\Big\}\bigg)\leq 2/d. \qquad (\text{D.20})$$

Recall that we define $\phi = \phi_1\vee\phi_2$. Then it holds that
$$\bar{\mathbb{P}}_0(\phi = 1)\leq\bar{\mathbb{P}}_0(\phi_1 = 1)+\bar{\mathbb{P}}_0(\phi_2 = 1) = 4/d, \qquad (\text{D.21})$$

which is an upper bound of the type-I error of $\phi$. It now suffices to upper bound the type-II error of $\phi$. If (D.11) holds, we obtain that either $s\rho^2/4 \geq \tau_1$ or $\sqrt{\alpha^2 s\rho^2}/4 \geq \tau_2$ for a sufficiently large $n$. We denote by

$$v^* \in \operatorname*{argmax}_{v\in\mathcal{N}(1/2)} \mathbb{E}_{\mathbb{P}_{\beta^*}}\big[q_{1,v}(Y,X)\big], \quad u^* \in \operatorname*{argmax}_{v\in\mathcal{N}(1/2)} \mathbb{E}_{\mathbb{P}_{\beta^*}}\big[q_{2,v}(Y,X)\big].$$

If it holds that $s\rho^2/4 \geq \tau_1$, then following from Lemma D.3, we obtain that

$$\bar{\mathbb{P}}_{\beta^*}(\phi_1 = 0) = \bar{\mathbb{P}}_{\beta^*}\bigg(\sup_{v\in\mathcal{N}(1/2)}\bar{Z}_{1,v}<\tau_1\bigg)\leq\bar{\mathbb{P}}_{\beta^*}(\bar{Z}_{1,v^*}<\tau_1)$$

$$\leq\bar{\mathbb{P}}_{\beta^*}\Big(\bar{Z}_{1,v^*}<\mathbb{E}_{\mathbb{P}_{\beta^*}}\big[q_{1,v^*}(Y,X)\big]-\tau_1\Big) \qquad (\text{D.22})$$

$$\leq\bar{\mathbb{P}}_{\beta^*}\Big(\big|\bar{Z}_{1,v^*}-\mathbb{E}_{\mathbb{P}_{\beta^*}}\big[q_{1,v^*}(Y,X)\big]\big|>\tau_{q_{1,v^*}}\Big), \qquad (\text{D.23})$$

where the last inequality follows from the fact that $\tau_1 > \tau_{q_{1,v^*}}$. Therefore, following from (2.11) with $\xi = 1/d$, we obtain that the right-hand side of (D.22) is upper bounded by $2/d$. Similarly, if it

holds that $\sqrt{\alpha^2 s \rho^2}/4 \geq \tau_2$, we obtain

$$\bar{\mathbb{P}}_{\beta^*}(\phi_2 = 0) = \bar{\mathbb{P}}_{\beta^*}\left(\sup_{\mathrm{v} \in \mathcal{N}(1/2)} \bar{Z}_{1,\mathrm{v}} < \tau_1\right) \leq \bar{\mathbb{P}}_{\beta^*}(\bar{Z}_{2,\mathrm{u}^*} < \tau_2)$$

$$\leq \bar{\mathbb{P}}_{\beta^*}\left(\left|\bar{Z}_{2,\mathrm{u}^*} - \mathbb{E}_{\mathbb{P}_{\beta^*}}[q_{2,\mathrm{u}^*}(Y, X)]\right| > \tau_{q_{2,\mathrm{u}^*}}\right), \tag{D.24}$$

where the last inequality follows from the fact that $\tau_1 > \tau_{q_{1,\mathrm{u}^*}}$. Therefore, following from (2.11) with $\xi = 1/d$, we obtain that the right-hand side of (D.24) is upper bounded by $2/d$. Note that (D.22) and (D.24) holds for all $(\beta^*, \sigma) \in \mathcal{G}_1(s, \gamma_n)$ if (D.11) holds. Therefore, we conclude that

$$\sup_{(\beta^*, \sigma) \in \mathcal{G}_1} \bar{\mathbb{P}}_{\beta^*}(\phi = 0) \leq \sup_{(\beta^*, \sigma) \in \mathcal{G}_1} \left\{\bar{\mathbb{P}}_{\beta^*}(\phi_1 = 0) \wedge \bar{\mathbb{P}}_{\beta^*}(\phi_2 = 0)\right\} \leq 2/d. \tag{D.25}$$

Combining (D.21) and (D.25), we obtain that if (D.11) holds, the risk of $\phi$ is $O(1/d)$, which concludes the proof. $\qquad\square$

## D.2 Proof of Theorem D.2

*Proof.* The proof is similar to that of Theorem A.3 in §B.4. Recall that we denote by $\mathbb{P}_0$ and $\mathbb{P}_{\beta^*}$ the distributions of $Z = (Y, X)$ under the null and alternative hypotheses, respectively. The following lemma holds, which characterizes the expection of $q_{1,j}(Y, X)$ and $q_{2,j}(Y, X)$ under the null and alternative hypotheses, respectively.

**Lemma D.4.** For any $\beta^* \in \mathcal{H}(s, \gamma_n)$ and

$$\gamma_n = \Omega\left((\log n)^{1+1/\nu} \cdot \sqrt{\frac{s^2 \log d}{n}} \bigwedge \frac{(\log n)^{1+2/\nu}}{\alpha^2} \cdot \frac{s \log d}{n}\right),$$

it holds that

$$\sup_{j \in [d]} \mathbb{E}_{\mathbb{P}_0}[q_{1,j}(Y, X)] \leq 1/n, \quad \sup_{j \in [d]} \mathbb{E}_{\mathbb{P}_0}[q_{2,j}(Y, X)] \leq 1/n. \tag{D.26}$$

In addition, it holds that

$$\sup_{j \in [d]} \mathbb{E}_{\mathbb{P}_{\beta^*}}[q_{1,j}(Y, X)] \geq \rho^2/2 \text{ if } \gamma_n = \Omega\left((\log n)^{1+1/\nu} \cdot \sqrt{\frac{s^2 \log d}{n}}\right),$$

$$\sup_{j \in [d]} \left|\mathbb{E}_{\mathbb{P}_{\beta^*}}[q_{2,j}(Y, X)]\right| \geq \alpha\rho/2 \text{ if } \gamma_n = \Omega\left(\frac{(\log n)^{1+2/\nu}}{\alpha^2} \cdot \frac{s \log d}{n}\right). \tag{D.27}$$

*Proof.* See §D.4 for a detailed proof. $\qquad\square$

In what follows, we upper bound the risk of $\widetilde{\phi} = \widetilde{\phi}_1 \vee \widetilde{\phi}_2$ where $\widetilde{\phi}_1$ and $\widetilde{\phi}_2$ are defined in (D.13). Recall that we define the threshold $\widetilde{\tau}_1$ and $\widetilde{\tau}_2$ as

$$\widetilde{\tau}_1 = CR^{2+1/\nu}(\log n)^{1+1/\nu} \cdot \sqrt{\frac{\log d}{n}}, \quad \widetilde{\tau}_2 = C'R^{1+1/\nu}(\log n)^{1/2+1/\nu} \cdot \sqrt{\frac{\log d}{n}}, \tag{D.28}$$

where $C$ and $C'$ are absolute constants. Note that for $\widetilde{\phi}$, the capacity of query functions is $2d$. Therefore, following from (2.12) with $\xi = 1/d$, it holds for a sufficiently large $n$ that

$$\tau_{q_{1,j}} \leq C_1 R^{2+1/\nu}(\log n)^{1/2+1/\nu} \cdot \sqrt{\frac{\log d}{n}}, \quad \tau_{q_{2,j}} \leq C_2 R^{1+1/\nu}(\log n)^{1/2+1/\nu} \cdot \sqrt{\frac{\log d}{n}}, \tag{D.29}$$

where $C_1$ and $C_2$ are positive absolute constants. We fix $C$ and $C'$ in (D.28) such that $\widetilde{\tau}_1 > \tau_{q_{1,j}} + 1/n$ and $\tau_2 > \tau_{1_{2,j}} + 1/n$ for a sufficiently large $n$. Recall that we denote by $\bar{Z}_{1,j}$ and $\bar{Z}_{2,j}$ the responses of the statistical oracle to the query functions $q_{1,j}$ and $q_{2,j}$. We denote by $\bar{\mathbb{P}}_0$ and $\bar{\mathbb{P}}_{\beta^*}$ the distributions of response of the statistical oracle to the query functions when the true distribution of the data is $\mathbb{P}_0$ and $\mathbb{P}_{\beta^*}$. Following from Lemma D.3, it holds for $j \in [d]$ and $i \in \{1, 2\}$ that

$$\bar{\mathbb{P}}_0(\bar{Z}_{i,j} \geq \widetilde{\tau}_1) \leq \bar{\mathbb{P}}_0\left(\left|\bar{Z}_{i,j} - \mathbb{E}_{\mathbb{P}_0}[q_{i,j}(Y, X)]\right| \geq \tau_{q_{i,j}}\right). \tag{D.30}$$

Therefore, following from (2.11) with $\xi = 1/d$, it holds for $i \in \{1, 2\}$ that

$$\bar{\mathbb{P}}_0(\widetilde{\phi}_i = 1) = \bar{\mathbb{P}}_0\left(\sup_{j \in [d]} \bar{Z}_{i,j} > \widetilde{\tau}_i\right)$$

$$\leq \bar{\mathbb{P}}_0\left(\bigcup_{j \in [d]}\left\{\left|\bar{Z}_{i,j} - \mathbb{E}_{\mathbb{P}_0}\left[q_{i,j}(Y, X)\right]\right| > \tau_{q_{i,j}}\right\}\right) \leq 2/d, \qquad (\text{D.31})$$

which further shows that

$$\bar{\mathbb{P}}_0(\widetilde{\phi} = 1) \leq \bar{\mathbb{P}}_0(\widetilde{\phi}_1 = 1) + \bar{\mathbb{P}}_0(\widetilde{\phi}_2 = 1) \leq 4/d. \qquad (\text{D.32})$$

In other words, it holds that the type-I error of $\widetilde{\phi}$ is asymptotically upper bounded by $4/d$. It remains to upper bound the type-II error of $\widetilde{\phi}$. Note that if (D.15) holds, it holds that either $\rho^2/4 \geq \widetilde{\tau}_1$ or $\alpha\rho/4 \geq \widetilde{\tau}_2$ for a sufficiently large $n$. We denote by

$$j^* \in \operatorname*{argmax}_{j \in [d]} \mathbb{E}_{\mathbb{P}_{\beta^*}}\left[q_{1,j}(Y, X)\right], \quad k^* \in \operatorname*{argmax}_{j \in [d]}\left|\mathbb{E}_{\mathbb{P}_{\beta^*}}\left[q_{2,j}(Y, X)\right]\right|.$$

If it holds that $\rho^2/4 \geq \widetilde{\tau}_1$, following from Lemma D.4, we obtain that

$$\bar{\mathbb{P}}_{\beta^*}(\widetilde{\phi}_1 = 0) \leq \bar{\mathbb{P}}_{\beta^*}\left(\sup_{j \in [d]} \bar{Z}_{1,j} < \widetilde{\tau}_2\right) \leq \bar{\mathbb{P}}_{\beta^*}(\bar{Z}_{1,j^*} < \widetilde{\tau}_1)$$

$$\leq \bar{\mathbb{P}}_{\beta^*}\left(\bar{Z}_{1,j^*} < \mathbb{E}_{\mathbb{P}_{\beta^*}}\left[q_{1,j^*}(Y, X)\right] - \widetilde{\tau}_1\right)$$

$$\leq \bar{\mathbb{P}}_{\beta^*}\left(\left|\bar{Z}_{2,j^*} - \mathbb{E}_{\mathbb{P}_\mathrm{v}}\left[q_{2,j^*}(Y, X)\right]\right| > \tau_{q_{2,j^*}}\right) \leq 2/d, \qquad (\text{D.33})$$

where the fourth inequality follows from the fact that $\widetilde{\tau}_1 > \tau_{q_{1,j^*}}$, and the last inequality following from (2.11) with $\xi = 1/d$. If it holds that $\alpha\rho/4 \geq \widetilde{\tau}_2$, following from Lemma D.4, we obtain that either $\mathbb{E}_{\mathbb{P}_{\beta^*}}\left[q_{2,k^*}(Y, X)\right] \geq \alpha\rho/2$ or $\mathbb{E}_{\mathbb{P}_{\beta^*}}\left[q_{2,k^*}(Y, X)\right] \leq -\alpha\rho/2$. If $\mathbb{E}_{\mathbb{P}_{\beta^*}}\left[q_{2,k^*}(Y, X)\right] \geq \alpha\rho/2$, we obtain that

$$\bar{\mathbb{P}}_{\beta^*}(\widetilde{\phi}_2 = 0) \leq \bar{\mathbb{P}}_{\beta^*}\left(\sup_{j \in [d]} \bar{Z}_{2,j} < \widetilde{\tau}_2\right) \leq \bar{\mathbb{P}}_{\beta^*}(\bar{Z}_{2,k^*} < \widetilde{\tau}_2)$$

$$\leq \bar{\mathbb{P}}_{\beta^*}\left(\bar{Z}_{2,k^*} < \mathbb{E}_{\mathbb{P}_{\beta^*}}\left[q_{2,k^*}(Y, X)\right] - \widetilde{\tau}_2\right)$$

$$\leq \bar{\mathbb{P}}_{\beta^*}\left(\left|\bar{Z}_{2,k^*} - \mathbb{E}_{\mathbb{P}_\mathrm{v}}\left[q_{2,k^*}(Y, X)\right]\right| > \tau_{q_{2,k^*}}\right) \leq 2/d, \qquad (\text{D.34})$$

where the fourth inequality follows from the fact that $\widetilde{\tau}_2 > \tau_{q_{2,k^*}}$, and the last inequality follows from (2.11) with $\xi = 1/d$. If it holds that $\mathbb{E}_{\mathbb{P}_{\beta^*}}\left[q_{2,k^*}(Y, X)\right] \leq -\alpha\rho/2$, we obtain that

$$\bar{\mathbb{P}}_{\beta^*}(\widetilde{\phi}_2 = 0) \leq \bar{\mathbb{P}}_{\beta^*}\left(\inf_{j \in [d]} \bar{Z}_{2,j} > -\widetilde{\tau}_2\right) \leq \bar{\mathbb{P}}_{\beta^*}(\bar{Z}_{2,k^*} > -\widetilde{\tau}_2)$$

$$\leq \bar{\mathbb{P}}_{\beta^*}\left(\bar{Z}_{2,k^*} > \mathbb{E}_{\mathbb{P}_{\beta^*}}\left[q_{2,k^*}(Y, X)\right] + \widetilde{\tau}_2\right)$$

$$\leq \bar{\mathbb{P}}_{\beta^*}\left(\left|\bar{Z}_{2,k^*} - \mathbb{E}_{\mathbb{P}_\beta^*}\left[q_{2,k^*}(Y, X)\right]\right| > \tau_{q_{2,k^*}}\right) \leq 2/d, \qquad (\text{D.35})$$

where the fourth inequality follows from the fact that $\widetilde{\tau}_2 > \tau_{q_{2,k^*}}$, and the last inequality follows from (2.11) with $\xi = 1/d$. Note that (D.33), (D.34), and (D.35) holds for all $(\beta^*, \sigma) \in \mathcal{G}_1(s, \gamma_n)$ if (D.15) holds. Therefore, we obtain that

$$\sup_{(\beta^*, \sigma) \in \mathcal{G}_1} \bar{\mathbb{P}}_{\beta^*}(\widetilde{\phi} = 0) \leq \sup_{(\beta^*, \sigma) \in \mathcal{G}_1}\left\{\bar{\mathbb{P}}_{\beta^*}(\widetilde{\phi}_1 = 0) \wedge \bar{\mathbb{P}}_{\beta^*}(\widetilde{\phi}_2 = 0)\right\} \leq 2/d. \qquad (\text{D.36})$$

Combining (D.32) and (D.36), we obtain that if (D.15) holds, the risk of $\widetilde{\phi}$ is $O(1/d)$, which concludes the proof of Theorem D.2. $\qquad \square$

## D.3 Proof of Lemma D.3

*Proof.* In the following proof, we denote by $C$ and $C'$ absolute constants, the value of which may vary from lines to lines. We define the following query functions,

$$\widetilde{q}_{1,\mathrm{v}}(Y, X) = \psi(Y) \cdot \left[(\mathrm{v}^\top X)^2 - 1\right] \cdot \mathbb{1}\left\{|\psi(Y)| \leq (R \cdot \log n)^{1/\nu}\right\}, \quad \mathrm{v} \in \bar{\mathcal{G}}(s),$$

$$\widetilde{q}_{2,\mathrm{v}}(Y, X) = Y \cdot (\mathrm{v}^\top X) \cdot \mathbb{1}\left\{|Y| \leq (R \cdot \log n)^{1/\nu}\right\}, \quad \mathrm{v} \in \bar{\mathcal{G}}(s). \qquad (\text{D.37})$$

Following from (D.7) and (D.37), we conclude that

$$\widetilde{q}_{1,\mathrm{v}} - q_{1,\mathrm{v}} = \psi(Y) \cdot \left[(\mathrm{v}^\top X)^2 - 1\right] \cdot \mathbb{1}\left\{|\psi(Y)| \leq (R \cdot \log n)^{1/\nu}\right\} \cdot \mathbb{1}\left\{|\mathrm{v}^\top X| > R \cdot \sqrt{\log n}\right\},$$

$$\widetilde{q}_{2,\mathrm{v}} - q_{2,\mathrm{v}} = Y \cdot (\mathrm{v}^\top X) \cdot \mathbb{1}\left\{|Y| \leq (R \cdot \log n)^{1/\nu}\right\} \cdot \mathbb{1}\left\{|\mathrm{v}^\top X| > R \cdot \sqrt{\log n}\right\}. \tag{D.38}$$

Therefore, following from the Cauchy-Schwartz inequality, we obtain from (D.38) that

$$\left|\mathbb{E}_{\mathbb{P}_0}\left[\widetilde{q}_{1,\mathrm{v}}(Y,X) - q_{1,\mathrm{v}}(Y,X)\right]\right|^2$$
$$\leq \mathbb{E}_{\mathbb{P}_0}\left[\psi^2(Y) \cdot \left[(\mathrm{v}^\top X)^2 - 1\right]^2\right] \cdot \mathbb{P}_0\left(|\mathrm{v}^\top X| \geq R \cdot \sqrt{\log n}\right). \tag{D.39}$$

Further note that under the null hypothesis, $Y$ is independent of $X$ and $X \sim N(0, I_d)$. Therefore, for $\mathrm{v} \in \mathcal{N}(1/2)$, it holds that $\mathrm{v}^\top X \sim N(0,1)$. Meanwhile, following from Assumption A.1, $Y$ has bounded fourth moment. Therefore, we obtain from (D.39) and the tail bound of standard Gaussian distribution in (C.54) that

$$\left|\mathbb{E}_{\mathbb{P}_0}\left[\widetilde{q}_{1,\mathrm{v}}(Y,X) - q_{1,\mathrm{v}}(Y,X)\right]\right|^2 \leq C \cdot \exp(-R^2 \log n), \tag{D.40}$$

where $C$ is a positive absolute constant. Similarly, it holds under the alternative hypothesis that

$$\left|\mathbb{E}_{\mathbb{P}_\beta^*}\left[\widetilde{q}_{1,\mathrm{v}}(Y,X) - q_{1,\mathrm{v}}(Y,X)\right]\right|^2$$
$$\leq \mathbb{E}_{\mathbb{P}_\beta^*}\left[\psi^2(Y) \cdot \left[(\mathrm{v}^\top X)^2 - 1\right]^2\right] \cdot \mathbb{P}_0\left(|\mathrm{v}^\top X| \geq R \cdot \sqrt{\log n}\right)$$
$$\leq \left(\mathbb{E}_{\mathbb{P}_\beta^*}\left[\psi^4(Y)\right] \cdot \mathbb{E}_{\mathbb{P}_\beta^*}\left[\left[(\mathrm{v}^\top X)^2 - 1\right]^4\right]\right)^{1/2} \cdot \mathbb{P}_0\left(|\mathrm{v}^\top X| \geq R \cdot \sqrt{\log n}\right), \tag{D.41}$$

where the above inequalities follow from the Cauchy-Schwartz inequality. Then following from Assumption A.1 and the fact that $X \sim N(0, I_d)$ under the alternative hypothesis, we conclude that

$$\left|\mathbb{E}_{\mathbb{P}_\beta^*}\left[\widetilde{q}_{1,\mathrm{v}}(Y,X) - q_{1,\mathrm{v}}(Y,X)\right]\right|^2 \leq C' \cdot \mathbb{P}_{\beta^*}\left(|\mathrm{v}^\top X| \geq R \cdot \sqrt{\log n}\right)$$
$$\leq C' \cdot \exp(-R^2 \log n), \tag{D.42}$$

where $C'$ is a positive absolute constant, and the last inequality follows from the tail bound of standard Gaussian distribution in (C.54). Similar argument holds for the query functions $q_{2,\mathrm{v}}(Y,X)$ and $\widetilde{q}_{2,\mathrm{v}}(Y,X)$. We conclude from (D.40), (D.42) and a similar argument on $q_{2,\mathrm{v}}(Y,X)$ and $\widetilde{q}_{2,\mathrm{v}}(Y,X)$ that

$$\left|\mathbb{E}_{\mathbb{P}_{\beta^*}}\left[q_{1,\mathrm{v}}(Y,X) - \widetilde{q}_{1,\mathrm{v}}(Y,X)\right]\right| \vee \left|\mathbb{E}_{\mathbb{P}_0}\left[q_{1,\mathrm{v}}(Y,X) - \widetilde{q}_{1,\mathrm{v}}(Y,X)\right]\right| \leq 1/n,$$

$$\left|\mathbb{E}_{\mathbb{P}_{\beta^*}}\left[q_{2,\mathrm{v}}(Y,X) - \widetilde{q}_{2,\mathrm{v}}(Y,X)\right]\right| \vee \left|\mathbb{E}_{\mathbb{P}_0}\left[q_{2,\mathrm{v}}(Y,X) - \widetilde{q}_{2,\mathrm{v}}(Y,X)\right]\right| \leq 1/n, \tag{D.43}$$

which holds for $\mathrm{v} \in \mathcal{N}(1/2)$, $\beta^* \in \mathcal{H}(s, \gamma_n)$, and sufficiently large $n$ and constant $R$. Note that under the null hypothesis, it holds that $X \sim N(0, I_d)$ and $Y$ is independent of $X$. Therefore, it follows from (D.37) that

$$\mathbb{E}_0\left[\widetilde{q}_{1,\mathrm{v}}(Y,X)\right] = \mathbb{E}_0\left[\widetilde{q}_{2,\mathrm{v}}(Y,X)\right] = 0, \tag{D.44}$$

which holds for all $\mathrm{v} \in \mathcal{N}(1/2)$. Meanwhile, following from the definition of $\mathcal{N}(1/2)$ in (D.5), it holds that for any $\beta^* \in \mathcal{H}(s, \gamma_n)$, there exist a $\mathrm{v}^* \in \mathcal{N}(1/2)$ such that

$$\|\beta^*/\sqrt{s\rho^2} - \mathrm{v}^*\|_2^2 \leq 1/4,$$

which is equivalent to

$$\mathrm{v}^{*\top}\beta^* \geq 7/8 \cdot \sqrt{s\rho^2}. \tag{D.45}$$

Therefore, following from (A.3) and (D.45), it holds that

$$49/64 \cdot s\rho^2 - \mathbb{E}_{\mathbb{P}_{\beta^*}}\left[\widetilde{q}_{1,\mathrm{v}^*}(Y,X)\right] \leq (\mathrm{v}^{*\top}\beta^*)^2 - \mathbb{E}_{\mathbb{P}_{\beta^*}}\left[\widetilde{q}_{1,\mathrm{v}^*}(Y,X)\right]$$
$$\leq \mathbb{E}_{\mathbb{P}_{\beta^*}}\left[\psi(Y) \cdot \left((\mathrm{v}^{*\top}X)^2 - 1\right) - \widetilde{q}_{1,\mathrm{v}}(Y,X)\right]$$
$$= \mathbb{E}_{\mathbb{P}_{\beta^*}}\left[\psi(Y) \cdot \left((\mathrm{v}^{*\top}X)^2 - 1\right) \cdot \mathbb{1}\left\{|\psi(Y)| > (R \cdot \log n)^{1/\nu}\right\}\right]$$
$$\leq \sqrt{\mathbb{E}_{\mathbb{P}_{\beta^*}}\left[\psi^2(Y) \cdot \left((\mathrm{v}^{*\top}X)^2 - 1\right)^2\right]} \cdot \sqrt{\mathbb{P}_{\beta^*}\left(|\psi(Y)| > (R \cdot \log n)^{1/\nu}\right)}, \tag{D.46}$$

where the last inequality follows from the Cauchy-Schwartz inequality. It then follows from the Cauchy-Schwartz inequality and Assumption A.1 that

$$49/64 \cdot s\rho^2 - \mathbb{E}_{\mathbb{P}_{\beta^*}}\left[\widetilde{q}_{1,\mathrm{v}^*}(Y,X)\right] \leq C \cdot \exp(-R/2 \cdot \log n), \tag{D.47}$$

where $C$ is a positive absolute constant. If it holds that $s\rho^2/\sigma^2 = \Omega(\sqrt{s\log d}/n)$, we obtain that for sufficiently large $n$ and constant $R$, it holds that $s\rho^2/64 > 1/n$ and

$$49/64 \cdot s\rho^2 - \mathbb{E}_{\mathbb{P}_{\beta*}}\big[\widetilde{q}_{1,\mathrm{v}^*}(Y, X)\big] \le 1/64 \cdot s\rho^2. \tag{D.48}$$

In other words, it holds that $\mathbb{E}_{\mathbb{P}_{\beta*}}[\widetilde{q}_{1,\mathrm{v}^*}(Y, X)] \ge 3/4 \cdot s\rho^2$. Similarly, following from (A.4) and (D.45), we obtain

$$
\begin{aligned}
7/8 \cdot \sqrt{\alpha^2 s\rho^2} - \mathbb{E}_{\mathbb{P}_{\beta*}}\big[\widetilde{q}_{2,\mathrm{v}^*}(Y, X)\big] &\le \alpha \cdot \mathrm{v}^{*\top}\beta^* - \mathbb{E}_{\mathbb{P}_{\beta*}}\big[\widetilde{q}_{2,\mathrm{v}^*}(Y, X)\big] \\
&\le \mathbb{E}_{\mathbb{P}_{\beta*}}\big[Y \cdot (\mathrm{v}^{*\top} X) - \widetilde{q}_{1,\mathrm{v}}(Y, X)\big] \\
&= \mathbb{E}_{\mathbb{P}_{\beta*}}\big[Y \cdot (\mathrm{v}^{*\top} X) \cdot \mathbb{1}\big\{|Y| > (R \cdot \log n)^{1/\nu}\big\}\big] \\
&\le \sqrt{\mathbb{E}_{\mathbb{P}_{\beta*}}\big[Y^2 \cdot (\mathrm{v}^{*\top}X)^2\big]} \cdot \sqrt{\mathbb{P}_{\beta*}\big(|Y| > (R \cdot \log n)^{1/\nu}\big)}. 
\end{aligned} \tag{D.49}
$$

Then following from the Cauchy-Schwartz inequality and Assumption A.1, we obtain that

$$7/8 \cdot \sqrt{\alpha^2 s\rho^2} - \mathbb{E}_{\mathbb{P}_{\beta*}}\big[\widetilde{q}_{2,\mathrm{v}^*}(Y, X)\big] \le C' \cdot \exp(-R/2 \cdot \log n), \tag{D.50}$$

where $C'$ is a positive absolute constant. If it holds that $s\rho^2/\sigma^2 = \Omega(1/\alpha \cdot s\log d/n)$, we obtain that for sufficiently large $n$ and constant $R$, it holds that $\sqrt{\alpha^2 s\rho^2}/8 > 1/n$ and

$$7/8 \cdot \sqrt{\alpha^2 s\rho^2} - \mathbb{E}_{\mathbb{P}_{\beta*}}\big[\widetilde{q}_{2,\mathrm{v}^*}(Y, X)\big] \le 1/8 \cdot \sqrt{\alpha^2 s\rho^2}. \tag{D.51}$$

In other words, it holds that $\mathbb{E}_{\mathbb{P}_{\beta*}}[\widetilde{q}_{2,\mathrm{v}^*}(Y, X)] \ge 3/4 \cdot \sqrt{\alpha^2 s\rho^2}$. Combining (D.43), (D.48), and (D.51), we conclude that for sufficiently large $n$ and constant $R$, it holds that

$$\mathbb{E}_{\mathbb{P}_0}\big[q_{1,\mathrm{v}}(Y, X)\big] \le 1/n, \quad \mathbb{E}_{\mathbb{P}_0}\big[q_{2,\mathrm{v}}(Y, X)\big] \le 1/n.$$

Furthermore, it holds for sufficiently large $n$ and constant $R$ that

$$\sup_{\mathrm{v}\in\mathcal{N}(1/2)} \mathbb{E}_{\mathbb{P}_{\beta*}}\big[q_{1,\mathrm{v}}(Y, X)\big] \ge \mathbb{E}_{\mathbb{P}_{\beta*}}\big[q_{1,\mathrm{v}^*}(Y, X)\big] \ge s\rho^2/2, \ \ \text{if } s\rho^2/\sigma^2 = \Omega(\sqrt{s\log d}/n),$$

$$\sup_{\mathrm{v}\in\mathcal{N}(1/2)} \mathbb{E}_{\mathbb{P}_{\beta*}}\big[q_{2,\mathrm{v}}(Y, X)\big] \ge \mathbb{E}_{\mathbb{P}_{\beta*}}\big[q_{2,\mathrm{v}^*}(Y, X)\big] \ge \sqrt{\alpha^2 s\rho^2}/2, \ \ \text{if } s\rho^2/\sigma^2 = \Omega(1/\alpha^2 \cdot s\log d/n),$$

which concludes the proof of Lemma D.3. $\qquad\square$

## D.4   Proof of Lemma D.4

*Proof.* In the following proof, we denote by $C$ and $C'$ absolute constants, the value of which may vary from lines to lines. We define the following query functions,

$$\widetilde{q}_{1,j}(Y, X) = \psi(Y) \cdot (X_j^2 - 1) \cdot \mathbb{1}\big\{|\psi(Y)| \le (R \cdot \log n)^{1/\nu}\big\}, \quad j \in [d],$$

$$\widetilde{q}_{2,j}(Y, X) = YX_j \cdot \mathbb{1}\big\{|Y| \le (R \cdot \log n)^{1/\nu}\big\}, \quad j \in [d]. \tag{D.52}$$

Following from (D.13) and the Cauchy-Schwartz inequality, it holds that

$$\big|\mathbb{E}_{\mathbb{P}_0}\big[\widetilde{q}_{1,j}(Y, X) - q_{1,j}(Y, X)\big]\big|^2 \le \mathbb{E}_{\mathbb{P}_0}\big[\psi^2(Y) \cdot (X_j^2 - 1)^2\big] \cdot \mathbb{P}_0\big(|X_j| \ge R \cdot \sqrt{\log n}\big). \tag{D.53}$$

Note that under the null hypothesis, $Y$ is independent of $X$ and $X \sim N(0, I_d)$. Then following from Assumption A.1 and the tail bound of standard Gaussian distribution in (C.54), it holds that

$$\big|\mathbb{E}_{\mathbb{P}_0}\big[\widetilde{q}_{1,j}(Y, X) - q_{1,j}(Y, X)\big]\big|^2 \le C \cdot \exp(-R^2 \cdot \log n), \tag{D.54}$$

where $C$ is a positive absolute constant. Under the alternative hypothesis, it holds that

$$\big|\mathbb{E}_{\mathbb{P}_{\beta*}}\big[\widetilde{q}_{1,j}(Y, X) - q_{1,j}(Y, X)\big]\big|^2 \le \mathbb{E}_{\mathbb{P}_{\beta*}}\big[\psi^2(Y) \cdot (X_j^2 - 1)^2\big] \cdot \mathbb{P}_{\beta*}\big(|X_j| \ge R \cdot \sqrt{\log n}\big) \tag{D.55}$$

$$\le \sqrt{\mathbb{E}_{\mathbb{P}_{\beta*}}\big[\psi^4(Y)\big] \cdot \mathbb{E}_{\mathbb{P}_{\beta*}}\big[(X_j^2 - 1)^4\big]} \cdot \mathbb{P}_{\beta*}\big(|X_j| \ge R \cdot \sqrt{\log n}\big),$$

where the above inequalities follows from the Cauchy-Schwartz inequality. Note that under the alternative hypothesis, we have $X \sim N(0, I_d)$. Then following from Assumption A.1 and the tail bound of standard Gaussian distribution in (C.54), it holds that

$$\big|\mathbb{E}_{\mathbb{P}_{\beta*}}\big[\widetilde{q}_{1,j}(Y, X) - q_{1,j}(Y, X)\big]\big|^2 \le C' \cdot \exp(-R^2 \cdot \log n), \tag{D.56}$$

where $C'$ is a positive absolute constant. Similar argument holds for $q_{2,j}(Y,X)$. Combining (D.54), (D.56), and a similar argument on $q_{2,j}(Y,X)$, we obtain that

$$\left| \mathbb{E}_{\mathbb{P}_0}\left[ q_{1,j}(Y,X) - \widetilde{q}_{1,j}(Y,X) \right] \right| \vee \left| \mathbb{E}_{\mathbb{P}_\beta^*}\left[ q_{1,j}(Y,X) - \widetilde{q}_{1,j}(Y,X) \right] \right| \leq 1/n,$$

$$\left| \mathbb{E}_{\mathbb{P}_0}\left[ q_{2,j}(Y,X) - \widetilde{q}_{2,j}(Y,X) \right] \right| \vee \left| \mathbb{E}_{\mathbb{P}_\beta^*}\left[ q_{2,j}(Y,X) - \widetilde{q}_{2,j}(Y,X) \right] \right| \leq 1/n, \qquad (D.57)$$

which holds for $j \in [d]$, $\beta^* \in \mathcal{H}(s, \gamma_n)$, and sufficiently large $n$ and constant $R$. Note that under the null hypothesis, it holds that $X \sim N(0, I_d)$ and $Y$ is independent of $X$. Therefore, following from (D.52), we obtain

$$\mathbb{E}_{\mathbb{P}_0}\left[ \widetilde{q}_{1,j}(Y,X) \right] = \mathbb{E}_{\mathbb{P}_0}\left[ \widetilde{q}_{2,j}(Y,X) \right] = 0. \qquad (D.58)$$

Meanwhile, under the alternative hypothesis, it follows from (A.3) that

$$\beta_j^{*\,2} - \mathbb{E}_{\mathbb{P}_{\beta^*}}\left[ \widetilde{q}_{1,j}(Y,X) \right]$$

$$\leq \mathbb{E}_{\mathbb{P}_{\beta^*}}\left[ \psi(Y) \cdot (X_j^2 - 1) \cdot \mathbb{1}\left\{ |\psi(Y)| > (R \cdot \log n)^{1/\nu} \right\} \right]$$

$$\leq \sqrt{ \mathbb{E}_{\mathbb{P}_{\beta^*}}\left[ \psi^2(Y) \cdot (X_j^2 - 1)^2 \right] } \cdot \sqrt{ \mathbb{P}_{\beta^*}\left( |\psi(Y)| > (R \cdot \log n)^{1/\nu} \right) }$$

$$\leq \left( \mathbb{E}_{\mathbb{P}_{\beta^*}}\left[ \psi^4(Y) \right] \cdot \mathbb{E}_{\mathbb{P}_{\beta^*}}\left[ (X_j^2 - 1)^4 \right] \right)^{1/4} \cdot \sqrt{ \mathbb{P}_{\beta^*}\left( |\psi(Y)| > (R \cdot \log n)^{1/\nu} \right) }, \qquad (D.59)$$

where we denote by $\beta_j^*$ the $j$-th entry of $\beta^*$, and the above inequalities follow from the Cauchy-Schwartz inequality. Then following from Assumption A.1 and the fact that $X \sim N(0, I_d)$ under the alternative hypothesis, we obtain that

$$\beta_j^{*\,2} - \mathbb{E}_{\mathbb{P}_{\beta^*}}\left[ \widetilde{q}_{1,j}(Y,X) \right] \leq C \cdot \exp(-R/2 \cdot \log n), \qquad (D.60)$$

where $C$ is a positive absolute constant. Note that $\|\beta^*\|_2^2 = s\rho^2$ and $\|\beta^*\|_0 = s$. Therefore, we obtain that

$$\sup_{j \in [d]} |\beta_j^*| \geq \rho. \qquad (D.61)$$

Following from (D.60) and (D.61), if it holds that $s\rho^2/\sigma^2 = \Omega(\sqrt{s^2 \log d/n})$, then for sufficiently large $n$ and constant $R$, we obtain that $\rho^2/4 > 1/n$ and

$$\sup_{j \in [d]} \mathbb{E}_{\mathbb{P}_{\beta^*}}\left[ \widetilde{q}_{1,j}(Y,X) \right] \geq 3\rho^2/4. \qquad (D.62)$$

Similar argument holds for $\widetilde{q}_{2,j}(Y,X)$. Following from (A.4), we obtain that under the alternative hypothesis, it holds that

$$\alpha\beta_j^* - \mathbb{E}_{\mathbb{P}_{\beta^*}}\left[ \widetilde{q}_{2,j}(Y,X) \right] = \mathbb{E}_{\mathbb{P}_{\beta^*}}\left[ \psi(Y) \cdot X_j \cdot \mathbb{1}\left\{ |\psi(Y)| > (R \cdot \log n)^{1/\nu} \right\} \right]. \qquad (D.63)$$

Meanwhile, it follows from the Cauchy-Schwartz inequality that

$$\left| \mathbb{E}_{\mathbb{P}_{\beta^*}}\left[ Y \cdot X_j \cdot \mathbb{1}\left\{ |Y| > (R \cdot \log n)^{1/\nu} \right\} \right] \right|^2 \leq \mathbb{E}_{\mathbb{P}_{\beta^*}}\left[ Y^2 \cdot X_j^2 \right] \cdot \mathbb{P}_{\beta^*}\left( |Y| > (R \cdot \log n)^{1/\nu} \right)$$

$$\leq \sqrt{ \mathbb{E}_{\mathbb{P}_{\beta^*}}\left[ Y^4 \right] \cdot \mathbb{E}_{\mathbb{P}_{\beta^*}}\left[ X_j^4 \right] } \cdot \mathbb{P}_{\beta^*}\left( |Y| > (R \cdot \log n)^{1/\nu} \right)$$

$$\leq C' \cdot \exp(-R \log n), \qquad (D.64)$$

where the last inequality follows from Assumption A.1 and the fact that $X \sim N(0, I_d)$ under the alternative hypothesis. Combining (D.61), (D.63), and (D.64), we obtain that for $s\rho^2/\sigma^2 = \Omega(1/\alpha^2 \cdot s \log d/n)$, it holds for sufficiently large $n$ and constant $R$ that $\alpha\rho/4 > 1/n$ and

$$\sup_{j \in [d]} \left| \mathbb{E}_{\mathbb{P}_{\beta^*}}\left[ \widetilde{q}_{2,j}(Y,X) \right] \right| \geq 3\alpha\rho/4. \qquad (D.65)$$

Combining (D.57), (D.62), and (D.65), we obtain that for sufficiently large $n$ and constant $R$, it holds that

$$\sup_{j \in [d]} \mathbb{E}_{\mathbb{P}_0}\left[ q_{1,j}(Y,X) \right] \leq 1/n, \quad \sup_{j \in [d]} \mathbb{E}_{\mathbb{P}_0}\left[ q_{1,j}(Y,X) \right] \leq 1/n. \qquad (D.66)$$

Moreover, for sufficiently large $n$ and constant $R$, it holds that

$$\sup_{j \in [d]} \mathbb{E}_{\mathbb{P}_{\beta^*}}\left[ q_{1,j}(Y,X) \right] \geq \rho^2/2 \ \text{ if } s\rho^2/\sigma^2 = \Omega(\sqrt{s \log d/n}),$$

$$\sup_{j \in [d]} \mathbb{E}_{\mathbb{P}_{\beta^*}}\left[ q_{2,j}(Y,X) \right] \geq \alpha\rho/2 \ \text{ if } s\rho^2/\sigma^2 = \Omega(1/\alpha^2 \cdot s \log d/n), \qquad (D.67)$$

which concludes the proof of Lemma D.4. $\qquad \square$

[Supplementary Material 2]

# A  Upper bounds

473 In this section, we establish upper bounds that attain the lower bounds obtained in Proposition 3.1 and
474 Theorem A.2 up to logarithmic factors. Based on the lower bounds and upper bounds, we obtain the
475 minimax and computational minimax separation rates defined in Definitions 2.2 and 2.4, respectively.

476 Recall that the hypothesis testing problem in (2.7) takes the form

$$H_0 \colon Y = \epsilon_0 \quad \text{versus} \quad H_1 \colon Y = \begin{cases} f_1(X^\top \beta^*) + \epsilon, & \text{with probability } \alpha, \\ f_2(X^\top \beta^*) + \epsilon, & \text{with probability } 1 - \alpha. \end{cases} \tag{A.1}$$

477 Here $\epsilon$ is a Gaussian noise with variance $\sigma^2$ and $\epsilon_0$ is a noise such that the variances of $Y$ under the
478 null and alternative hypotheses are the same. Besides, $f_1 \in \mathcal{C}_1 \cap \mathcal{C}(\psi)$ and $f_2 \in \mathcal{C}_2 \cap \mathcal{C}(\psi)$ are two
479 unknown link functions, where $\mathcal{C}_1(\psi)$, $\mathcal{C}_2(\psi)$, and $\mathcal{C}(\psi)$ are defined in (2.4) and (2.5). Meanwhile,
480 we set $X \sim N(0, I_d)$ and $\beta^*$ to be $s$-sparse. For the simplicity of the following discussions, we
481 restrict to the set of $\beta^*$ such that $\beta^* = \rho \cdot \mathrm{v}^*$, where $\mathrm{v}^* \in \bar{\mathcal{G}}(s) = \{\mathrm{v} \in \{-1, 0, 1\}^d : \|\mathrm{v}\|_0 = s\}$.
482 We further define

$$\bar{\mathcal{G}}_1(s, \gamma_n) = \big\{ (\beta^*, \sigma) \in \mathbb{R}^{d+1} \colon \beta^* = \rho \cdot \mathrm{v}^*, \mathrm{v}^* \in \bar{\mathcal{G}}(s), \kappa(\beta^*, \sigma) \geq \gamma_n \big\}.$$

483 We highlight the fact that such a restricted parameter set is sufficient to characterize the difficulty of
484 the hypothesis testing problem in (2.7), and defer the proof of the general case to §D.

485 Let $Z = (Y, X)$ and $\mathbb{P}_0$, $\mathbb{P}_{\mathrm{v}^*}$ be the distributions of $Z$ under the null and alternative hypotheses,
486 respectively. We introduce the following assumption on $Y$ and $\psi(Y)$ under the alternative hypothesis,
487 which regulates the tail and moment of $Y$ and $\psi(Y)$.

**Assumption A.1.** 488 We assume that $Y$ and $\psi(Y)$ have bounded fourth moments. We further assume
489 that under the alternative hypothesis, $Y$ and $\psi(Y)$ have desired tail bounds in the form of

$$\mathbb{P}_{\mathrm{v}^*}(|Y| \geq R) \leq C \exp(-R^\nu), \quad \mathbb{P}_{\mathrm{v}^*}(|\psi(Y)| \geq R) \leq C' \exp(-R^\nu), \tag{A.2}$$

490 which holds for a sufficiently large $R$ and positive absolute constants $C$, $C'$, and $\nu$.

491 Assumption A.1 is required only for the upper bounds. It is needed to construct bounded query
492 functions defined in Definition 2.3. Such an assumption is a mild regularity condition in the sense
493 that it holds for the linear regression model and most of the phase retrieval models. For instance, let
494 $(Y, X)$ be generated by the mixed regression model and $\psi(Y) = Y^2$. Then $Y$ follows the mixture of
495 Gaussian distributions. Therefore, $Y$ has bounded fourth moment and Gaussian tail, and $\psi(Y) = Y^2$
496 is sub-exponential under the alternative hypothesis with bounded fourth moment. Hence, the tail
497 bound stated in (A.2) holds for $Y$ and $\psi(Y)$ with $\nu = 1$. Similar arguments hold for the linear
498 regression model and the phase retrieval models $Y = |X^\top \beta^*| + \epsilon$ and $Y = (X^\top \beta^*)^2 + \epsilon$.

499 In what follows, we design the test function $\phi$ based on the first-order and second-order Stein's
500 identities in (2.2) and (2.3). Following from (2.5), it holds that $S_2(Y, \psi) \geq \|\beta^*\|_2^4$ under the
501 alternative hypothesis. It then follows from the second-order Stein's identity in (2.3) that $\mathbb{E}_{\mathbb{P}_{\mathrm{v}^*}}[\psi(Y) \cdot$
502 $(XX^\top - I)] \succeq \beta^* \beta^{*\top}$ under the alternative hypothesis. Meanwhile, under the null hypothesis, $\psi(Y)$
503 is independent of $X$. Therefore, it holds that

$$\mathbb{E}_{\mathbb{P}_{\mathrm{v}^*}}\big[\mathrm{v}^\top \psi(Y) \cdot (XX^\top - I)\mathrm{v}\big] \geq (\mathrm{v}^\top \beta^*)^2, \quad \mathbb{E}_{\mathbb{P}_0}\big[\psi(Y) \cdot (XX^\top - I)\big] = 0. \tag{A.3}$$

504 Meanwhile, following from (2.4), it holds that $\mathbb{E}[Y_1 X] = \beta^*$ with $Y_1 = f_1(X^\top \beta^*, \epsilon)$. Therefore, it
505 follows from the first-order Stein's identity in (2.2) that

$$\mathbb{E}_{\mathbb{P}_{\mathrm{v}^*}}[\mathrm{v}^\top Y X] = \alpha \cdot \mathrm{v}^\top \beta^*, \quad \mathbb{E}_{\mathbb{P}_0}[YX] = 0. \tag{A.4}$$

506 We introduce the following query functions,

$$q_{1,\mathrm{v}}(Y, X) = \psi(Y) \cdot \big[s^{-1}(\mathrm{v}^\top X)^2 - 1\big] \cdot \mathbb{1}\big\{|\psi(Y)| \leq (R \log n)^{1/\nu}\big\} \cdot \mathbb{1}\big\{|\mathrm{v}^\top X| \leq R \cdot \sqrt{s \log n}\big\},$$

$$q_{2,\mathrm{v}}(Y, X) = Y \cdot (s^{-1/2}\mathrm{v}^\top X) \cdot \mathbb{1}\big\{|Y| \leq (R \log n)^{1/\nu}\big\} \cdot \mathbb{1}\big\{|\mathrm{v}^\top X| \leq R \cdot \sqrt{s \log n}\big\}. \tag{A.5}$$

507 We denote by $\bar{Z}_{1,\mathrm{v}}$ and $\bar{Z}_{2,\mathrm{v}}$ the responses of the statistical oracle to query functions $q_{1,\mathrm{v}}$ and $q_{2,\mathrm{v}}$, as
508 defined in Definition 2.3. We define the test functions $\phi_1$ and $\phi_2$ as

$$\phi_1 = \mathbb{1}\Big\{ \sup_{\mathrm{v} \in \bar{\mathcal{G}}(s)} \bar{Z}_{1,\mathrm{v}} \geq \tau_1 \Big\}, \quad \phi_2 = \mathbb{1}\Big\{ \sup_{\mathrm{v} \in \bar{\mathcal{G}}(s)} \bar{Z}_{2,\mathrm{v}} \geq \tau_2 \Big\}, \tag{A.6}$$

509 where we set the thresholds $\tau_1$ and $\tau_2$ to be

$$\tau_1 = CR^{2+1/\nu} \cdot (\log n)^{1+1/\nu} \cdot \sqrt{\frac{s \log d}{n}}, \quad \tau_2 = C'R^{1+1/\nu} \cdot (\log n)^{1/2+1/\nu} \cdot \sqrt{\frac{s \log d}{n}}. \tag{A.7}$$

Here $C$ and $C'$ are absolute constants (which are specified in §B.3). We define the test function as $\phi = \phi_1 \vee \phi_2$. The following theorem characterizes an upper bound for the minimax separation rate by quantifying the SNR for $\phi$ to be asymptotically powerful, which attains the information-theoretic lower bound in Proposition 3.1 up to logarithmic factors.

**Theorem A.2.** We consider the hypothesis testing problem in (A.1) under Assumption A.1. For

$$\gamma_n = \Omega\left((\log n)^{1+1/\nu} \cdot \sqrt{\frac{s \log d}{n}} \bigwedge \frac{(\log n)^{1+2/\nu}}{\alpha^2} \cdot \frac{s \log d}{n}\right), \tag{A.8}$$

it holds that $R_n(\phi; \mathcal{G}_0, \bar{\mathcal{G}}_1) = O(1/d)$. In other words, $\phi$ is asymptotically powerful.

*Proof.* See §B.3 for a detailed proof. $\qquad\square$

It follows from Theorem A.2 that any sequence satisfying (i) of Definition 2.2 is asymptotically upper bounded by any sequence that satisfies (A.8). As a result, it holds that

$$\gamma_n^* = o\left((\log n)^{1+1/\nu} \cdot \sqrt{\frac{s \log d}{n}} \bigwedge \frac{(\log n)^{1+2/\nu}}{\alpha^2} \cdot \frac{s \log d}{n}\right). \tag{A.9}$$

Based on (3.2) and (A.9), up to logarithmic factors, the minimax separation rate defined in Definition 2.2 takes the form

$$\gamma_n^* = \sqrt{\frac{s \log d}{n}} \bigwedge \frac{1}{\alpha^2} \cdot \frac{s \log d}{n}. \tag{A.10}$$

Note that the query functions in (A.5) have exponential oracle complexity, since searching over the parameter set $\bar{\mathcal{G}}(s)$ requires querying the statistical oracle $T = \binom{d}{s} \cdot 2^s$ rounds. To construct a computationally tractable test, we design query functions that access each entry $X_j$ of $X$,

$$q_{1,j}(Y, X) = \psi(Y) \cdot (X_j^2 - 1) \cdot \mathbb{1}\{|\psi(Y)| \le (R \log n)^{1/\nu}\} \cdot \mathbb{1}\{|X_j| \le R\sqrt{\log n}\}, \quad j \in [d]$$

$$q_{2,j}(Y, X) = Y \cdot X_j \cdot \mathbb{1}\{|Y| \le (R \log n)^{1/\nu}\} \cdot \mathbb{1}\{|X_j| \le R\sqrt{\log n}\}, \quad j \in [d]. \tag{A.11}$$

We denote by $\bar{Z}_{1,j}$ and $\bar{Z}_{2,j}$ the responses of the statistical oracle to the query functions $q_{1,j}$ and $q_{2,j}$, as defined in Definition 2.3 . We define the test functions $\widetilde{\phi}_1$ and $\widetilde{\phi}_2$ as

$$\widetilde{\phi}_1 = \mathbb{1}\left\{\sup_{j \in [d]} \bar{Z}_{1,j} \ge \widetilde{\tau}_1\right\}, \quad \widetilde{\phi}_2 = \mathbb{1}\left\{\sup_{j \in [d]} \bar{Z}_{2,j} \ge \widetilde{\tau}_2\right\} \bigvee \mathbb{1}\left\{\inf_{j \in [d]} \bar{Z}_{2,j} \le -\widetilde{\tau}_2\right\}, \tag{A.12}$$

where we set the thresholds $\widetilde{\tau}_1$ and $\widetilde{\tau}_2$ to be

$$\widetilde{\tau}_1 = CR^{2+1/\nu}(\log n)^{1+1/\nu} \cdot \sqrt{\frac{\log d}{n}}, \quad \widetilde{\tau}_2 = C'R^{1+1/\nu}(\log n)^{1/2+1/\nu} \cdot \sqrt{\frac{\log d}{n}}. \tag{A.13}$$

Finally, we define the test function to be $\widetilde{\phi} = \widetilde{\phi}_1 \vee \widetilde{\phi}_2$. By the definition of $\phi_1$ and $\phi_2$ in (A.12), the test function $\widetilde{\phi}$ is computationally tractable with query complexity $T = 2d$. The following theorem characterizes an upper bound for the computational minimax separation rate, which attains the computational lower bound in Theorem 3.2 up to logarithmic factors.

**Theorem A.3.** We consider the hypothesis testing problem in (A.1) under Assumption A.1. For

$$\gamma_n = \Omega\left((\log n)^{1+1/\nu} \cdot \sqrt{\frac{s^2 \log d}{n}} \bigwedge \frac{(\log n)^{1+2/\nu}}{\alpha^2} \cdot \frac{s \log d}{n}\right), \tag{A.14}$$

it holds that $\bar{R}_n(\widetilde{\phi}; \mathcal{G}_0, \bar{\mathcal{G}}_1) = O(1/d)$. In other words, $\widetilde{\phi}$ is asymptotically powerful.

*Proof.* See §B.4 for a detailed proof. $\qquad\square$

It follows from Theorem A.3 that any sequence satisfying (i) of Definition 2.4 is asymptotically upper bounded by any sequence that satisfies (A.14). As a result, it holds that

$$\bar{\gamma}_n^* = o\left((\log n)^{1+1/\nu} \cdot \sqrt{\frac{s^2 \log d}{n}} \bigwedge \frac{(\log n)^{1+2/\nu}}{\alpha^2} \cdot \frac{s \log d}{n}\right). \tag{A.15}$$

Based on (3.5) and (A.15), up to logarithmic factors, the computational minimax separation rate defined in Definition 2.4 takes the form

$$\bar{\gamma}_n^* = \sqrt{\frac{s^2}{n}} \bigwedge \frac{1}{\alpha^2} \cdot \frac{s \log d}{n}. \tag{A.16}$$

# B  Proof of Main Results

539  In this section, we lay out the proofs of the main results in §3 and §A.

540  **B.1  Proof of Proposition 3.1**

541  *Proof.* We have the following lower bound of minimax risk,

$$R_n^*(\mathcal{G}_0, \mathcal{G}_1) = \inf_\phi \sup_{f_1, f_2, \psi} R_n(\phi; \mathcal{G}_0, \mathcal{G}_1) \geq \inf_\phi R_n(\phi; \mathcal{G}_0, \mathcal{G}_1)$$

$$= \inf_\phi \Big\{ \sup_{\theta^* \in \mathcal{G}_0} \mathbb{P}_{\theta^*}(\phi = 1) + \sup_{\theta^* \in \mathcal{G}_1} \mathbb{P}_{\theta^*}(\phi = 0) \Big\}.$$

542  where the first inequality is obtained by restricting $f_1$, $f_2$, and $\psi$ in the testing problem in (2.7)
543  as follows. We set $\psi(y) = y^2$ and the sample $\{z_i\}_{i \in [n]}$ to be generated from a mixture of the
544  linear regression model $Y_1 = f_1(X^\top \beta^*) + \epsilon = X^\top \beta^* + \epsilon$ and the mixed regression model
545  $Y_2 = f_2(X^\top \beta^*) + \epsilon = \eta \cdot X^\top \beta^* + \epsilon$. Here we set $\epsilon \sim N(0, \sigma^2)$ and $\eta$ to be a Rademacher
546  random variable, which is independent of both $X$ and $\epsilon$. Since $S_1(Y_1) = \|\beta^*\|_2^2$, $S_1(Y_2) = 0$, and
547  $S_2(Y_1, \psi) = S_2(Y_2, \psi) = 2\|\beta^*\|_2^4$, we have $f_1 \in \mathcal{C}_1 \cap \mathcal{C}(\psi)$ and $f_2 \in \mathcal{C}_2 \cap \mathcal{C}(\psi)$, where $\mathcal{C}_1$, $\mathcal{C}_2$, and
548  $\mathcal{C}(\psi)$ are defined in (2.4) and (2.5).

549  We further restrict the parameter space of $\theta^* = (\beta^*, \sigma)$ as follows. Let $\beta^* \in \{\beta = \rho \cdot v \colon v \in \mathcal{G}(s)\}$,
550  where $\rho$ is a positive constant and $\mathcal{G}(s) = \{v \in \{0,1\}^d \colon \|v\|_0 = s\}$. Therefore, the original
551  hypothesis testing problem is reduced to

$$H_0 \colon Y = \epsilon_0 \quad \text{versus} \quad H_1 \colon Y = \begin{cases} X^\top \beta^* + \epsilon, & \text{with probability } \alpha, \\ \eta \cdot X^\top \beta^* + \epsilon, & \text{with probability } 1 - \alpha, \end{cases} \tag{B.1}$$

552  where under $H_0$ we have $\epsilon_0 \sim N(0, \sigma^2 + s\rho^2)$ and under $H_1$ we have $\epsilon \sim N(0, \sigma^2)$. We denote by
553  $\mathbb{P}_0$ and $\mathbb{P}_{v^*}$ the probability distributions of $Z = (Y, X)$ under the null and alternative hypotheses
554  with $\beta^* = \rho \cdot v^*$, respectively. In addition, we define $\overline{\mathbb{P}} = |\mathcal{G}(s)|^{-1} \sum_{v \in \mathcal{G}(s)} \mathbb{P}_v^n$, where we use the
555  superscript $n$ to denote the $n$-fold product probability measure. By Neyman-Pearson lemma, we have

$$R_n^*(\mathcal{G}_0, \mathcal{G}_1) \geq \inf_\phi \big[ \mathbb{P}_0^n(\phi = 1) + \overline{\mathbb{P}}(\phi = 0) \big] = 1 - 1/2 \cdot \mathbb{E}_{\mathbb{P}_0^n} \big[ |d\overline{\mathbb{P}}/d\mathbb{P}_0^n - 1| \big]$$

$$\geq 1 - 1/2 \cdot \Big( \big( \mathbb{E}_{\mathbb{P}_0^n} \big[ d\overline{\mathbb{P}}/d\mathbb{P}_0^n \big] \big)^2 - 1 \Big)^{1/2}, \tag{B.2}$$

556  where the second inequality follows from the Cauchy-Schwarz inequality. In what follows, we show
557  that $\mathbb{E}_{\mathbb{P}_0^n}[d\overline{\mathbb{P}}/d\mathbb{P}_0^n]^2 = 1 + o(1)$ under the condition in (3.1), which implies $\liminf_{n \to \infty} R_n^*(\mathcal{G}_0, \mathcal{G}_1) \geq$
558  $1 - o(1)$ by (B.2). Note that on the right-hand side of (B.2), we have

$$\big( \mathbb{E}_{\mathbb{P}_0^n} \big[ d\overline{\mathbb{P}}/d\mathbb{P}_0^n \big] \big)^2 = \frac{1}{|\mathcal{G}(s)|^2} \sum_{v, v' \in \mathcal{G}(s)} \mathbb{E}_{\mathbb{P}_0^n} \Big[ \frac{d\mathbb{P}_v^n}{d\mathbb{P}_0^n} \frac{d\mathbb{P}_{v'}^n}{d\mathbb{P}_0^n} (Z_1, \ldots, Z_n) \Big], \tag{B.3}$$

559  where $Z_i$ are independent copies of $Z = (Y, X)$. The following lemma establishes an upper bound
560  of the right-hand side of (B.3).

561  **Lemma B.1.** For any $v_1, v_2 \in \mathcal{G}(s)$, if $s\rho^2 = o(1)$, it holds that

$$\mathbb{E}_{\mathbb{P}_0} \Big[ \frac{d\mathbb{P}_{v_1}}{d\mathbb{P}_0} \frac{d\mathbb{P}_{v_2}}{d\mathbb{P}_0} (Z) \Big] \leq \cosh \Big( \frac{2\rho^2 \langle v_1, v_2 \rangle}{\sigma^2 + s\beta^2} \Big) + \alpha^2 \sinh \Big( \frac{2\rho^2 \langle v_1, v_2 \rangle}{\sigma^2 + s\rho^2} \Big). \tag{B.4}$$

562  *Proof.* See §C.1 for a detailed proof. □

563  Following from Lemma B.1, it holds that

$$\mathbb{E}_{\mathbb{P}_0} \Big[ \frac{d\mathbb{P}_{v_1}^n}{d\mathbb{P}_0^n} \frac{d\mathbb{P}_{v_2}^n}{d\mathbb{P}_0^n} (Z_1, \ldots, Z_n) \Big] = \Big( \mathbb{E}_{\mathbb{P}_0} \Big[ \frac{d\mathbb{P}_{v_1}}{d\mathbb{P}_0} \frac{d\mathbb{P}_{v_2}}{d\mathbb{P}_0} (Z) \Big] \Big)^n$$

$$\leq \Big[ \cosh \Big( \frac{2\rho^2 \langle v_1, v_2 \rangle}{\sigma^2 + s\rho^2} \Big) + \alpha^2 \sinh \Big( \frac{2\rho^2 \langle v_1, v_2 \rangle}{\sigma^2 + s\rho^2} \Big) \Big]^n, \tag{B.5}$$

564  where $Z_i$ are independent copies of $Z = (Y, X)$. The following lemma by [62] establishes an upper
565  bound of the right-hand side in (B.5).

**Lemma B.2** ([62]). For any $x \geq 0$ and $0 \leq k \leq 1$, we have,
$$\cosh(x) + k\sinh(x) \leq \exp(2kx) \vee \cosh(2x).$$

*Proof.* See the appendix of [62] for a detailed proof. $\qquad\square$

Following from (B.3), (B.5), and Lemma B.2, we conclude
$$\left(\mathbb{E}_{\mathbb{P}_0^n}\left[\mathrm{d}\bar{\mathbb{P}}/\mathrm{d}\mathbb{P}_0^n\right]\right)^2 \leq \frac{1}{|\mathcal{G}(s)|^2} \sum_{\mathrm{v}_1,\mathrm{v}_2 \in \mathcal{G}(s)} \left[\exp\left(\frac{4\alpha^2\rho^2\langle \mathrm{v}_1, \mathrm{v}_2\rangle}{\sigma^2 + s\rho^2}\right) \vee \cosh\left(\frac{4\rho^2\langle \mathrm{v}_1, \mathrm{v}_2\rangle}{\sigma^2 + s\rho^2}\right)\right]^n. \quad \text{(B.6)}$$

The following lemma shows that the right-hand side of (B.6) is of order $1 + o(1)$.

**Lemma B.3** ([62]). For
$$\gamma_n = o\left(\sqrt{\frac{s\log d}{n}} \bigwedge \frac{1}{\alpha^2} \cdot \frac{s\log d}{n}\right),$$

if $s = o(d^{1/2-\delta})$ for some absolute constant $\delta > 0$, it then holds that
$$\frac{1}{|\mathcal{G}(s)|^2} \sum_{\mathrm{v}_1,\mathrm{v}_2 \in \mathcal{G}(s)} \left[\exp\left(\frac{4\alpha^2\rho^2\langle \mathrm{v}_1, \mathrm{v}_2\rangle}{\sigma^2 + s\rho^2}\right) \bigvee \cosh\left(\frac{4\rho^2\langle \mathrm{v}_1, \mathrm{v}_2\rangle}{\sigma^2 + s\rho^2}\right)\right]^n = 1 + o(1). \quad \text{(B.7)}$$

*Proof.* See §C.2 for a detailed proof. $\qquad\square$

Combining Lemma B.3 and (B.6), we conclude that for $\gamma_n = o(\sqrt{s\log d/n} \wedge 1/\alpha^2 \cdot s\log d/n)$, it holds that $\left(\mathbb{E}_{\mathbb{P}_0^n}\left[\mathrm{d}\bar{\mathbb{P}}/\mathrm{d}\mathbb{P}_0^n\right]\right)^2 - 1 = o(1)$. Then following from (B.2), we have $\liminf_{n\to\infty} R_n^*(\mathcal{G}_0, \mathcal{G}_1) \geq 1$, which concludes the proof of Proposition 3.1. $\qquad\square$

## B.2 Proof of Theorem 3.2

*Proof.* It follows from Definition 2.2 that for $\gamma_n = o(\gamma_n^*)$, any hypothesis testing problem in (2.7) is asymptotically powerless. It remains to show that for $\gamma_n = o(\sqrt{s^2/n} \wedge 1/\alpha^2 \cdot s/n)$, any computationally tractable test is asymptotically powerless. First, we restrict the original estimation problem to the following hypothesis testing problem,
$$H_0: Y = \epsilon \quad \text{versus} \quad H_1: Y = \begin{cases} X^\top\beta^* + \epsilon, & \text{with probability } \alpha \\ \eta \cdot X^\top\beta^* + \epsilon, & \text{with probability } 1 - \alpha \end{cases}. \quad \text{(B.8)}$$

In (B.8), we restrict $\beta^*$ to the set $\beta^* \in \{\rho \cdot \mathrm{v} : \mathrm{v} \in \mathcal{G}(s)\}$ with $\mathcal{G}(s) = \{\mathrm{v} \in \{0,1\}^d : \|\mathrm{v}\|_0 = s\}$. We set $\epsilon \sim N(0, \sigma^2 + s\rho^2)$ under $H_0$ and $\epsilon \sim N(0, \sigma^2)$ under $H_1$ so that straightforward tests based on mean and variance are not able to detect the existence of a nonzero parameter $\beta^*$.

By restricting the parameter space, we obtain a lower bound for the minimax risk. Recall that we denote by $\bar{\mathbb{P}}_0$ and $\bar{\mathbb{P}}_\mathrm{v}$ the distributions of $Z_q$, which denotes the response of the oracle to the query $q$ when the true distributions of the data are $\mathbb{P}_0$ and $\mathbb{P}_\mathrm{v}$, correspondingly. We have
$$\bar{R}_n^*[\mathcal{G}_0, \mathcal{G}_1; \mathscr{A}, r] \geq \inf_{\phi \in \mathcal{H}(\mathscr{A}, r)} \left\{\bar{\mathbb{P}}_0(\phi = 1) + \sup_{\mathrm{v} \in \mathcal{G}(s)} \bar{\mathbb{P}}_\mathrm{v}(\phi = 0)\right\}. \quad \text{(B.9)}$$

To show that any computationally tractable test is asymptotically powerless, it suffices to show that the right-hand side of (B.9) is asymptotically lower bounded by one. By Theorem 4.2 of [53], we know that this holds true if
$$T \cdot \sup_{q \in \mathcal{Q}} |\mathcal{C}(q)|/|\mathcal{G}(s)| = o(1),$$

where $\mathcal{C}(q)$ is defined as
$$\mathcal{C}(q) = \left\{\mathrm{v} \in \mathcal{G}(s) : \left|\mathbb{E}_{\mathbb{P}_\mathrm{v}}\left[q(Z)\right] - \mathbb{E}_{\mathbb{P}_0}\left[q(Z)\right]\right| > \tau_q\right\}.$$

Here $\tau_q$ is the tolerance parameter defined in Definition 2.3, with $(Y, X)$ following $\mathbb{P}_\mathrm{v}$. The following lemma shows that $T \cdot \sup_{q \in \mathcal{Q}} |\mathcal{C}(q)|/|\mathcal{G}(s)| = o(1)$ if $\gamma_n$ is sufficiently small.

**Lemma B.4** ([53]). For $s = o(d^{1/2-\delta})$, $T = O(d^\mu)$, and
$$\gamma_n = o\left(\frac{s^2}{n} \bigwedge \frac{1}{\alpha^2} \cdot \frac{s}{n}\right),$$

it holds that

$$T \cdot \sup_{q \in \mathcal{Q}} |\mathcal{C}(q)|/|\mathcal{G}(s)| = o(1). \tag{B.10}$$

*Proof.* See §C.3 for a detailed proof. □

By combining Theorem 4.2 of [53] and Lemma B.4, we conclude that the right-hand side of (B.9) is asymptotically lower bounded by one. Therefore, it holds that $\liminf_{n \to \infty} \bar{R}_n^*[\mathcal{G}_0, \mathcal{G}_1; \mathscr{A}, r] \geq 1$, which concludes the proof of Theorem 3.2. □

## B.3 Proof of Theorem A.2

*Proof.* Recall that we denote by $Z = (Y, X)$ and $\mathbb{P}_0$, $\mathbb{P}_{\mathrm{v}^*}$ the distributions of $Z$ under the null and alternative hypotheses with $\beta^* = \rho \cdot \mathrm{v}^*$, respectively. For the hypothesis testing problem in (A.1), the following lemma characterizes the expectations of the query functions defined in (A.5).

**Lemma B.5.** For any $\mathrm{v}, \mathrm{v}^* \in \bar{\mathcal{G}}(s)$ and

$$\gamma_n = \Omega\bigg( (\log n)^{1+1/\nu} \cdot \sqrt{\frac{s \log d}{n}} \bigwedge \frac{(\log n)^{1+2/\nu}}{\alpha^2} \cdot \frac{s \log d}{n} \bigg),$$

it holds that

$$\mathbb{E}_{\mathbb{P}_0}\big[q_{1,\mathrm{v}}(Y, X)\big] \leq 1/n, \quad \mathbb{E}_{\mathbb{P}_0}\big[q_{2,\mathrm{v}}(Y, X)\big] \leq 1/n. \tag{B.11}$$

In addition, it holds that

$$\mathbb{E}_{\mathbb{P}_{\mathrm{v}^*}}\big[q_{1,\mathrm{v}^*}(Y, X)\big] \geq s\rho^2/2 \ \text{ if } \ \gamma_n = \Omega\bigg( (\log n)^{1+1/\nu} \cdot \sqrt{\frac{s \log d}{n}} \bigg),$$

$$\mathbb{E}_{\mathbb{P}_{\mathrm{v}^*}}\big[q_{2,\mathrm{v}^*}(Y, X)\big] \geq \sqrt{\alpha^2 s\rho^2}/2 \ \text{ if } \ \gamma_n = \Omega\bigg( \frac{(\log n)^{1+2/\nu}}{\alpha^2} \cdot \frac{s \log d}{n} \bigg). \tag{B.12}$$

*Proof.* See §C.4 for a detailed proof. □

In what follows, we establish an upper bound of the risk of $\phi = \phi_1 \vee \phi_2$. Recall that we define the test functions $\phi_1$ and $\phi_2$ in (A.6) with parameters

$$\tau_1 = CR^{2+1/\nu} \cdot (\log n)^{1+1/\nu} \cdot \sqrt{\frac{s \log d}{n}}, \quad \tau_2 = C'R^{1+1/\nu} \cdot (\log n)^{1/2+1/\nu} \cdot \sqrt{\frac{s \log d}{n}}. \tag{B.13}$$

where $C$ and $C'$ are absolute constants. Note that the total number of query functions $\{q_{1,\mathrm{v}}\}_{\mathrm{v} \in \mathcal{G}(s)}$ and $\{q_{2,\mathrm{v}}\}_{\mathrm{v} \in \mathcal{G}(s)}$ is $|\mathcal{Q}_\phi| = 2 \cdot \binom{d}{s} \cdot 2^s$. Therefore, following from (2.12) with $\xi = 1/d$, for sufficiently large $d$ and $n$, it holds that

$$\tau_{q_{1,\mathrm{v}}} \leq C_0 R^{2+1/\nu} (\log n)^{1/2+1/\nu} \cdot \sqrt{\frac{s \log d}{n}}, \quad \tau_{q_{2,\mathrm{v}}} \leq C_1 R^{1+1/\nu} (\log n)^{1/2+1/\nu} \cdot \sqrt{\frac{s \log d}{n}}, \tag{B.14}$$

where $\tau_{q_{1,\mathrm{v}}}$ and $\tau_{q_{2,\mathrm{v}}}$ are the tolerance parameters of $q_{1,\mathrm{v}}$ and $q_{2,\mathrm{v}}$ defined in Definition 2.3, and $C_0$, $C_1$ are positive absolute constants. We fix $C$ and $C'$ in (B.13) such that $\tau_1 \geq \tau_{q_{1,\mathrm{v}}} + 1/n$ and $\tau_2 \geq \tau_{q_{2,\mathrm{v}}} + 1/n$. Recall that we denote by $\bar{Z}_{1,\mathrm{v}}$ and $\bar{Z}_{2,\mathrm{v}}$ the responses of the statistical oracle to the query functions $q_{1,\mathrm{v}}$ and $q_{2,\mathrm{v}}$. Further recall that we denote by $\bar{\mathbb{P}}_0$ and $\bar{\mathbb{P}}_{\mathrm{v}^*}$ the distributions of response of the statistical oracle to the query functions when the true distribution of the data is $\mathbb{P}_0$ and $\mathbb{P}_{\mathrm{v}^*}$. Following from Lemma B.5, it holds for any $\mathrm{v} \in \mathcal{G}(s)$ and $i \in \{1, 2\}$ that

$$\bar{\mathbb{P}}_0\big(\bar{Z}_{i,\mathrm{v}} \geq \tau_i\big) \leq \bar{\mathbb{P}}_0\Big(\big|\bar{Z}_{i,\mathrm{v}} - \mathbb{E}_{\mathbb{P}_0}\big[q_{i,\mathrm{v}}(Y, X)\big]\big| \geq \tau_{q_{i,\mathrm{v}}}\Big).$$

Based on (2.11) with $\xi = 1/d$, it holds for $i \in \{1, 2\}$ that

$$\bar{\mathbb{P}}_0(\phi_i = 1) = \bar{\mathbb{P}}_0\bigg( \sup_{\mathrm{v} \in \mathcal{G}(s)} \bar{Z}_{i,\mathrm{v}} > \tau_i \bigg)$$

$$\leq \bar{\mathbb{P}}_0\bigg( \bigcup_{\mathrm{v} \in \mathcal{G}(s)} \Big\{ \big|\bar{Z}_{i,\mathrm{v}} - \mathbb{E}_{\mathbb{P}_0}\big[q_{i,\mathrm{v}}(Y, X)\big]\big| > \tau_{q_{i,\mathrm{v}}} \Big\} \bigg) \leq 2/d. \tag{B.15}$$

Recall that we define $\phi = \phi_1 \vee \phi_2$. Therefore, we obtain from (B.15) that

$$\bar{\mathbb{P}}_0(\phi = 1) \leq \bar{\mathbb{P}}_0(\phi_1 = 1) + \bar{\mathbb{P}}_0(\phi_2 = 1) = 4/d. \tag{B.16}$$

In other words, the type-I error of $\phi$ is upper bounded by $4/d$. It remains to upper bound the type-II error of $\phi$. Following from the lower bound of SNR in (A.8), it holds that either $s\rho^2/4 \geq \tau_1$ or $\sqrt{\alpha^2 s\rho^2}/4 \geq \tau_2$ for a sufficiently large $n$. Following from Lemma B.5, if $s\rho^2/4 \geq \tau_1$, it holds that

$$\bar{\mathbb{P}}_{\mathrm{v}^*}\big(\bar{Z}_{1,\mathrm{v}^*} \leq \tau_1\big) \leq \bar{\mathbb{P}}_{\mathrm{v}^*}\big(\bar{Z}_{1,\mathrm{v}^*} \leq \mathbb{E}_{\mathbb{P}_{\mathrm{v}^*}}\big[q_{1,\mathrm{v}^*}(Y,X)\big] - \tau_1\big)$$
$$\leq \bar{\mathbb{P}}_{\mathrm{v}^*}\Big(\big|\bar{Z}_{1,\mathrm{v}^*} - \mathbb{E}_{\mathbb{P}_{\mathrm{v}^*}}\big[q_{1,\mathrm{v}^*}(Y,X)\big]\big| \geq \tau_{q_{1,\mathrm{v}^*}}\Big), \tag{B.17}$$

where the last inequality holds since $\tau_1 > \tau_{q_{1,\mathrm{v}^*}}$. Therefore, it follows from (2.11) with $\xi = 1/d$ that

$$\bar{\mathbb{P}}_{\mathrm{v}^*}(\phi_1 = 0) = \bar{\mathbb{P}}_{\mathrm{v}^*}\Big(\sup_{\mathrm{v} \in \mathcal{G}(s)} \bar{Z}_{1,\mathrm{v}} < \tau_1\Big) \leq \bar{\mathbb{P}}_{\mathrm{v}^*}\big(\bar{Z}_{1,\mathrm{v}^*} < \tau_1\big)$$
$$\leq \bar{\mathbb{P}}_{\mathrm{v}^*}\Big(\big|\bar{Z}_{1,\mathrm{v}^*} - \mathbb{E}_{\mathbb{P}_{\mathrm{v}^*}}\big[q_{1,\mathrm{v}^*}(Y,X)\big]\big| > \tau_{q_{1,\mathrm{v}^*}}\Big) \leq 2/d. \tag{B.18}$$

Similarly, following from Lemma B.5, if $\sqrt{\alpha^2 s\rho^2}/4 \geq \tau_2$, it holds that,

$$\bar{\mathbb{P}}_{\mathrm{v}^*}(\phi_2 = 0) = \bar{\mathbb{P}}_{\mathrm{v}^*}\Big(\sup_{\mathrm{v} \in \mathcal{G}(s)} \bar{Z}_{2,\mathrm{v}} < \tau_2\Big) \leq \bar{\mathbb{P}}_{\mathrm{v}^*}\big(\bar{Z}_{2,\mathrm{v}^*} < \tau_2\big)$$
$$\leq \bar{\mathbb{P}}_{\mathrm{v}^*}\Big(\big|\bar{Z}_{2,\mathrm{v}^*} - \mathbb{E}_{\mathbb{P}_{\mathrm{v}^*}}\big[q_{2,\mathrm{v}^*}(Y,X)\big]\big| > \tau_{q_{2,\mathrm{v}^*}}\Big) \leq 2/d, \tag{B.19}$$

where the last inequality holds since $\tau_2 > \tau_{q_{2,\mathrm{v}^*}}$. Note that (B.18) and (B.19) holds for any $(\beta^*, \sigma) \in \bar{\mathcal{G}}_1(s, \gamma_n)$ if (A.8) holds. Therefore, by combining (B.18) and (B.19), we have

$$\sup_{(\beta^*, \sigma) \in \bar{\mathcal{G}}_1(s, \gamma_n)} \bar{\mathbb{P}}_{\mathrm{v}^*}(\phi = 0) \leq \sup_{(\beta^*, \sigma) \in \bar{\mathcal{G}}_1(s, \gamma_n)} \big\{\bar{\mathbb{P}}_{\mathrm{v}^*}(\phi_1 = 0) \wedge \bar{\mathbb{P}}_{\mathrm{v}^*}(\phi_2 = 0)\big\} \leq 2/d. \tag{B.20}$$

In other words, the type-II error of $\phi$ is upper bounded by $2/d$. By combining (B.16) and (B.20), we conclude that if (A.8) holds, the risk for $\phi$ is of order $O(1/d)$, which completes the proof of Theorem A.2. $\qquad\square$

## B.4 Proof of Theorem A.3

*Proof.* The proof is similar to that of Theorem A.2 in §B.3. Recall that we denote by $Z = (Y, X)$ and $\mathbb{P}_0, \mathbb{P}_{\mathrm{v}^*}$ the distributions of $Z$ under the null and alternative hypotheses with $\beta^* = \rho \cdot \mathrm{v}^*$, respectively. The following lemma characterizes the expectations of the query functions defined in (A.11).

**Lemma B.6.** For any $\mathrm{v}^* \in \bar{\mathcal{G}}(s)$ and

$$\gamma_n = \Omega\bigg((\log n)^{1+1/\nu} \cdot \sqrt{\frac{s^2 \log d}{n}} \bigwedge \frac{(\log n)^{1+2/\nu}}{\alpha^2} \cdot \frac{s \log d}{n}\bigg),$$

it holds that

$$\sup_{j \in [d]} \mathbb{E}_{\mathbb{P}_0}\big[q_{1,j}(Y,X)\big] \leq 1/n, \quad \sup_{j \in [d]} \mathbb{E}_{\mathbb{P}_0}\big[q_{2,j}(Y,X)\big] \leq 1/n. \tag{B.21}$$

In addition, it holds that

$$\sup_{j \in [d]} \mathbb{E}_{\mathbb{P}_{\mathrm{v}^*}}\big[q_{1,j}(Y,X)\big] \geq \rho^2/2 \ \text{if} \ \gamma_n = \Omega\bigg((\log n)^{1+1/\nu} \cdot \sqrt{\frac{s^2 \log d}{n}}\bigg),$$

$$\sup_{j \in [d]} \big|\mathbb{E}_{\mathbb{P}_{\mathrm{v}^*}}\big[q_{2,j}(Y,X)\big]\big| \geq \alpha\rho/2 \ \text{if} \ \gamma_n = \Omega\bigg(\frac{(\log n)^{1+2/\nu}}{\alpha^2} \cdot \frac{s \log d}{n}\bigg). \tag{B.22}$$

*Proof.* See §C.5 for a detailed proof. $\qquad\square$

In what follows, we upper bound the risk of the test function $\widetilde{\phi} = \widetilde{\phi}_1 \vee \widetilde{\phi}_2$. Recall that we define the test functions $\widetilde{\phi}_1$ and $\widetilde{\phi}_2$ in (A.11) with parameters

$$\widetilde{\tau}_1 = C R^{2+1/\nu} \cdot (\log n)^{1+1/\nu} \cdot \sqrt{\frac{\log d}{n}}, \quad \widetilde{\tau}_2 = C' R^{1+1/\nu} \cdot (\log n)^{1/2+1/\nu} \cdot \sqrt{\frac{\log d}{n}}, \tag{B.23}$$

where $C, C'$ are absolute constants. Note that the total number of query functions $\{q_{1,j}\}_{j \in [d]}$ and $\{q_{2,j}\}_{j \in [d]}$ is $|\mathcal{Q}_{\widetilde{\phi}}| = 2d$. Therefore, following from Definition 2.3 with $\xi = 1/d$, for sufficiently

large $d$ and $n$, the tolerance parameters of $q_{1,j}$ and $q_{2,j}$ are upper bounded as follows,

$$\tau_{q_{1,j}} \leq C_0' R^{2+1/\nu} (\log n)^{1/2+1/\nu} \cdot \sqrt{\frac{\log d}{n}}, \quad \tau_{q_{2,j}} \leq C_1' R^{1+1/\nu} (\log n)^{1/2+1/\nu} \cdot \sqrt{\frac{\log d}{n}},$$
(B.24)

where $C_0'$ and $C_1'$ are positive absolute constants. We fix $C$ and $C'$ in (B.13) such that $\widetilde{\tau}_1 \geq \tau_{q_{1,j}} + 1/n$ and $\widetilde{\tau}_2 \geq \tau_{q_{2,j}} + 1/n$. Recall that we denote by $\bar{Z}_{1,j}$ and $\bar{Z}_{2,j}$ the responses of the statistical oracle to the query functions $q_{1,j}$ and $q_{2,j}$, respectively. Further recall that we denote by $\bar{\mathbb{P}}_0$ and $\bar{\mathbb{P}}_{\mathrm{v}^*}$ the distributions of response of the statistical oracle to the query functions when the true distribution of the data is $\mathbb{P}_0$ and $\mathbb{P}_{\mathrm{v}^*}$. Following from Lemma B.6, for any $j \in [d]$ and $i \in \{1,2\}$, it holds that

$$\bar{\mathbb{P}}_0 \big( \bar{Z}_{i,j} \geq \widetilde{\tau}_1 \big) \leq \bar{\mathbb{P}}_0 \Big( \big| \bar{Z}_{i,j} - \mathbb{E}_{\mathbb{P}_0}[q_{i,j}(Y,X)] \big| \geq \tau_{q_{i,j}} \Big).$$

Based on (2.11) with $\xi = 1/d$, it holds for $i \in \{1,2\}$ that

$$\bar{\mathbb{P}}_0(\widetilde{\phi}_i = 1) = \bar{\mathbb{P}}_0 \bigg( \sup_{j \in [d]} \bar{Z}_{i,j} > \widetilde{\tau}_i \bigg)$$

$$\leq \bar{\mathbb{P}}_0 \bigg( \bigcup_{j \in [d]} \Big\{ \big| \bar{Z}_{i,j} - \mathbb{E}_{\mathbb{P}_0}[q_{i,j}(Y,X)] \big| > \tau_{q_{i,j}} \Big\} \bigg) \leq 2/d, \tag{B.25}$$

Recall that we define $\widetilde{\phi} = \widetilde{\phi}_1 \vee \widetilde{\phi}_2$. Therefore, we obtain from (B.25) that

$$\bar{\mathbb{P}}_0(\widetilde{\phi} = 1) \leq \bar{\mathbb{P}}_0(\widetilde{\phi}_1 = 1) + \bar{\mathbb{P}}_0(\widetilde{\phi}_2 = 1) = 4/d. \tag{B.26}$$

In other words, the type-I error of $\widetilde{\phi}$ is upper bounded by $4/d$. It remains to upper bound the type-II error of $\phi$. Following from the lower bound on SNR in (A.14), it holds that either $\rho^2/4 \geq \widetilde{\tau}_1$ or $\alpha\rho/4 \geq \widetilde{\tau}_2$ with a sufficiently large $n$. For any $\mathrm{v}^* \in \bar{\mathcal{G}}(s)$, let $j^* = \operatorname{argmax}_{j \in [d]} \mathbb{E}_{\mathbb{P}_{\mathrm{v}^*}}[q_{1,j}(Y,X)]$. Following from Lemma B.5, if $\rho^2/4 \geq \widetilde{\tau}_1$, it holds that

$$\bar{\mathbb{P}}_{\mathrm{v}^*} \big( \bar{Z}_{1,j^*} \leq \widetilde{\tau}_1 \big) \leq \bar{\mathbb{P}}_{\mathrm{v}^*} \Big( \bar{Z}_{1,j^*} \leq \mathbb{E}_{\mathbb{P}_{\mathrm{v}^*}}[q_{1,j^*}(Y,X)] - \widetilde{\tau}_1 \Big)$$

$$\leq \bar{\mathbb{P}}_{\mathrm{v}^*} \Big( \big| \bar{Z}_{1,j^*} - \mathbb{E}_{\mathbb{P}_{\mathrm{v}^*}}[q_{1,j^*}(Y,X)] \big| \geq \tau_{q_{1,j^*}} \Big), \tag{B.27}$$

where the last inequality holds since $\widetilde{\tau}_1 > \tau_{q_{1,j^*}}$. Therefore, we conclude from (2.11) with $\xi = 1/d$ that

$$\bar{\mathbb{P}}_{\mathrm{v}^*}(\widetilde{\phi}_1 = 0) = \bar{\mathbb{P}}_{\mathrm{v}^*} \bigg( \sup_{j \in [d]} \bar{Z}_{1,j} < \widetilde{\tau}_1 \bigg) \leq \bar{\mathbb{P}}_{\mathrm{v}^*} \big( \bar{Z}_{1,j^*} < \widetilde{\tau}_1 \big)$$

$$\leq \bar{\mathbb{P}}_{\mathrm{v}^*} \Big( \big| \bar{Z}_{1,j^*} - \mathbb{E}_{\mathbb{P}_{\mathrm{v}^*}}[q_{1,j^*}(Y,X)] \big| > \tau_{q_{1,j^*}} \Big) \leq 2/d. \tag{B.28}$$

Similarly, for any $\mathrm{v}^* \in \bar{\mathcal{G}}(s)$, let $k^* = \operatorname{argmax}_{j \in [d]} \mathbb{E}_{\mathbb{P}_{\mathrm{v}^*}}[q_{2,j}(Y,X)]$ and $\ell^* = \operatorname{argmin}_{j \in [d]} \mathbb{E}_{\mathbb{P}_{\mathrm{v}^*}}[q_{2,j}(Y,X)]$. Following from Lemma B.5, if $\alpha\rho/4 \geq \widetilde{\tau}_2$, it holds that either $\mathbb{E}[q_{2,k^*}(Y,X)] \geq \alpha\rho/2$ or $\mathbb{E}[q_{2,\ell^*}(Y,X)] \leq -\alpha\rho/2$. If it holds that $\mathbb{E}_{\mathbb{P}_{\mathrm{v}^*}}[q_{2,k^*}(Y,X)] \geq \alpha\rho/2 \geq 2\widetilde{\tau}_2$, we have

$$\bar{\mathbb{P}}_{\mathrm{v}^*}(\widetilde{\phi}_2 = 0) \leq \bar{\mathbb{P}}_{\mathrm{v}^*} \bigg( \sup_{j \in [d]} \bar{Z}_{2,j} < \widetilde{\tau}_2 \bigg) \leq \bar{\mathbb{P}}_{\mathrm{v}^*} \big( \bar{Z}_{2,k^*} < \widetilde{\tau}_2 \big)$$

$$\leq \bar{\mathbb{P}}_{\mathrm{v}^*} \Big( \big| \bar{Z}_{2,k^*} - \mathbb{E}_{\mathbb{P}_{\mathrm{v}}}[q_{2,k^*}(Y,X)] \big| > \tau_{q_{2,k^*}} \Big) \leq 2/d, \tag{B.29}$$

where the last inequality holds since $\widetilde{\tau}_2 > \tau_{q_{2,k^*}}$. If it holds that $\mathbb{E}_{\mathbb{P}_{\mathrm{v}^*}}[q_{2,\ell^*}(Y,X)] \leq -\alpha\rho/2 \leq -2\widetilde{\tau}_2$, we have

$$\bar{\mathbb{P}}_{\mathrm{v}^*}(\widetilde{\phi}_2 = 0) \leq \bar{\mathbb{P}}_{\mathrm{v}^*} \bigg( \inf_{j \in [d]} \bar{Z}_{2,j} > -\widetilde{\tau}_2 \bigg) \leq \bar{\mathbb{P}}_{\mathrm{v}^*} \big( \bar{Z}_{2,\ell^*} > -\widetilde{\tau}_2 \big)$$

$$\leq \bar{\mathbb{P}}_{\mathrm{v}^*} \Big( \big| \bar{Z}_{2,\ell^*} - \mathbb{E}_{\mathbb{P}_{\mathrm{v}}}[q_{2,\ell^*}(Y,X)] \big| > \tau_{q_{2,\ell^*}} \Big) \leq 2/d, \tag{B.30}$$

where the last inequality holds since $\widetilde{\tau}_2 > \tau_{q_{2,\ell^*}}$. Note that (B.28), (B.29), and (B.30) holds for any $(\beta^*, \sigma) \in \bar{\mathcal{G}}_1(s, \gamma_n)$ if (A.14) holds. Therefore, by combining (B.28), (B.29), and (B.30), we have

$$\sup_{(\beta^*,\sigma)\in\bar{\mathcal{G}}_1(s,\gamma_n)} \bar{\mathbb{P}}_{\mathrm{v}^*}(\widetilde{\phi} = 0) \leq \sup_{(\beta^*,\sigma)\in\bar{\mathcal{G}}_1(s,\gamma_n)} \big\{ \bar{\mathbb{P}}_{\mathrm{v}^*}(\widetilde{\phi}_1 = 0) \wedge \bar{\mathbb{P}}_{\mathrm{v}^*}(\widetilde{\phi}_2 = 0) \big\} \leq 2/d. \tag{B.31}$$

In other words, the type-II error of $\phi$ is upper bounded by $2/d$. By combining (B.26) and (B.31), we conclude that if (A.14) holds, the risk for $\widetilde{\phi}$ is of order $O(1/d)$, which completes the proof of Theorem A.3. $\qquad\square$

## B.5 Proof of Theorem 3.3

*Proof.* We prove by contradiction in the following. We assume that there exist an absolute constant $\eta$ and an algorithm $\mathscr{A} \in \mathcal{A}(T)$ with $T = O(d^\eta)$ that estimates $\beta^*$ in (2.6), such that for any given oracle $r \in \mathcal{R}[\xi, n, T, \eta(\mathcal{Q})]$, it holds that

$$\bar{\mathbb{P}}\big(\|\widehat{\beta} - \beta^*\|_2^2/\sigma^2 \geq \gamma_n/16\big) = o(1), \tag{B.32}$$

where $\widehat{\beta}$ is the estimator of $\beta^*$. In other words, it holds that $\|\widehat{\beta} - \beta^*\|_2^2/\sigma^2 \leq \gamma_n/16$ with probability $1 - o(1)$. Recall that we set $\|\beta^*\|^2/\sigma^2 = \gamma_n$. Based on (B.32), it holds with probability $1 - o(1)$ that

$$\|\widehat{\beta} + \beta^*\|_2^2 \leq (\|\widehat{\beta} - \beta^*\|_2 + 2\|\beta^*\|_2)^2 \leq 2\|\widehat{\beta} - \beta^*\|_2^2 + 8\|\beta^*\|_2^2 \leq (1/8 + 8) \cdot \sigma^2\gamma_n. \tag{B.33}$$

Combining (B.32) and (B.33), it follows from the Cauchy-Schwartz inequality that

$$\big|\|\widehat{\beta}\|_2^2 - \|\beta^*\|_2^2\big|^2 = \big|(\widehat{\beta} - \beta^*)^\top(\widehat{\beta} + \beta^*)\big|^2 \leq \|\widehat{\beta} - \beta^*\|_2^2 \cdot \|\widehat{\beta} + \beta^*\|_2^2 \leq 5/8 \cdot \sigma^4\gamma_n^2, \tag{B.34}$$

which holds with probability $1 - o(1)$. In what follows, we construct an asymptotically powerful test with $T = O(d^\eta)$ query complexity for the hyppthesis testing problem in (2.7). We set $\phi = \mathbb{1}\{\|\widehat{\beta}\|_2^2 \geq \gamma_n/5\}$, where $\widehat{\beta}$ is the estimator of $\beta^*$ given the algorithm $\mathscr{A}$. Following from (B.32), it holds with probability $1 - o(1)$ that $\|\widehat{\beta}\|_2^2/\sigma^2 \leq \gamma_n/16$ under the null hypothesis with $\beta^* = 0$. Meanwhile, following from (B.34), it holds with probability $1 - o(1)$ that $\|\widehat{\beta}\|_2^2/\sigma^2 \geq \gamma_n/5$ under the alternative hypothesis with $\beta^* \neq 0$ and $\|\beta^*\|^2/\sigma^2 = \gamma_n$. In other words, $\phi$ is asymptotically powerful and computationally tractable with $\gamma_n = o(\sqrt{s^2/n} \wedge 1/\alpha^2 \cdot s \log d/n)$, which contradicts the computational minimax separation rate in (A.16). $\qquad\square$

## C  Proof of Lemmas

In this section, we lay out the proof of the lemmas in §B.

## C.1  Proof of Lemma B.1

*Proof.* It follows from the model in (B.1) that under the alternative hypothesis,

$$Z = (Y, X) \sim \alpha \cdot N\big(0, \Sigma(\mathrm{v})\big) + \frac{1 - \alpha}{2} \cdot N\big(0, \Sigma(\mathrm{v})\big) + \frac{1 - \alpha}{2} \cdot N\big(0, \Sigma(-\mathrm{v})\big),$$

$$\sim \frac{1 + \alpha}{2} \cdot N\big(0, \Sigma(\mathrm{v})\big) + \frac{1 - \alpha}{2} \cdot N\big(0, \Sigma(-\mathrm{v})\big),$$

where $\Sigma(\mathrm{v})$ is the covariance matrix

$$\Sigma(\mathrm{v}) = \begin{bmatrix} \sigma^2 + s\rho^2 & \rho\mathrm{v}^\top \\ \rho\mathrm{v} & I_d \end{bmatrix} \in \mathbb{R}^{(d+1)\times(d+1)}. \tag{C.1}$$

Meanwhile, we have $Z = (Y, X) \sim N(0, \Sigma_0)$ under the null hypothesis, where we denote by $\Sigma_0 = \Sigma(0)$. Recall that we denote by $\mathbb{P}_\mathrm{v}$ and $\mathbb{P}_0$ the distributions of $Z$ under the alternative and null hypotheses, respectively. Therefore, it holds that

$$\frac{d\mathbb{P}_\mathrm{v}}{d\mathbb{P}_0}(Z) = \frac{1 + \alpha}{2} \cdot \sqrt{\frac{\det(\Sigma_0)}{\det\big(\Sigma(\mathrm{v})\big)}} \cdot \exp\left(-\frac{Z\big(\Sigma^{-1}(\mathrm{v}) - \Sigma_0^{-1}\big)Z^\top}{2}\right)$$

$$+ \frac{1 - \alpha}{2} \cdot \sqrt{\frac{\det(\Sigma_0)}{\det\big(\Sigma(-\mathrm{v})\big)}} \cdot \exp\left(-\frac{Z\big(\Sigma^{-1}(-\mathrm{v}) - \Sigma_0^{-1}\big)Z^\top}{2}\right), \tag{C.2}$$

where we denote by $\Sigma^{-1}(\mathrm{v})$ the inverse matrix of $\Sigma(\mathrm{v})$. We denote by $\xi$ the Bernoulli random variable with distribution

$$\mathbb{P}(\xi = 1) = \frac{1 + \alpha}{2}, \quad \mathbb{P}(\xi = -1) = \frac{1 - \alpha}{2}. \tag{C.3}$$

692 Therefore, it follows from (C.2) that

$$\frac{d\mathbb{P}_v}{d\mathbb{P}_0}(Z) = \mathbb{E}_\xi\left[\sqrt{\frac{\det(\Sigma_0)}{\det(\Sigma(\xi v))}} \cdot \exp\left(-\frac{Z\left(\Sigma^{-1}(\xi v) - \Sigma_0^{-1}\right)Z^\top}{2}\right)\right]. \tag{C.4}$$

693 Following from (C.4), for $v_1$ and $v_2$ in $\mathcal{G}(s)$, we have

$$\mathbb{E}_{\mathbb{P}_0}\left[\frac{d\mathbb{P}_{v_1}}{d\mathbb{P}_0}\frac{d\mathbb{P}_{v_2}}{d\mathbb{P}_0}(Z)\right] = \mathbb{E}_{\mathbb{P}_0}\mathbb{E}_{\xi_1,\xi_2}\left[\frac{\det(\Sigma_0)}{\sqrt{\det\left(\Sigma(\xi_1 v_1)\right) \cdot \det\left(\Sigma(\xi_2 v_2)\right)}} \right. \tag{C.5}$$

$$\left. \cdot \exp\left(-1/2 \cdot Z^\top\left(\Sigma^{-1}(\xi_1 v_1) + \Sigma^{-1}(\xi_1 v_2) - 2\Sigma_0^{-1}\right)Z\right)\right],$$

694 where $\xi_1$ and $\xi_2$ are independent copies of $\xi$ defined in (C.3). In what follows, we calculate the
695 right-hand side of (C.5) by invoking Fubini's theorem. We first calculate the right-hand side of (C.5)
696 by integrating under $\mathbb{P}_0$ and obtain that

$$\mathbb{E}_{\mathbb{P}_0}\left[\exp\left(-1/2 \cdot Z^\top\left(\Sigma^{-1}(\xi_1 v_1) + \Sigma^{-1}(\xi_1 v_2) - 2\Sigma_0^{-1}\right)Z\right)\right]$$

$$= \frac{1}{\sqrt{(2\pi)^{d+1} \cdot \det(\Sigma_0)}} \cdot \int_{z\in\mathbb{R}^{d+1}} \exp\left(-1/2 \cdot z^\top\left(\Sigma^{-1}(\xi_1 v_1) + \Sigma^{-1}(\xi_1 v_2) - \Sigma_0^{-1}\right)z\right)d\mathbb{P}_0(z)$$

$$= \left(\det\left(\Sigma^{-1}(\xi_1 v_1) + \Sigma^{-1}(\xi_1 v_2) - \Sigma_0^{-1}\right) \cdot \det(\Sigma_0)\right)^{-1/2}. \tag{C.6}$$

697 By plugging (C.6) into (C.5), we obtain

$$\mathbb{E}_{\xi_1,\xi_2}\mathbb{E}_{\mathbb{P}_0}\left[\frac{\det(\Sigma_0)}{\sqrt{\det\left(\Sigma(\xi_1 v_1)\right) \cdot \det\left(\Sigma(\xi_2 v_2)\right)}} \cdot \exp\left(-1/2 \cdot Z^\top\left(\Sigma^{-1}(\xi_1 v_1) + \Sigma^{-1}(\xi_1 v_2) - 2\Sigma_0^{-1}\right)Z\right)\right]$$

$$= \mathbb{E}_{\xi_1,\xi_2}\left[\frac{\det(\Sigma_0)}{\sqrt{\det\left(\Sigma(\xi_1 v_1)\right) \cdot \det\left(\Sigma(\xi_2 v_2)\right)}} \cdot \left(\det\left(\Sigma^{-1}(\xi_1 v_1) + \Sigma^{-1}(\xi_1 v_2) - \Sigma_0^{-1}\right)\det(\Sigma_0)\right)^{-1/2}\right]$$

$$= \sqrt{\det(\Sigma_0)} \cdot \mathbb{E}_{\xi_1,\xi_2}\left[\det\left(\Sigma(\xi_1 v_1) + \Sigma(\xi_2 v_2) - \Sigma(\xi_1 v_1)\Sigma_0^{-1}\Sigma(\xi_2 v_2)\right)^{-1/2}\right]. \tag{C.7}$$

698 Meanwhile, by (C.1) it holds that $\det(\Sigma_0) = \sigma^2 + s\rho^2$ and

$$\Sigma(\xi_1 v_1) + \Sigma(\xi_2 v_2) - \Sigma(\xi_1 v_1) \cdot \Sigma_0^{-1} \cdot \Sigma(\xi_2 v_2)$$

$$= \begin{bmatrix} \sigma^2 + s\rho^2(1 - \xi_1\xi_2 \cdot v_1^\top v_2) & 0 \\ 0 & I_d - (\rho^2\xi_1\xi_2)/(\sigma^2 + s\rho^2) \cdot v_1 v_2^\top \end{bmatrix}. \tag{C.8}$$

699 Therefore, we are able to calculate the right-hand side of (C.7) explicitly. Combining (C.5) and (C.7)
700 and apply Fubini's theorem, we obtain that

$$\mathbb{E}_{\mathbb{P}_0}\left[\frac{d\mathbb{P}_{v_1}}{d\mathbb{P}_0}\frac{d\mathbb{P}_{v_2}}{d\mathbb{P}_0}(Z)\right] = \mathbb{E}_{\xi_1,\xi_2}\left[1 - \frac{\rho^2\xi_1\xi_2}{\sigma^2 + s\rho^2} \cdot \langle v_1, v_2\rangle\right]. \tag{C.9}$$

701 Recall that $\xi_1$ and $\xi_2$ are independent copies of $\xi$ defined in (C.3), it then holds that

$$\mathbb{E}_{\mathbb{P}_0}\left[\frac{d\mathbb{P}_{v_1}}{d\mathbb{P}_0}\frac{d\mathbb{P}_{v_2}}{d\mathbb{P}_0}(Z)\right] = \frac{1 + \alpha^2(\sigma^2 + s\rho^2)^{-1}\rho^2 \cdot \langle v_1, v_2\rangle}{1 - (\sigma^2 + s\rho^2)^{-2}\rho^4 \cdot \langle v_1, v_2\rangle^2}. \tag{C.10}$$

702 Meanwhile, for $0 \le x < 1/2$ and $0 \le k \le 1$, we have

$$\frac{1 + kx}{1 - x^2} \le \cosh(2x) + k \cdot \sinh(2x).$$

703 Therefore, following from (C.10) with $s\rho^2 = o(1)$, we obtain that

$$\mathbb{E}_{\mathbb{P}_0}\left[\frac{d\mathbb{P}_{v_1}}{d\mathbb{P}_0}\frac{d\mathbb{P}_{v_2}}{d\mathbb{P}_0}(Z)\right] \le \cosh\left(\frac{2\rho^2 \cdot \langle v_1, v_2\rangle}{\sigma^2 + s\rho^2}\right) + \alpha^2 \cdot \sinh\left(\frac{2\rho^2 \cdot \langle v_1, v_2\rangle}{\sigma^2 + s\rho^2}\right), \tag{C.11}$$

704 which concludes the proof of Lemma B.1. □

## C.2   Proof of Lemma B.3

*Proof.* In what follows, we establish the upper bound of the following sum,

$$S = \frac{1}{|\mathcal{G}(s)|^2} \sum_{v_1, v_2 \in \mathcal{G}(s)} \left[ \exp\left( \frac{4\alpha^2 \rho^2 \cdot \langle v_1, v_2 \rangle}{\sigma^2 + s\rho^2} \right) \bigvee \cosh\left( \frac{4\rho^2 \cdot \langle v_1, v_2 \rangle}{\sigma^2 + s\rho^2} \right) \right]^n. \tag{C.12}$$

In specific, we show that $S = 1 + o(1)$ if it holds that

$$\gamma_n = o\left( \sqrt{\frac{s \log d}{n}} \bigwedge \frac{1}{\alpha^2} \cdot \frac{s \log d}{n} \right).$$

The proof strategy is similar to that of Theorem 3.1 by [62]. We define $\mathcal{V}(s)$ the class of index set as follows,

$$\mathcal{V}(s) = \{ \mathcal{S} \subseteq [d] : |\mathcal{S}| = s \}.$$

We further denote by $\mathcal{S}_1$ and $\mathcal{S}_2$ two independent random variables, which are uniformly distributed over $\mathcal{V}(s)$ and

$$T = |\mathcal{S}_1 \cap \mathcal{S}_2|.$$

We obtain from (C.12) the following upper bound of $S$,

$$S \leq \mathbb{E}_T \left[ \left\{ \exp\left( \frac{4\alpha^2 \rho^2 T}{\sigma^2 + s\rho^2} \right) \bigvee \cosh\left( \frac{4\rho^2 T}{\sigma^2 + s\rho^2} \right) \right\}^n \right]. \tag{C.13}$$

Let $\{\eta_i\}_{i \in [n]}$ be $n$ independent Rademacher random variables and $U$ be their sum. Following from (C.13) and the fact that $\cosh(x) = \mathbb{E}_{\eta_i}[\exp(\eta_i x)]$, we obtain

$$S \leq \mathbb{E}_T \left[ \exp\left( \frac{4n\alpha^2 \rho^2 T}{\sigma^2 + s\rho^2} \right) \bigvee \mathbb{E}_U \left[ \exp\left( \frac{4\rho^2 UT}{\sigma^2 + s\rho^2} \right) \right] \right]$$

$$= \mathbb{E}_T \mathbb{E}_U \left[ \exp\left( \frac{4n\alpha^2 \rho^2 T}{\sigma^2 + s\rho^2} \right) \bigvee \exp\left( \frac{4\rho^2 UT}{\sigma^2 + s\rho^2} \right) \right]. \tag{C.14}$$

We apply Fubini's theorem to calculate the right-hand side of (C.14). We first calculate the expectation with respect to $T$. Recall that we denote by $T = |\mathcal{S}_1 \cap \mathcal{S}_2|$. Therefore, it holds that

$$\mathbb{E}_T \left[ \exp\left( \frac{4n\alpha^2 \rho^2 T}{\sigma^2 + s\rho^2} \right) \bigvee \exp\left( \frac{4\rho^2 UT}{\sigma^2 + s\rho^2} \right) \right]$$

$$= \mathbb{E}_T \left[ \left\{ \exp\left( \frac{4n\alpha^2 \rho^2}{\sigma^2 + s\rho^2} \right) \bigvee \exp\left( \frac{4\rho^2 U}{\sigma^2 + s\rho^2} \right) \right\}^T \right]$$

$$\leq \sup_{\mathcal{S} \in \mathcal{V}(s)} \mathbb{E}_{\mathcal{S}_2} \left[ \left\{ \exp\left( \frac{4n\alpha^2 \rho^2}{\sigma^2 + s\rho^2} \right) \bigvee \exp\left( \frac{4\rho^2 U}{\sigma^2 + s\rho^2} \right) \right\}^{|\mathcal{S} \cap \mathcal{S}_2|} \right], \tag{C.15}$$

where the last inequality holds since $\mathcal{S}_1$ is uniformly distributed over $\mathcal{V}(s)$. We fix an arbitrary $\mathcal{S} \in \mathcal{V}(s)$ and denote by $|\mathcal{S} \cap \mathcal{S}_2| = \sum_{i \in \mathcal{V}} v_i$, where $\{v_i\}_{i \in \mathcal{V}}$ are random variables that takes value one if $i \in \mathcal{S} \cap \mathcal{S}_2$ and zero otherwise. Recall that $\mathcal{S}_2$ is uniformly distributed over $\mathcal{C}(s)$. Therefore, $v_i$ takes value one with probability $s/d$ and zero otherwise. Meanwhile, for $i \neq j$, $v_i$ and $v_j$ are negatively associated with each other. Thus, it holds that

$$\mathbb{E}_{\mathcal{S}_2} \left[ \left\{ \exp\left( \frac{4n\alpha^2 \rho^2}{\sigma^2 + s\rho^2} \right) \bigvee \exp\left( \frac{4\rho^2 U}{\sigma^2 + s\rho^2} \right) \right\}^{|\mathcal{S} \cap \mathcal{S}_2|} \right]$$

$$\leq \prod_{i \in \mathcal{V}} \mathbb{E}_{v_i} \left[ \left\{ \exp\left( \frac{4n\alpha^2 \rho^2}{\sigma^2 + s\rho^2} \right) \bigvee \exp\left( \frac{4\rho^2 U}{\sigma^2 + s\rho^2} \right) \right\}^{v_i} \right]$$

$$= \left( s/d \cdot \left[ \exp\left( \frac{4n\alpha^2 \rho^2}{\sigma^2 + s\rho^2} \right) \bigvee \exp\left( \frac{4\rho^2 U}{\sigma^2 + s\rho^2} \right) \right] + 1 - s/d \right)^s. \tag{C.16}$$

Since the inequality in (C.16) holds for any $\mathcal{S} \in \mathcal{V}(s)$, it holds for the supreme over $\mathcal{V}(s)$. By plugging (C.16) into (C.15), we obtain that

$$
\mathbb{E}_T\left[\exp\left(\frac{4n\alpha^2\rho^2 T}{\sigma^2 + s\rho^2}\right) \bigvee \exp\left(\frac{4\rho^2 U T}{\sigma^2 + s\rho^2}\right)\right]
$$

$$
\leq 1 + \sum_{k=1}^{s}\binom{s}{k}\left(\frac{s}{d}\right)^k \cdot \left[\exp\left(\frac{4n\alpha^2\rho^2}{\sigma^2 + s\rho^2}\right) \bigvee \exp\left(\frac{4\rho^2 U}{\sigma^2 + s\rho^2}\right) - 1\right]^k. \tag{C.17}
$$

Finally, by combining (C.14) and (C.17), we obtain from Fubini's theorem that

$$
S - 1 \leq \sum_{k=1}^{s}\binom{s}{k}\left(\frac{s}{d}\right)^k \cdot \mathbb{E}_U\left[\left\{\exp\left(\frac{4n\alpha^2\rho^2}{\sigma^2 + s\rho^2}\right) \bigvee \exp\left(\frac{4\rho^2 U}{\sigma^2 + s\rho^2}\right) - 1\right\}^k\right]
$$

$$
\leq \sum_{k=1}^{s}\binom{s}{k}\left(\frac{s}{d}\right)^k \cdot \left[\exp\left(\frac{4n\alpha^2\rho^2}{\sigma^2 + s\rho^2}\right) - 1\right]^k
$$

$$
+ \binom{s}{k}\left(\frac{s}{d}\right)^k \cdot \mathbb{E}_U\left[\left\{\exp\left(\frac{4\rho^2 U}{\sigma^2 + s\rho^2}\right) - 1\right\}^k \;\middle|\; U \geq n\alpha^2\right]. \tag{C.18}
$$

It now suffices to show that the right-hand side of (C.18) is of order $o(1)$. The following lemma upper bounds the first term on the right-hand side of (C.18).

**Lemma C.1** ([62]). For $\gamma_n = s\rho^2/\sigma^2 = o(1/\alpha^2 \cdot s\log d/n)$, it holds that

$$
\sum_{k=1}^{s}\binom{s}{k}\left(\frac{s}{d}\right)^k \cdot \left[\exp\left(\frac{4n\alpha^2\rho^2}{\sigma^2 + s\rho^2}\right) - 1\right]^k = o(1). \tag{C.19}
$$

*Proof.* See §C.6 for a detailed proof. □

We denote by $Q = 4\rho^2 U/(\sigma^2 + s\rho^2)$. Note that $\exp(x) - 1 \leq 2x$ for $0 < x < 1$. Therefore, the following upper bound of the second term on the right-hand side of (C.18) holds,

$$
\sum_{k=1}^{s}\binom{s}{k}\left(\frac{s}{d}\right)^k \cdot \mathbb{E}_U\left[\left\{\exp\left(\frac{4\rho^2 U}{\sigma^2 + s\rho^2}\right) - 1\right\}^k \;\middle|\; U \geq 0\right]
$$

$$
\leq \sum_{k=1}^{s}\left(\frac{s^2 e}{kd}\right)^k \cdot \mathbb{E}_U\left[(2|Q|)^k + \exp(k|Q|) \cdot \mathbb{1}\{|Q| \geq 1\}\right]
$$

$$
\leq \underbrace{\sum_{k=1}^{s}\mathbb{E}_U\left[\frac{2s^2 e|Q|}{kd}\right]^k}_{(i)} + \underbrace{\sum_{k=1}^{s}\left(\frac{s^2 e}{kd}\right)^k \cdot \mathbb{E}_U\left[\exp(k|Q|) \cdot \mathbb{1}\{|Q| \geq 1\}\right]}_{(ii)}. \tag{C.20}
$$

The following Lemma establishes the upper bounds of terms (i) and (ii) in (C.20).

**Lemma C.2** ([62]). For $\gamma_n = s\rho^2/\sigma^2 = o(\sqrt{s\log d/n})$, it holds that

$$
T_1 = \sum_{k=1}^{s}\mathbb{E}_U\left[\frac{2s^2 e|Q|}{kd}\right]^k = o(1),
$$

$$
T_2 = \sum_{k=1}^{s}\left(\frac{s^2 e}{kd}\right)^k \cdot \mathbb{E}_U\left[\exp(k|Q|) \cdot \mathbb{1}(|Q| \geq 1)\right] = o(1). \tag{C.21}
$$

*Proof.* See §C.7 for a detailed proof. □

By combining (C.18) and (C.20), we obtain from Lemmas C.1 and C.2 that $S - 1 = o(1)$ for

$$
\gamma_n = o\left(\sqrt{\frac{s\log d}{n}} \bigwedge \frac{1}{\alpha^2} \cdot \frac{s\log d}{n}\right),
$$

which concludes the proof of Lemma B.3. □

 **C.3   Proof of Lemma B.4**

*Proof.* In what follows, we prove that $T \cdot \sup_{q \in \mathcal{Q}} |\mathcal{C}(q)|/|\mathcal{G}(s)| = o(1)$ under the assumptions of Lemma B.4. Our proof strategy is similar to that of Theorem 5.3 by [53]. As $|\mathcal{G}(s)|$ is given, we focus on upper bounding $|\mathcal{C}(q)|$. We first partition $\mathcal{C}(q)$ into two parts, namely, $\mathcal{C}_1(q)$ and $\mathcal{C}_2(q)$, where

$$\mathcal{C}_1(q) = \left\{ \mathrm{v} \in \mathcal{G}(s) : \mathbb{E}_{\mathbb{P}_0}\big[q(Z)\big] - \mathbb{E}_{\mathbb{P}_\mathrm{v}}\big[q(Z)\big] > \tau_q \right\},$$

and $\mathcal{C}_2(q) = \mathcal{C}(q)\backslash\mathcal{C}_1(q)$. It holds that

$$\sup_{q \in \mathcal{Q}} |\mathcal{C}(q)| \leq \sup_{q \in \mathcal{Q}} |\mathcal{C}_1(q)| + \sup_{q \in \mathcal{Q}} |\mathcal{C}_2(q)|. \tag{C.22}$$

We introduce the following distributions,

$$\mathbb{P}_{\mathcal{C}_1(q)} = \frac{1}{|\mathcal{C}_1(q)|} \sum_{\mathrm{v} \in \mathcal{C}_1(q)} \mathbb{P}_\mathrm{v}, \quad \mathbb{P}_{\mathcal{C}_2(q)} = \frac{1}{|\mathcal{C}_2(q)|} \sum_{\mathrm{v} \in \mathcal{C}_2(q)} \mathbb{P}_\mathrm{v}.$$

We further denote by

$$\bar{\mathcal{C}}_\ell(q, \mathrm{v}) = \underset{\mathcal{C}}{\mathrm{argmax}} \left\{ \frac{1}{|\mathcal{C}|} \sum_{\mathrm{v}' \in \mathcal{C}} \mathbb{E}_{\mathbb{P}_0} \left[ \frac{\mathrm{d}\mathbb{P}_\mathrm{v}}{\mathrm{d}\mathbb{P}_0} \frac{\mathrm{d}\mathbb{P}_{\mathrm{v}'}}{\mathrm{d}\mathbb{P}_0}(X) \right] - 1 \,\Big|\, |\mathcal{C}| = |\mathcal{C}_\ell(q)| \right\} \subseteq \mathcal{G}(s) \tag{C.23}$$

for $\ell \in \{1, 2\}$. It then holds that

$$D_{\chi^2}(\mathbb{P}_{\mathcal{C}_\ell(q)}, \mathbb{P}_0) = \mathbb{E}_{\mathbb{P}_0} \left[ \left( \frac{\mathrm{d}\mathbb{P}_{\mathcal{C}_\ell(q)}}{\mathrm{d}\mathbb{P}_0}(Z) - 1 \right)^2 \right] = \frac{1}{\mathcal{C}_\ell(q)} \sum_{\mathrm{v}, \mathrm{v}' \in \mathcal{C}_\ell(q)} \mathbb{E}_{\mathbb{P}_0} \left[ \frac{\mathrm{d}\mathbb{P}_\mathrm{v}}{\mathrm{d}\mathbb{P}_0} \frac{\mathrm{d}\mathbb{P}_{\mathrm{v}'}}{\mathrm{d}\mathbb{P}_0}(Z) \right] - 1$$

$$\leq \sup_{\mathrm{v} \in \mathcal{C}_\ell(q)} \frac{1}{|\mathcal{C}_\ell(q)|} \sum_{\mathrm{v}' \in \mathcal{C}_\ell(q)} \mathbb{E}_{\mathbb{P}_0} \left[ \frac{\mathrm{d}\mathbb{P}_\mathrm{v}}{\mathrm{d}\mathbb{P}_0} \frac{\mathrm{d}\mathbb{P}_{\mathrm{v}'}}{\mathrm{d}\mathbb{P}_0}(Z) \right] - 1$$

$$\leq \sup_{\mathrm{v} \in \mathcal{C}_\ell(q)} \frac{1}{|\mathcal{C}_\ell(q)|} \sum_{\mathrm{v}' \in \bar{\mathcal{C}}_\ell(q, \mathrm{v})} \mathbb{E}_{\mathbb{P}_0} \left[ \frac{\mathrm{d}\mathbb{P}_\mathrm{v}}{\mathrm{d}\mathbb{P}_0} \frac{\mathrm{d}\mathbb{P}_{\mathrm{v}'}}{\mathrm{d}\mathbb{P}_0}(Z) \right] - 1, \tag{C.24}$$

where the last inequality follows from the definition of $\bar{\mathcal{C}}_\ell(q, \mathrm{v})$ in (C.23). By Lemma B.1, it holds that

$$\mathbb{E}_{\mathbb{P}_0} \left[ \frac{\mathrm{d}\mathbb{P}_\mathrm{v}}{\mathrm{d}\mathbb{P}_0} \frac{\mathrm{d}\mathbb{P}_{\mathrm{v}'}}{\mathrm{d}\mathbb{P}_0}(Z) \right] \leq \cosh\left( \frac{2\rho^2 \cdot \langle \mathrm{v}, \mathrm{v}' \rangle}{\sigma^2 + s\rho^2} \right) + \alpha^2 \cdot \sinh\left( \frac{2\rho^2 \cdot \langle \mathrm{v}, \mathrm{v}' \rangle}{\sigma^2 + s\rho^2} \right). \tag{C.25}$$

Combining (C.24) and (C.25), we conclude that

$$1 + D_{\chi^2}(\mathbb{P}_{\mathcal{C}_\ell(q)}, \mathbb{P}_0)$$

$$\leq \sup_{\mathrm{v} \in \mathcal{C}_\ell(q)} \left\{ \frac{1}{|\mathcal{C}_\ell(q)|} \sum_{\mathrm{v}' \in \bar{\mathcal{C}}_\ell(q, \mathrm{v})} \cosh\left( \frac{2\rho^2 \cdot \langle \mathrm{v}, \mathrm{v}' \rangle}{\sigma^2 + s\rho^2} \right) + \alpha^2 \cdot \sinh\left( \frac{2\rho^2 \cdot \langle \mathrm{v}, \mathrm{v}' \rangle}{\sigma^2 + s\rho^2} \right) \right\}. \tag{C.26}$$

In what follows, we calculate the sum on the right-hand side of (C.26). To achieve this, we calculate the sum based on the value of $\langle \mathrm{v}, \mathrm{v}' \rangle$. We denote by

$$\mathcal{C}_j(\mathrm{v}) = \big\{ \mathrm{v}' \in \mathcal{G}(s) : \langle \mathrm{v}, \mathrm{v}' \rangle = s - j \big\}.$$

Then for any choice of $\ell$, $q$, and $\mathrm{v} \in \mathcal{C}_\ell(q)$, there exists an integer $k_\ell(q, \mathrm{v})$ such that

$$\bar{\mathcal{C}}_\ell(q, \mathrm{v}) = \mathcal{C}_0(\mathrm{v}) \cup \cdots \cup \mathcal{C}_{k_\ell(q, \mathrm{v})-1} \cup \mathcal{C}'_\ell(q, \mathrm{v}),$$

where $\mathcal{C}'_\ell(q, \mathrm{v}) = \bar{\mathcal{C}}_\ell(q, \mathrm{v})\backslash\bigcup_{j=0}^{k_\ell(q, \mathrm{v})-1} \mathcal{C}_j(\mathrm{v})$. Note that we have

$$|\mathcal{C}'_\ell(q, \mathrm{v})| = |\mathcal{C}_\ell(q)| - \sum_{j=0}^{k_\ell(q, \mathrm{v})-1} |\mathcal{C}_j(\mathrm{v})| < |\mathcal{C}_{k_\ell(q, \mathrm{v})}(\mathrm{v})|.$$

Hence, the cardinality of $\bar{\mathcal{C}}_\ell(q, \mathrm{v})$ is between $\sum_{j=0}^{k_\ell(q, \mathrm{v})-1} |\mathcal{C}_j(\mathrm{v})|$ and $\sum_{j=0}^{k_\ell(q, \mathrm{v})} |\mathcal{C}_j(\mathrm{v})|$. Following form (C.26), we have

$$1 + D_{\chi^2}(\mathbb{P}_{\mathcal{C}_\ell(q)}, \mathbb{P}_0) \leq \frac{\sum_{j=0}^{k_\ell(q, \mathrm{v})-1} h_\alpha(j) \cdot |\mathcal{C}_j(\mathrm{v})| + h_\alpha\big(k_\ell(q, \mathrm{v})\big) \cdot |\mathcal{C}'_\ell(q, \mathrm{v})|}{\sum_{j=0}^{k_\ell(q, \mathrm{v})-1} |\mathcal{C}_j(\mathrm{v})| + |\mathcal{C}'_\ell(q, \mathrm{v})|}, \tag{C.27}$$

where we denote by $h_\alpha(j)$ the right-hand side of (C.25) when $\mathrm{v}' \in \mathcal{C}_j(\mathrm{v})$. In other words, it holds that

$$h_\alpha(j) = \cosh\left(\frac{2\rho^2(s-j)}{\sigma^2 + s\rho^2}\right) + \alpha^2 \cdot \sinh\left(\frac{2\rho^2(s-j)}{\sigma^2 + s\rho^2}\right). \tag{C.28}$$

Note that $h_\alpha(j)$ is monotonically decreasing as $j$ increases. Therefore, it follows from (C.27) that

$$1 + D_{\chi^2}(\mathbb{P}_{\mathcal{C}_\ell(q)}, \mathbb{P}_0) \le \frac{\sum_{j=0}^{k_\ell(q,\mathrm{v})-1} h_\alpha(j) \cdot |\mathcal{C}_j(\mathrm{v})|}{\sum_{j=0}^{k_\ell(q,\mathrm{v})-1} |\mathcal{C}_j(\mathrm{v})|}. \tag{C.29}$$

Further note that $|\mathcal{C}_j(\mathrm{v})| = \binom{s}{s-j}\binom{d-s}{j}$. Therefore, it holds that

$$|\mathcal{C}_{j+1}(\mathrm{v})|/|\mathcal{C}_j(\mathrm{v})| = (s-j)(d-s-j)/(j+1)^2 \ge d/2s^2,$$

where $j \in \{0, \ldots, s-1\}$, $\mathrm{v} \in \mathcal{G}(s)$, and $s = o(d^{1/2-\delta})$. We denote by $\zeta = d/2s^2$, which satisfies $\zeta^{-1} = o(1)$ by the assumption that $s = o(d^{1/2-\delta})$. It then holds that

$$|\mathcal{C}_\ell(q)| \le \sum_{j=0}^{k_\ell(q,\mathrm{v})} |\mathcal{C}_j(\mathrm{v})| \le |\mathcal{C}_s(\mathrm{v})| \cdot \sum_{j=0}^{k_\ell(q,\mathrm{v})} \zeta^{j-s}$$

$$\le \frac{\zeta^{-(s-k_\ell(q,\mathrm{v}))} \cdot |\mathcal{G}(s)|}{1 - \zeta^{-1}} \le 2\zeta^{-(s-k_\ell(q,\mathrm{v}))} \cdot |\mathcal{G}(s)|. \tag{C.30}$$

For any integer $k \ge 1$ and two positive sequences $\{w_i\}_{i=0}^\infty$ and $\{u_i\}_{i=0}^\infty$ such that $w_i/w_{i-1} \ge u_i/u_{i-1} > 1$, it holds that

$$\frac{\sum_{j=0}^k w_j \cdot h_\alpha(j)}{\sum_{i=0}^k w_j} \le \frac{\sum_{j=0}^k u_j \cdot h_\alpha(j)}{\sum_{j=0}^k u_j}. \tag{C.31}$$

Therefore, by setting $w_j = |\mathcal{C}_j(\mathrm{v})|$ and $u_j = \zeta^j$, we conclude from (C.29) and (C.31) that

$$1 + D_{\chi^2}(\mathbb{P}_{\mathcal{C}_\ell(q)}, \mathbb{P}_0) \le \frac{\sum_{j=0}^{k_\ell(q,\mathrm{v})-1} \zeta^j \cdot h_\alpha(j)}{\sum_{j=0}^{k_\ell(q,\mathrm{v})-1} \zeta^j} \tag{C.32}$$

$$= \left[\sum_{j=0}^{k_\ell(q,\mathrm{v})-1} \zeta^j \cdot \cosh\left(\frac{2\rho^2(s-j)}{\sigma^2 + s\rho^2}\right) + \alpha^2 \cdot \sinh\left(\frac{2\rho^2(s-j)}{\sigma^2 + s\rho^2}\right)\right] \bigg/ \sum_{j=0}^{k_\ell(q,\mathrm{v})-1} \zeta^j$$

$$\le \sum_{j=0}^{k_\ell(q,\mathrm{v})-1} \zeta^j \cdot \left\{\cosh\left(\frac{4\rho^2(s-j)}{\sigma^2 + s\rho^2}\right) \bigvee \exp\left(\frac{4\alpha^2\rho^2(s-j)}{\sigma^2 + s\rho^2}\right)\right\} \bigg/ \sum_{j=0}^{k_\ell(q,\mathrm{v})-1} \zeta^j,$$

where the last inequality follows from Lemma B.1. In what follows, we denote by

$$f(j) = \cosh\left(\frac{4\rho^2(s-j)}{\sigma^2 + s\rho^2}\right), \quad g(j) = \exp\left(\frac{4\alpha^2\rho^2(s-j)}{\sigma^2 + s\rho^2}\right) \tag{C.33}$$

for notational simplicity. Note that

$$f(j-1)/f(j) \ge \cosh\left(\frac{4\rho^2}{\sigma^2 + s\rho^2}\right).$$

Therefore, it holds for $j \in \{0, 1, \ldots, k_\ell(q,\mathrm{v})-1\}$ that

$$f(j) \le f\big(k_\ell(q,\mathrm{v})-1\big) \cdot \left\{\cosh\left(\frac{4\rho^2}{\sigma^2 + s\rho^2}\right)\right\}^{k_\ell(q,\mathrm{v})-j-1}. \tag{C.34}$$

Meanwhile, we have

$$g(j) = \exp\big(4\alpha^2\rho^2(s-j)\sigma^2 + s\rho^2\big) = g\big(k_\ell(q,\mathrm{v})-1\big) \cdot \left\{\exp\left(\frac{4\alpha^2\rho^2}{\sigma^2 + s\rho^2}\right)\right\}^{k_\ell(q,\mathrm{v})-j-1}. \tag{C.35}$$

We denote by

$$\Gamma(s,\rho) = \exp\left(\frac{4\alpha^2\rho^2}{\sigma^2 + s\rho^2}\right) \bigvee \cosh\left(\frac{4\rho^2}{\sigma^2 + s\rho^2}\right). \tag{C.36}$$

Combining (C.34) and (C.35), we conclude that

$$f(j) \vee g(j) \le \left\{f\big(k_\ell(q,\mathrm{v})-1\big) \vee g\big(k_\ell(q,\mathrm{v})-1\big)\right\} \cdot \big(\Gamma(s,\rho)\big)^{k_\ell(q,\mathrm{v})-j-1}. \tag{C.37}$$

768 Following from (C.32) and (C.37), it holds that

$$1 + D_{\chi^2}(\mathbb{P}_{\mathcal{C}_\ell(q)}, \mathbb{P}_0)$$

$$\leq \left\{ f\big(k_\ell(q,\mathrm{v})-1\big) \vee g\big(k_\ell(q,\mathrm{v})-1\big) \right\} \cdot \frac{\sum_{j=0}^{k_\ell(q,\mathrm{v})-1} \zeta^j \cdot \big(\Gamma(s,\rho)\big)^{k_\ell(q,\mathrm{v})-j-1}}{\sum_{j=0}^{k_\ell(q,\mathrm{v})-1} \zeta^j}. \tag{C.38}$$

769 By direct calculation, we obtain

$$\frac{\sum_{j=0}^{k_\ell(q,\mathrm{v})-1} \zeta^j \cdot \big(\Gamma(s,\rho)\big)^{k_\ell(q,\mathrm{v})-j-1}}{\sum_{j=0}^{k_\ell(q,\mathrm{v})-1} \zeta^j} = \frac{\zeta^{k_\ell(q,\mathrm{v})-1} \cdot \sum_{j=0}^{k_\ell(q,\mathrm{v})-1} \big(\Gamma(s,\rho)/\zeta\big)^{k_\ell(q,\mathrm{v})-j-1}}{\zeta^{k_\ell(q,\mathrm{v})-1} \cdot \sum_{j=0}^{k_\ell(q,\mathrm{v})-1} \zeta^{-(k_\ell(q,\mathrm{v})-j-1)}}$$

$$= \frac{1 - \big(\Gamma(s,\rho)/\zeta\big)^{k_\ell(q,\mathrm{v})}}{1 - \zeta^{-k_\ell(q,\mathrm{v})}} \cdot \frac{1 - \zeta^{-1}}{1 - \Gamma(s,\rho)/\zeta}. \tag{C.39}$$

770 Note that $\Gamma(s,\rho) \geq 1$. Therefore, the following upper bound of the right-hand side of (C.39) holds,

$$\frac{1 - \big(\Gamma(s,\rho)/\zeta\big)^{k_\ell(q,\mathrm{v})}}{1 - \zeta^{-k_\ell(q,\mathrm{v})}} \cdot \frac{1 - \zeta^{-1}}{1 - \Gamma(s,\rho)/\zeta} \leq \frac{1 - \zeta^{-1}}{1 - \Gamma(s,\rho)/\zeta}. \tag{C.40}$$

771 Combining (C.38), (C.39), and (C.40), we conclude that

$$1 + D_{\chi^2}(\mathbb{P}_{\mathcal{C}_\ell(q)}, \mathbb{P}_0) \leq \left\{ f\big(k_\ell(q,\mathrm{v})-1\big) \vee g\big(k_\ell(q,\mathrm{v})-1\big) \right\} \cdot \frac{1 - \zeta^{-1}}{1 - \Gamma(s,\rho)/\zeta}, \tag{C.41}$$

772 where $f(j)$ and $g(j)$ are defined in (C.33). Meanwhile, by Lemma 4.5 of [53], it holds that

$$D_{\chi^2}(\mathbb{P}_{\mathcal{C}_\ell(q)}, \mathbb{P}_0) \geq \log(T/\xi)/n. \tag{C.42}$$

773 We denote by $\tau^2$ the right-hand side of (C.42). Combining (C.41) and (C.42), we have

$$\tau^2 + 1 \leq \left\{ f\big(k_\ell(q,\mathrm{v})-1\big) \vee g\big(k_\ell(q,\mathrm{v})-1\big) \right\} \cdot \frac{1 - \zeta^{-1}}{1 - \Gamma(s,\rho)/\zeta}.$$

774 Therefore, one of the following inequalities holds,

$$(1 + \tau^2) \cdot \frac{1 - \Gamma(s,\rho)/\zeta}{1 - \zeta^{-1}} \leq g\big(k_\ell(q,\mathrm{v})-1\big) = \exp\left( \frac{4\alpha^2 \rho^2 \cdot \big(s - k_\ell(q,\mathrm{v})+1\big)}{\sigma^2 + s\rho^2} \right),$$

$$(1 + \tau^2) \cdot \frac{1 - \Gamma(s,\rho)/\zeta}{1 - \zeta^{-1}} \leq f\big(k_\ell(q,\mathrm{v})-1\big) \leq \exp\left( \frac{2\rho^4 \cdot \big(s - k_\ell(q,\mathrm{v})+1\big)^2}{(\sigma^2 + s\rho^2)^2} \right), \tag{C.43}$$

775 where the second inequality holds because of the fact that $\cosh(x) \leq \exp(x^2/2)$. We take the
776 logarithm of (C.43) and obtain that one of the following inequalities holds,

$$\log(1 + \tau^2) + \log\left( \frac{1 - \zeta^{-1}}{1 - \Gamma(s,\rho)/\zeta} \right) \leq \frac{4\alpha^2 \rho^2 \cdot \big(s - k_\ell(q,\mathrm{v})+1\big)}{\sigma^2 + s\rho^2},$$

$$\log(1 + \tau^2) + \log\left( \frac{1 - \zeta^{-1}}{1 - \Gamma(s,\rho)/\zeta} \right) \leq \frac{2\rho^4 \cdot \big(s - k_\ell(q,\mathrm{v})+1\big)^2}{(\sigma^2 + s\rho^2)^2}. \tag{C.44}$$

777 Following from the definition of $\Gamma(s,\rho)$ in (C.36), we have $\Gamma(s,\rho)/\zeta = o(1)$. By Taylor's expansion,
778 it holds that

$$\log\left( \frac{1 - \zeta^{-1}}{1 - \Gamma(s,\rho)/\zeta} \right) = \log\left( 1 - \zeta^{-1} \cdot \frac{1 - \Gamma(s,\rho)}{1 - \Gamma(s,\rho)/\zeta} \right) = O(\zeta^{-1}\rho^4 \vee \zeta^{-1}\alpha^2\rho^2). \tag{C.45}$$

779 For $\gamma_n = s\rho^2/\delta^2 = o(\sqrt{s^2/n} \wedge 1/\alpha^2 \cdot s/n)$, where $\sigma^2$ is a constant, it holds that $\alpha^2\rho^2 \vee \rho^4 = o(1/n)$.
780 Hence, the right-hand side of (C.45) is negligible compared with $\log(1 + \tau^2)$. Then following form
781 (C.44), it holds that

$$s - k_\ell(q,\mathrm{v}) + 1 \geq \sqrt{\frac{(\sigma^2 + s\rho^2)^2 \cdot \log(1 + \tau^2)}{2\rho^4}} \bigwedge \sqrt{\frac{(\sigma^2 + s\rho^2) \cdot \log(1 + \tau^2)}{4\alpha^2\rho^2}}. \tag{C.46}$$

Note that $\log(1 + \tau^2) \geq \tau^2/2 = \log(T/\xi)/(2n)$ for $\tau < 1$. Therefore, by combining (C.30) and (C.46), we conclude that

$$T \cdot \frac{\sup_{q \in \mathcal{Q}} |\mathcal{C}(q)|}{|\mathcal{G}(s)|} \tag{C.47}$$

$$\leq 4T \cdot \exp\left(-\log\zeta \cdot \left\{\sqrt{\frac{(\sigma^2 + s\rho^2)^2 \cdot \log(T/\xi)}{4n\rho^4}} - 1 \bigvee \sqrt{\frac{(\sigma^2 + s\rho^2) \cdot \log(T/\xi)}{8n\alpha^2\rho^2}} - 1\right\}\right).$$

Note that $\rho^4 \cdot n \vee \alpha^2\rho^2 \cdot n = o(1)$ for $s\rho^2/\sigma^2 = o(\sqrt{s^2/n} \wedge 1/\alpha^2 \cdot s/n)$. We choose an absolute constant $C > 0$ satisfying $\delta(C - 1) > \mu$, where $\mu$ and $\delta$ are absolute constants such that $T = O(d^\mu)$ and $s = o(d^{1/2-\delta})$. Then it holds for a sufficiently large $n$ that

$$\sqrt{\frac{(\sigma^2 + s\rho^2)^2 \cdot \log(T/\xi)}{4n\rho^4}} \bigvee \sqrt{\frac{(\sigma^2 + s\rho^2) \cdot \log(T/\xi)}{8n\alpha^2\rho^2}}$$

$$\geq \sqrt{\frac{(\sigma^2 + s\rho^2)^2 \cdot \log(1/\xi)}{4n\rho^4}} \bigvee \sqrt{\frac{(\sigma^2 + s\rho^2) \cdot \log(1/\xi)}{8n\alpha^2\rho^2}} \geq C. \tag{C.48}$$

Note that $\zeta = d/(2s^2) = \Omega(d^\delta)$ for $s = o(d^{1/2-\delta})$, where $\delta > 0$ is an absolute constant. Finally, combining (C.47) and (C.48), we obtain that for $T = O(d^\mu)$,

$$T \cdot \sup_{q \in \mathcal{Q}} |\mathcal{C}(q)|/|\mathcal{G}(s)| \leq \mathcal{O}(d^\mu \cdot \zeta^{-(C-1)}) = \mathcal{O}(d^{\mu-\delta(C-1)}) = o(1), \tag{C.49}$$

which concludes the proof of Lemma B.4. $\qquad\square$

## C.4  Proof of Lemma B.5

*Proof.* In the following proof, we denote by $C$ and $C'$ absolute constants, the value of which may vary from lines to lines. We define the following unbounded query functions,

$$\widetilde{q}_{1,\mathrm{v}}(Y, X) = \psi(Y) \cdot \left[s^{-1}(\mathrm{v}^\top X)^2 - 1\right] \cdot \mathbb{1}\left\{|\psi(Y)| \leq (R \cdot \log n)^{1/\nu}\right\}, \quad \mathrm{v} \in \bar{\mathcal{G}}(s),$$

$$\widetilde{q}_{2,\mathrm{v}}(Y, X) = Y \cdot (s^{-1/2}\mathrm{v}^\top X) \cdot \mathbb{1}\left\{|Y| \leq (R \cdot \log n)^{1/\nu}\right\}, \quad \mathrm{v} \in \bar{\mathcal{G}}(s). \tag{C.50}$$

In the sequel, we first upper bound the difference between the query functions in (A.5) and the query functions in (C.50). We then characterize the two expectations $\mathbb{E}_{\mathbb{P}_\mathrm{v}}[q_{i,\mathrm{v}}(Y, X)]$ and $\mathbb{E}_{\mathbb{P}_0}[q_{i,\mathrm{v}}(Y, X)]$ using the corresponding expectations of $\widetilde{q}_{i,\mathrm{v}}(Y, X)$. Following from (A.5) and (C.50), it holds that

$$\widetilde{q}_{1,\mathrm{v}} - q_{1,\mathrm{v}} = \psi(Y) \cdot \left[s^{-1}(\mathrm{v}^\top X)^2 - 1\right] \cdot \mathbb{1}\left\{|\psi(Y)| \leq (R \cdot \log n)^{1/\nu}\right\} \cdot \mathbb{1}\left\{|\mathrm{v}^\top X| > R \cdot \sqrt{s \log n}\right\},$$

$$\widetilde{q}_{2,\mathrm{v}} - q_{2,\mathrm{v}} = Y \cdot (s^{-1/2}\mathrm{v}^\top X) \cdot \mathbb{1}\left\{|Y| \leq (R \cdot \log n)^{1/\nu}\right\} \cdot \mathbb{1}\left\{|\mathrm{v}^\top X| > R \cdot \sqrt{s \log n}\right\}. \tag{C.51}$$

Then following from the Cauchy-Schwartz inequality, it holds for $q_{1,\mathrm{v}}$ and $\widetilde{q}_{1,\mathrm{v}}$ that

$$\left|\mathbb{E}_{\mathbb{P}_0}\left[q_{1,\mathrm{v}}(Y, X) - \widetilde{q}_{1,\mathrm{v}}(Y, X)\right]\right|^2$$

$$\leq \left|\mathbb{E}_{\mathbb{P}_0}\left[\psi(Y) \cdot \left(s^{-1}(\mathrm{v}^\top X)^2 - 1\right)\right]\right|^2 \cdot \mathbb{P}_0\left(|\mathrm{v}^\top X| > R \cdot \sqrt{s \log n}\right). \tag{C.52}$$

Note that under $H_0$, $X \sim N(0, I_d)$ is the standard Gaussian distribution, which is independent of $Y$. Therefore, it holds that $\mathbb{E}_{\mathbb{P}_0}[(s^{-1}(X^\top \mathrm{v})^2 - 1)^2] = 2$. Then following from the Cauchy-Schwartz inequality, we obtain that

$$\left|\mathbb{E}_{\mathbb{P}_0}\left[\psi(Y) \cdot \left(s^{-1}(\mathrm{v}^\top X)^2 - 1\right)\right]\right|^2 \cdot \mathbb{P}_0\left(|\mathrm{v}^\top X| > R \cdot \sqrt{s \log n}\right)$$

$$\leq \mathbb{E}_{\mathbb{P}_0}\left[\psi^2(Y)\right] \cdot \mathbb{E}_{\mathbb{P}_0}\left[\left(s^{-1}(X^\top \mathrm{v})^2 - 1\right)^2\right] \cdot \mathbb{P}_0\left(|\mathrm{v}^\top X| > R \cdot \sqrt{s \log n}\right)$$

$$= C \cdot \mathbb{P}_0\left(|\mathrm{v}^\top X| > R \cdot \sqrt{s \log n}\right) \tag{C.53}$$

for a positive absolute constant $C$. Note that $X^\top \mathrm{v}/\sqrt{s} \sim N(0, 1)$ under the null hypothesis. Following from the tail bound of standard Gaussian distribution, it holds for any $t \geq 1$ that

$$\mathbb{P}_0\left(|X^\top \mathrm{v}/\sqrt{s}| \geq t\right) \leq 2\exp(-t^2/2). \tag{C.54}$$

Combining (C.52), (C.53), and (C.54), we obtain that

$$\left|\mathbb{E}_{\mathbb{P}_0}\left[q_{1,\mathrm{v}}(Y, X) - \widetilde{q}_{1,\mathrm{v}}(Y, X)\right]\right|^2 \leq C \cdot \mathbb{P}(|\mathrm{v}^\top X| > R \cdot s\sqrt{\log n})$$

$$\leq C \cdot \exp(-R^2 \cdot \log n/2). \tag{C.55}$$

In the following, we upper bound the distance between $q_{1,\mathrm{v}}(Y,X)$ and $\widetilde{q}_{1,\mathrm{v}}(Y,X)$ under $\mathbb{P}_\mathrm{v}$. Following from the Cauchy-Schwartz inequality, it holds that

$$
\begin{aligned}
&\left|\mathbb{E}_{\mathbb{P}_{\mathrm{v}^*}}\left[q_{1,\mathrm{v}}(Y,X) - \widetilde{q}_{1,\mathrm{v}}(Y,X)\right]\right|^2 \\
&\qquad \leq \mathbb{E}_{\mathbb{P}_{\mathrm{v}^*}}\left[\psi^2(Y)\cdot\left(s^{-1}(\mathrm{v}^\top X)^2 - 1\right)^2\right]\cdot\mathbb{P}_{\mathrm{v}^*}\left(|\mathrm{v}^\top X| > R\cdot\sqrt{s\log n}\right) \\
&\qquad \leq \sqrt{\mathbb{E}_{\mathbb{P}_{\mathrm{v}^*}}\left[\psi^4(Y)\right]\cdot\mathbb{E}_{\mathbb{P}_{\mathrm{v}^*}}\left[\left(s^{-1}(\mathrm{v}^\top X)^2 - 1\right)^4\right]}\cdot\mathbb{P}_{\mathrm{v}^*}\left(|\mathrm{v}^\top X| > R\cdot\sqrt{s\log n}\right). \qquad \text{(C.56)}
\end{aligned}
$$

Note that under Assumption A.1, $\mathbb{E}_{\mathbb{P}_{\mathrm{v}^*}}[\psi^4(Y)]$ is upper bounded. Meanwhile, we have that $X^\top \mathrm{v}/\sqrt{s} \sim N(0,1)$. Therefore, it holds for an absolute constant $C$ that

$$
\left|\mathbb{E}_{\mathbb{P}_{\mathrm{v}^*}}\left[q_{1,\mathrm{v}}(Y,X) - \widetilde{q}_{1,\mathrm{v}}(Y,X)\right]\right|^2 \leq C\cdot\exp(-R^2\cdot\log n/2). \qquad \text{(C.57)}
$$

Similar arguments apply to $q_{2,\mathrm{v}}(Y,X)$ and $\widetilde{q}_{2,\mathrm{v}}(Y,X)$. Under the null hypothesis, it holds for an absolute constant $C'$ that

$$
\begin{aligned}
\left|\mathbb{E}_{\mathbb{P}_0}\left[q_{2,\mathrm{v}}(Y,X) - \widetilde{q}_{2,\mathrm{v}}(Y,X)\right]\right|^2 &\leq \mathbb{E}_{\mathbb{P}_0}[Y^2]\cdot\mathbb{E}_{\mathbb{P}_0}\left[s^{-1}(X^\top\mathrm{v})^2\right]\cdot\mathbb{P}\left(|\mathrm{v}^\top X| > R\cdot\sqrt{s\log n}\right) \\
&\leq C'\cdot\exp(-R^2\cdot\log n/2), \qquad \text{(C.58)}
\end{aligned}
$$

which also holds under the alternative hypothesis with distribution $\mathbb{P}_{\mathrm{v}^*}$. Therefore, following from (C.55), (C.57), and (C.58), it holds for a sufficiently large constant $R$ that

$$
\begin{aligned}
\left|\mathbb{E}_{\mathbb{P}_{\mathrm{v}^*}}\left[q_{1,\mathrm{v}}(Y,X) - \widetilde{q}_{1,\mathrm{v}}(Y,X)\right]\right| \vee \left|\mathbb{E}_{\mathbb{P}_0}\left[q_{1,\mathrm{v}}(Y,X) - \widetilde{q}_{1,\mathrm{v}}(Y,X)\right]\right| \leq 1/n, \\
\left|\mathbb{E}_{\mathbb{P}_{\mathrm{v}^*}}\left[q_{2,\mathrm{v}}(Y,X) - \widetilde{q}_{2,\mathrm{v}}(Y,X)\right]\right| \vee \left|\mathbb{E}_{\mathbb{P}_0}\left[q_{2,\mathrm{v}}(Y,X) - \widetilde{q}_{2,\mathrm{v}}(Y,X)\right]\right| \leq 1/n, \qquad \text{(C.59)}
\end{aligned}
$$

which holds for any $\mathrm{v}\in\bar{\mathcal{G}}(s)$. In what follows, we characterize the expectations of $\widetilde{q}_{i,\mathrm{v}}(Y,X)$ under the null and alternative hypotheses for $i\in\{1,2\}$. We then obtain the desired bounds of $q_{i,\mathrm{v}}(Y,X)$ based on $\widetilde{q}_{i,\mathrm{v}}(Y,X)$. Note that under the null hypothesis, $Y$ is independent of $X$. Then, following from (C.50) and the fact that $X\sim N(0,I_d)$, it holds that

$$
\mathbb{E}_{\mathbb{P}_0}\left[\widetilde{q}_{1,\mathrm{v}}(Y,X)\right] = \mathbb{E}_{\mathbb{P}_0}\left[\widetilde{q}_{2,\mathrm{v}}(Y,X)\right] = 0. \qquad \text{(C.60)}
$$

Following from (A.3), we have

$$
\begin{aligned}
s\rho^2 - \mathbb{E}_{\mathbb{P}_{\mathrm{v}^*}}\left[\widetilde{q}_{1,\mathrm{v}^*}(Y,X)\right] &\leq \mathbb{E}_{\mathbb{P}_{\mathrm{v}^*}}\left[\psi(Y)\cdot\left(s^{-1}(\mathrm{v}^{*\top}X)^2 - 1\right) - \widetilde{q}_{1,\mathrm{v}}(Y,X)\right] \qquad \text{(C.61)} \\
&= \mathbb{E}_{\mathbb{P}_{\mathrm{v}^*}}\left[\psi(Y)\cdot\left(s^{-1}(\mathrm{v}^{*\top}X)^2 - 1\right)\cdot\mathbb{1}\left\{|\psi(Y)| > (R\cdot\log n)^{1/\nu}\right\}\right] \\
&\leq \sqrt{\mathbb{E}_{\mathbb{P}_{\mathrm{v}^*}}\left[\psi^2(Y)\cdot\left(s^{-1}(\mathrm{v}^{*\top}X)^2 - 1\right)^2\right]}\cdot\sqrt{\mathbb{P}_{\mathrm{v}^*}\left(|\psi(Y)| > (R\cdot\log n)^{1/\nu}\right)},
\end{aligned}
$$

where the last inequality follows from the Cauchy-Schwartz inequality. It then follows from Assumption A.1 that

$$
\mathbb{P}_{\mathrm{v}^*}\left(|\psi(Y)| > (R\cdot\log n)^{1/\nu}\right) \leq C\cdot\exp(-R\cdot\log n). \qquad \text{(C.62)}
$$

Meanwhile, following from the Cauchy-Schwartz inequality, it holds that

$$
\mathbb{E}_{\mathbb{P}_{\mathrm{v}^*}}\left[\psi^2(Y)\cdot\left(s^{-1}(\mathrm{v}^{*\top}X)^2 - 1\right)^2\right] \leq \sqrt{\mathbb{E}_{\mathbb{P}_{\mathrm{v}^*}}\left[\psi^4(Y)\right]\cdot\mathbb{E}_{\mathbb{P}_{\mathrm{v}^*}}\left[\left(s^{-1}(\mathrm{v}^{*\top}X)^2 - 1\right)^4\right]}, \qquad \text{(C.63)}
$$

which is upper bounded by an absolute constant. Combining (C.61), (C.62), and (C.63), if it holds that $s\rho^2/\sigma^2 = \Omega(\sqrt{s\log d/n})$, then for sufficiently large $n$ and constant $R$, we obtain that $1/n \leq s\rho^2/4$ and

$$
s\rho^2 - \mathbb{E}_{\mathbb{P}_{\mathrm{v}^*}}\left[\widetilde{q}_{1,\mathrm{v}}(Y,X)\right] \leq s\rho^2/4. \qquad \text{(C.64)}
$$

In other words, it holds that $\mathbb{E}_{\mathbb{P}_{\mathrm{v}^*}}\left[\widetilde{q}_{1,\mathrm{v}}(Y,X)\right] \geq 3s\rho^2/4$. Similar arguments hold for the query function $\widetilde{q}_{2,\mathrm{v}}(Y,X)$. If it holds that $s\rho^2/\sigma^2 = \Omega(1/\alpha^2\cdot s\log d/n)$, then for sufficiently large $n$ and constant $R$, we obtain that $1/n \leq \sqrt{\alpha^2 s\rho^2}/4$ and

$$
\mathbb{E}_{\mathbb{P}_{\mathrm{v}^*}}\left[\widetilde{q}_{2,\mathrm{v}}(Y,X)\right] \geq 3\sqrt{\alpha^2 s\rho^2}/4. \qquad \text{(C.65)}
$$

Combining (C.59), (C.60), (C.64), and (C.65), it holds for sufficiently large $n$ and constant $R$ that

$$
\mathbb{E}_{\mathbb{P}_0}\left[q_{1,\mathrm{v}}(Y,X)\right] \leq 1/n, \quad \mathbb{E}_{\mathbb{P}_0}\left[q_{2,\mathrm{v}}(Y,X)\right] \leq 1/n.
$$

Furthermore, it holds for sufficiently large $n$ and constant $R$ that

$$
\begin{aligned}
\mathbb{E}_{\mathbb{P}_{\mathrm{v}^*}}\left[q_{1,\mathrm{v}^*}(Y,X)\right] &\geq s\rho^2/2, \ \ \text{if } s\rho^2/\sigma^2 = \Omega(\sqrt{s\log d/n}), \\
\mathbb{E}_{\mathbb{P}_{\mathrm{v}^*}}\left[q_{2,\mathrm{v}^*}(Y,X)\right] &\geq \sqrt{\alpha^2 s\rho^2}/2, \ \ \text{if } s\rho^2/\sigma^2 = \Omega(1/\alpha^2\cdot s\log d/n),
\end{aligned}
$$

which concludes the proof of Lemma B.5. $\qquad\qquad\square$

 **C.5 Proof of Lemma B.6**

829 *Proof.* In the following proof, we denote by $C$ and $C'$ absolute constants, the value of which may
830 vary from lines to lines. We define the following unbounded query functions,

$$\widetilde{q}_{1,j}(Y,X) = \psi(Y) \cdot (X_j^2 - 1) \cdot \mathbb{1}\{|\psi(Y)| \leq (R \cdot \log n)^{1/\nu}\}, \quad j \in [d],$$

$$\widetilde{q}_{2,j}(Y,X) = YX_j \cdot \mathbb{1}\{(|Y| \leq (R \cdot \log n)^{1/\nu}\}, \quad j \in [d]. \tag{C.66}$$

831 The proof is similar to the proof of Lemma B.5 in §C.4. Following from (C.66) and (A.11), it holds
832 that

$$\left|\mathbb{E}_{\mathbb{P}_0}\left[\widetilde{q}_{1,j}(Y,X) - q_{1,j}(Y,X)\right]\right|^2 \leq \mathbb{E}_{\mathbb{P}_0}\left[\psi^2(Y) \cdot (X_j^2 - 1)^2\right] \cdot \mathbb{P}_0\left(|X_j| \geq R \cdot \sqrt{\log n}\right), \tag{C.67}$$

833 where the inequality follows from the Cauchy-Schwartz inequality. Under the null hypothesis, $Y$ is
834 independent of $X$. Meanwhile, it holds that $X \sim N(0, I_d)$. Thus, we have $X_j \sim N(0,1)$. Following
835 from the Gaussian tail bound in (C.54), we have

$$\left|\mathbb{E}_{\mathbb{P}_0}\left[\widetilde{q}_{1,j}(Y,X) - q_{1,j}(Y,X)\right]\right|^2 \leq C \cdot \exp(-R^2 \cdot \log n/2). \tag{C.68}$$

836 Therefore, for a sufficiently large constant $R$, the right-hand side of (C.68) is upper bounded by $1/n^2$.
837 Under the alternative hypothesis, it follows from the Cauchy-Schwartz inequality that

$$\left|\mathbb{E}_{\mathbb{P}_{v^*}}\left[\widetilde{q}_{1,j}(Y,X) - q_{1,j}(Y,X)\right]\right|^2 \leq \mathbb{E}_{\mathbb{P}_{v^*}}\left[\psi^2(Y) \cdot (X_j^2 - 1)^2\right] \cdot \mathbb{P}_{v^*}\left(|X_j| \geq R \cdot \sqrt{\log n}\right)$$

$$\tag{C.69}$$

$$\leq \sqrt{\mathbb{E}_{\mathbb{P}_{v^*}}\left[\psi^4(Y)\right] \cdot \mathbb{E}_{\mathbb{P}_{v^*}}\left[(X_j^2 - 1)^4\right]} \cdot \mathbb{P}_{v^*}\left(|X_j| \geq R \cdot \sqrt{\log n}\right).$$

838 Following from Assumption A.1, it holds that $\mathbb{E}_{\mathbb{P}_v}[\psi^4(Y)]$ is upper bounded under the alternative
839 hypothesis. Meanwhile, it holds that $X_j \sim N(0,1)$ under the alternative hypothesis. Therefore, for a
840 sufficiently large constant $R$, the right-hand side of (C.69) is upper bounded by $1/n^2$.

841 For $q_{2,j}(X,Y)$, we follow similar arguments. By the Cauchy-Schwartz inequality, it holds under the
842 null hypothesis that

$$\left|\mathbb{E}_{\mathbb{P}_0}\left[\widetilde{q}_{2,j}(Y,X) - q_{2,j}(Y,X)\right]\right|^2 \leq \mathbb{E}_{\mathbb{P}_0}[Y^2 X_j^2] \cdot \mathbb{P}_0\left(|X_j| \geq R \cdot \sqrt{\log n}\right). \tag{C.70}$$

843 Note that $Y$ is independent of $X$ and $X_j \sim N(0,1)$ under the null hypothesis. Thus, following from
844 the Gaussian tail bound, it holds for a sufficiently large constant $R$ that

$$\left|\mathbb{E}_{\mathbb{P}_0}\left[\widetilde{q}_{2,j}(Y,X) - q_{2,j}(Y,X)\right]\right|^2 \leq 1/n^2. \tag{C.71}$$

845 Meanwhile, it holds under the alternative hypothesis that

$$\left|\mathbb{E}_{\mathbb{P}_{v^*}}\left[\widetilde{q}_{1,j}(Y,X) - q_{1,j}(Y,X)\right]\right|^2 \leq \mathbb{E}_{\mathbb{P}_{v^*}}[Y^2 X_j^2] \cdot \mathbb{P}_{v^*}\left(|X_j| \geq R \cdot \sqrt{\log n}\right)$$

$$\leq \sqrt{\mathbb{E}_{\mathbb{P}_{v^*}}[Y^2] \cdot \mathbb{E}_{\mathbb{P}_{v^*}}[X_j^4]} \cdot \mathbb{P}_{v^*}\left(|X_j| \geq R \cdot \sqrt{\log n}\right), \tag{C.72}$$

846 where the above inequalities follow from the Cauchy-Schwartz inequality. Also, by Assumption A.1,
847 it holds that $\mathbb{E}_{\mathbb{P}_{v^*}}[Y^4]$ is upper bounded under the alternative hypothesis. Therefore, the right-hand
848 side of (C.72) is upper bounded by $1/n^2$ with a sufficiently large constant $R$. In conclusion, it holds
849 for a sufficiently large constant $R$ that

$$\left|\mathbb{E}_{\mathbb{P}_0}\left[q_{1,j}(Y,X) - \widetilde{q}_{1,j}(Y,X)\right]\right| \vee \left|\mathbb{E}_{\mathbb{P}_v}\left[q_{1,j}(Y,X) - \widetilde{q}_{1,j}(Y,X)\right]\right| \leq 1/n,$$

$$\left|\mathbb{E}_{\mathbb{P}_0}\left[q_{2,j}(Y,X) - \widetilde{q}_{2,j}(Y,X)\right]\right| \vee \left|\mathbb{E}_{\mathbb{P}_v}\left[q_{2,j}(Y,X) - \widetilde{q}_{2,j}(Y,X)\right]\right| \leq 1/n. \tag{C.73}$$

850 It remains to characterize the expectations of $\widetilde{q}_{1,j}(Y,X)$ and $\widetilde{q}_{2,j}(Y,X)$ under the null and alternative
851 hypotheses. Note that under the null hypothesis, it holds that $Y$ is independent of $X$ and $X_j \sim$
852 $N(0,1)$. Therefore, we have $\mathbb{E}_{\mathbb{P}_0}[X_j^2 - 1] = 0$ and $\mathbb{E}_{\mathbb{P}_0}[X_j] = 0$, which imply

$$\mathbb{E}_{\mathbb{P}_0}\left[\widetilde{q}_{1,j}(Y,X)\right] = \mathbb{E}_{\mathbb{P}_0}\left[\psi(Y) \cdot (X_j^2 - 1) \cdot \mathbb{1}\{|\psi(Y)| \leq (R \cdot \log n)^{1/\nu}\}\right] = 0,$$

$$\mathbb{E}_{\mathbb{P}_0}\left[\widetilde{q}_{2,j}(Y,X)\right] = \mathbb{E}_{\mathbb{P}_0}\left[YX_j \cdot \mathbb{1}\{|Y| \leq (R \cdot \log n)^{1/\nu}\}\right] = 0. \tag{C.74}$$

853 Under the alternative hypothesis, it follows from (A.3) and (A.4) that

$$\mathbb{E}_{\mathbb{P}_{v^*}}\left[\psi(Y) \cdot (X_j^2 - 1)\right] \geq \rho^2 \mathrm{v}_j^{*2}, \quad \mathbb{E}_{\mathbb{P}_v}[YX_j] = \alpha\rho\mathrm{v}_j^*, \tag{C.75}$$

where $\mathrm{v}_j^* \in \{-1, 0, 1\}$ is the $j$-th entry of $\mathrm{v}^* \in \bar{\mathcal{G}}(s)$. For the query function $q_{1,j}(Y, X)$, it holds that

$$\rho^2 \mathrm{v}_j^{*2} - \mathbb{E}_{\mathbb{P}_{\mathrm{v}^*}}\big[\widetilde{q}_{1,j}(Y, X)\big] \leq \mathbb{E}_{\mathbb{P}_{\mathrm{v}^*}}\Big[Y^2(X_j^2 - 1) \cdot \mathbb{1}\big\{|Y| > (R \cdot \log n)^{1/\nu}\big\}\Big]$$

$$\leq \sqrt{\mathbb{E}_{\mathbb{P}_{\mathrm{v}^*}}\big[Y^4(X_j^2 - 1)^2\big]} \cdot \sqrt{\mathbb{P}_{\mathrm{v}^*}\big(|Y| > (R \cdot \log n)^{1/\nu}\big)}$$

$$\leq C \cdot \exp(-R \cdot \log n), \tag{C.76}$$

where $C$ is a positive absolute constant and the last inequality follows from Assumption A.1. We fix an index $k$ such that $\mathrm{v}_k^* \neq 0$. Therefore, if $s\rho^2/\sigma^2 = \Omega(\sqrt{s \log d/n})$, it holds for a sufficiently large constant $R$ that

$$\rho^2 - \mathbb{E}_{\mathbb{P}_{\mathrm{v}}}\big[\widetilde{q}_{1,k}(Y, X)\big] \leq \rho^2/4. \tag{C.77}$$

In other words, it holds that $\sup_{j \in [d]} \mathbb{E}_{\mathbb{P}_{\mathrm{v}}}[\widetilde{q}_{1,j}(Y, X)] \geq 3\rho^2/4$. Similarly, we have

$$\rho \mathrm{v}_j^* - \mathbb{E}_{\mathbb{P}_{\mathrm{v}^*}}\big[\widetilde{q}_{1,j}(Y, X)\big] = \mathbb{E}_{\mathbb{P}_{\mathrm{v}^*}}\Big[YX_j \cdot \mathbb{1}\big\{|Y| > (R \cdot \log n)^{1/\nu}\big\}\Big]. \tag{C.78}$$

Meanwhile, if $s\rho^2/\sigma^2 = \Omega(1/\alpha^2 \cdot s \log d/n)$, it holds for a sufficiently large constant $R$ that

$$\Big|\mathbb{E}_{\mathbb{P}_{\mathrm{v}^*}}\Big[YX_j \cdot \mathbb{1}\big\{|Y| > (R \cdot \log n)^{1/\nu}\big\}\Big]\Big|$$

$$\leq \sqrt{\mathbb{E}_{\mathbb{P}_{\mathrm{v}^*}}[Y^2 X_j^2]} \cdot \sqrt{\mathbb{P}_{\mathrm{v}^*}\big(|Y| > (R \cdot \log n)^{1/\nu}\big)} \leq \alpha\rho/4. \tag{C.79}$$

Recall that $\mathrm{v}_j^* \in \{-1, 0, 1\}$ is the $j$-th entry of $\mathrm{v}^* \in \bar{\mathcal{G}}(s)$. Following from (C.78) and (C.79), we obtain that

$$\sup_{j \in [d]}\big|\mathbb{E}_{\mathbb{P}_{\mathrm{v}^*}}\big[\widetilde{q}_{1,j}(Y, X)\big]\big| \geq 3\alpha\rho/4. \tag{C.80}$$

Combining (C.73), (C.74), (C.77), and (C.80), we conclude that for sufficiently large $n$ and constant $R$, it holds that

$$\sup_{j \in [d]} \mathbb{E}_{\mathbb{P}_0}\big[q_{1,j}(Y, X)\big] \leq 1/n, \quad \sup_{j \in [d]} \mathbb{E}_{\mathbb{P}_0}\big[q_{1,j}(Y, X)\big] \leq 1/n. \tag{C.81}$$

Moreover, for sufficiently large $n$ and constant $R$, it holds that

$$\sup_{j \in [d]} \mathbb{E}_{\mathbb{P}_{\mathrm{v}}^*}\big[q_{1,j}(Y, X)\big] \geq \rho^2/2 \text{ if } s\rho^2/\sigma^2 = \Omega(\sqrt{s \log d/n}),$$

$$\sup_{j \in [d]} \mathbb{E}_{\mathbb{P}_{\mathrm{v}}^*}\big[q_{2,j}(Y, X)\big] \geq \alpha\rho/2 \text{ if } s\rho^2/\sigma^2 = \Omega(1/\alpha^2 \cdot s \log d/n), \tag{C.82}$$

which concludes the proof of Lemma B.6. $\qquad\square$

## C.6 Proof of Lemma C.1

*Proof.* In what follows, we show that for $\gamma_n = s\rho^2/\sigma^2 = o(1/\alpha^2 \cdot s \log d/n)$, we have

$$T = \sum_{k=1}^s \binom{s}{k}\Big(\frac{s}{d}\Big)^k \cdot \exp\Big(\frac{4nk\alpha^2\rho^2}{\sigma^2 + s\rho^2}\Big) = o(1).$$

Note that if $\gamma_n = s\rho^2/\sigma^2 = o(1/\alpha^2 \cdot s \log d/n)$, it holds that $\rho^2/(\sigma^2 + s\rho^2) = o(1/\alpha^2 \cdot \log d/n)$, where $\sigma^2$ is a constant. Therefore, we have

$$\Big(\frac{s}{d}\Big)^k \cdot \exp\Big(\frac{4nk\alpha^2\rho^2}{\sigma^2 + s\rho^2}\Big) \leq \Big(\frac{s}{d}\Big)^k \cdot \exp(C \cdot k \log d) = (s \cdot d^{C-1})^k, \tag{C.83}$$

which holds for an arbitrary positive absolute constant $C$ and a sufficiently large $n$, respectively. Meanwhile, note that $s = o(d^{1/2-\delta})$ for an absolute constant $\delta > 0$ and $\binom{s}{k} \leq (es/k)^k$. By (C.83), it holds that

$$\binom{s}{k}\Big(\frac{s}{d}\Big)^k \leq (s^2 e/k \cdot d^{C-1})^k \leq (e/k \cdot d^{C-2\delta})^k. \tag{C.84}$$

Since $C$ is arbitrary, we fix $C \leq \delta$. Following from (C.84), we obtain that

$$T = \sum_{k=1}^s \binom{s}{k}\Big(\frac{s}{d}\Big)^k \cdot \exp\Big(\frac{4nk\alpha^2\rho^2}{\sigma^2 + s\rho^2}\Big) \leq \sum_{k=1}^s (e/k \cdot d^{C-2\delta})^k = o(1),$$

which concludes the proof of Lemma C.1. $\qquad\square$

 **C.7  Proof of Lemma C.2**

*Proof.* In the following proof, we denote by $C$, $C'$, and $C''$ absolute constants, the value of which may vary from lines to lines. We first show that for $\gamma_n = s\rho^2/\sigma^2 = o(\sqrt{s\log d/n})$, it holds that

$$T_1 = \sum_{k=1}^{s} \mathbb{E}_U\left[\left(\frac{2s^2 eQ}{kd}\right)^k\right] = o(1),$$

where $Q = 4\rho^2 U/(\sigma^2 + s\rho^2)$. Recall that $U$ is the sum of $n$ independent Rademacher random variables with Orlicz $\psi_2$-norm equal to one. Therefore, it holds that $\|U\|_{\psi_2} \le C\sqrt{n}$ for an absolute constant $C$. It then follows from the definition of Orlicz $\psi_2$-norm [51] that

$$\mathbb{E}_U\left[|Q|^k\right] \le \left(\frac{\sqrt{k} \cdot 4\rho^2 \cdot \|U\|_{\psi_2}}{\sigma^2 + s\rho^2}\right)^k \le \left(\frac{C\rho^2\sqrt{nk}}{\sigma^2 + s\rho^2}\right)^k. \tag{C.85}$$

Following from (C.85), it holds that

$$T_1 \le \sum_{k=1}^{s} \mathbb{E}_U\left[\frac{2s^2 e|Q|}{kd}\right]^k \le \sum_{k=1}^{s}\left(Ce \cdot \frac{s^2\rho^2\sqrt{n}}{\sigma^2 d\sqrt{k}}\right)^k. \tag{C.86}$$

For $s\rho^2/\sigma^2 = o(\sqrt{s\log d/n})$ and $s = o(d^{1/2-\delta})$, it holds that

$$s\sqrt{n}/d \cdot s\rho^2/\sigma^2 = o(s/d \cdot \sqrt{s\log d}) = o(1). \tag{C.87}$$

Combining (C.86) and (C.87), we obtain that $T_1 = o(1)$. It remains to show that

$$T_2 = \sum_{k=1}^{s}\left(\frac{s^2 e}{kd}\right)^k \cdot \mathbb{E}_U\left[\exp(k|Q|) \cdot \mathbb{1}\{|Q| \ge 1\}\right] = o(1).$$

By integration by parts, we have

$$\mathbb{E}\left[\exp(k|Q|) \cdot \mathbb{1}\{|Q| \ge 1\}\right] = \exp(k) \cdot \mathbb{P}(|Q| \ge 1) + \int_1^\infty k \cdot \exp(tk) \cdot \bar{F}_{|Q|}(t)\mathrm{d}t. \tag{C.88}$$

Note that $Q = 4\rho^2 U/(\sigma^2 + s\rho^2)$ is symmetric and sub-Gaussian with Orlicz $\psi_2$-norm upper bounded by $\|Q\|_{\psi_2} \le C\rho^2\sqrt{n}/(\sigma^2 + s\rho^2)$ for an absolute constant $C$. Thus, it holds that

$$\mathbb{P}(Q \ge t) \le C_1 \cdot \exp\left(-\frac{C_2 \cdot t^2(\sigma^2 + s\rho^2)^2}{\rho^4 n}\right), \tag{C.89}$$

where $C_1$ and $C_2$ are positive absolute constants. Then for the right-hand side of (C.88), it holds that

$$\int_1^\infty k \cdot \exp(tk) \cdot \bar{F}_{|Q|}(t)\mathrm{d}t$$

$$\le C_1 k \cdot \exp\left(\frac{k^2\rho^4 n}{4C_2(\sigma^2 + s\rho^2)^2}\right) \cdot \int_1^\infty \exp\left(-\frac{C_2(\sigma^2 + s\rho^2)^2}{\rho^4 n} \cdot \left(t - \frac{k\rho^4 n}{2C_2(\sigma^2 + s\rho^2)}\right)^2\right)\mathrm{d}t$$

$$\le Ck \cdot \exp\left(\frac{k^2\rho^4 n}{4C_2(\sigma^2 + s\rho^2)^2}\right) \cdot \frac{\rho^2\sqrt{n}}{\sigma^2 + s\rho^2}, \tag{C.90}$$

where $C$ is a positive absolute constant. Meanwhile, for $s\rho^2/\sigma^2 = o(\sqrt{s\log d/n})$, it holds for the right-hand side of (C.90) that

$$\exp\left(\frac{k^2\rho^4 n}{4C_2(\sigma^2 + s\rho^2)^2}\right) \cdot \frac{\rho^2\sqrt{n}}{\sigma^2 + s\rho^2} \le C'\sqrt{\log d/s} \cdot \exp(C_0 k^2 \log d/s), \tag{C.91}$$

which holds for an arbitrary positive absolute constant $C_0$ and a sufficiently large $n$, respectively. Here $C'$ is a positive absolute constant. Combining (C.88), (C.90), and (C.91), we conclude that

$$T_2 = \sum_{k=1}^{s}\left(\frac{s^2 e}{kd}\right)^k \cdot \mathbb{E}_U\left[\exp(k|Q|) \cdot \mathbb{1}\{|Q| \ge 1\}\right]$$

$$\le C_1 \sum_{k=1}^{s}\left(\frac{s^2 e^2}{kd}\right)^k + C''\sqrt{\log d/s} \cdot \sum_{k=1}^{s} k \cdot \left(\frac{s^2 e^2}{kd} \cdot \exp(C_0 k \log d/s)\right)^k. \tag{C.92}$$

Note that $s = o(d^{1/2-\delta})$ for a positive absolute constant $\delta$. Thus, it holds that $s^2 e^2/(kd) = o(1)$ for $0 \leq k \leq s$, which implies that

$$\sum_{k=1}^{s} \left( \frac{s^2 e^2}{kd} \right)^k = o(1). \tag{C.93}$$

Meanwhile, it holds for any $1 \leq k \leq s$ that

$$\frac{s^2 e^2}{kd} \cdot \exp(C_0 k \log d/s) \leq \frac{s^2 e^2}{kd} \cdot \exp(C_0 \log d) \leq e^2/d^{2\delta - C_0}. \tag{C.94}$$

Since $C_0$ is arbitrary, we fix $C_0 > 2\delta$. It then holds for a positive absolute constant $C$ that

$$\sqrt{\log d/s} \cdot \sum_{k=1}^{s} k \cdot \left( \frac{s^2 e^2}{kd} \cdot \exp(C_0 k \log d/s) \right)^k \leq C \cdot \sqrt{\log d/s} \cdot e^2/d^{2\delta - C_0} = o(1). \tag{C.95}$$

Combining (C.92), (C.93), and (C.95), we obtain that $T_2 = o(1)$, which concludes the proof of Lemma C.2. $\qquad\square$

# D   Upper Bounds for General Cases

In this section, we characterize the upper bounds for the hypothesis testing problem in (A.1) under the general setting. In specific, we consider the hypothesis testing problem that takes the form

$$H_0 : Y = \epsilon_0 \quad \text{versus} \quad H_1 : Y = \begin{cases} f_1(X^\top \beta^*) + \epsilon, & \text{with probability } \alpha, \\ f_2(X^\top \beta^*) + \epsilon, & \text{with probability } 1 - \alpha. \end{cases} \tag{D.1}$$

Here $\epsilon$ is a Gaussian noise with variance $\sigma^2$, $\epsilon_0$ is a noise such that the variances of $Y$ under the null and alternative hypotheses are the same. Besides, $f_1 \in \mathcal{C}_1 \cap \mathcal{C}(\psi)$ and $f_2 \in \mathcal{C}_2 \cap \mathcal{C}(\psi)$ are two unknown link functions, where $\mathcal{C}_1(\psi)$, $\mathcal{C}_2(\psi)$, and $\mathcal{C}(\psi)$ are defined in (2.4) and (2.5). Meanwhile, we set $X \sim N(0, I_d)$ and

$$(\beta^*, \sigma) \in \mathcal{G}_1(s, \gamma_n) = \left\{ (\beta^*, \sigma) \in \mathbb{R}^{d+1} : \|\beta^*\|_0 = s, \kappa(\beta^*, \sigma) \geq \gamma_n \right\} \tag{D.2}$$

under the alternative hypothesis, where $\kappa(\beta^*, \sigma) = \|\beta^*\|_2^2/\sigma^2$ is the SNR. We further denote by

$$\mathcal{H}(s, \gamma_n) = \left\{ \beta^* \in \mathbb{R}^d : \|\beta^*\|_2^2/\sigma^2 = s\rho^2/\sigma^2 \geq \gamma_n, \|\beta^*\|_0 = s \right\}. \tag{D.3}$$

We denote by $Z = (Y, X)$ and $\mathbb{P}_0$, $\mathbb{P}_{\beta^*}$ be the distributions of $Z$ under the null and alternative hypotheses, respectively. We assume that the Assumption A.1 holds. We denote by

$$\mathcal{V}(s) = \left\{ \mathcal{S} \in [d] : |\mathcal{S}| = s \right\}$$

the class of index sets. For each index set $\mathcal{S} \in \mathcal{V}(s)$, we denote by $\mathcal{B}(\mathcal{S})$ the $s$-sparse unit sphere that is supported on the index set $\mathcal{S}$. We further denote by $\mathcal{N}(\epsilon, \mathcal{S}) \subseteq \mathcal{B}(\mathcal{S})$ the minimum $\epsilon$-covering of the $s$-sparse unit sphere $\mathcal{B}(\mathcal{S})$. In other words, it holds for any $u \in \mathcal{B}(\mathcal{S})$ that $\|u - v\|_2 \leq \epsilon$ for some $v \in \mathcal{N}(\epsilon, \mathcal{S})$. Meanwhile, $\mathcal{N}(\epsilon, \mathcal{S})$ attains the smallest cardinality among the sets that have such a property. It then holds that

$$|\mathcal{N}(\epsilon, \mathcal{S})| \leq C_0 \cdot (1 + 2/\epsilon)^s, \tag{D.4}$$

where $C_0$ is a positive absolute constant. We define

$$\mathcal{N}(\epsilon) = \bigcup_{\mathcal{S} \in \mathcal{V}(s)} \mathcal{N}(\epsilon, \mathcal{S}). \tag{D.5}$$

Therefore, it holds that

$$|\mathcal{N}(\epsilon)| \leq C_0 \cdot (1 + 2/\epsilon)^s \cdot \binom{d}{s}. \tag{D.6}$$

In what follows, we construct test functions based on $v \in \mathcal{N}(1/2)$. We introduce the following query functions for $v \in \mathcal{N}(1/2)$,

$$q_{1,v}(Y, X) = \psi(Y) \cdot \left[ (v^\top X)^2 - 1 \right] \cdot \mathbb{1}\left\{ |\psi(Y)| \leq (R \log n)^{1/\nu} \right\} \cdot \mathbb{1}\left\{ |v^\top X| \leq R \cdot \sqrt{\log n} \right\},$$

$$q_{2,v}(Y, X) = Y \cdot (v^\top X) \cdot \mathbb{1}\left\{ |Y| \leq (R \log n)^{1/\nu} \right\} \cdot \mathbb{1}\left\{ |v^\top X| \leq R \cdot \sqrt{\log n} \right\}. \tag{D.7}$$

We denote by $\bar{Z}_{1,v}$ and $\bar{Z}_{2,v}$ the responses of the statistical oracle to query functions $q_{1,v}$ and $q_{2,v}$, as defined in Definition 2.3. We define the test functions $\phi_1$ and $\phi_2$ as

$$\phi_1 = \mathbb{1}\left\{ \sup_{v \in \bar{\mathcal{G}}(s)} \bar{Z}_{1,v} \geq \tau_1 \right\}, \quad \phi_2 = \mathbb{1}\left\{ \sup_{v \in \bar{\mathcal{G}}(s)} \bar{Z}_{2,v} \geq \tau_2 \right\}, \tag{D.8}$$

where we set the thresholds $\tau_1$ and $\tau_2$ to be

$$\tau_1 = CR^{2+1/\nu} \cdot (\log n)^{1+1/\nu} \cdot \sqrt{\frac{s \log d}{n}}, \quad \tau_2 = C'R^{1+1/\nu} \cdot (\log n)^{1/2+1/\nu} \cdot \sqrt{\frac{s \log d}{n}}, \quad \text{(D.9)}$$

where $C$ and $C'$ are positive absolute constants that will be specified in §D.1. We define the test function as $\phi = \phi_1 \vee \phi_2$. Following from (D.6), the capacity of $\mathcal{Q}_\phi$ is upper bounded as follows,

$$|\mathcal{Q}_\phi| \leq 2C_0 \cdot 5^s \cdot \binom{d}{s}. \quad \text{(D.10)}$$

The following theorem characterizes an upper bound for the minimax separation rate by quantifying the SNR for $\phi$ to be asymptotically powerful.

**Theorem D.1.** We consider the hypothesis testing problem in (D.1) under Assumption A.1. For

$$\gamma_n = \Omega\left((\log n)^{1+1/\nu} \cdot \sqrt{\frac{s \log d}{n}} \bigwedge \frac{(\log n)^{1+2/\nu}}{\alpha^2} \cdot \frac{s \log d}{n}\right), \quad \text{(D.11)}$$

it holds that $R_n(\phi; \mathcal{G}_0, \mathcal{G}_1) = O(1/d)$. In other words, $\phi$ is asymptotically powerful.

*Proof.* See §D.1 for a detailed proof. $\qquad \square$

To construct a computationally tractable test, we define query functions as follows,

$$q_{1,j}(Y,X) = \psi(Y) \cdot (X_j^2 - 1) \cdot \mathbb{1}\{|\psi(Y)| \leq (R \log n)^{1/\nu}\} \cdot \mathbb{1}\{|X_j| \leq R\sqrt{\log n}\}, \quad j \in [d]$$

$$q_{2,j}(Y,X) = Y \cdot X_j \cdot \mathbb{1}\{|Y| \leq (R \log n)^{1/\nu}\} \cdot \mathbb{1}\{|X_j| \leq R\sqrt{\log n}\}, \quad j \in [d]. \quad \text{(D.12)}$$

We denote by $\bar{Z}_{1,j}$ and $\bar{Z}_{2,j}$ the responses of the statistical oracle to the query functions $q_{1,j}$ and $q_{2,j}$, as defined in Definition 2.3 . We define the test functions $\widetilde{\phi}_1$ and $\widetilde{\phi}_2$ as

$$\widetilde{\phi}_1 = \mathbb{1}\left\{\sup_{j \in [d]} \bar{Z}_{1,j} \geq \widetilde{\tau}_1\right\}, \quad \widetilde{\phi}_2 = \mathbb{1}\left\{\sup_{j \in [d]} \bar{Z}_{2,j} \geq \widetilde{\tau}_2\right\} \bigvee \mathbb{1}\left\{\inf_{j \in [d]} \bar{Z}_{2,j} \leq -\widetilde{\tau}_2\right\}, \quad \text{(D.13)}$$

where we set the thresholds $\widetilde{\tau}_1$ and $\widetilde{\tau}_2$ to be

$$\widetilde{\tau}_1 = CR^{2+1/\nu}(\log n)^{1+1/\nu} \cdot \sqrt{\frac{\log d}{n}}, \quad \widetilde{\tau}_2 = C'R^{1+1/\nu}(\log n)^{1/2+1/\nu} \cdot \sqrt{\frac{\log d}{n}}. \quad \text{(D.14)}$$

We define the test function $\widetilde{\phi} = \widetilde{\phi}_1 \vee \widetilde{\phi}_2$. Therefore, the test function $\widetilde{\phi}$ is with capacity of query functions $|\mathcal{Q}_{\widetilde{\phi}}| = 2d$. The following theorem holds, which characterizes the minimum SNR required for the test function $\widetilde{\phi}$ to be asymptotically powerful.

**Theorem D.2.** We consider the hypothesis testing problem in (D.1) under Assumption A.1. For

$$\gamma_n = \Omega\left((\log n)^{1+1/\nu} \cdot \sqrt{\frac{s^2 \log d}{n}} \bigwedge \frac{(\log n)^{1+2/\nu}}{\alpha^2} \cdot \frac{s \log d}{n}\right), \quad \text{(D.15)}$$

it holds that $\bar{R}_n(\widetilde{\phi}; \mathcal{G}_0, \mathcal{G}_1) = O(1/d)$. In other words, $\widetilde{\phi}$ is asymptotically powerful.

*Proof.* See §D.2 for a detailed proof. $\qquad \square$

## D.1 Proof of Theorem D.1

*Proof.* The proof is similar to that of Theorem A.2 in §B.3. Recall that we denote by $\mathbb{P}_0$ and $\mathbb{P}_{\beta^*}$ the distributions of $Z = (Y, X)$ under the null and alternative hypotheses, respectively. The following lemma holds, which characterizes the expection of $q_{1,\mathrm{v}}$ and $q_{2,\mathrm{v}}$ under the null and alternative hypotheses, respectively.

**Lemma D.3.** For any $\mathrm{v} \in \mathcal{N}(1/2)$, $\beta^* \in \mathcal{H}(s, \gamma_n)$, and

$$\gamma_n = \Omega\left((\log n)^{1+1/\nu} \cdot \sqrt{\frac{s \log d}{n}} \bigwedge \frac{(\log n)^{1+2/\nu}}{\alpha^2} \cdot \frac{s \log d}{n}\right),$$

it holds that

$$\mathbb{E}_{\mathbb{P}_0}\left[q_{1,\mathrm{v}}(Y,X)\right] \leq 1/n, \quad \mathbb{E}_{\mathbb{P}_0}\left[q_{2,\mathrm{v}}(Y,X)\right] \leq 1/n. \quad \text{(D.16)}$$

In addition, it holds that

$$\sup_{v \in \mathcal{N}(1/2)} \mathbb{E}_{\mathbb{P}_{\beta^*}} \big[ q_{1,v}(Y, X) \big] \geq s\rho^2/2 \text{ if } \gamma_n = \Omega \bigg( (\log n)^{1+1/\nu} \cdot \sqrt{\frac{s \log d}{n}} \bigg),$$

$$\sup_{v \in \mathcal{N}(1/2)} \mathbb{E}_{\mathbb{P}_{\beta^*}} \big[ q_{2,v}(Y, X) \big] \geq \sqrt{\alpha^2 s\rho^2}/2 \text{ if } \gamma_n = \Omega \bigg( \frac{(\log n)^{1+2/\nu}}{\alpha^2} \cdot \frac{s \log d}{n} \bigg). \tag{D.17}$$

*Proof.* See §D.3 for a detailed proof. $\qquad\square$

It now suffices to upper bound the risk of $\phi = \phi_1 \vee \phi_2$, where $\phi_1$ and $\phi_2$ are defined in (D.8). Recall that we define the threshold $\tau_1$ and $\tau_2$ as

$$\tau_1 = CR^{2+1/\nu} \cdot (\log n)^{1+1/\nu} \cdot \sqrt{\frac{s \log d}{n}}, \quad \tau_2 = C'R^{1+1/\nu} \cdot (\log n)^{1/2+1/\nu} \cdot \sqrt{\frac{s \log d}{n}}, \tag{D.18}$$

where $C$ and $C'$ are positive absolute constants. Note that for the test function $\phi$, the capacity of query functions is upper bounded in (D.10). Therefore, following from (2.12) with $\xi = 1/d$, it holds for a sufficiently large $n$ that

$$\tau_{q_{1,v}} \leq C_1 R^{2+1/\nu} (\log n)^{1/2+1/\nu} \cdot \sqrt{\frac{s \log d}{n}},$$

$$\tau_{q_{2,v}} \leq C_2 R^{1+1/\nu} (\log n)^{1/2+1/\nu} \cdot \sqrt{\frac{s \log d}{n}}, \tag{D.19}$$

where $\tau_{q_{1,v}}$ and $\tau_{q_{2,v}}$ are the tolerance parameters of $q_{1,v}$ and $q_{2,v}$ defined in Definition 2.3, and $C_1$, $C_2$ are positive absolute constants. We fix $C$ and $C'$ in (D.18) such that $\tau_1 \geq \tau_{q_{1,v}} + 1/n$ and $\tau_2 \geq \tau_{q_{2,v}} + 1/n$. The rest of the proof then follows a similar argument in §B.3. Recall that we denote by $\bar{Z}_{1,v}$ and $\bar{Z}_{2,v}$ the responses of the statistical oracle to the query functions $q_{1,v}$ and $q_{2,v}$. We denote by $\bar{\mathbb{P}}_0$ and $\bar{\mathbb{P}}_{\beta^*}$ the distributions of response of the statistical oracle to the query functions when the true distribution of the data is $\mathbb{P}_0$ and $\mathbb{P}_{\beta^*}$. Following from Lemma D.3, it holds for any $v \in \mathcal{N}(1/2)$ that

$$\bar{\mathbb{P}}_0 \big( \bar{Z}_{i,v} \geq \tau_i \big) \leq \bar{\mathbb{P}}_0 \Big( \big| \bar{Z}_{i,v} - \mathbb{E}_{\mathbb{P}_0} \big[ q_{i,v}(Y, X) \big] \big| \geq \tau_{q_{i,v}} \Big), \quad i \in \{1, 2\}.$$

Therefore, following from (2.11) with $\xi = 1/d$, we obtain

$$\bar{\mathbb{P}}_0(\phi_i = 1) = \bar{\mathbb{P}}_0 \bigg( \sup_{v \in \mathcal{N}(1/2)} \bar{Z}_{i,v} > \tau_i \bigg)$$

$$\leq \bar{\mathbb{P}}_0 \bigg( \bigcup_{v \in \mathcal{N}(1/2)} \Big\{ \big| \bar{Z}_{i,v} - \mathbb{E}_{\mathbb{P}_0} \big[ q_{i,v}(Y, X) \big] \big| > \tau_{q_{i,v}} \Big\} \bigg) \leq 2/d. \tag{D.20}$$

Recall that we define $\phi = \phi_1 \vee \phi_2$. Then it holds that
$$\bar{\mathbb{P}}_0(\phi = 1) \leq \bar{\mathbb{P}}_0(\phi_1 = 1) + \bar{\mathbb{P}}_0(\phi_2 = 1) = 4/d, \tag{D.21}$$

which is an upper bound of the type-I error of $\phi$. It now suffices to upper bound the type-II error of $\phi$. If (D.11) holds, we obtain that either $s\rho^2/4 \geq \tau_1$ or $\sqrt{\alpha^2 s\rho^2}/4 \geq \tau_2$ for a sufficiently large $n$. We denote by
$$v^* \in \underset{v \in \mathcal{N}(1/2)}{\operatorname{argmax}} \mathbb{E}_{\mathbb{P}_{\beta^*}} \big[ q_{1,v}(Y, X) \big], \quad u^* \in \underset{v \in \mathcal{N}(1/2)}{\operatorname{argmax}} \mathbb{E}_{\mathbb{P}_{\beta^*}} \big[ q_{2,v}(Y, X) \big].$$

If it holds that $s\rho^2/4 \geq \tau_1$, then following from Lemma D.3, we obtain that

$$\bar{\mathbb{P}}_{\beta^*}(\phi_1 = 0) = \bar{\mathbb{P}}_{\beta^*} \bigg( \sup_{v \in \mathcal{N}(1/2)} \bar{Z}_{1,v} < \tau_1 \bigg) \leq \bar{\mathbb{P}}_{\beta^*}(\bar{Z}_{1,v^*} < \tau_1)$$

$$\leq \bar{\mathbb{P}}_{\beta^*} \Big( \bar{Z}_{1,v^*} < \mathbb{E}_{\mathbb{P}_{\beta^*}} \big[ q_{1,v^*}(Y, X) \big] - \tau_1 \Big) \tag{D.22}$$

$$\leq \bar{\mathbb{P}}_{\beta^*} \Big( \big| \bar{Z}_{1,v^*} - \mathbb{E}_{\mathbb{P}_{\beta^*}} \big[ q_{1,v^*}(Y, X) \big] \big| > \tau_{q_{1,v^*}} \Big), \tag{D.23}$$

where the last inequality follows from the fact that $\tau_1 > \tau_{q_{1,v^*}}$. Therefore, following from (2.11) with $\xi = 1/d$, we obtain that the right-hand side of (D.22) is upper bounded by $2/d$. Similarly, if it

holds that $\sqrt{\alpha^2 s \rho^2}/4 \geq \tau_2$, we obtain

$$\bar{\mathbb{P}}_{\beta^*}(\phi_2 = 0) = \bar{\mathbb{P}}_{\beta^*}\left(\sup_{v \in \mathcal{N}(1/2)} \bar{Z}_{1,v} < \tau_1\right) \leq \bar{\mathbb{P}}_{\beta^*}(\bar{Z}_{2,u^*} < \tau_2)$$

$$\leq \bar{\mathbb{P}}_{\beta^*}\left(\left|\bar{Z}_{2,u^*} - \mathbb{E}_{\mathbb{P}_{\beta^*}}[q_{2,u^*}(Y, X)]\right| > \tau_{q_{2,u^*}}\right), \tag{D.24}$$

where the last inequality follows from the fact that $\tau_1 > \tau_{q_{1,u^*}}$. Therefore, following from (2.11) with $\xi = 1/d$, we obtain that the right-hand side of (D.24) is upper bounded by $2/d$. Note that (D.22) and (D.24) holds for all $(\beta^*, \sigma) \in \mathcal{G}_1(s, \gamma_n)$ if (D.11) holds. Therefore, we conclude that

$$\sup_{(\beta^*,\sigma) \in \mathcal{G}_1} \bar{\mathbb{P}}_{\beta^*}(\phi = 0) \leq \sup_{(\beta^*,\sigma) \in \mathcal{G}_1} \{\bar{\mathbb{P}}_{\beta^*}(\phi_1 = 0) \wedge \bar{\mathbb{P}}_{\beta^*}(\phi_2 = 0)\} \leq 2/d. \tag{D.25}$$

Combining (D.21) and (D.25), we obtain that if (D.11) holds, the risk of $\phi$ is $O(1/d)$, which concludes the proof. □

## D.2  Proof of Theorem D.2

*Proof.* The proof is similar to that of Theorem A.3 in §B.4. Recall that we denote by $\mathbb{P}_0$ and $\mathbb{P}_{\beta^*}$ the distributions of $Z = (Y, X)$ under the null and alternative hypotheses, respectively. The following lemma holds, which characterizes the expection of $q_{1,j}(Y, X)$ and $q_{2,j}(Y, X)$ under the null and alternative hypotheses, respectively.

**Lemma D.4.** For any $\beta^* \in \mathcal{H}(s, \gamma_n)$ and

$$\gamma_n = \Omega\left((\log n)^{1+1/\nu} \cdot \sqrt{\frac{s^2 \log d}{n}} \bigwedge \frac{(\log n)^{1+2/\nu}}{\alpha^2} \cdot \frac{s \log d}{n}\right),$$

it holds that

$$\sup_{j \in [d]} \mathbb{E}_{\mathbb{P}_0}[q_{1,j}(Y, X)] \leq 1/n, \quad \sup_{j \in [d]} \mathbb{E}_{\mathbb{P}_0}[q_{2,j}(Y, X)] \leq 1/n. \tag{D.26}$$

In addition, it holds that

$$\sup_{j \in [d]} \mathbb{E}_{\mathbb{P}_{\beta^*}}[q_{1,j}(Y, X)] \geq \rho^2/2 \text{ if } \gamma_n = \Omega\left((\log n)^{1+1/\nu} \cdot \sqrt{\frac{s^2 \log d}{n}}\right),$$

$$\sup_{j \in [d]} \left|\mathbb{E}_{\mathbb{P}_{\beta^*}}[q_{2,j}(Y, X)]\right| \geq \alpha\rho/2 \text{ if } \gamma_n = \Omega\left(\frac{(\log n)^{1+2/\nu}}{\alpha^2} \cdot \frac{s \log d}{n}\right). \tag{D.27}$$

*Proof.* See §D.4 for a detailed proof. □

In what follows, we upper bound the risk of $\widetilde{\phi} = \widetilde{\phi}_1 \vee \widetilde{\phi}_2$ where $\widetilde{\phi}_1$ and $\widetilde{\phi}_2$ are defined in (D.13). Recall that we define the threshold $\widetilde{\tau}_1$ and $\widetilde{\tau}_2$ as

$$\widetilde{\tau}_1 = CR^{2+1/\nu}(\log n)^{1+1/\nu} \cdot \sqrt{\frac{\log d}{n}}, \quad \widetilde{\tau}_2 = C'R^{1+1/\nu}(\log n)^{1/2+1/\nu} \cdot \sqrt{\frac{\log d}{n}}, \tag{D.28}$$

where $C$ and $C'$ are absolute constants. Note that for $\widetilde{\phi}$, the capacity of query functions is $2d$. Therefore, following from (2.12) with $\xi = 1/d$, it holds for a sufficiently large $n$ that

$$\tau_{q_{1,j}} \leq C_1 R^{2+1/\nu}(\log n)^{1/2+1/\nu} \cdot \sqrt{\frac{\log d}{n}}, \quad \tau_{q_{2,j}} \leq C_2 R^{1+1/\nu}(\log n)^{1/2+1/\nu} \cdot \sqrt{\frac{\log d}{n}}, \tag{D.29}$$

where $C_1$ and $C_2$ are positive absolute constants. We fix $C$ and $C'$ in (D.28) such that $\widetilde{\tau}_1 > \tau_{q_{1,j}} + 1/n$ and $\tau_2 > \tau_{1_{2,j}} + 1/n$ for a sufficiently large $n$. Recall that we denote by $\bar{Z}_{1,j}$ and $\bar{Z}_{2,j}$ the responses of the statistical oracle to the query functions $q_{1,j}$ and $q_{2,j}$. We denote by $\bar{\mathbb{P}}_0$ and $\bar{\mathbb{P}}_{\beta^*}$ the distributions of response of the statistical oracle to the query functions when the true distribution of the data is $\mathbb{P}_0$ and $\mathbb{P}_{\beta^*}$. Following from Lemma D.3, it holds for $j \in [d]$ and $i \in \{1, 2\}$ that

$$\bar{\mathbb{P}}_0(\bar{Z}_{i,j} \geq \widetilde{\tau}_1) \leq \bar{\mathbb{P}}_0\left(\left|\bar{Z}_{i,j} - \mathbb{E}_{\mathbb{P}_0}[q_{i,j}(Y, X)]\right| \geq \tau_{q_{i,j}}\right). \tag{D.30}$$

Therefore, following from (2.11) with $\xi = 1/d$, it holds for $i \in \{1, 2\}$ that

$$\bar{\mathbb{P}}_0(\widetilde{\phi}_i = 1) = \bar{\mathbb{P}}_0\left(\sup_{j \in [d]} \bar{Z}_{i,j} > \widetilde{\tau}_i\right)$$

$$\leq \bar{\mathbb{P}}_0\left(\bigcup_{j \in [d]} \left\{\left|\bar{Z}_{i,j} - \mathbb{E}_{\mathbb{P}_0}\left[q_{i,j}(Y, X)\right]\right| > \tau_{q_{i,j}}\right\}\right) \leq 2/d, \tag{D.31}$$

which further shows that

$$\bar{\mathbb{P}}_0(\widetilde{\phi} = 1) \leq \bar{\mathbb{P}}_0(\widetilde{\phi}_1 = 1) + \bar{\mathbb{P}}_0(\widetilde{\phi}_2 = 1) \leq 4/d. \tag{D.32}$$

In other words, it holds that the type-I error of $\widetilde{\phi}$ is asymptotically upper bounded by $4/d$. It remains to upper bound the type-II error of $\widetilde{\phi}$. Note that if (D.15) holds, it holds that either $\rho^2/4 \geq \widetilde{\tau}_1$ or $\alpha\rho/4 \geq \widetilde{\tau}_2$ for a sufficiently large $n$. We denote by

$$j^* \in \operatorname*{argmax}_{j \in [d]} \mathbb{E}_{\mathbb{P}_{\beta^*}}\left[q_{1,j}(Y, X)\right], \quad k^* \in \operatorname*{argmax}_{j \in [d]}\left|\mathbb{E}_{\mathbb{P}_{\beta^*}}\left[q_{2,j}(Y, X)\right]\right|.$$

If it holds that $\rho^2/4 \geq \widetilde{\tau}_1$, following from Lemma D.4, we obtain that

$$\bar{\mathbb{P}}_{\beta^*}(\widetilde{\phi}_1 = 0) \leq \bar{\mathbb{P}}_{\beta^*}\left(\sup_{j \in [d]} \bar{Z}_{1,j} < \widetilde{\tau}_2\right) \leq \bar{\mathbb{P}}_{\beta^*}(\bar{Z}_{1,j^*} < \widetilde{\tau}_1)$$

$$\leq \bar{\mathbb{P}}_{\beta^*}\left(\bar{Z}_{1,j^*} < \mathbb{E}_{\mathbb{P}_{\beta^*}}\left[q_{1,j^*}(Y, X)\right] - \widetilde{\tau}_1\right)$$

$$\leq \bar{\mathbb{P}}_{\beta^*}\left(\left|\bar{Z}_{2,j^*} - \mathbb{E}_{\mathbb{P}_v}\left[q_{2,j^*}(Y, X)\right]\right| > \tau_{q_{2,j^*}}\right) \leq 2/d, \tag{D.33}$$

where the fourth inequality follows from the fact that $\widetilde{\tau}_1 > \tau_{q_{1,j^*}}$, and the last inequality following from (2.11) with $\xi = 1/d$. If it holds that $\alpha\rho/4 \geq \widetilde{\tau}_2$, following from Lemma D.4, we obtain that either $\mathbb{E}_{\mathbb{P}_{\beta^*}}\left[q_{2,k^*}(Y, X)\right] \geq \alpha\rho/2$ or $\mathbb{E}_{\mathbb{P}_{\beta^*}}\left[q_{2,k^*}(Y, X)\right] \leq -\alpha\rho/2$. If $\mathbb{E}_{\mathbb{P}_{\beta^*}}\left[q_{2,k^*}(Y, X)\right] \geq \alpha\rho/2$, we obtain that

$$\bar{\mathbb{P}}_{\beta^*}(\widetilde{\phi}_2 = 0) \leq \bar{\mathbb{P}}_{\beta^*}\left(\sup_{j \in [d]} \bar{Z}_{2,j} < \widetilde{\tau}_2\right) \leq \bar{\mathbb{P}}_{\beta^*}(\bar{Z}_{2,k^*} < \widetilde{\tau}_2)$$

$$\leq \bar{\mathbb{P}}_{\beta^*}\left(\bar{Z}_{2,k^*} < \mathbb{E}_{\mathbb{P}_{\beta^*}}\left[q_{2,k^*}(Y, X)\right] - \widetilde{\tau}_2\right)$$

$$\leq \bar{\mathbb{P}}_{\beta^*}\left(\left|\bar{Z}_{2,k^*} - \mathbb{E}_{\mathbb{P}_v}\left[q_{2,k^*}(Y, X)\right]\right| > \tau_{q_{2,k^*}}\right) \leq 2/d, \tag{D.34}$$

where the fourth inequality follows from the fact that $\widetilde{\tau}_2 > \tau_{q_{2,k^*}}$, and the last inequality follows from (2.11) with $\xi = 1/d$. If it holds that $\mathbb{E}_{\mathbb{P}_{\beta^*}}\left[q_{2,k^*}(Y, X)\right] \leq -\alpha\rho/2$, we obtain that

$$\bar{\mathbb{P}}_{\beta^*}(\widetilde{\phi}_2 = 0) \leq \bar{\mathbb{P}}_{\beta^*}\left(\inf_{j \in [d]} \bar{Z}_{2,j} > -\widetilde{\tau}_2\right) \leq \bar{\mathbb{P}}_{\beta^*}(\bar{Z}_{2,k^*} > -\widetilde{\tau}_2)$$

$$\leq \bar{\mathbb{P}}_{\beta^*}\left(\bar{Z}_{2,k^*} > \mathbb{E}_{\mathbb{P}_{\beta^*}}\left[q_{2,k^*}(Y, X)\right] + \widetilde{\tau}_2\right)$$

$$\leq \bar{\mathbb{P}}_{\beta^*}\left(\left|\bar{Z}_{2,k^*} - \mathbb{E}_{\mathbb{P}^*_\beta}\left[q_{2,k^*}(Y, X)\right]\right| > \tau_{q_{2,k^*}}\right) \leq 2/d, \tag{D.35}$$

where the fourth inequality follows from the fact that $\widetilde{\tau}_2 > \tau_{q_{2,k^*}}$, and the last inequality follows from (2.11) with $\xi = 1/d$. Note that (D.33), (D.34), and (D.35) holds for all $(\beta^*, \sigma) \in \mathcal{G}_1(s, \gamma_n)$ if (D.15) holds. Therefore, we obtain that

$$\sup_{(\beta^*, \sigma) \in \mathcal{G}_1} \bar{\mathbb{P}}_{\beta^*}(\widetilde{\phi} = 0) \leq \sup_{(\beta^*, \sigma) \in \mathcal{G}_1} \left\{\bar{\mathbb{P}}_{\beta^*}(\widetilde{\phi}_1 = 0) \wedge \bar{\mathbb{P}}_{\beta^*}(\widetilde{\phi}_2 = 0)\right\} \leq 2/d. \tag{D.36}$$

Combining (D.32) and (D.36), we obtain that if (D.15) holds, the risk of $\widetilde{\phi}$ is $O(1/d)$, which concludes the proof of Theorem D.2. $\qquad\square$

## D.3 Proof of Lemma D.3

*Proof.* In the following proof, we denote by $C$ and $C'$ absolute constants, the value of which may vary from lines to lines. We define the following query functions,

$$\widetilde{q}_{1,v}(Y, X) = \psi(Y) \cdot \left[(v^\top X)^2 - 1\right] \cdot \mathbb{1}\left\{|\psi(Y)| \leq (R \cdot \log n)^{1/\nu}\right\}, \quad v \in \bar{\mathcal{G}}(s),$$

$$\widetilde{q}_{2,v}(Y, X) = Y \cdot (v^\top X) \cdot \mathbb{1}\left\{|Y| \leq (R \cdot \log n)^{1/\nu}\right\}, \quad v \in \bar{\mathcal{G}}(s). \tag{D.37}$$

Following from (D.7) and (D.37), we conclude that

$$\widetilde{q}_{1,\mathrm{v}} - q_{1,\mathrm{v}} = \psi(Y) \cdot \left[(\mathrm{v}^\top X)^2 - 1\right] \cdot \mathbb{1}\left\{|\psi(Y)| \leq (R \cdot \log n)^{1/\nu}\right\} \cdot \mathbb{1}\left\{|\mathrm{v}^\top X| > R \cdot \sqrt{\log n}\right\},$$

$$\widetilde{q}_{2,\mathrm{v}} - q_{2,\mathrm{v}} = Y \cdot (\mathrm{v}^\top X) \cdot \mathbb{1}\left\{|Y| \leq (R \cdot \log n)^{1/\nu}\right\} \cdot \mathbb{1}\left\{|\mathrm{v}^\top X| > R \cdot \sqrt{\log n}\right\}. \qquad (D.38)$$

Therefore, following from the Cauchy-Schwartz inequality, we obtain from (D.38) that

$$\left|\mathbb{E}_{\mathbb{P}_0}\left[\widetilde{q}_{1,\mathrm{v}}(Y,X) - q_{1,\mathrm{v}}(Y,X)\right]\right|^2$$

$$\leq \mathbb{E}_{\mathbb{P}_0}\left[\psi^2(Y) \cdot \left[(\mathrm{v}^\top X)^2 - 1\right]^2\right] \cdot \mathbb{P}_0\left(|\mathrm{v}^\top X| \geq R \cdot \sqrt{\log n}\right). \qquad (D.39)$$

Further note that under the null hypothesis, $Y$ is independent of $X$ and $X \sim N(0, I_d)$. Therefore, for $\mathrm{v} \in \mathcal{N}(1/2)$, it holds that $\mathrm{v}^\top X \sim N(0,1)$. Meanwhile, following from Assumption A.1, $Y$ has bounded fourth moment. Therefore, we obtain from (D.39) and the tail bound of standard Gaussian distribution in (C.54) that

$$\left|\mathbb{E}_{\mathbb{P}_0}\left[\widetilde{q}_{1,\mathrm{v}}(Y,X) - q_{1,\mathrm{v}}(Y,X)\right]\right|^2 \leq C \cdot \exp(-R^2 \log n), \qquad (D.40)$$

where $C$ is a positive absolute constant. Similarly, it holds under the alternative hypothesis that

$$\left|\mathbb{E}_{\mathbb{P}_\beta^*}\left[\widetilde{q}_{1,\mathrm{v}}(Y,X) - q_{1,\mathrm{v}}(Y,X)\right]\right|^2$$

$$\leq \mathbb{E}_{\mathbb{P}_\beta^*}\left[\psi^2(Y) \cdot \left[(\mathrm{v}^\top X)^2 - 1\right]^2\right] \cdot \mathbb{P}_0\left(|\mathrm{v}^\top X| \geq R \cdot \sqrt{\log n}\right)$$

$$\leq \left(\mathbb{E}_{\mathbb{P}_\beta^*}\left[\psi^4(Y)\right] \cdot \mathbb{E}_{\mathbb{P}_\beta^*}\left[\left[(\mathrm{v}^\top X)^2 - 1\right]^4\right]\right)^{1/2} \cdot \mathbb{P}_0\left(|\mathrm{v}^\top X| \geq R \cdot \sqrt{\log n}\right), \qquad (D.41)$$

where the above inequalities follow from the Cauchy-Schwartz inequality. Then following from Assumption A.1 and the fact that $X \sim N(0, I_d)$ under the alternative hypothesis, we conclude that

$$\left|\mathbb{E}_{\mathbb{P}_\beta^*}\left[\widetilde{q}_{1,\mathrm{v}}(Y,X) - q_{1,\mathrm{v}}(Y,X)\right]\right|^2 \leq C' \cdot \mathbb{P}_{\beta^*}\left(|\mathrm{v}^\top X| \geq R \cdot \sqrt{\log n}\right)$$

$$\leq C' \cdot \exp(-R^2 \log n), \qquad (D.42)$$

where $C'$ is a positive absolute constant, and the last inequality follows from the tail bound of standard Gaussian distribution in (C.54). Similar argument holds for the query functions $q_{2,\mathrm{v}}(Y,X)$ and $\widetilde{q}_{2,\mathrm{v}}(Y,X)$. We conclude from (D.40), (D.42) and a similar argument on $q_{2,\mathrm{v}}(Y,X)$ and $\widetilde{q}_{2,\mathrm{v}}(Y,X)$ that

$$\left|\mathbb{E}_{\mathbb{P}_{\beta^*}}\left[q_{1,\mathrm{v}}(Y,X) - \widetilde{q}_{1,\mathrm{v}}(Y,X)\right]\right| \vee \left|\mathbb{E}_{\mathbb{P}_0}\left[q_{1,\mathrm{v}}(Y,X) - \widetilde{q}_{1,\mathrm{v}}(Y,X)\right]\right| \leq 1/n,$$

$$\left|\mathbb{E}_{\mathbb{P}_{\beta^*}}\left[q_{2,\mathrm{v}}(Y,X) - \widetilde{q}_{2,\mathrm{v}}(Y,X)\right]\right| \vee \left|\mathbb{E}_{\mathbb{P}_0}\left[q_{2,\mathrm{v}}(Y,X) - \widetilde{q}_{2,\mathrm{v}}(Y,X)\right]\right| \leq 1/n, \qquad (D.43)$$

which holds for $\mathrm{v} \in \mathcal{N}(1/2)$, $\beta^* \in \mathcal{H}(s, \gamma_n)$, and sufficiently large $n$ and constant $R$. Note that under the null hypothesis, it holds that $X \sim N(0, I_d)$ and $Y$ is independent of $X$. Therefore, it follows from (D.37) that

$$\mathbb{E}_0\left[\widetilde{q}_{1,\mathrm{v}}(Y,X)\right] = \mathbb{E}_0\left[\widetilde{q}_{2,\mathrm{v}}(Y,X)\right] = 0, \qquad (D.44)$$

which holds for all $\mathrm{v} \in \mathcal{N}(1/2)$. Meanwhile, following from the definition of $\mathcal{N}(1/2)$ in (D.5), it holds that for any $\beta^* \in \mathcal{H}(s, \gamma_n)$, there exist a $\mathrm{v}^* \in \mathcal{N}(1/2)$ such that

$$\|\beta^*/\sqrt{s\rho^2} - \mathrm{v}^*\|_2^2 \leq 1/4,$$

which is equivalent to

$$\mathrm{v}^{*\top}\beta^* \geq 7/8 \cdot \sqrt{s\rho^2}. \qquad (D.45)$$

Therefore, following from (A.3) and (D.45), it holds that

$$49/64 \cdot s\rho^2 - \mathbb{E}_{\mathbb{P}_{\beta^*}}\left[\widetilde{q}_{1,\mathrm{v}^*}(Y,X)\right] \leq (\mathrm{v}^{*\top}\beta^*)^2 - \mathbb{E}_{\mathbb{P}_{\beta^*}}\left[\widetilde{q}_{1,\mathrm{v}^*}(Y,X)\right]$$

$$\leq \mathbb{E}_{\mathbb{P}_{\beta^*}}\left[\psi(Y) \cdot \left((\mathrm{v}^{*\top}X)^2 - 1\right) - \widetilde{q}_{1,\mathrm{v}}(Y,X)\right]$$

$$= \mathbb{E}_{\mathbb{P}_{\beta^*}}\left[\psi(Y) \cdot \left((\mathrm{v}^{*\top}X)^2 - 1\right) \cdot \mathbb{1}\left\{|\psi(Y)| > (R \cdot \log n)^{1/\nu}\right\}\right]$$

$$\leq \sqrt{\mathbb{E}_{\mathbb{P}_{\beta^*}}\left[\psi^2(Y) \cdot \left((\mathrm{v}^{*\top}X)^2 - 1\right)^2\right]} \cdot \sqrt{\mathbb{P}_{\beta^*}\left(|\psi(Y)| > (R \cdot \log n)^{1/\nu}\right)}, \qquad (D.46)$$

where the last inequality follows from the Cauchy-Schwartz inequality. It then follows from the Cauchy-Schwartz inequality and Assumption A.1 that

$$49/64 \cdot s\rho^2 - \mathbb{E}_{\mathbb{P}_{\beta^*}}\left[\widetilde{q}_{1,\mathrm{v}^*}(Y,X)\right] \leq C \cdot \exp(-R/2 \cdot \log n), \qquad (D.47)$$

where $C$ is a positive absolute constant. If it holds that $s\rho^2/\sigma^2 = \Omega(\sqrt{s\log d}/n)$, we obtain that for sufficiently large $n$ and constant $R$, it holds that $s\rho^2/64 > 1/n$ and

$$49/64 \cdot s\rho^2 - \mathbb{E}_{\mathbb{P}_{\beta^*}}\big[\widetilde{q}_{1,\mathrm{v}^*}(Y,X)\big] \leq 1/64 \cdot s\rho^2. \tag{D.48}$$

In other words, it holds that $\mathbb{E}_{\mathbb{P}_{\beta^*}}[\widetilde{q}_{1,\mathrm{v}^*}(Y,X)] \geq 3/4 \cdot s\rho^2$. Similarly, following from (A.4) and (D.45), we obtain

$$\begin{aligned}
7/8 \cdot \sqrt{\alpha^2 s\rho^2} - \mathbb{E}_{\mathbb{P}_{\beta^*}}\big[\widetilde{q}_{2,\mathrm{v}^*}(Y,X)\big] &\leq \alpha \cdot \mathrm{v}^{*\top}\beta^* - \mathbb{E}_{\mathbb{P}_{\beta^*}}\big[\widetilde{q}_{2,\mathrm{v}^*}(Y,X)\big] \\
&\leq \mathbb{E}_{\mathbb{P}_{\beta^*}}\big[Y \cdot (\mathrm{v}^{*\top}X) - \widetilde{q}_{1,\mathrm{v}}(Y,X)\big] \\
&= \mathbb{E}_{\mathbb{P}_{\beta^*}}\Big[Y \cdot (\mathrm{v}^{*\top}X) \cdot \mathbb{1}\big\{|Y| > (R \cdot \log n)^{1/\nu}\big\}\Big] \\
&\leq \sqrt{\mathbb{E}_{\mathbb{P}_{\beta^*}}\big[Y^2 \cdot (\mathrm{v}^{*\top}X)^2\big]} \cdot \sqrt{\mathbb{P}_{\beta^*}\big(|Y| > (R \cdot \log n)^{1/\nu}\big)}. \tag{D.49}
\end{aligned}$$

Then following from the Cauchy-Schwartz inequality and Assumption A.1, we obtain that

$$7/8 \cdot \sqrt{\alpha^2 s\rho^2} - \mathbb{E}_{\mathbb{P}_{\beta^*}}\big[\widetilde{q}_{2,\mathrm{v}^*}(Y,X)\big] \leq C' \cdot \exp(-R/2 \cdot \log n), \tag{D.50}$$

where $C'$ is a positive absolute constant. If it holds that $s\rho^2/\sigma^2 = \Omega(1/\alpha \cdot s\log d/n)$, we obtain that for sufficiently large $n$ and constant $R$, it holds that $\sqrt{\alpha^2 s\rho^2}/8 > 1/n$ and

$$7/8 \cdot \sqrt{\alpha^2 s\rho^2} - \mathbb{E}_{\mathbb{P}_{\beta^*}}\big[\widetilde{q}_{2,\mathrm{v}^*}(Y,X)\big] \leq 1/8 \cdot \sqrt{\alpha^2 s\rho^2}. \tag{D.51}$$

In other words, it holds that $\mathbb{E}_{\mathbb{P}_{\beta^*}}[\widetilde{q}_{2,\mathrm{v}^*}(Y,X)] \geq 3/4 \cdot \sqrt{\alpha^2 s\rho^2}$. Combining (D.43), (D.48), and (D.51), we conclude that for sufficiently large $n$ and constant $R$, it holds that

$$\mathbb{E}_{\mathbb{P}_0}\big[q_{1,\mathrm{v}}(Y,X)\big] \leq 1/n, \quad \mathbb{E}_{\mathbb{P}_0}\big[q_{2,\mathrm{v}}(Y,X)\big] \leq 1/n.$$

Furthermore, it holds for sufficiently large $n$ and constant $R$ that

$$\sup_{\mathrm{v}\in\mathcal{N}(1/2)} \mathbb{E}_{\mathbb{P}_{\beta^*}}\big[q_{1,\mathrm{v}}(Y,X)\big] \geq \mathbb{E}_{\mathbb{P}_{\beta^*}}\big[q_{1,\mathrm{v}^*}(Y,X)\big] \geq s\rho^2/2, \;\; \text{if } s\rho^2/\sigma^2 = \Omega(\sqrt{s\log d}/n),$$

$$\sup_{\mathrm{v}\in\mathcal{N}(1/2)} \mathbb{E}_{\mathbb{P}_{\beta^*}}\big[q_{2,\mathrm{v}}(Y,X)\big] \geq \mathbb{E}_{\mathbb{P}_{\beta^*}}\big[q_{2,\mathrm{v}^*}(Y,X)\big] \geq \sqrt{\alpha^2 s\rho^2}/2, \;\; \text{if } s\rho^2/\sigma^2 = \Omega(1/\alpha^2 \cdot s\log d/n),$$

which concludes the proof of Lemma D.3. $\qquad\square$

## D.4 Proof of Lemma D.4

*Proof.* In the following proof, we denote by $C$ and $C'$ absolute constants, the value of which may vary from lines to lines. We define the following query functions,

$$\widetilde{q}_{1,j}(Y,X) = \psi(Y) \cdot (X_j^2 - 1) \cdot \mathbb{1}\big\{|\psi(Y)| \leq (R \cdot \log n)^{1/\nu}\big\}, \quad j \in [d],$$

$$\widetilde{q}_{2,j}(Y,X) = Y X_j \cdot \mathbb{1}\big\{|Y| \leq (R \cdot \log n)^{1/\nu}\big\}, \quad j \in [d]. \tag{D.52}$$

Following from (D.13) and the Cauchy-Schwartz inequality, it holds that

$$\big|\mathbb{E}_{\mathbb{P}_0}\big[\widetilde{q}_{1,j}(Y,X) - q_{1,j}(Y,X)\big]\big|^2 \leq \mathbb{E}_{\mathbb{P}_0}\big[\psi^2(Y) \cdot (X_j^2 - 1)^2\big] \cdot \mathbb{P}_0\big(|X_j| \geq R \cdot \sqrt{\log n}\big). \tag{D.53}$$

Note that under the null hypothesis, $Y$ is independent of $X$ and $X \sim N(0, I_d)$. Then following from Assumption A.1 and the tail bound of standard Gaussian distribution in (C.54), it holds that

$$\big|\mathbb{E}_{\mathbb{P}_0}\big[\widetilde{q}_{1,j}(Y,X) - q_{1,j}(Y,X)\big]\big|^2 \leq C \cdot \exp(-R^2 \cdot \log n), \tag{D.54}$$

where $C$ is a positive absolute constant. Under the alternative hypothesis, it holds that

$$\big|\mathbb{E}_{\mathbb{P}_{\beta^*}}\big[\widetilde{q}_{1,j}(Y,X) - q_{1,j}(Y,X)\big]\big|^2 \leq \mathbb{E}_{\mathbb{P}_{\beta^*}}\big[\psi^2(Y) \cdot (X_j^2 - 1)^2\big] \cdot \mathbb{P}_{\beta^*}\big(|X_j| \geq R \cdot \sqrt{\log n}\big) \tag{D.55}$$

$$\leq \sqrt{\mathbb{E}_{\mathbb{P}_{\beta^*}}\big[\psi^4(Y)\big] \cdot \mathbb{E}_{\mathbb{P}_{\beta^*}}\big[(X_j^2 - 1)^4\big]} \cdot \mathbb{P}_{\beta^*}\big(|X_j| \geq R \cdot \sqrt{\log n}\big),$$

where the above inequalities follows from the Cauchy-Schwartz inequality. Note that under the alternative hypothesis, we have $X \sim N(0, I_d)$. Then following from Assumption A.1 and the tail bound of standard Gaussian distribution in (C.54), it holds that

$$\big|\mathbb{E}_{\mathbb{P}_{\beta^*}}\big[\widetilde{q}_{1,j}(Y,X) - q_{1,j}(Y,X)\big]\big|^2 \leq C' \cdot \exp(-R^2 \cdot \log n), \tag{D.56}$$

where $C'$ is a positive absolute constant. Similar argument holds for $q_{2,j}(Y,X)$. Combining (D.54), (D.56), and a similar argument on $q_{2,j}(Y,X)$, we obtain that

$$\left|\mathbb{E}_{\mathbb{P}_0}\big[q_{1,j}(Y,X) - \widetilde{q}_{1,j}(Y,X)\big]\right| \vee \left|\mathbb{E}_{\mathbb{P}_\beta^*}\big[q_{1,j}(Y,X) - \widetilde{q}_{1,j}(Y,X)\big]\right| \leq 1/n,$$

$$\left|\mathbb{E}_{\mathbb{P}_0}\big[q_{2,j}(Y,X) - \widetilde{q}_{2,j}(Y,X)\big]\right| \vee \left|\mathbb{E}_{\mathbb{P}_\beta^*}\big[q_{2,j}(Y,X) - \widetilde{q}_{2,j}(Y,X)\big]\right| \leq 1/n, \tag{D.57}$$

which holds for $j \in [d]$, $\beta^* \in \mathcal{H}(s, \gamma_n)$, and sufficiently large $n$ and constant $R$. Note that under the null hypothesis, it holds that $X \sim N(0, I_d)$ and $Y$ is independent of $X$. Therefore, following from (D.52), we obtain

$$\mathbb{E}_{\mathbb{P}_0}\big[\widetilde{q}_{1,j}(Y,X)\big] = \mathbb{E}_{\mathbb{P}_0}\big[\widetilde{q}_{2,j}(Y,X)\big] = 0. \tag{D.58}$$

Meanwhile, under the alternative hypothesis, it follows from (A.3) that

$$\beta_j^{*2} - \mathbb{E}_{\mathbb{P}_{\beta^*}}\big[\widetilde{q}_{1,j}(Y,X)\big]$$
$$\leq \mathbb{E}_{\mathbb{P}_{\beta^*}}\Big[\psi(Y) \cdot (X_j^2 - 1) \cdot \mathbb{1}\big\{|\psi(Y)| > (R \cdot \log n)^{1/\nu}\big\}\Big]$$
$$\leq \sqrt{\mathbb{E}_{\mathbb{P}_{\beta^*}}\big[\psi^2(Y) \cdot (X_j^2 - 1)^2\big]} \cdot \sqrt{\mathbb{P}_{\beta^*}\big(|\psi(Y)| > (R \cdot \log n)^{1/\nu}\big)}$$
$$\leq \Big(\mathbb{E}_{\mathbb{P}_{\beta^*}}\big[\psi^4(Y)\big] \cdot \mathbb{E}_{\mathbb{P}_{\beta^*}}\big[(X_j^2 - 1)^4\big]\Big)^{1/4} \cdot \sqrt{\mathbb{P}_{\beta^*}\big(|\psi(Y)| > (R \cdot \log n)^{1/\nu}\big)}, \tag{D.59}$$

where we denote by $\beta_j^*$ the $j$-th entry of $\beta^*$, and the above inequalities follow from the Cauchy-Schwartz inequality. Then following from Assumption A.1 and the fact that $X \sim N(0, I_d)$ under the alternative hypothesis, we obtain that

$$\beta^*{}_j^2 - \mathbb{E}_{\mathbb{P}_{\beta^*}}\big[\widetilde{q}_{1,j}(Y,X)\big] \leq C \cdot \exp(-R/2 \cdot \log n), \tag{D.60}$$

where $C$ is a positive absolute constant. Note that $\|\beta^*\|_2^2 = s\rho^2$ and $\|\beta^*\|_0 = s$. Therefore, we obtain that

$$\sup_{j \in [d]} |\beta_j^*| \geq \rho. \tag{D.61}$$

Following from (D.60) and (D.61), if it holds that $s\rho^2/\sigma^2 = \Omega(\sqrt{s^2 \log d/n})$, then for sufficiently large $n$ and constant $R$, we obtain that $\rho^2/4 > 1/n$ and

$$\sup_{j \in [d]} \mathbb{E}_{\mathbb{P}_{\beta^*}}\big[\widetilde{q}_{1,j}(Y,X)\big] \geq 3\rho^2/4. \tag{D.62}$$

Similar argument holds for $\widetilde{q}_{2,j}(Y,X)$. Following from (A.4), we obtain that under the alternative hypothesis, it holds that

$$\alpha\beta_j^* - \mathbb{E}_{\mathbb{P}_{\beta^*}}\big[\widetilde{q}_{2,j}(Y,X)\big] = \mathbb{E}_{\mathbb{P}_{\beta^*}}\Big[\psi(Y) \cdot X_j \cdot \mathbb{1}\big\{|\psi(Y)| > (R \cdot \log n)^{1/\nu}\big\}\Big]. \tag{D.63}$$

Meanwhile, it follows from the Cauchy-Schwartz inequality that

$$\left|\mathbb{E}_{\mathbb{P}_{\beta^*}}\Big[Y \cdot X_j \cdot \mathbb{1}\big\{|Y| > (R \cdot \log n)^{1/\nu}\big\}\Big]\right|^2 \leq \mathbb{E}_{\mathbb{P}_{\beta^*}}[Y^2 \cdot X_j^2] \cdot \mathbb{P}_{\beta^*}\big(|Y| > (R \cdot \log n)^{1/\nu}\big)$$
$$\leq \sqrt{\mathbb{E}_{\mathbb{P}_{\beta^*}}[Y^4] \cdot \mathbb{E}_{\mathbb{P}_{\beta^*}}[X_j^4]} \cdot \mathbb{P}_{\beta^*}\big(|Y| > (R \cdot \log n)^{1/\nu}\big)$$
$$\leq C' \cdot \exp(-R \log n), \tag{D.64}$$

where the last inequality follows from Assumption A.1 and the fact that $X \sim N(0, I_d)$ under the alternative hypothesis. Combining (D.61), (D.63), and (D.64), we obtain that for $s\rho^2/\sigma^2 = \Omega(1/\alpha^2 \cdot s \log d/n)$, it holds for sufficiently large $n$ and constant $R$ that $\alpha\rho/4 > 1/n$ and

$$\sup_{j \in [d]} \left|\mathbb{E}_{\mathbb{P}_{\beta^*}}\big[\widetilde{q}_{2,j}(Y,X)\big]\right| \geq 3\alpha\rho/4. \tag{D.65}$$

Combining (D.57), (D.62), and (D.65), we obtain that for sufficiently large $n$ and constant $R$, it holds that

$$\sup_{j \in [d]} \mathbb{E}_{\mathbb{P}_0}\big[q_{1,j}(Y,X)\big] \leq 1/n, \quad \sup_{j \in [d]} \mathbb{E}_{\mathbb{P}_0}\big[q_{1,j}(Y,X)\big] \leq 1/n. \tag{D.66}$$

Moreover, for sufficiently large $n$ and constant $R$, it holds that

$$\sup_{j \in [d]} \mathbb{E}_{\mathbb{P}_{\beta^*}}\big[q_{1,j}(Y,X)\big] \geq \rho^2/2 \ \text{ if } s\rho^2/\sigma^2 = \Omega(\sqrt{s \log d/n}),$$

$$\sup_{j \in [d]} \mathbb{E}_{\mathbb{P}_{\beta^*}}\big[q_{2,j}(Y,X)\big] \geq \alpha\rho/2 \ \text{ if } s\rho^2/\sigma^2 = \Omega(1/\alpha^2 \cdot s \log d/n), \tag{D.67}$$

which concludes the proof of Lemma D.4. $\qquad\square$