[Reviews · NeurIPS 2019]

Reviewer 1



1. The computational-statistical tradeoff shown in this paper is novel, which seems a significant contribution to understanding the single index models. 2. The paper is highly technical, and it is difficult to understand every single piece of the proofs. Therefore for the main paper, I would like to see a more gentle description of model (2.6) and the associated hypothesis testing problem. In the current flow of the paper, it is hard to gain much intuition into why we are interested in such a model and how the lower bounds of the hypothesis testing can imply the lower bound for parameter estimation.

Reviewer 2



The paper first introduces first-order and second-order Stein's identity and then defines two function sets, C1 and C2, characterized by the covariance between f and X^T\beta^*. Further, authors define a common function set C(psi), which includes all link functions such that the second-order Stein's identity does not vanish under transformation psi. Then, authors propose a mixed model in 2.6 using two link functions f1\in C1\cap C(\psi) and f2\in C2\cap C(\psi). This model is finally used to derive lower bound. This is reasonable since true beta with link function f1 is easy to estimate (using first-order Stein's identity), while true beta with f2 is indistinguishable. The minimax rate is established in Prop 3.1. Analogously, the paper also discusses the computational minimax rate via oracle computational model in the second part. The paper is not in a good quality due to many typos. But my main concern is about the significance of this paper and contribution. 1, Stein's association in 2.1 is more like a proof technique, which is used to define the function sets and further define the mixed model. The usefulness and necessity of a general transformation psi are not clear. One only need very specific psi and also specific f1 and f2, e.g. psi(y) = y^2, to derive lower bound. It would be better if authors discuss the importance of generalization of psi in Def 2.1, otherwise the this directly comes from [42]. 2, Why we need normalization constant \|beta^*\| in C1, instead of directly assuming \|beta^*\| = 1 (the constant can be absorbed in f). 3, It seems the mixed model defined in 148-150 is closely related with [16]. Authors should distinguish this work from [16]. What's the main difference (and difficulty) for using this mixed model to derive lower bound. Are these two works basically solving the same problem? 4, In the main paper, authors emphasize the lower bound only, without upper bound. So writing the contribution paragraph need be careful. It seems the upper bound has different conditions as lower bound (A.2). It would be better to clarify this in main context. 5, Comparing with proof in [16, 53, 62], I don't get the significance of two lower bounds in paper, which I thought are main contributions. Is the proof same as [62]. Although we have index model here but we can always choose specific link function when showing lower bounds. Thus, as my understanding, the problem will reduce to mixed linear regression, studied in those related work. 6, minor things: 23: has--> have 84: second method --> The second method 98: estimate --> estimates 103: Stein's identity comes from Stein's work, not 58, 59. 148: redefine f1

Reviewer 3



This reads a long paper that is shortened to fit the 8-page limit of NeuIPS. Such manipulation hurts the reading pace a lot since many materials are moved to supplement, and discussions are not thorough. However, I overall still find it an interesting read. My comments are minor in this regard: (1) It should be highlighted throughout that you are studying a simple single index model with a design known to the practitioner (BTW, is "design known" equivalent to saying "design known to be standard multivariate normal"?). (2) For the main results, I noticed that only lower bounds are discussed, and the matching upper bounds are moved to the supplement? I recommend the authors to at least comment in Section 3 that these lower bounds can be matched by the corresponding upper bounds, and point the readers to the exact locations where these results are put.

Reviewer 4



The introduction is very well written, with a clear presentation of the topic. However, I think that the descriptions of the statistical model and the important definitions are rather unclear (and with many typos). In general, I think that even though this paper should be very relevant to the ML community, its presentation should be revised by the authors: Some of the concepts should be described more precisely for the readers who are unfamiliar with them; The main ideas of the proofs should be explained in the main text; The reasons for the introduction of the first and second order Stein operators are not very clear in the main text; etc. Some more specific comments are given below. I have not read all the proofs, but I have not found any major issue. 1. Abstract, line 8: A star is missing twice on \beta. 2. Line 23, has -> have 3. Line 45: Some words seem to be missing... 4. Line 66: has -> have 5. Line 107: I could not understand what the authors meant when explaining that these quantities are pivotal. Actually, to me, the whole paragraph from line 107 to line 113 is really unclear. Also, the epsilon should be outside of f, in all occurrences. 6. Line 125: I do not understand the introduction of the functions \psi. The paragraph in lines 136-143 is really unclear to me. The choice of \psi does not determine f, so I do not understand the use of the words “reduces” and “characterizes” in lines 136-137. Moreover, in the definition of the minimax risk (et. 2.10), there is a supremum over \psi (on which class?), so \psi does not seem to be fixed. Hence, can the results be used, e.g., for the phase retrieval problem? 7. Line 148, just write f_1(u)=u^2+u and f_2(u)=u^2 (or else, use \beta or \beta^* consistently). 8. The notation for the minimax risk in Eq 2.10 is strange, there is a supremum over f_1, f_2, \psi, of a quantity that does not seem to depend on these functions... 9. Line 180: Remove “is” 10. Line 182: Remove the ‘s’ of “functions” 11. Definition 2.3 and the next paragraph: What exactly is r? For somebody who is not used to this terminology (such as “statistical oracle”), the definition is not very clear. It could be useful to provide some examples, maybe. Moreover, the choice of \tau_q seems arbitrary: Does it mean that one allows the algorithm to only make queries that yield sub-Gaussian errors? Why make this restriction in your model? 12. Line 249, replace T with d^\mu 13. I think that the whole paragraph 3.2 could be replaced with a table, which would be much more readable. And Figure 1 is not informative at all. 14. Could the authors discuss in more details the choice of their mixture model (eq. 2.6)? This is not what one usually thinks of a single index model, since here the function f is random (it randomly takes the values f_1 or f_2). 15. I think that the whole paragraph 3.3 could be removed. Maybe one short comment could be made about the estimation lower bound, but estimation requires many more assumptions (just regarding the identifiability issues), so I do not think that this paragraph adds anything deeply relevant to the paper. Moreover, I think that the authors should make some space in the paper in order to address the issues that I pointed out above.

[Author Response · NeurIPS 2019]

We appreciate the valuable comments from reviewers on paper presentation and typos. We will revise our work accordingly. In what follows, we first address some common concerns as follows.

(Intuition Behind the Model Design. ) Our model is a mixture of two single index models with $\alpha$ as the mixing probability. The intuition behind the design is to interpolate between two classes of the link functions, which exploit the first- and second-order Stein's identities for the estimation of $\beta^*$, respectively. Such a model design allows us to control the magnitude of first-order Stein's identity by the mixing probability $\alpha$ and study the effect of the first-order Stein's identity to the recovery of $\beta^*$.

(Intuition Behind the Function Class $\mathcal{C}(\psi)$. ) The function class $\mathcal{C}(\psi)$ includes all the link functions with nonzero second-order Stein's identity under the marginal transformation $\psi$. Our work studies the minimax separation rate, which corresponds to the minimum signal to noise ratio (SNR) needed for the best algorithm when applying to the hardest model. Here $\psi$ is one of the parameters that specify the model. Hence, by introducing the function class $\mathcal{C}(\psi)$ with $\psi$ being arbitrary (yet smooth), we investigate a large family of models when searching for the hardest model.

**Reviewer 1**

(1). (Intuition Behind Our Model. ) Please refer to the intuition behind the model design in the common concerns.

(2). (Lower Bound For Parameter Estimation Implied By Testing. ) Intuitively, an algorithm that estimates $\beta^*$ with sufficient accuracy can also be used to test nonzero $\beta^*$ if the SNR is sufficiently large. Hence, by investigating the lower bound in the testing of nonzero $\beta^*$, we can characterize the minimal possible statistical error in the estimation of $\beta^*$. We refer to §3.3 for a detailed investigation of such an intuition.

**Reviewer 2** (1). (Specify $\psi$, $f_1$, and $f_2$? ) We refer to common concerns for the intuition behind $\psi$. We highlight that our goals is to characterizes the minimax separation rate, which corresponds to the minimal SNR needed for the best algorithm in solving the hardest model. Fixing arbitrary $f_1$, $f_2$, and $\psi$ does not suffice, as they may not corresponds to the hardest model. In specific, we derive an upper bound that matches the lower bound we obtained by our selection of the parameters $f_1$, $f_2$, and $\psi$. Our work does not follow [2]. In specific, we study the phase transition in the minimax separation rate corresponding to the first-order Stein's association, whereas [2] studies a misspecified phase retrieval algorithm that estimates $\beta^*$ via solely exploiting the second-order Stein's association, and their technique does not apply to our work. Moreover, the effect of computational tractability on the sample complexity is not studied in [2], which is a major concern of our work.

2. (Normalizing $\beta^*$? ) Our work investigates the minimax separation rate, which corresponds to the minimum SNR needed for the hardest recovery of $\beta^*$. In specific, our work fixes the scale of link functions and the noise, and characterize the hardness of the recovery of $\beta^*$ by the SNR $\|\beta^*\|_2^2/\sigma^2$. Hence we cannot normalize the quantity $\|\beta^*\|_2^2$, as such a quantity characterizes the hardness of recovering $\beta^*$, which is a major concern of our work.

(3). (Similarity to [1]? ) Our model is a single index model where the link function is unknown, whereas [1] study the mixed linear regression model. They also show that phase retrieval model with link function $f(u) = |u|$ can be reduced to mixed linear regression and thus obtain the lower bound by resorting to the hardness of mixed linear regression. Thus, their detour approach cannot be used for showing the computational barriers in general single index models. Instead, our results cover their phase retrieval model as a special case.

(4).(Similarity to [3] and [4]? ) Our work studies the statistical-computational tradeoffs in the single index model, whereas [3] study mean detection problems only. Moreover, [4] study the binary classification problem with the corrupted label. The model we consider are more complicated than their models and thus their analysis does not apply.

(5).(Technical Challenges.) Compared with these related work, we consider a larger model class. Our lower bound requires careful analysis of the $\chi_2$-divergence the interpolated model introduced in (B.8). Moreover, our upper bounds holds for general single index models are thus handling the unknown nonlinear function $f$.

**Reviewer 4** We thank the reviewer for the numerous advice in the layout and content of our work. We will revise accordingly. We move the results on upper bounds to the appendix due to the page limit. We will briefly introduce the upper bounds in the revision.

**Reviewer 5** (1). (Intuition Behind The Function $\psi$. ) Please refer to intuition behind the function class $\mathcal{C}(\psi)$ in the common concerns. Our result applies to the function class $\mathcal{C}(\psi)$, which contains the link functions that correspond to a large family of phase retrieval problems.

(2). (Explanation Of The Statistical Oracle. ) Under the statistical query model, the sample complexity $n$ becomes a parameter of the statistical oracle, and the sub-Gaussian error in the response of a query function is a natural extension of $n$ independent realizations of the query function.

[1] Fan et al. " Curse of heterogeneity: Computational barriers in sparse mixture models and phase retrieval" (2018).

[2] Neykov et al. "Agnostic estimation for misspecified phase retrieval models" (2016).

[3] Wang et al. "Sharp computational-statistical phase transitions via oracle computational model" (2015).

[4] Yi et al. "More supervision, less computation: Statistical-computational tradeoffs in weakly supervised learning"


[Meta-Review · NeurIPS 2019]

Congratulations! The reviewers mostly recommended acceptance but made several suggestions in their reviews. It would also be helpful to further clarify precisely the contributions in the computational lower bound relative to known SQ lower bounds (for sparse phase-retrieval).